# Basis functions for complex social decisions in dorsomedial frontal cortex

Marco K. Wittmann[1,2,3 ✉], Yongling Lin[1,4], Deng Pan[3], Moritz N. Braun[3,5], Cormac Dickson[2], Lisa Spiering[3], Shuyi Luo[3], Caroline Harbison[3], Ayat Abdurahman[3], Sorcha Hamilton[3,6,7], Nadira S. Faber[3,8,9], Nima Khalighinejad[3], Patricia L. Lockwood[3,10,11] & Matthew F. S. Rushworth[3,12]

Navigating social environments is a fundamental challenge for the brain. It has been established that the brain solves this problem, in part, by representing social information in an agent-centric manner; knowledge about others' abilities or attitudes is tagged to individuals such as 'oneself' or the 'other'[1–6]. This intuitive approach has informed the understanding of key nodes in the social parts of the brain, the dorsomedial prefrontal cortex (dmPFC) and the anterior cingulate cortex (ACC)[7–9]. However, the patterns or combinations in which individuals might interact with one another is as important as the identities of the individuals. Here, in four studies using functional magnetic resonance imaging, behavioural experiments and a social group decision-making task, we show that the dmPFC and ACC represent the combinatorial possibilities for social interaction afforded by a given situation, and that they do so in a compressed format resembling the basis functions used in spatial, visual and motor domains[10–12]. The basis functions align with social interaction types, as opposed to individual identities. Our results indicate that there are deep analogies between abstract neural coding schemes in the visual and motor domain and the construction of our sense of social identity.

Social environments routinely comprise multiple people[13,14], who have an exponential number of relationships between them, and these are typically characterized by several distinct social interactions in quick succession, such as when people take turns in a group discussion. Humans have a ubiquitous sense of themselves as living in just such social environments, in which patterns of cooperation and competition occur simultaneously and change frequently, for example at work, playing sport or in global politics[14]. In these contexts, agent-centric coding, which requires tracking of all identities and relationships, becomes computationally demanding, so a more flexible solution is needed[15].

Encoding the potential combinatorial patterns afforded by a given situation may constitute the flexible solution that is needed. Combinatorial patterns are widely used in sensory, motor and spatial domains. For example, single neurons in the pre-supplementary motor area fire when a macaque prepares a movement, but only when the movement comprises a particular sequence of elements in a particular order, such as push→pull→turn[11]. The neurons are silent when single elements of this sequence are prepared or when the elements are executed in a different order. More generally, the planning of motor sequences and perception of sequences of visual cues are accompanied by abstract coding of sequential positions (first cue, second cue and so on), independent of specific stimuli and specific actions[16,17]. Computational theories indicate that such combinatorial and sequential codes are abstractions from sensory information that can act as a scaffold for the acquisition of new information[12,18].

This concept of combinatorial patterns in neural coding aligns with, and extends to, the idea of basis functions. For example, it has been suggested that to solve the problem of visually guided motor actions, parietal neurons compress multiple input variables (such as retinal position, eye position and object orientation) into single neuronal responses[19]. These compressed responses can be understood as basis functions that can flexibly be combined in a linear manner to guide motor actions[19]. Basis functions can effectively reduce the dimensionality by representing a limited set of feature dimensions that efficiently summarize a behavioural repertoire or task[12]. In the motor domain, action sequences are represented in such multidimensional neural spaces defined by basis functions[20,21]. Similarly, basis functions explain face processing in the macaque ventral stream, in which neural signals vary along the axes of a multidimensional space defined by the basis functions, but not along 'null' axes[10].

Here we demonstrate that an analogous, limited set of basis functions is computed in dmPFC and the adjacent ACC during social cognition

[1]Department of Experimental Psychology, University College London, London, UK. [2]Max Planck UCL Centre for Computational Psychiatry and Ageing Research, University College London, London, UK. [3]Wellcome Integrative Neuroimaging (WIN), Department of Experimental Psychology, University of Oxford, Oxford, UK. [4]State Key Laboratory of Cognitive Neuroscience and Learning, Faculty of Psychology, Beijing Normal University, Beijing, China. [5]Department of Psychology, Saarland University, Saarbrücken, Germany. [6]Department of Psychiatry, University of Oxford, Oxford, UK. [7]Department of Psychology, University of Bath, Bath, UK. [8]Department of Psychology, University of Bremen, Bremen, Germany. [9]Uehiro Oxford Institute, University of Oxford, Oxford, UK. [10]Centre for Human Brain Health, School of Psychology, University of Birmingham, Birmingham, UK. [11]Institute for Mental Health, School of Psychology, University of Birmingham, Birmingham, UK. [12]Wellcome Integrative Neuroimaging (WIN), Centre for Functional MRI of the Brain (MRI), Nuffield Department of Clinical Neurosciences, John Radcliffe Hospital, University of Oxford, Oxford, UK. ✉e-mail: m.wittmann@ucl.ac.uk

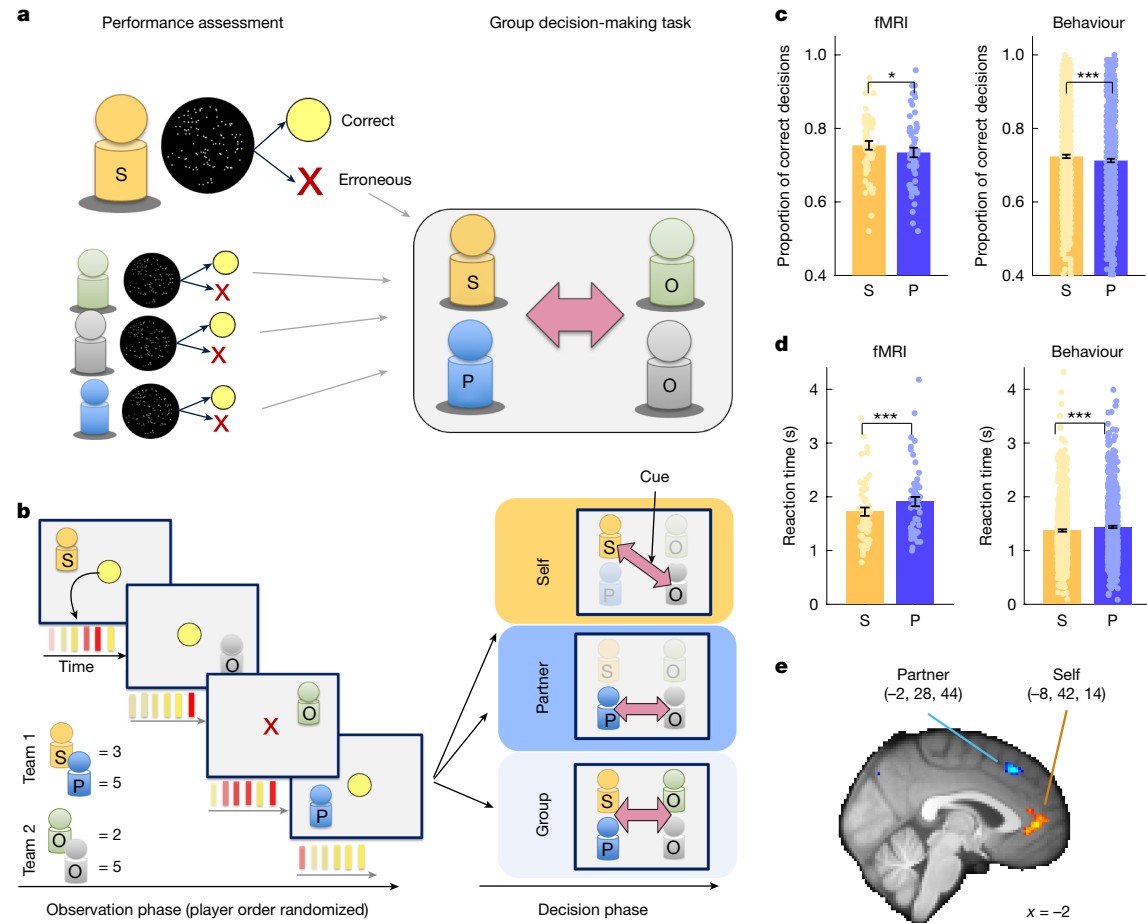

**Fig. 1 | Group decision task and agent-centric representations. a**, We collected performance data from participants in a perceptual task (left). During the subsequent fMRI group decision-making experiment (right), participants' veridical perceptual performance data were presented to them alongside the veridical performances of three other previous participants, grouped in two teams, one comprising self (S) and partner (P), and the other having two opponents, O. **b**, Simplified trial timeline. Successful (yellow) and erroneous (red) performances from the pre-experiment were shown in the trial's initial observation phase (lasting about 8 s) for all four players in random order. In the decision phase (lasting about 3 s), participants made one of three types of binary decision depending on the cued players (S versus O1 or O2; partner versus O1 or O2; or S + P versus O1 + O2 for group decisions). Self, partner and group decisions were equally frequent. In the example, the correct self decision is to indicate that the grey opponent performed better than oneself. Each trial comprised two subsequent decisions relating to the same observation phase. **c**,**d**, Participants were more accurate in self decisions than in partner decisions (**c**) (study 1, $P = 0.041$; study 2, $P < 0.001$) and also responded faster (**d**) (both $P < 0.001$). This result from our fMRI experiment was replicated in an independent behavioural experiment. **e**, In self decisions, the perigenual ACC represented the performance of oneself (parametric regressor indicating trial-wise performance scores for self), whereas a more dorsomedial region represented partner performance in partner decisions, replicating previous results on self/other coding in the prefrontal cortex (peak MNI coordinates of the effects are in brackets). Both signals were positively signed and extended bilaterally. Sample sizes: $n = 56$ for the social fMRI experiment (study 1); $n = 795$ for the behavioural experiment (study 2); error bars show s.e.m.; two-sided paired $t$-tests were used in **c** and **d**; all MRI results cluster are family-wise error (FWE) corrected at $Z > 3.1$, $P = 0.05$; *$P < 0.05$, ***$P < 0.001$.

(study 1, a social functional magnetic resonance imaging (fMRI) experiment). The basis functions summarize the possible social interactions in a compressed format tailored to the decision task at hand. As such, basis functions are specific to the decision problems that are expected in a given context. Such a representational format provides a flexible basis for constantly changing the multi-agent interactions that humans and other group-living animals must negotiate[7,22,23]. In accordance with common ideas about social interaction and its neural representation, however, we show that the basis functions are referenced to one's own place in the social world[24–26]. Basis functions can then be used to construct estimates of the values of cooperation/competition choices in an adjacent brain area, the ventromedial prefrontal cortex (vmPFC), which is important for decision making[27]. We show that the basis functions evident in neural activity leave several observable behavioural traces (study 2: a behavioural experiment) and that they might be relevant for solving computational challenges that share a similar structure in the non-social domain (study 3: a control fMRI experiment).

We devised a multi-person social decision-making task to test whether the brain uses a combinatorial or an agent-centric frame of reference to guide social decisions (Fig. 1 and Extended Data Fig. 1). To this end, in each trial we presented information in a randomized order about four players[6,25,28–30] whom participants subsequently made decisions about. The scenario approximately corresponds to real-life situations in which it is clear that several people are potentially relevant, but exactly which decision will need to be made about which subset of individuals only becomes clear later. This allowed us to disentangle the neural signals related to player identity and to player position in the sequence. The presented information comprised abstract performance scores that ranged from 0 to 6 for each player. To pre-empt the use of simple heuristics, we presented performance scores as brief sequences of six binary cues per player (indicating either successful performance or erroneous performance) that participants aggregated. The performance scores reflected veridical task performances by each player and were recorded before scanning in an arbitrary two-alternative

forced-choice task (Fig. 1a,b and Extended Data Fig. 1; note that the task also comprised a non-social bonus that we controlled for in all analyses). The experimental schedule was constructed from the pre-recorded performance using a procedure that ensured that insight into one's pre-recorded performance was not helpful for correctly responding in the main experiment (Extended Data Fig. 2). We imposed a specific social structure by assigning players to two groups. The participant (self, S) and a designated partner (P) constituted one group, and two opponents (O1 and O2) constituted another group. Cues presented after the performance sequence instructed participants to make a dyadic comparison of performances of two individuals from each group (either S versus one of the opponents (self decisions) or P versus one of the opponents (partner decisions)) or to compare the two groups (S + P versus O1 + O2 (group decisions); Fig. 1b). During the binary comparison of relevant players, participants decided whether their own team had performed better (that is, making an 'engage' choice) or whether the opponent team had performed better (that is, making an 'avoid' choice). Participants obtained rewards for correct engage choices proportional to the true team difference. By contrast, avoid choices always resulted in a zero outcome and were useful to avoid losses proportional to the true team difference when the other team had performed better. This asymmetric pay-off scheme incentivised accurate responding while ensuring that the participants favoured their own over the opponent team (Methods). Each trial comprised two different decisions relating to the same prior observation phase. The rapid presentation of performances during the observation phase (0.2 s duration per performance cue) hindered optimal performance and required participants to encode information efficiently to respond accurately in each possible subsequent decision type.

We considered two ways in which the brain could solve this task. First, participants could use an agent-centric frame of reference by simply remembering the performance scores tagged to each player and discarding any sequential information about the order in which the performances were presented. However, people could also use a sequential frame of reference by representing performance as linked to the sequential position when it was shown (irrespective of identity). We found that people do both.

First, in an agent-centric frame of reference, performance information was linked to four players with identities that were a function of their relationship to the participant. Behaviour and neural signals revealed signatures of agent-centric representations. Participants made decisions about their own performances more accurately ($t_{55} = 2.094$, $P = 0.041$; Fig. 1c) than decisions about their partner's performances, and they made these decisions more quickly for themselves than they did for the partner ($t_{55} = 5.299$, $P < 0.001$; Fig. 1d). This was true even though self and partner exhibited comparable performance in the observation phase, were equally often decision relevant and had the same influence on reward pay-off. We replicated both effects in a follow-up behavioural study (study 2; accuracy, $t_{794} = 4.369$, $P < 0.001$; reaction time, $t_{794} = 10.447$, $P < 0.001$; Fig. 1c,d). Further decision analyses indicated that participants performed similarly, if not better, in group decisions than dyadic ones (Extended Data Fig. 3). This is difficult to explain for agent-centric decision models that assume that decision errors accumulate the more elements there are to be considered. However, it is consistent with our model that assumes that decisions are guided by combinatorial representations, such as the overall difference in performance between groups.

Moreover, in line with an agent-centric perspective, neural data revealed a previously characterized distinction of self-related and other-related activity in the medial prefrontal cortex in which perigenual ACC and dmPFC activity reflected self and partner performances, respectively[6,9,31] (Fig. 1e). These signals were apparent during the second decision phase at the end of each trial. Region of interest (ROI) analyses also revealed opponent-related signals in the dmPFC (Extended Data Fig. 4).

However, by going beyond a two-person paradigm and examining the social representations underlying multi-person group situations[13,32], we were able to investigate whether the brain exclusively entertains agent-centric representations or whether it also relies on identity-independent, sequential representations tailored to the specific requirements of the task. A critical design feature allowed us to dissociate the two formats of representation: player positions during the observation phase (position 1, position 2 and so on) were independent of player identity (self, partner and so on; Fig. 2a). Because all sequence positions are occupied by each player equally often and in every possible sequence combination, it was possible to analyse signals related to sequence position, or our sequential basis functions, irrespective of player identity.

This meant that decisions could not be analysed simply by comparing performances in different player identities (the agent-centric frame of reference), but also by comparing performances from different points in the sequence (the sequential frame of reference; Fig. 2b). The former perspective meant that all decision-relevant information was represented along four axes, each reflecting the parametric performance score of one specific player (Fig. 2c). Knowing the point defined along the four axes (for example, self-score of 3, green opponent score of 2) meant knowing the decision variables (DVs) for identity comparisons (for example, self was 1 better than the green opponent). Analogously, the sequential perspective entailed a 4D representation along the axes of the four sequential positions, in which the length of each axis reflects the performance for the player occupying this sequence position, independent of identity (Fig. 2d). We confirmed that sequential expectations did indeed affect decisions in our paradigm in a large behavioural study conducted online (study 4; Extended Data Fig. 5).

Critically, all sequential DVs lay in a 3D subspace of the 4D sequential performance space (Fig. 2e). This was a consequence of the team structure and the decisions of our task. The subspace was defined by three basis functions. The four sequential performance scores could be projected onto the three basis functions through a linear combination (dot product) of the performance scores and the basis functions. For example, one basis function, $\mathbf{w}_2 = [1\ 1\ -1\ -1]$, weights the first two performances positively and the last two negatively, and combining $\mathbf{w}_2$ with the performance sequence results in the performance difference between these pairs of positions (Fig. 2f). The subspace was independent of a null axis[10] that reflected variation in performance scores that was irrelevant for the task and not useful for responding to any decision that could be encountered. The null axis reflected the sum of performances across players. Knowledge about the absolute sum was irrelevant in our task because all decisions involved relative comparisons between a member or members of one team versus the other. The three comparison vectors for the sequential group decisions (Fig. 2f) formed an ideal set of basis functions for the transformation into the subspace: Pairwise, linear combinations of these vectors, which specify all possible dyadic decision weight vectors, when combined with the sequential performance scores, can guide any decisions about dyadic interactions (Fig. 2g). Just one of the three basis functions is sufficient, when combined with the sequential performance scores, to guide decisions about any groupwise interaction that may occur.

In simple terms, in each trial, projecting performance scores onto the basis functions results in three numbers indicating specific differences between pairs of performances. For example, $b_2 = 1$ in Fig. 2 relates to $\mathbf{w}_2$ and indicates that the combined performances of the first two players was one point better than the combined performances of players three and four. Similar to neurons encoding not just a single movement but a particular combination of movement elements in a sequence[11,33], basis functions organize information along axes that are not linked to a single sequential element, but to a weighted pattern of several of them; activity along $\mathbf{w}_2$ is highest when the two players at either end of the sequence performed maximally differently. The basis function projection $b_2 = 1$ directly maps onto the DV when the

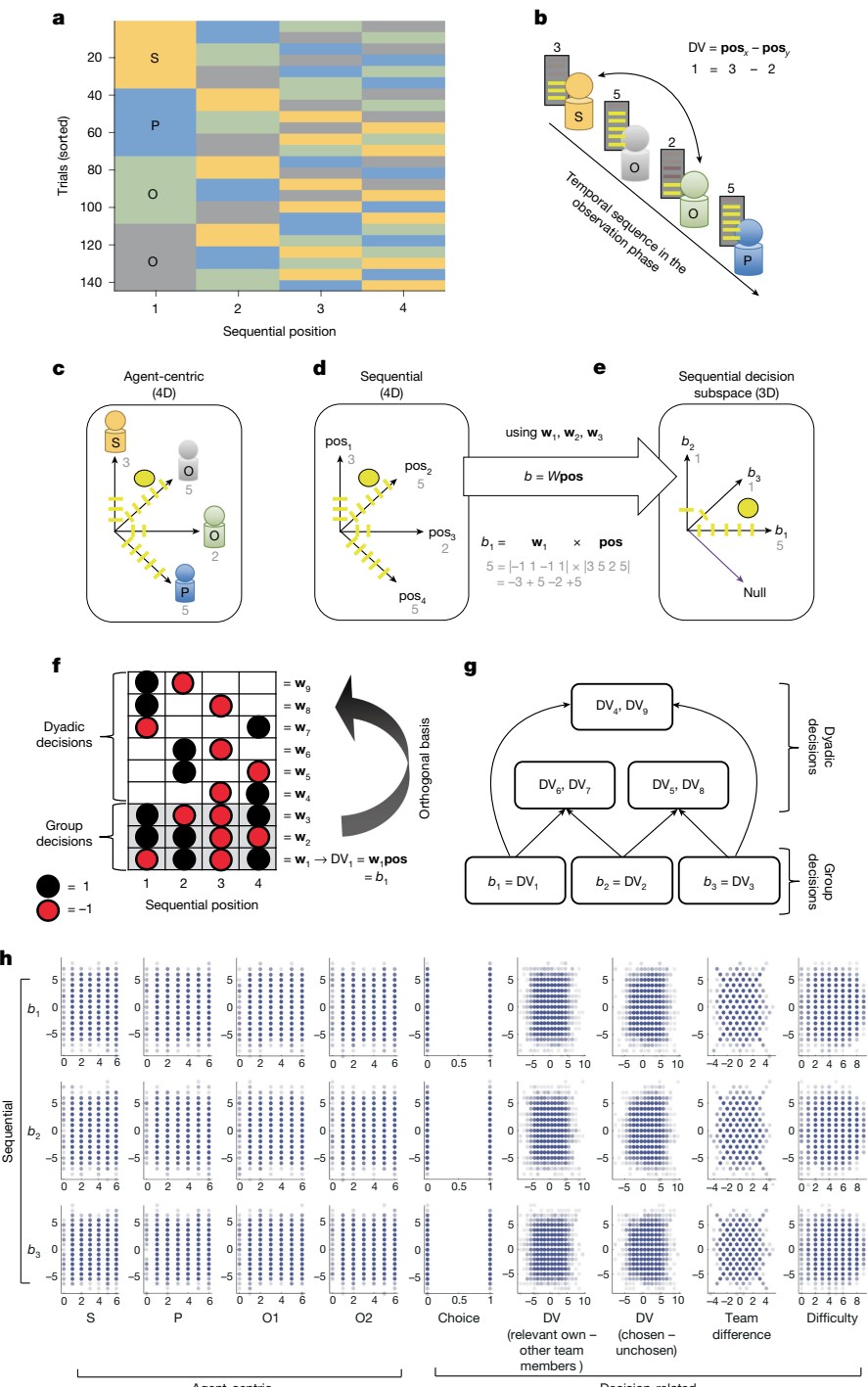

**Fig. 2 | Sequential basis functions define a low-dimensional decision space. a**, Player positions during the observation phase were carefully counterbalanced. **b**, A sequential frame of reference meant that decisions could be made by comparing different positions (**pos**) of the performance sequence. Counterbalanced player order allowed us to dissociate agent-centric and sequentially encoded neural activity. **c**, From an agent-centric perspective, the trial-wise decision-relevant information can be described as points in a four-dimensional (4D) space along the axes of each player's identity with the observed performance scores defining a unique point in the space. **d**, The same performances are displayed in a sequential reference frame along axes that represent the performance for each player position (pos$_1$, pos$_2$ and so on). **e**, Critically, all DVs of the sequential frame of reference lie in a 3D subspace. The axes of the subspace are the basis functions $\mathbf{w}_1$, $\mathbf{w}_2$ and $\mathbf{w}_3$; $b_1$, $b_2$ and $b_3$, are projections of the sequential performance scores onto the basis functions.

This subspace is orthogonal to a task-irrelevant null axis. **f**, Weight combinations for all possible decisions in this experiment in a sequential frame of reference. Rows represent possible sequential comparison in the decision phase. Basis functions $\mathbf{w}_1$, $\mathbf{w}_2$ and $\mathbf{w}_3$ are an orthogonal basis for this space, meaning that these weight vectors are pairwise independent and can be combined to construct all other decision-relevant weight vectors. Note that the sign of each $w$ is arbitrary. The sequential DV for each decision is a simple projection onto these basis functions. **g**, Projections $b_1$, $b_2$ and $b_3$ reflect the DVs for group decisions. Every dyadic DV can be constructed with a linear combination of precisely two projections. **h**, Basis function projections are unrelated to agent-centric and decision-related variables. Correlation plots of $b_1$, $b_2$, $b_3$ (rows) with agent-centric and decision-related (columns) variables. Data are collapsed over participants ($n = 56$; study 1).

decision is to compare one set of group members, who either appear at the beginning or the end of the sequence, with the other team, who appear at the other end of the sequence (for example, compare positions 1/2 with 3/4). Whereas group decisions can be guided by a single basis function, linear combinations of precisely two basis functions, in combination with the performance scores, are used to construct the DV for each type of dyadic decision (Fig. 2g). Therefore, in a sequential frame of reference, the basis functions afford a low-dimensional representation of all possible DVs. They allow sequential task representations that are both compact in format and flexible in terms of the task space[34,35]. Such sequential representations were clearly distinct from representations that simply associate the performance with each player's identity (agent-centric coding as S, P, O1 and O2), but they were also clearly distinct from decision-related representations (choice (0/1), social DV (relevant own minus relevant other group members), DV (chosen−unchosen), overall team difference (own−other), choice difficulty, or choice difficulty (inverse of absolute social DV); Fig. 2h). None of these variables shared more than 0.01 of variance ($r^2$) with the basis function projections, and all neural analyses controlled for these related variables (Methods).

Using fMRI, we studied the brain for evidence of a sequential neural code in our task (study 1: social fMRI experiment, $n = 56$). As described above, such a code can be useful to compress social information and prepare a decision-maker for possible decisions. We focused our analyses on the end of the performance observation phase (note that Fig. 1e above related to a different phase of the trial) and searched for neural correlates of the projections onto the basis functions (as in all subsequent analyses, we correlated the neural signals with the trialwise variation in the projections onto the basis functions). Crucially, we performed this analysis while simultaneously controlling for agent-centric representations in the same general linear model (Fig. 3a). Moreover, we accounted for the fact that the basis functions should be related to the participant's role in the task in some way to translate the sequentially organized basis functions into a social DV that could guide engage/avoid decisions. We tested whether the brain encoded the sequential position of oneself (the S-position) in the observed sequence, as well as to $b_1$, $b_2$ and $b_3$.

We discovered that activity in the dmPFC extending into the adjacent ACC gyrus (Fig. 3b) covaried with such a sequential neural code in a whole-brain analysis. The location is consistent with previous reports of grid cells in humans[36,37] but is also close to areas engaged during theory of mind tasks[1,9] and reports of a specific subregion of the ACC that may be especially important for social cognition[5,38,39]. We discovered this activation using an ROI approach based on independent neural effects time locked to a different trial phase discovered in Fig. 1 (yellow activation in Fig. 1e; Montreal Neurological Institute (MNI) coordinates = (−8, 42, 14)). In this independent ROI, we found significant effects for all three basis function projections (using the weight vectors from Fig. 2f; Methods): $b_1$ ($t_{55} = 2.886$, $P = 0.006$), $b_2$, ($t_{55} = 2.749$, $P = 0.008$) and $b_3$ ($t_{55} = 3.081$, $P = 0.003$) (aggregate mean effect of all three projections, $t_{55} = 5.02$, $P < 0.001$; Cohen's $d = 0.671$), but, as predicted, not by the null vector ($t_{55} = 0.229$, $P = 0.820$; Bayesian evidence for the null, $BF_{01} = 6.684$; Fig. 3c and Extended Data Fig. 4). The S-position showed a negative deflection at the same time ($t_{55} = −2.379$, $P = 0.021$) that was preceded by a positive signal ($t_{55} = 2.051$, $P = 0.045$; Fig. 3d and Extended Data Fig. 4). The whole-brain analysis subsequently confirmed that these effects were most prevalent in the dmPFC/ACC region (Fig. 3b). Further neural simulations showed that these signals were clearly dissociable from agent-centric signals (Extended Data Fig. 6), and Bayesian model comparison confirmed that the neural activity in the dmPFC/ACC was better explained by our basis function model than by an agent-centric model (Extended Data Fig. 6). This indicates that the dmPFC/ACC represents performance along all task-relevant dimensions in the flexible, temporally structured format before the choice.

Basis functions can be used to guide social decisions in a simple and efficient manner. They can be sorted by relevance and combined serially in time to guide the different decisions that follow the performance observations. As noted above, one basis function alone defines the sequential DV in group decisions, and we refer to this as the primary basis function (Fig. 3e). Which of the three basis functions is primary depends on the specific trial and is known after the position of each player has been revealed in the observation phase. For example, $w_2$ comparing players 1 and 2 with players 3 and 4 is the primary basis function if the sequence is P→S→O1→O2 or O1→O2→P→S, but not if the sequence is S→O1→O2→P. Group decisions can be solved sequentially by using primary basis functions alone. Dyadic decisions (self decisions and partner decisions) require the comparison of one player per team while ignoring another player per team. This is achieved by first identifying the two teams through the primary basis function, which effectively serves as a prior in a hierarchical decision process[40,41]. Then, the primary basis function is combined with a secondary basis function that specifies the relevant players in each team. In contrast to the primary basis function, the secondary basis function is known only once a specific dyadic decision is cued in the decision phase (Fig. 3f). Therefore, sorting basis functions on the basis of relevance enables participants to efficiently partition the decision process in time.

We tested whether the brain organizes the basis function in this way and performed whole-brain analyses using relevance-sorted basis functions during dyadic decisions. We found evidence that a large region along the cingulate sulcus encoded the combination of the basis function projections using a sequential frame of reference (Extended Data Fig. 6). These signals exist in parallel with the inverted/social basis function projections that use an agent-centric frame of the two teams (own minus other team; Extended Data Fig. 7). We found that both primary and secondary inverted basis functions captured brain activity in distinct regions of the frontal cortex and subcortical regions, such as the dmPFC, the lateral prefrontal cortex and the ventral striatum (Extended Data Fig. 7). The combined effect of both primary and secondary inverted basis functions is shown in Fig. 3g. These signals seem to reflect an intermediate stage of information organization in the brain, and they are in strong contrast with the activation in the vmPFC that seemed to be linked to the final choice (Fig. 3g). The vmPFC is a different prefrontal region that is robustly linked to decision making, regardless of whether there is a social dimension[42]. In contrast to basis function-related activity, during dyadic decisions, vmPFC activity tracked the value difference between chosen and unchosen options, regardless of which teams generated those values (Extended Data Fig. 7).

We wanted to understand the implications of basis functions for decision-making. We therefore tested one key prediction implied by our model, namely whether participants compartmentalize the decision-making process serially by first considering the primary basis function and then combining it with the secondary one serially in time. We confirmed this prediction using a version of the drift diffusion model[43]. We found that secondary basis projections exerted their influence on choice significantly later than primary basis functions (both already inverted to an agent-centric frame of reference; Fig. 3h) in both self decisions ($t_{55} = 3.71$, $P < 0.001$) and partner decisions ($t_{55} = 2.20$, $P = 0.032$), and the model predicted dyadic decisions better than a model without such onset asynchrony (likelihood ratio test, $\chi^2(1) = 23.77$, $P < 0.001$). We replicated this sequential prioritization in a model-free way (Extended Data Fig. 8). The net effect of such an onset asynchrony is likely to be that primary basis functions exert a bigger overall influence on decisions than secondary basis functions because they influence choices for a longer time. We examined the consequences of this in computational simulations in which we fit simulated choices with a logistic general linear model (GLM) that predicted choice as a function of both relevant (S or Or in self decisions) and irrelevant (P and Oi in self decisions) players. We found that

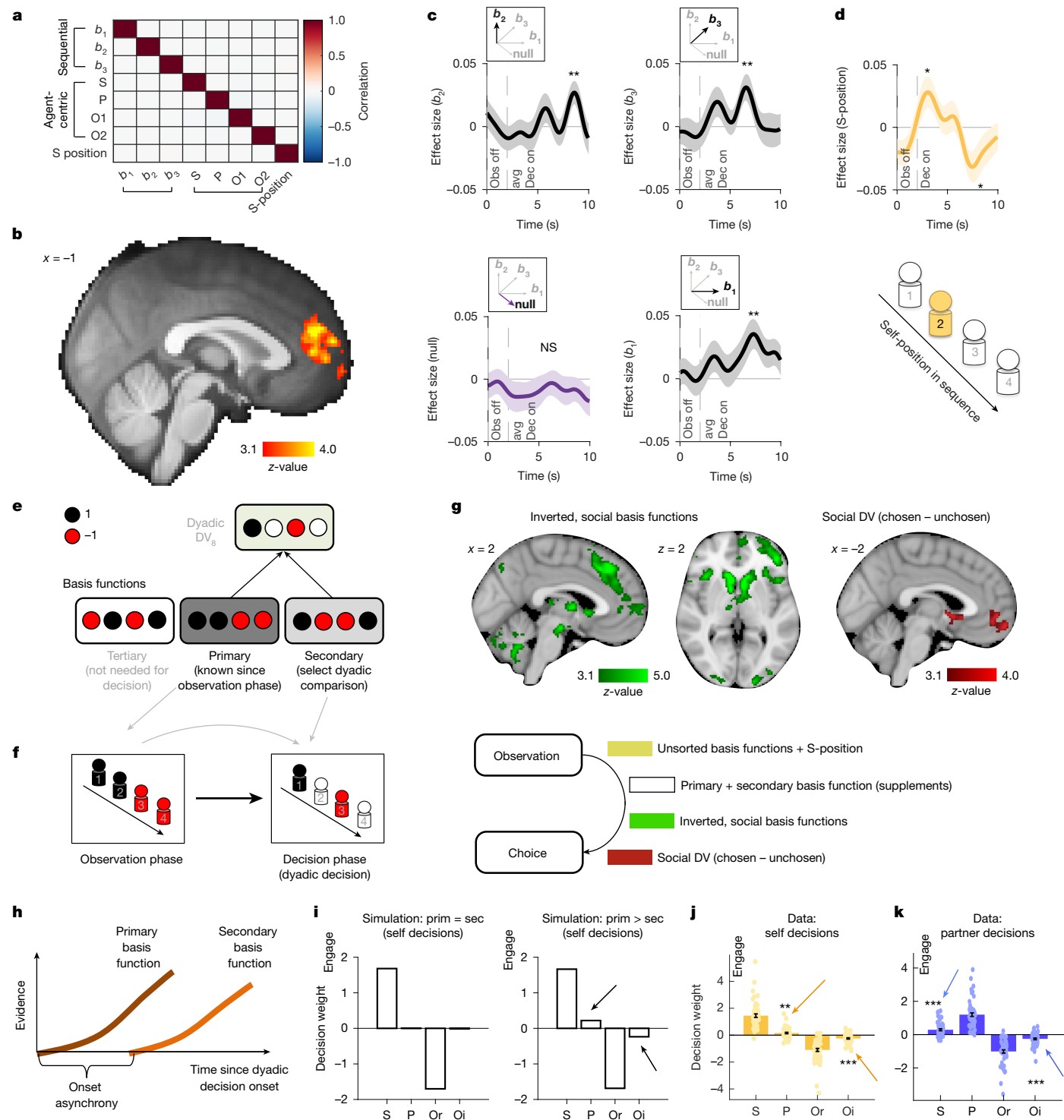

**Fig. 3 | Basis functions in medial prefrontal cortex (study 1). a**, Correlation matrix showing uncorrelated agent-centric and basis function fMRI regressors. **b**, The dmPFC encoding the average effects of the projections onto the three basis functions and the S-position (peak is MNI = (2, 48, 18)). **c,d**, ROI timecourse effects of $b_1$ ($P = 0.006$), $b_2$ ($P = 0.008$), $b_3$ ($P = 0.003$) (**c**) and a biphasic S-position signal (**d**) (positive $P = 0.045$; negative $P = 0.021$) time locked to the observation-phase offset. As predicted, effects of the null vector were absent ($P = 0.820$; based on an independent ROI from Fig. 1e, MNI = (−8, 42, 14)). **e**, Example primary, secondary and tertiary basis functions illustrating the construction of $DV_8$ from Fig. 2. **f**, Example of primary and secondary basis function combination. This sequential relevance sorting of basis functions enables an efficient partitioning of the decision process in time. **g**, Neural activity signalled a combination of inverted primary and secondary social basis function projections. By contrast, the final social decision variable was encoded in the vmPFC. **h**, Drift diffusion

modelling indicated that primary and secondary basis functions were integrated sequentially with a significant onset asynchrony, leading to overweighting of primary basis functions. **i**, Simulations demonstrated that overweighting primary basis functions (primary (prim) > secondary (sec)) predicted irrelevant player effects in line with group membership (indicated by arrows). A GLM model predicting self decisions (engage or avoid) as a function of the performance scores of S, P, Or and Oi. For example, positive P effects increase the likelihood to engage in a self decision, despite P being irrelevant. **j,k**, As predicted, we found small but highly significant effects of irrelevant players in self decisions (P, $P = 0.003$; Oi, $P < 0.001$) and in partner decisions (for S and Oi, $P < 0.001$) in precisely the predicted directions. Sample sizes: $n = 56$; study 1, social fMRI experiment; all MRI results cluster corrected at $Z > 3.1$; $P$ values determined by two-sided one-sample $t$-tests; error bars are s.e.m; $*P < 0.05$, $**P < 0.01$, $***P < 0.001$; NS, not significant ($P > 0.05$).

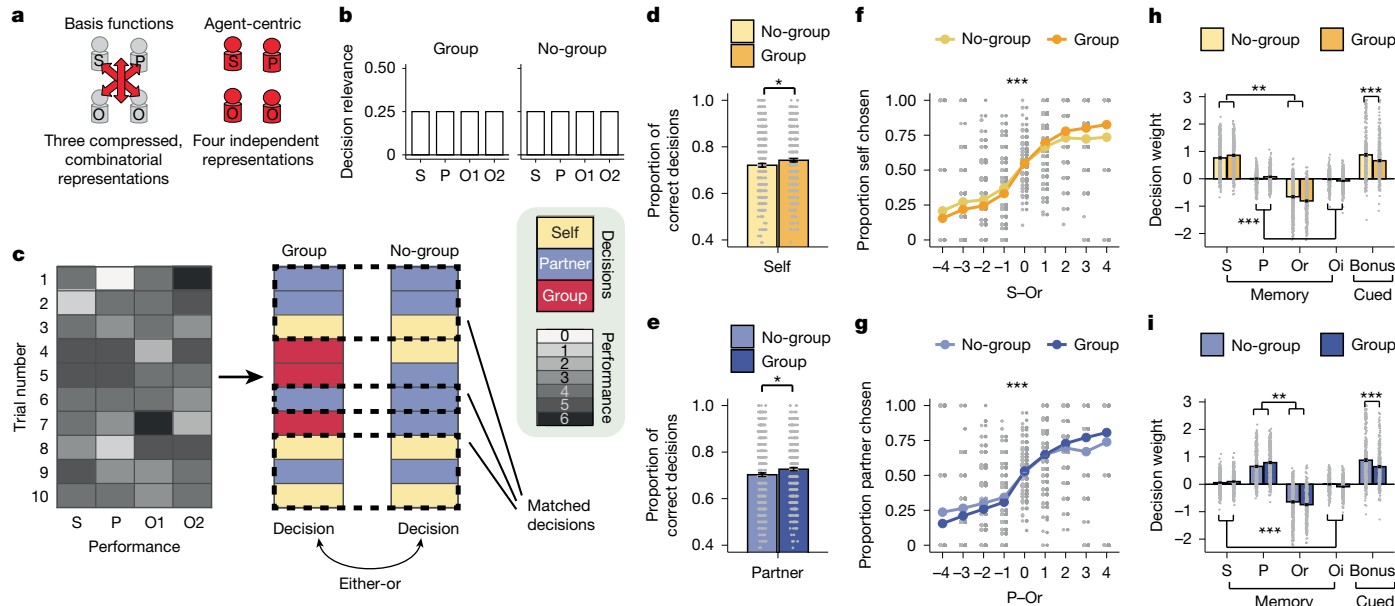

**Fig. 4 | Group decisions incentivise efficient representation compression (study 2). a**, In a between-subject behavioural study, we tested whether decision accuracy improved under conditions conducive to the formation of basis functions but that were more complex from a four-element agent-centric perspective. **b**, We devised two conditions (group and no-group). **c**, The observed performances were identical across the conditions. Both conditions comprised self and partner decisions, but only the group condition also comprised group decisions. The latter were replaced by extra self and partner decisions in the no-group condition. Importantly, we analysed only matched self and partner decisions, with their only difference being the decision context. **d**,**e**, Participants' first decisions (self, **d**; partner, **e**) after the observation phase were significantly more correct in the group than in the no-group condition; self/partner decisions: $P = 0.038/P = 0.024$. **f**,**g**, Participants adjusted their decisions more strongly based on the decision variable (S–Or in self decisions (**f**)

and P–Or in partner decisions (**g**)) in the group condition compared with the no group condition ($P < 0.001$ for both). **h**, Decision weights of a logistic regression on choice in self decisions showed increased impacts of the differences between both relevant (S–Or, $P = 0.002$) and irrelevant players (P–Oi, $P < 0.001$) effects in the group compared with the no group condition. By contrast, participants from the no group condition relied more on the cued bonus ($P < 0.001$; see Methods 'fMRI experiment, decision phase') that was displayed on the screen. **i**, We obtained analogous results in partner decisions (P–Or, $P = 0.003$; S–Oi, $P < 0.001$; bonus, $P < 0.001$). S, self performance; P, partner performance; Or, relevant opponent performance; Oi, irrelevant opponent performance; all data from behavioural experiment [study 2]; *$P < 0.05$, **$P < 0.005$, ***$P < 0.001$; error bars are s.e.m.; $P$ values determined by two-sided independent samples $t$-tests (**d**,**h**,**e**,**i**) and repeated measures ANOVA (**f**,**g**); $N_{total} = 795$; $N_{group} = 396$; $N_{no-group} = 399$.

overweighting primary over secondary basis functions predicted a characteristic pattern of choices in which players irrelevant to dyadic decisions have an inappropriate effect on decisions in line with their team membership (Fig. 3i). Indeed, analysing player-specific effects on choice revealed precisely these effects of irrelevant players in self decisions (P, $t_{55} = 3.093$, $P = 0.003$; Oi, $t_{55} = -5.131$, $P < 0.001$; combined P–Oi effect, $t_{55} = 5.803$, $P < 0.001$; Cohen's $d = 0.775$; Fig. 3j) and partner decisions (S, $t_{55} = 4.867$, $P < 0.001$; Oi, $t_{55} = -5.104$, $P < 0.001$; combined S–Oi effect, $t_{55} = 6.027$, $P < 0.001$; Cohen's $d = 0.805$; Fig. 3k). Formal information-theoretic metrics confirmed the higher model accuracy of the basis function model over the agent-centric model (paired $t$-test on model accuracy, $t_{55} = 8.111$, $P < 0.001$; Cohen's $d = 1.084$; Extended Data Fig. 8). Furthermore, we found subtle but significant effects of the inversion of the basis functions on reaction times. Such effects are predicted specifically by our basis function approach, and we replicated them in a subsequent online study (Extended Data Fig. 9).

Our neural and behavioural results indicate that the brain organizes social information along the basis functions to solve the decision-making task efficiently. However, it is possible to transform information without compressing it, just as it is possible to change a coordinate system without losing information. Our neural results indicate that the brain compresses task information to simplify the task and solve it more efficiently. This is because the neural signatures of basis functions in the mPFC are organized along relevant task dimensions, but not along an irrelevant null dimension. Nevertheless, we also tested behaviourally whether people rely on a compressed code in our experiment by conducting a large-scale behavioural study (study 2, behavioural experiment).

Specifically, we designed a behavioural study (study 2, $n = 795$) to test whether our task could be better solved by holding four agent-centric representations or by forming compressed representations using basis functions (Fig. 4a). We devised two versions of the task that, from an agent-centric perspective, posed similar demands (Fig. 4b). Both versions comprised both self decisions and partner decisions. However, the group condition included extra group decisions (as in study 1), whereas the other version (the no-group condition) did not. The performance score information in each trial and the decisions were identical in both conditions, except that the group decisions were replaced, in equal numbers, by more self and partner decisions in the no-group condition (Fig. 4c and Extended Data Fig. 10). Although the presence of group decisions might initially seem to make the task more complex, it made the task simpler from a basis function perspective; as noted above, group decisions correspond to single basis functions (Fig. 2f), so including group decisions is likely to be conducive to the use of basis functions. To compare behaviour across conditions fairly, we compared only matched decisions (self and partner decisions) that were identical across the group and no-group conditions in terms of when they were encountered in the experiment and which performance scores they related to. This constitutes a particularly tough test of our hypothesis; participants in the no-group condition dealt with a simpler task space (fewer possible decision types to expect)[34,35] and received more training in relevant decision types (more self and partner decisions in the no-group condition). If people solved the task using only four independent agent-centric representations, there should be no advantage in embedding matched self and partner decisions in group decisions.

However, we found the opposite, namely that the benefits of compressing information along the basis functions in the group condition resulted in a net increase in decision accuracy that outweighed the effect of the reduced task space and increased training in specific decision types (self or partner decisions) experienced in the no-group condition. Choice accuracy differed on matched self and partner decisions, depending on decision number (first or second) and condition (group or no-group). A three-way analysis of variance (ANOVA) test revealed a group × decision number interaction ($F_{1,793} = 8.723$, $P = 0.003$). This was because participants performed significantly better on first self decisions ($t_{793} = 2.074$, $P = 0.038$; Fig. 4d) and first partner decisions ($t_{793} = 2.254$, $P = 0.024$; Fig. 4e and Extended Data Fig. 10) in the group condition. It was at precisely the same time period (the first decisions) when the neural evidence for basis functions was clearest in the fMRI experiment (Fig. 3b). We examined the origins of this improvement by plotting how accurately participants made decisions in group and no-group conditions (Fig. 4f,g). This showed that participants became more accurate in matched self (observed player performance difference × group interaction, $F_{8,6,344} = 9.339$, $P < 0.001$) and partner ($F_{8,6,344} = 10.036$, $P < 0.001$) decisions. Next, we confirmed this result using the same logistic GLM analysis that we had applied to the fMRI data. Relevant player information exerted a stronger influence on decisions in the group condition than in the no-group condition; the difference in beta weights of relevant players was increased in self decisions (S–Or, $t_{793} = 3.058$, $P = 0.002$; Fig. 4h) and partner decisions (P–Or, $t_{793} = 3.021$, $P = 0.003$; Fig. 4i) that occurred in the context of group decisions in the group condition. However, simultaneously, the effects of the irrelevant players increased in the group condition relative to the no-group condition (P–Oi in self decisions, $t_{793} = 4.471$, $P < 0.001$, Cohen's $d = 0.317$; S–Oi in partner decisions: $t_{793} = 4.317$, $P < 0.001$, Cohen's $d = 0.306$). This resulted in the same characteristic pattern of irrelevant player effects in the group condition that we had observed in the fMRI study, which also contained group decisions (P–Oi in self decisions, $t_{395} = 7.241$, $P < 0.001$, Cohen's $d = 0.364$; S–Oi in partner decisions, $t_{395} = 9.253$, $P < 0.001$, Cohen's $d = 0.465$). This further supports the contention that such mergence effects[6,25] reflect adaptive compression of representations along the basis functions. By contrast, we found that participants in the no-group condition relied more strongly on a non-social bonus, a visual cue indicating a small handicap (study 1, Methods; in self decisions, $t_{793} = 4.239$, $P < 0.001$; in partner decisions, $t_{793} = 4.658$, $P < 0.001$). This indicates that the inclusion of group decisions selectively improved the retention and use of social agent information, but that participants in the no-group condition increasingly relied on other information in the stimulus array that did not need to be retrieved from memory.

Overall, participants relied more on correct information about relevant players in the group condition. This is strong evidence that a compressed code along the basis functions improves decision making, and it illustrates how the use of the basis function changes depending on decision context. So, despite a superficially more complex decision space and less training, but predicted by our basis function model, participants seemed to access a more accurate representation of the relevant players' performance in the group condition. Rather than purely confusing or grouping individuals[25,44–47], the systematic effects of irrelevant players seemed to index an adaptive factorization of social information. These findings cannot be explained by a four-element agent-centric code. Instead, they indicate that information is compressed, and that this enables more efficient decision making.

Representing social information along the basis functions allows a decision maker to simplify decision problems and solve them sequentially. However, it is possible that such a representational format is not unique to social situations. Instead, it might guide decisions in a variety of domains in which multiple pieces of information need to be tracked and strong priors exist about which combinations of these pieces are possible and which are not. Therefore, we conducted a non-social fMRI

control experiment (study 3, control fMRI experiment, $n = 32$; Fig. 5). We predicted that removing the social framing would eliminate signatures of socially specific representations, but that an underlying combinatorial code would still guide choices and be measurable in behaviour, even if the precise neural implementation might differ slightly[8]. We tested and confirmed these predictions in the following way.

Instead of tracking social information and cueing decisions comparing players (Fig. 5a), in the control fMRI experiment, participants tracked and compared motor actions assigned to individual fingers of the left and right hand (study 3; Fig. 5b). However, formal aspects of the task remained similar across both social and motor tasks. The control experiment also involved an observation and a decision phase. We arbitrarily mapped the formal roles of the four players from the social fMRI experiment onto movements made by the index and middle fingers of both hands. We assigned S and P to the left hand, and the fingers (motor–S and motor–P) of the right hand occupied the same roles as two opponents, using the same input schedules as in study 1. Participants now decided which hand had been active more often in the observation phase. The decision phase comprised, in equal numbers, motor–self decisions, motor–partner decisions and motor–group decisions. All the statistical contingencies, timings, schedules and pay-off scheme were identical to the social fMRI experiment. The difference lay entirely in the manner that information was visually displayed and in the cover story (Methods and Extended Data Fig. 11). This enabled us to use the identical analyses that we had used in the social fMRI experiment to test for the presence or absence of socially specific representations and a compressed basis function code in behaviour and the brain.

As expected, socially specific representations were absent. When the task was no longer framed in social terms, participants were no longer more correct or faster for the cue (formerly) corresponding to self than for partner (Fig. 5c; percentage correct, $t_{31} = -0.3$, $P = 0.766$; Bayesian evidence for the null, $BF_{01} = 5.978$ (values bigger than 1 provide evidence for the null hypothesis); reaction times, $t_{31} = -1.078$, $P = 0.289$; $BF_{01} = 3.112$). However, despite the non-social nature of the task, behavioural signatures related to the basis functions were still visible. Our basis function model indicates that people use a compressed and combinatorial code that predicts that irrelevant information affects decisions in a characteristic pattern. We found precisely this pattern in the control fMRI experiment. In motor–self decisions, motor–P had positive effects (Fig. 5d; $t_{31} = 2.783$, $P = 0.009$) and motor–Oi had negative effects ($t_{31} = -4.490$, $P < 0.001$; combined motor–P/motor–Oi, $t_{31} = 4.629$, $P < 0.001$, Cohen's $d = 0.818$). The pattern of effects in motor–partner decisions provided further evidence for this claim (motor–S effect, $t_{31} = 3.564$, $P = 0.001$; motor–Oi effect, $t_{31} = -3.844$, $P < 0.001$; combined motor–S/motor–Oi, $t_{31} = 4.166$, $P < 0.001$, Cohen's $d = 0.736$).

We then looked for socially specific representations and signatures of basis functions in the brain. We applied the same fMRI GLMs as before (fMRI GLM1 and fMRI GLM2); importantly, this ensured that the analyses had the same degrees of freedom. Note that, owing to our tightly matched design, these models were not only conceptually but also numerically identical to our social GLMs. However, regressors that previously identified self-related and partner-related signals in the pgACC and dmPFC (Fig. 1e), respectively, failed to capture significant effects in these regions in the control fMRI experiment. ROI analyses centred on whole-brain significant peaks in the social study revealed no effects of motor–S in pgACC (Fig. 5f; MNI = (−8, 42, 14); $t_{31} = 0.129$, $P = 0.899$, $BF_{01} = 5.255$) and no effects of motor–P in dmPFC (Fig. 5f; MNI = (−2, 28, 14); $t_{31} = 1.36$, $P = 0.184$, $BF_{01} = 2.291$; Extended Data Fig. 12).

Nevertheless, the same neural signatures of basis functions were visible in the same pgACC region where we found them in study 1 (Fig. 5g). We looked for an average effect of the three basis function projections, as we had done in study 1. The contrast was the same, but we omitted self-position from it, because this variable had lost its meaning (and accordingly, had no significant neural effect; see the analyses in

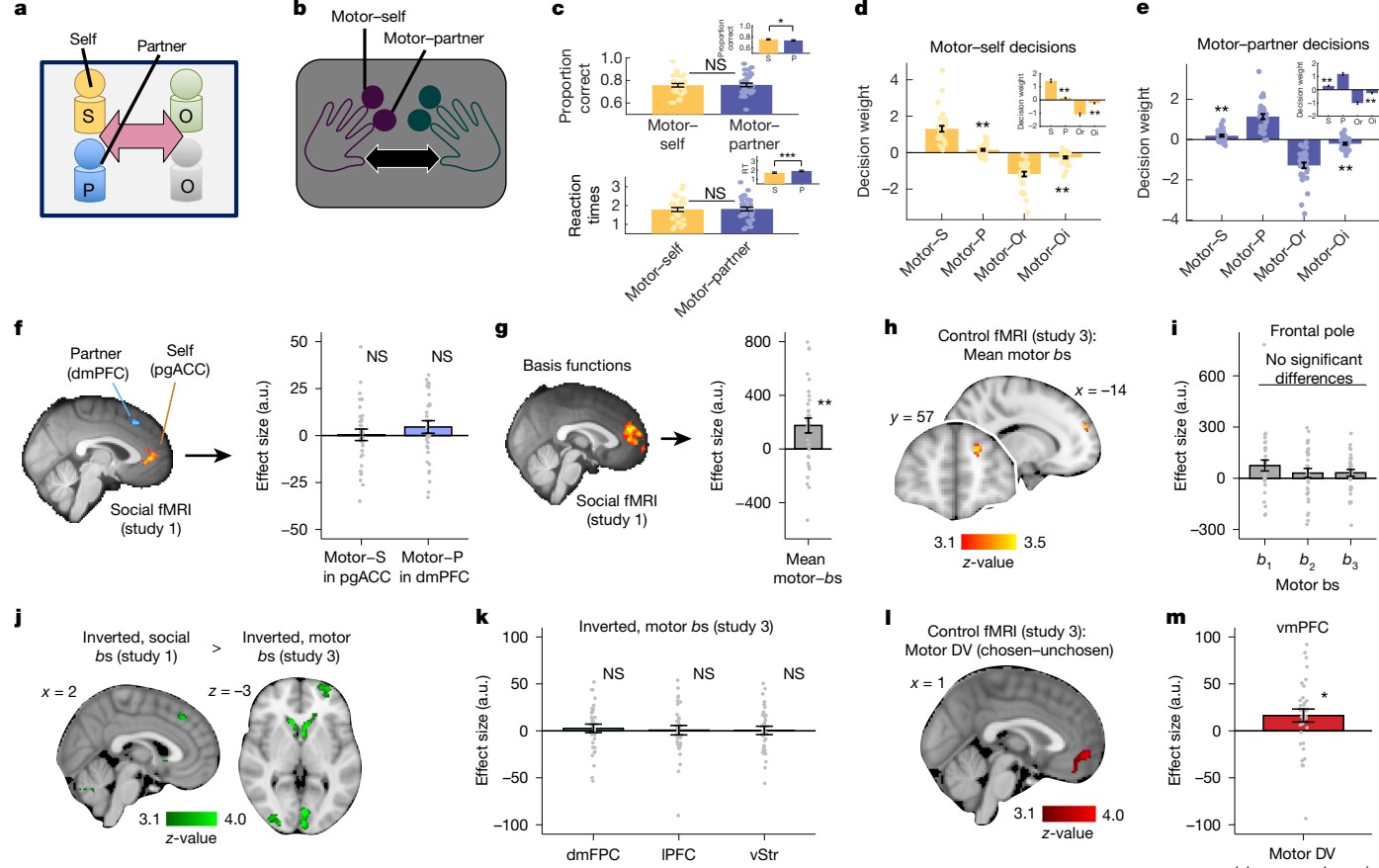

**Fig. 5 | Control fMRI experiment revealed behavioural and neural signatures of basis functions (study 3). a**, Social fMRI experiment (study 1). The task was to compare the performance of cued players and decide which team was better. **b**, In study 3, participants compared finger-tap counts instead of player performance. The self player's scores were reassigned to the left middle finger (motor–self) and the partner's scores to the left index finger (motor–partner). The task was to compare the indicated finger taps for the cued fingers and decide which hand tapped more often. **c**, As predicted, the self-biases in accuracy ($P = 0.766$) and reaction times ($P = 0.289$) were abolished (compare with the insets in Fig. 1c,d). **d**,**e**, By contrast, the effects of irrelevant information predicted by our basis function model were replicated (motor–self decisions (**d**): motor–P/motor–Oi: $P = 0.009/P < 0.001$; motor–P decisions (**e**): $P < 0.001$ for both; compare with the insets to Fig. 3j,k). **f**, Motor–S and motor–P signals in the pgACC and dmPFC, respectively, were absent (ROIs from Fig. 1e). **g**, Significant average effect of motor–$b_1$, $b_2$, $b_3$ in the same pgACC ROI ($P = 0.003$),

replicating previous basis function effects. **h**, These effects were most pronounced in the frontal pole (pre-threshold masked, $z > 3.1$, $P = 0.05$ FWE-corrected). **i**, Basis function effects were uniformly positive; no main effect in one-way ANOVA; MNI = $(-14, 56, 24)$. **j**, Regions in which inverted, social basis functions (study 1; Fig. 3g) were significantly more strongly represented than inverted motor basis functions ($z > 3.1$; $P = 0.05$, FWE-corrected; dmPFC is pre-threshold masked). **k**, ROI analyses of the motor study showed no inverted basis function signals in dmPFC, vStr or lPFC (independent ROIs from study 1: lPFC = $(-44, 42, 4)$, vStr: combined = $(-6, 14, -2)/(8, 18, -2)$, dmPFC = $(-2, 28, 44)$). By contrast, we replicated DV signals in the vmPFC. **l**,**m**, Whole brain (**l**) and ROI (**m**) analyses ($P = 0.025$; independent ROI from social study 1 (Fig. 3g). MNI = $(-2, 60, -12)$) (study 1, $n = 56$; study 3, $n = 32$; $P$ values determined by one-sample and paired $t$-tests, two-sided; *$P < 0.05$, **$P < 0.005$, ***$P < 0.001$, error bars are s.e.m.; all results are whole brain cluster corrected with $z > 3.1$ and $P = 0.05$ FWE-corrected unless otherwise noted).

Extended Data Fig. 4). Moreover, the behavioural analyses reported above indicated that the basis functions were no longer anchored to a self representation as they were in the social fMRI experiment. There was a strongly positive mean effect for all three basis function projections in pgACC (Fig. 5g; the same ROI as in the social study; MNI = $(-8, 42, 14)$, $t_{31} = 3.168$, $P = 0.003$, Cohen's $d = 0.560$; Extended Data Fig. 12). However, a subsequent test of where the activation was most prominent revealed an adjacent brain region in the frontal pole. This was revealed by a whole-brain analysis looking for an average effect of the basis function projections in a sphere of radius 20 mm around the pgACC coordinate (pre-threshold masked, $z > 3.1$; $P = 0.05$ FWE (family-wise error)-corrected; Extended Data Fig. 12). ROI analyses of this cluster indicated that here the basis function projections had uniformly positive effects (Fig. 5i; ROI on significant peak of cluster, MNI = $(-14, 56, 24)$, $F_{2,62} = 0.718$, $P = 0.492$, BF$_{01} = 5.036$, indicating an absence of differences; Extended Data Fig. 12). As expected, replicating previous results, the null vector reflecting variation in the BOLD signal that was not along

the basis function axes had no significant effects in the frontal pole (frontal pole, $t_{31} = 0.274$, $P = 0.786$, BF$_{01} = 5.113$).

We tested for neural effects of inverted basis functions in the control fMRI experiment, proposing that dissociations in activity patterns from those seen in study 1 might emerge after inversion to agent-centric versus motor frames of reference (fMRI GLM 3). Indeed whole-brain analyses comparing the combined inverted basis function representations (Fig. 3g, green activity) revealed significantly stronger activations in the social study (Fig. 5j; $z > 3.1$, $P = 0.05$, FWE-corrected) in the ventral striatum (vStr) and lateral prefrontal cortex (lPFC), and also in the dmPFC ($z > 3.1$, $P = 0.05$, FWE-corrected, pre-threshold masked in sphere around partner-related activity from Fig. 1e). Corresponding signals were absent in each of the three regions in the motor study (all $t_{31} < 0.575$, all $P > 0.570$, all BF$_{01} > 4.544$; Fig. 3k). By contrast, activation was stronger in lateral primary motor regions in the motor study (Extended Data Fig. 10). This indicates similar basis function-related computations across domains with differing neural

implementations[8], particularly at intermediate stages of processing. However, decision-related activity in the vmPFC was present in both experiments (motor DV(chosen–unchosen); Fig. 5l; $z > 3.1$, $P = 0.05$, FWE-corrected; compare with social DV in Fig. 3g), even using the same ROI ($t_{31} = 2.351$, $P = 0.025$; Fig. 5m). This indicates a common final pathway for decision making across domains.

Taken together, our results demonstrate that the brain tracks combinatorial patterns of social interactions in a compressed format, leaving observable traces in social decision making. We have shown that, prior to choice, dmPFC and ACC signals vary along the axes of a set of basis functions that efficiently summarize the social task space navigated by the participants in our experiment[35] (Fig. 3). During decision making, projections along these axes reorganize according to their relevance for choice along primary, secondary and tertiary axes. These sequential representations exist in addition to individualized representations of self and others, which become apparent late in a trial (Fig. 1). Note that our basis function effects show moderate effect sizes[48], although they are numerically small, and future research should establish their relative importance in social processes. The basis functions may act as a computational scaffold[12,18] for the construction of player identities. The use of basis functions allows efficient compression and sequencing of the decision process serially in time, and explain the improved decision accuracy under conditions of a superficially more complex decision space (Figs. 3 and 4). The use of the basis function may be a general mechanism to organize decision making in settings in which multiple pieces of information have to be flexibly maintained in the memory and strong priors exist about how these pieces of information are related and may become decision relevant (Fig. 5).

These codes can be thought of as complementing a suite of temporal lobe brain structures that enable primates to process visual information to recognize faces and to retrieve individual-specific identity information[10,49]. Representations of faces and individuals are essential for social cognition. However, such representations in isolation are insufficient to encode the relationships between individuals, and knowledge, both of the individuals and their interrelationships, is needed to guide adaptive social decisions. Here we suggest that representations of social environments using a set of basis functions define the structure of a space for social interaction. This proposal is consistent with recent evidence of abstract task space encoding by the adjacent anterior medial frontal cortex[36,50–52] and offline playing out of sequential operations across maps of a physical or abstract task space[16,41].

More broadly, we propose that, even in the social domain, the brain applies coding principles that are observed in visual, motor and spatial domains[10,11,17,53]. Just as representations of faces might be constructed by the combination of highly abstract visual dimensions of which we are not aware[10], our results indicate that our subjective sense of personal identities is a consequence of a series of transformations in an abstract social feature space.

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

## Methods

We conducted a total of four studies: a social fMRI experiment (study 1, $n = 56$; Fig. 3); a behavioural experiment (study 2, $n = 795$; Fig. 4); a control fMRI experiment (study 3, $n = 32$; Fig. 5); and a supplementary behavioural experiment (study 4, $n = 1,022$; Extended Data Fig. 5). All studies used variants of the same experimental paradigm.

### Study 1: social fMRI experiment

**Participants.** There were initially 59 participants in the social fMRI experiment (study 1). However, two participants did not complete the scanning session and one participant repeatedly fell asleep during the experiment; these were removed from the sample, so the final sample contained 56 participants (age range 18–38 years, 33 of them female). Participants received £50 for taking part in in the experiment, as well as extra earnings that were allocated according to their performance in the tasks. The ethics committee of the University of Oxford approved the study and all participants provided informed consent (MSD reference number: R60547/RE001).

**Experimental procedures.** At the start of the experimental procedures, and before entering the MRI scanner, participants performed a behavioural pre-experiment. They were informed that the purpose of the pre-experiment was a performance assessment to record their performance in a perceptual decision-making task (a random dot motion task[54]) and that the performance would be used in the subsequent fMRI experiment. This was indeed the case, and the experimental procedure involved no deception. We used an automated computer algorithm to pair up the recorded performance of the participant with the performance of three previous participants. The algorithm was used after the pre-experiment to create the experimental schedule, and ensured that experimental schedules were comparable across participants, balanced with respect to some key features (for example, each of the four players had approximately similar performance) and decorrelated with respect to key variables of interest (such as performance estimates across players). To ensure careful balancing, participants performed many more trials than necessary given the number of trials in the main fMRI experiment, allowing the algorithm to repeatedly subsample performances until the above-mentioned key criteria for schedules were reached. We used the schedule-generation algorithm after participants performed the pre-experiment but before they entered the MRI scanner. The schedules comprising veridical performances from the participant and three other players were then transferred to the computer operating the MRI experiment. The performance of the three other players in each experiment (the partner and the two opponents) were taken from log files of previous participants. For our first few participants of the fMRI experiment (when there were no preceding participants), these log files came from participants who had taken part in pilot experiments. The fMRI experiment lasted approximately 60 min. Afterwards, participants were debriefed, filled in some questionnaires that were unrelated to the current study and left.

The behavioural pre-experiment that took place before scanning and the fMRI experiment were programmed in Matlab using Psychtoolbox-3 (ref. 55; http://psychtoolbox.org). For the presentation of random dot motion stimuli we used the Variable coherence random dot motion toolbox (version 2; https://shadlenlab.columbia.edu/resources/VCRDM.html).

**Behavioural pre-experiment.** During the pre-experiment, participants judged the motion direction of a random dot-motion kinematogram (RDK) stimulus. The participants pressed the left or right button to indicate the congruent leftwards or rightwards motion directions of the RDK stimuli. Participants were made aware that they would perform these motion judgements at varying levels of coherence, making the detection of the RDK stimuli easier or more difficult. The pre-experiment comprised 972 RDK trials using motion coherences of 3.2%, 12.8% and 25.6%. Each RDK stimulus was presented for 0.45 s and valid responses had to be made within 0.6 s after RDK offset. Failure to respond in this time window was counted as incorrect performance and indicated by a "Missed!" message on the screen. Participants were made aware of this and were instructed to avoid having missed trials. Importantly, participants did not receive performance feedback (except for missed trials) for their RDK direction judgements. The reason for this was that participants would witness and learn about their recorded, veridical performances in the subsequent fMRI experiment, so it was important not to give them this information this early in the experiment. The pre-experiment took approximately 25 min.

We used several measures to streamline the pre-experiment and the subsequent fMRI decision-making experiment in terms of the participants' subjective experience. First, participants performed a few practice trials of the pre-experiment, in which they were given explicit performance feedback, and this feedback was cued using the same cues that also indicated successful and erroneous performance in the main fMRI experiment (a yellow coin for successful performance and a red X for erroneous performance). For participants, this measure underscored the fact that performance indices presented during fMRI scanning related to the participants' own performances in the pre-experiment period. Second, during the pre-experiment, participants performed sequences of six RDK trials in a sequence, followed by an intertrial interval of 1 s. This blocking of trials corresponded to the fact that in the fMRI experiment, participants observed rapid performance sequences of six performance cues in a row for each participant on each trial. So, again, this measure was taken to align the participants' experience of the pre-experiment with the main fMRI experiment. Third, during the entire pre-experiment, the layout of the screen was similar to the screen of the main fMRI experiment in the following way. Cues referring to self, partner and the two opponents were distributed over the screen, with each player occupying either a top-left, top-right, bottom-left or bottom-right position. These cue positions were the same in the pre-experiment and the main fMRI experiment. They remained fixed throughout the entire experimental procedures for each participant, but they were randomized and balanced across participants, with the restriction that the partner position would always be adjacent to the self position (self and partner could occupy the two upper positions, for example, or the two left-sided positions). The cue for self was not shown most of the time during the pre-experiment; instead, participants saw the RDK stimuli that they were asked to respond to in this location. However, the cues indicating the other three players were shown throughout the pre-experiment, despite being irrelevant for the task performed during the pre-experiment. This measure was used to indicate during the pre-experiment that participants' performances would be paired with the performance of three other players in the main fMRI experiment, and to demonstrate the team pairings (self and partner versus two opponents). Hence, again, this measure served to ensure that the participants' experiences in the pre-experiment and main fMRI experiment were similar.

Finally, in both the pre-experiment and the main fMRI experiment, we used static RDK images as cues indicating player identity, and these cues were fixed to specific screen locations (see the previous point). The RDKs were identifiable by their spatial position, their colour (white, green, orange or purple) and by two letters on them (the participant's initials, for example MW, for self, Pa for partner and O1 and O2 for the two opponents). Static RDK images symbolized that every player's performance cues were derived from their veridical RDK performance. This was done to remind participants in the main fMRI experiment that the performances they would observe were taken from the RDK pre-experiment. In summary, several measures were taken to illustrate to participants that their pre-experimental performance assessment seamlessly fed into the main fMRI experiment, and that their performance was important in the social context of two competing teams.

**fMRI experiment.** The main fMRI experiment involved completing a memory-guided social decision-making task. Each trial comprised an observation phase and a decision phase, in which participants made decisions about the information presented in the observation phase from memory. Four players' performances were relevant in the fMRI experiment when the players were divided in two teams: the participant's own performance (self, S), the partner's performance (Pa) and the performance of two opponents (O1 and O2). Team membership was constant throughout the experiment, and it was predefined by the experimenters. No further information about the other players was given (such as their gender or age).

**Observation phase.** In the observation phase, participants observed performance cues for each player in a random but counter-balanced sequence (each player appeared first in the sequence for the same number of trials, for example). Participants' veridical performances from the pre-experiment were paired with veridical performances from three other previous participants and displayed during this phase. The insight into pre-recorded performance was unrelated to beating the other team in the main experiment (Extended Data Fig. 2). Player identities were cued by RDK images and displayed in the same spatial positions as in the pre-experiment to reinforce continuity between the pre-experiment and the main fMRI experiment and to illustrate that the displayed performances referred to the pre-experimental performance assessment.

In the observation phase, for each player, participants observed six brief performance cues. The performance cues were always displayed centrally on the screen at the same location for each player, and successful performances were indicated by yellow coin cues and erroneous performances were indicated by a red X. Participants already knew the meaning of these cues from the pre-experiment (see the 'Behavioural pre-experiment' section). Because the performance cues always appeared in the same location for all players, we cued the player to which of these performances related in the following way: 400 ms before the performance cue sequence started, the RDK image of the respective player appeared in its predefined location and its dots moved in a slow and coherent fashion in a direction (randomly left or right) for 2.1 s overall. Importantly, the active RDK was simply a means to indicate the relevant player.

Participants this time did not have to discern the movement direction of the dots (which would have been easy because the coherence of the RDK was very high). They simply needed to understand that the performance cues displayed centrally represented the performance recorded for this specific player. Using the moving RDK to indicate the relevant player was, like many other small manipulations in this study, a way to enable participants to make a link between the pre-experiment and the main fMRI experiment, and to make it plausible that the displayed performance cues related to the RDK performance of the respective player. While the player's RDK was moving, the sequence of six performance cues was shown centrally on screen. After the above-mentioned 400 ms, the first performance cue was presented for 200 ms. The remaining performance cues were shown subsequently with a delay of 100 ms between them, also for 200 ms each. During this time, the player's RDK was still active to indicate the relevant player, and the RDK movement ended precisely at the time the sequence of six performance cues ended as well.

For each of the four players, the performance cues were presented in exactly this fashion. A sequence of six centrally presented performance cues was shown while the player-specific RDK was active. After a player's performance was presented, their RDK stopped moving and remained static until the decision phase started. This meant that at the end of the performance phase, all four RDKs were shown statically on screen in the location associated with the respective player. Importantly, the order of the players was fully balanced (Fig. 2). At the end, after the last player's performance was shown and the observation phase had ended, after a Poisson-distributed jitter (1–5 s, with a mean of 2 s), the decision phase started.

**Decision phase.** In each trial, after the observation phase came the decision phase. During the decision phase, participants compared performance scores between players, which were displayed in the previous observation phase. The decision was between whether the relevant member(s) of one's own team had performed better in the observation or whether the relevant member(s) of the opponent team had performed better. It was unknown which players would have to be compared until the start of the decision. Participants therefore had to memorize the performance score of all players, that is, the number of successful performances. Each player's performance score ranged between 0 and 6 (in case no performances were successful or all of them were successful, respectively), and this had to be extracted from the series of six performance cues that were shown while that player's RDK was active (by accumulating the yellow success cues and discarding the red error cues). Importantly, each decision phase actually comprised two decisions, and both referred to the same set of performances that had just been seen in the observation phase of the trial. Both decisions followed precisely the same logic, which is why only the first of them is shown in Fig. 1 for illustration. Note that the fact that two decisions were given, and that both referred to the same information, meant that people had to remember the performances of all four players throughout the trial, beyond the first decision and until the second decision was made.

After the temporal jitter following the observation phase, the decision phase started by presenting an arrow cue that indicated which players to compare in the decision. Which decisions were possible was constrained by the team membership of the players. Three possible decision types were possible. In self decisions, the participants' own performance was compared with the performance of one of the two opponents (each of them equally often). The performance of the other two players (the partner and an irrelevant opponent) had to be ignored. In partner decisions, the partner's performance was compared with the performance of one of the two opponents (each of them equally often), ignoring the performances for self and the other, irrelevant opponent. Finally, in group decisions, the sum of the performances of both groups was compared. The arrow cue indicated these different decisions by pointing at the self and a relevant opponent (self decisions), the partner and a relevant opponent (partner decisions) or at both groups (group decisions), respectively. This means that for self and partner decisions, the opponents could be divided into a relevant opponent (Or) and an irrelevant opponent (Oi), and both opponents were relevant and irrelevant for the same number of trials for both self and partner decisions. In group decisions, by contrast, both opponents were relevant.

Decisions were made by comparing the performances of the cued players. However, participants also had to factor in a non-social bonus. The bonus was cued on top of the decision arrow indicating the relevant players. The bonus was displayed as yellow coins and half-coins for a positive bonus, and red coins and half-coins for a negative bonus. A positive bonus meant that points had to be added towards the performance of one's own team, and a negative bonus meant that points had to be added towards the performance of the opponent team. Including the bonus in decisions was useful because it meant that participants had to wait until the time of the decision cue before all the information about the decision was known. This ensured that participants would make the decisions at the time of the decision phase and not beforehand. Moreover, the bonus had a value of either −1.5, −0.5, +0.5 or +1.5. This was useful because it meant that there was always a correct response in each trial. Because the performance scores were always integers, the ±0.5 meant that one of the teams had to come out as the better one when factoring in the bonus, even if performance scores were identical. This meant that the full social decision variables (DV) for the self, partner and group were: $DV_{self} = S - Or + B$, $DV_{partner} = P - Or + B$ and

$DV_{group} = S + P - O1 - O2 + B$, where $S$ indicates one's own performance score (self performance), $P$ indicates the partner's performance score (P performance), Or indicates the relevant opponent's performance score (Or performance), and O1 and O2 indicate the performance scores of the two opponents (O1 performance and O2 performance). Note that these labels are used in group decisions instead of Or because both opponents are relevant in group decisions.

The decision was presented as an engage/avoid decision[6,25] using two buttons. The decision was to compare the performances of the relevant players (factoring in the bonus) and indicate whether the relevant member(s) of one's own team had performed better in the observation or whether the relevant member(s) of the opponent team had performed better. Deciding to engage meant choosing one's own team, and this was indicated, after the respective button press, by a large box appearing around the two RDKs symbolizing one's own team. Making that choice indicated that the relevant member(s) of one's own team was estimated as better in performance than the other team's relevant player(s) (also factoring in the bonus). The pay-off from the engage choice was the veridical DV for that trial. This meant that, if one's own team was indeed better, and the DV was, say, +2 (see the equation above for the DV calculation), then the outcome would be +2. However, if the choice was engage and the DV was, say, −1, then the outcome would be −1. By contrast, the outcome of choosing the opponent team (making the avoid choice) would always lead to a pay-off of zero. This pay-off scheme meant that it was always beneficial to make the correct choice to maximize the reward outcome of a trial. It meant choosing engage only if one's own team was indeed better, and choosing avoid when this was not the case to avoid losing points. Feedback about whether the correct choice was made was not given. Points were accumulated over the course of the experiment and translated into a small amount of extra bonus payment at the end of the experiment. Participants also received an additional pay-off that was proportional to how many points their partner had collected over the course of their experiment. The latter was based on a veridical readout of how many points their partner had accumulated when they had done the fMRI decision-making experiment.

Note that, in the context of the decision-making experiment, self and partner were equally important for the task. Both self and partner decisions were equally frequent in the experimental schedule. The pay-off from a trial with the same DV was the same whether it was a partner trial or a self trial. Therefore, estimating the partner's performance had precisely the same influence on performing the task successfully as estimating one's own performance correctly.

Each decision lasted until a response was given. Afterwards, for 0.5 s, a box around either one's own team or the opponent's team indicated whether an engage (a box around one's own team) or an avoid decision (a box around the opponent team) had been made. After the first decision, there was a temporal jitter of 2–8 s (Poisson distributed, mean = 3.5 s) until the second decision started. Note that the second decision could not be a comparison between the same players as the first decision. This meant that after a group decision, there could not be another group decision in the same trial. After a self decision with O1 as the relevant opponent, there could be another self decision with O2 as the relevant opponent (but not another one with self and O1). After the second decision, there was an intertrial interval of 1–5 s (mean = 2 s), and then the next trial started.

**Experimental schedule.** The experiment comprised 144 trials. Therefore, 288 decisions were made, and these were evenly distributed between self decisions, partner decisions and group decisions. As described above, generating schedules from the pre-experiment by using an algorithm ensured precisely balanced performances for each of the four players for all participants.

**Basis functions and their weight vectors.** A set of three sequential basis functions form a basis for the sequential decision space in our study. We define a matrix $W$ comprising three row vectors of basis functions ($\mathbf{w}_1, \mathbf{w}_2, \mathbf{w}_3$) as $W = (\mathbf{w}_1, \mathbf{w}_2, \mathbf{w}_3)^T$, where $T$ denotes the transpose operation. We refer to the projections of the sequential performances observed during the observation phase onto the basis functions as $\mathbf{b} = (b_1, b_2$ and $b_3)^T$. We refer to the sequential performances scores as $\mathbf{pos} = (pos_1, pos_2, pos_3, pos_4)^T$. For example, $pos_1$ is the performance score observed for the first player in the observation phase sequence irrespective of identity; it is a number between 0 and 6 on each trial, reflecting the aggregated performance scores. Performance scores per player have to be extracted from the series of performance cues indicating either successful or erroneous performance presented for each player. We used the following set of basis functions $\mathbf{w}_1$, $\mathbf{w}_2$ and $\mathbf{w}_3$:
1. $\mathbf{w}_1 = [-1, 1, -1, 1]$,
2. $\mathbf{w}_2 = [1, 1, -1, -1]$,
3. $\mathbf{w}_3 = [1, -1, -1, 1]$.

The position of the weight indicates the sequential position of the respective player in the sequence of performances that were presented at the beginning of every trial. For example, $\mathbf{w}_1(2)$ refers to the second player in the sequence, who is given a positive weight. Importantly, basis functions are defined sequentially, and not in an agent-centric frame of reference (the latter uses positive signs for one's own team and negative signs for the opponent's team). The projections onto the basis functions are the dot product of the weight matrix and the sequential performance scores: $\mathbf{b} = W\mathbf{pos}$ (see the main text and Fig. 2 for an example calculation). This means the basis function projections are defined as:
4. $b_1 = \mathbf{w}_1 \cdot \mathbf{pos}$,
5. $b_2 = \mathbf{w}_2 \cdot \mathbf{pos}$,
6. $b_3 = \mathbf{w}_3 \cdot \mathbf{pos}$.

The three basis functions ($\mathbf{w}_1, \mathbf{w}_2, \mathbf{w}_3$) have two important features. The first is that they are pairwise orthogonal, and the second is that all group and dyadic weight vectors can be derived from them (that is, they form a basis for sequential decision space in our task);
7. $\mathbf{w}_i \cdot \mathbf{w}_j = 0$ for all $i \neq j$.

That the three basis functions form a basis for sequential decision space means that all the possible sequential comparisons afforded by our experimental design can be defined with them. First, the three weight vectors $\mathbf{w}_1$, $\mathbf{w}_2$ and $\mathbf{w}_3$ already represent all possible group decisions made in a sequential frame of reference. This means that they capture all the possible pairings of a team of two players with a positive sign and a team of two other players with a negative sign in a four-player sequence. Note that the overall signs of the weight vectors are arbitrary (for example, whether it is $[1, 1, -1, -1]$ or $[-1, -1, 1, 1]$) because we only care about the comparison itself, irrespective of whether a team is one's own team or the opponent's team. Inverted contrasts can easily be constructed by multiplication with −1 and are therefore omitted from the list of contrasts here (for example, $[-1, -1, 1, 1] = -1 \times [1, 1, -1, -1]$).

The dyadic weight vectors require participants to ignore two players and compare only one player per team. Therefore, they are expressed as contrasts, such as $[0, 1, -1, 0]$, that contain two zeros (the irrelevant players), and one positive and one negative weight (the relevant players being compared). For example, the contrast $[0, 1, -1, 0]$ indicates that the performance presented at the second time point in the sequence must be compared with the third performance of the sequence ($pos_2$ versus $pos_3$). Again, inverted dyadic weight vectors (such as $[0, -1, 1, 0]$) can easily be constructed by multiplication with −1. Regarding the dyadic comparisons, below is the complete list of all possible sequential dyadic weight vectors and how these comparisons are linear combinations of the basis functions $\mathbf{w}_1$, $\mathbf{w}_2$ and $\mathbf{w}_3$. Again, we omit sign-inverted contrasts. Note that the numbering of the contrasts corresponds to the main text (see Fig. 2).

8. $\mathbf{w}_4 = [0, 0, -1, 1] = ([-1, 1, -1, 1] + [1, -1, -1, 1])/2 = (\mathbf{w}_1 + \mathbf{w}_3)/2$
9. $\mathbf{w}_5 = [0, 1, 0, -1] = ([1, 1, -1, -1] - [1, -1, -1, 1])/2 = (\mathbf{w}_2 - \mathbf{w}_3)/2$
10. $\mathbf{w}_6 = [0, 1, -1, 0] = ([-1, 1, -1, 1] + [1, 1, -1, -1])/2 = (\mathbf{w}_1 + \mathbf{w}_2)/2$
11. $\mathbf{w}_7 = [-1, 0, 0, 1] = ([-1, 1, -1, 1] - [1, 1, -1, -1])/2 = (\mathbf{w}_1 - \mathbf{w}_2)/2$
12. $\mathbf{w}_8 = [1, 0, -1, 0] = ([1, 1, -1, -1] + [1, -1, -1, 1])/2 = (\mathbf{w}_2 + \mathbf{w}_3)/2$
13. $\mathbf{w}_9 = [1, -1, 0, 0] = ([1, -1, -1, 1] - [-1, 1, -1, 1])/2 = (\mathbf{w}_3 - \mathbf{w}_1)/2$

For these reasons, $\mathbf{w}_1$, $\mathbf{w}_2$ and $\mathbf{w}_3$ form an orthogonal basis for all sequential decision contrasts made in this experiment. Just as the three basis functions $\mathbf{w}_1$, $\mathbf{w}_2$ and $\mathbf{w}_3$ are sufficient to define all the relevant sequential comparisons in the context in this task, so are the projections onto the basis functions sufficient to compute the actual performance differences (the decision variables, associated with those contrasts). Note that participants also factor in the non-social bonus as well as this sequential decision variable (see the 'Decision phase' section). Because the weight vectors $\mathbf{w}_1$ to $\mathbf{w}_9$ define all the possible sequential comparisons, the DVs associated with these contrasts can be calculated as:
14. $DV_i = \mathbf{w}_i \cdot \mathbf{pos}$, for $i \in \{1, \ldots, 9\}$.

Because the dot product is distributive over vector addition, the fact that the three basis functions $\mathbf{w}_1$, $\mathbf{w}_2$ and $\mathbf{w}_3$ can be linearly combined to construct all other weight vectors implies that the projections onto the basis functions can be combined in precisely the same way to construct all the DVs:
15. $b_i + b_j = \mathbf{w}_i \cdot \mathbf{pos} + \mathbf{w}_j \cdot \mathbf{pos} = (\mathbf{w}_i + \mathbf{w}_j) \cdot \mathbf{pos}$.

Specifically, this means:
16. $DV_k = a_i b_i + a_j b_j = a_i \mathbf{w}_i \cdot \mathbf{pos} + a_j \mathbf{w}_j \cdot \mathbf{pos} = (a_i \mathbf{w}_i + a_j \mathbf{w}_j) \cdot \mathbf{pos}$, for $k \in \{1, 2, 3\}$ with $i, j \in \{1, 2, 3\}$, $a_i, a_j \in \{0, 1\}$ and
17. $DV_k = (a_i b_i + a_j b_j)/2 = (a_i \mathbf{w}_i \cdot \mathbf{pos} + a_j \mathbf{w}_j \cdot \mathbf{pos})/2 = (a_i \mathbf{w}_i + a_j \mathbf{w}_j)/2 \cdot \mathbf{pos}$, for $k \in \{4, 5, \ldots, 9\}$, with $i, j \in \{1, 2, 3\}$ and $a_i, a_j \in \{-1, 1\}$.

**Sorting of basis functions and transformation to choice.** The basis functions can easily be used to derive sequential decision variables from the observed performance sequence. Basis functions can be sorted on the basis of relevance. We define the primary basis function as the one that coincides with the groupings of the two teams in the four-player sequence during the observation phase. The projection onto the primary basis function serves as a sequential DV for group decisions. The secondary basis function is defined as the one other basis function that in combination with the primary basis function results in the dyadic decision that is currently relevant. Secondary basis functions are defined only for dyadic decisions and are known only once the decision is revealed. Finally, tertiary basis functions are defined as the remaining basis function that is not relevant for a current dyadic decision.

To reach an agent-centric DV, both the primary and secondary basis functions need to be in the reference frame of the two teams. If this is already the case, the agent-centric decision variable is a simple linear addition of both primary and secondary basis functions (plus the non-social bonus). However, sometimes the basis functions have to be sign inverted to align with the agent-centric perspective. We refer to this as sign inversion and refer to the process of transforming the basis function from a sequential frame of reference to an agent-centric social frame of reference as inversion of the basis function projections. Regarding the primary basis function, this means that the weights of the corresponding weight vector must be in accordance with the player identities and assign positive weights to the players of one's own group and negative weights to the opponents' group. The agent-centred primary projection is therefore independent of any sequence information, and simply assigns positive weights to one's own team and negative weights to the opponents' team. Hence, the agent-centred primary basis function captures the difference in performance between one's own team minus the performance of the other team. Note that our neural analyses investigating primary and secondary basis function projections contain, as control variables, these same basis function projections, but transformed in this agent-centric reference frame (inverted).

The secondary basis function also needs to be transformed into an agent-centric space to arrive at an agent-centric DV for dyadic decisions. In the same manner as the agent-centred primary basis function, the agent-centred secondary basis function assigns positive weights to the relevant player from one's own group, and negative weights for the relevant player from the opponents' group. However, different from the agent-centred primary basis function, the agent-centred secondary basis function assigns a negative weight to the irrelevant player from one's own group and a positive weight to the irrelevant player from the opponents' team. For example, in a self trial, the following set of weights is required to make the correct decision:
18. $\mathbf{w}_{DEC-self} = [S, P, Or, Oi] = [1, 0, -1, 0]$
(note that positions in this vector do not denote sequence positions, but simply refer to self ($S$), partner ($P$), relevant opponent (Or) and irrelevant opponent (Oi)).

The inverted (that is, agent-centric social) primary and secondary weight vectors for this comparison are:
19. $\mathbf{w}_{primary-inverted} = [\mathbf{S}, \mathbf{P}, Or, Oi] = [1, 1, -1, -1]$,
20. $\mathbf{w}_{secondary-inverted} = [\mathbf{S}, \mathbf{P}, Or, Oi] = [1, -1, -1, 1]$.

Note that, again, in these equations, positions inside the vectors do not refer to the sequential position in the observation phase, but simply denote a player's identity. In this manner, the irrelevant players cancel out when linearly combining the two agent-centred basis functions (compare with the example given in Fig. 3). For example, for self decisions, in which partner and one of the opponents is irrelevant:
21. $\mathbf{w}_{DEC-self} = [\mathbf{S}, \mathbf{P}, Or, Oi]$

$$= (\mathbf{w}_{primary-inverted} + \mathbf{w}_{secondary-inverted})/2$$

$$= ([1, 1, -1, -1] + [1, -1, -1, 1])/2$$

$$= [2, 0, -2, 0]/2$$

$$= [1, 0, -1, 0].$$

As the dot product is distributive over vector addition (see 'Basis functions and their weight vectors' section), it follows that the primary and secondary basis projections, when averaged together, provide a simple route to the agent-centric decision variable for dyadic decisions (note that the non-social bonus still needs to be added and this is considered accordingly in all analyses).

**Imaging data acquisition and preprocessing.** Imaging data were acquired using a 3-Tesla Siemens MRI scanner with a 64-channel head coil. T1 weighted structural images were collected with an echo time (TE) of 3.97 ms, a repetition time (TR) of 1.9 s and a voxel size of 1 mm × 1 mm × 1 mm. Functional images were collected using a multiband T2*-weighted echo planar imaging sequence with an acceleration factor of two with TE = 30 ms, TR = 1.2 s, a voxel size of 2.4 mm × 2.4 mm × 2.4 mm, a 60° flip angle, a field of view of 216 mm and 60 slices per volume. Most scanning data was collected with an oblique angle of 30° to the PC–AC line to avoid signal dropout in orbitofrontal regions[56]. Two field-map scans (sequence parameters: TE1, 4.92 ms; TE2, 7.38 ms; TR, 4482 ms; flip angle, 46°; voxel size, 2 mm × 2 mm × 2 mm) of the B0 field were also acquired and used to assist distortion–correction.

The FMRIB Software Library (FSL) was used to analyse the imaging data[57]. We preprocessed the data through field-map correction, and temporal (3 dB cut-off, 100 s) and spatial filtering (Gaussian using a full-width half-maximum of 5 mm) and using the FSL MCFLIRT to correct for motion. The functional scans were registered to standard MNI space using a two-step process: first, the registration of subjects' whole-brain EPI to T1 structural image was done using BBR with

(nonlinear) field-map distortion–correction; and second, the registration of the subjects' T1 structural scan to a 1 mm standard space was done using an affine transformation followed by nonlinear registration. We used the FSL MELODIC to filter out noise components after visual inspection.

**fMRI whole-brain analysis.** We used FSL FEAT for first-level analysis. First, data were pre-whitened with FSL FILM to account for temporal autocorrelations[57]. Temporal derivatives and standard motion parameters were included in the model and we used a double gamma HRF[58,59]. Results were calculated using automatic outlier-deweighting and FSL FLAME 1 with a cluster-correction threshold of $z > 3.1$ and $P < 0.05$.

For all whole-brain analyses, all non-constant regressors were normalized to a mean of zero and a standard deviation of 1. In self and partner decisions, we refer to O1 and O2 as the Or and Oi, depending on whether participants were asked to compare their performance or not.

**fMRI GLM1.** In a first GLM (fMRI GLM1), we modelled each RDK as a 2-s constant event time-locked to its onset. This constant captured the player-unspecific variance in the BOLD signal for all random dot motion events. As well as this constant, we specified four parametric regressors that were specific to the performance of each of the players and captured their parametric performance score for this trial (0–6). These regressors also had a duration of 2 s to match the constant's duration and were time-locked to the onset of the corresponding players' RDK. Related to participants' decisions, we constructed six regressors to capture the main activation for decisions, binned by condition and decision number (first or second after the RDK). This meant we had one constant for self decisions that came first (S1) and one constant for self decisions that came second (S2), and did the same for partner decisions and group decisions (termed P1, P2, G1 and G2). These constants had a duration of 2 s, which was the average time participants took to make choices. Furthermore, parametric regressors of interest were time-locked to the same constant effects. For self and partner decisions, we used the following parametric regressors: S performance (indicating performance score associated with self); P performance (indicating performance score associated with partner); Or performance (indicating performance score associated with the relevant opponent); Oi performance (indicating performance score associated with the irrelevant opponent); and bonus.

This meant that we used four sets of these parametric regressors, which were each time-locked to the onsets of S1, P1, S2 and P2. The duration of these regressors were also set to 2 s to match the main effects. For group decisions, we used the following set of parametric regressors, each of a duration of 2 s: S performance, P performance, O1 performance, O2 performance and bonus.

Using the same logic as for the other trial types, we used two sets of these regressors, separately time-locked to G1 and G2. Note that O1 and O2, the two opponents, were clearly identifiable because the letters O1 and O2 were overlaid over their cues. We coded the fMRI regressors in line with these identities, even though other features of the opponents, such as their position on screen and colour, were randomized across participants (see Extended Data Fig. 1 for details of the visual presentation). Finally, as regressors of no-interest, we modelled button responses as regressors time-locked to all button responses, setting the duration to a standard duration of 0.1 s.

In Fig. 1, we present the effects of S performance during S2 and P performance during P2.

**fMRI GLM 2.** In the second GLM (fMRI GLM2), we focused on the representation of the basis functions towards the end of the observation phase. As in all analyses related to the basis functions, we tested the parametric effects of the trialwise projections onto the basis functions. We modelled the constant effect of RDKs by time-locking a stick function (duration of 0.1 s) to a time 2 s after the offset of the last RDK in the sequence of four RDKs that were presented at the start of each trial. This time point coincided precisely with the average onsets of the first

decision. We time-locked several parametric regressors to the same time point, each with the same standard duration of 0.1 s: $b_1$, $b_2$ $b_3$, S performance, P performance, O1 performance and O2 performance.

Again, each parametric regressor was normalized. We also used two parametric regressors related to the position of self and partner in the sequence (S-position and P-position). Each could have a value between 1 and 4 depending on the sequential position of that player. We time-locked these latter two parametric effects to the offset of the last RDK in the sequence when all performances have been presented.

We took care to include regressors that account for decision-related activity. We coded the different decision types as three constants, each with a duration of 2 s, as in the previous design: S, P and G decisions. Each constant was accompanied by parametric regressors of the same timing that captured decision related activations: DV, the DV relevant for the current decision, including bonus; DV × C, DV in interaction with choice (engage or avoid on the current trial); choice, a binary variable coded as engage/avoid; DVi, the performance difference of the irrelevant players, coded as own team member versus opponent team member (only defined for self and partner, not group decisions); and DVi × C, DVi in interaction with choice.

All interactions were calculated by normalizing both components of the interaction to a mean of zero and a standard deviation of 1, and then multiplying both. Finally, as regressors of no-interest, we modelled button responses as a regressors time-locked to all button responses, setting the duration to a standard duration of 0.1 s.

On the contrast level, we combined all basis function projections ($b_1$, $b_2$ and $b_3$) and the S-position regressor, each weighted evenly ([1, 1, 1, 1] contrast; Fig. 3).

**fMRI GLM 3.** In this design (fMRI GLM3), we again modelled each random dot-motion kinematogram as a 2 s constant event time-locked to its onset. We combined self and partner decisions to one category (dyadic trials, DY), but split by number of decisions (DY1 for the first decision of both self and partner decisions, and DY2 analogously). The duration of the decision events was set to 2 s, as in the other designs. We time-locked parametric regressors to the DY trials, but separately to DY1 and DY2. These regressors had the same timing parameters as the respective decision constants and they were:

1. primary basis function;
2. secondary basis function;
3. tertiary basis function;
4. inverted primary basis function (the primary basis function transformed to an agent-centric, social frame of reference; see above);
5. inverted secondary code inversion (yes or no);
6. inverted primary basis function in interaction with choice;
7. inverted secondary basis function in interaction with choice;
8. bonus;
9. bonus in interaction with choice; and
10. inverted secondary basis function (transformed to an agent-centric, social frame of reference; see above).

We calculated the combined effect across both DY1 and DY2 for the inverted primary and the inverted secondary basis function (see 4 and 10 in above list) using a [1, 1] contrast. We then averaged both of these combined contrasts to estimate the overall effect of inverted primary and secondary basis function combined. Furthermore, we modelled group decisions as a separate constant regressor, collapsed over both the first and second decisions. The duration of this regressor was set to 2 s and we time-locked the following regressors to it: primary basis function; inverted primary basis function; inverted primary basis function in interaction with choice; bonus; and bonus in interaction with choice.

For all the above regressors in the GLM, if they are related to the basis functions, they refer to the trialwise projections onto the basis functions. On the contrast level, for DY1 trials (dyadic decisions that came first), we combined the first two basis function projections linearly ([1, 1]; primary + secondary basis function; regressors 1 and 2 in the

above list). We also contrasted them with the tertiary basis function projection ([1, 1, −1]; primary + secondary − tertiary function; regressors 1, 2 and 3 in the above list). We also calculated the dyadic decision variable in the reference frame of choice (as chosen versus unchosen). We did this by combining regressors 6, 7 and 9 in the list above over both DY1 and DY2 trials.

**ROI analyses.** ROIs had a radius of three voxels and were centred on peak voxels of significant clusters. To guarantee statistical independence, we analysed only those variables that were independent of ROI selection and only epochs that were temporally dissociated from the time period that served for ROI selection. For ROI time-course analyses, we extracted the preprocessed BOLD time courses from each ROI and averaged over all voxels of each volume. The time courses were normalized (per session, as for subsequent analyses), oversampled by a factor of ten (using cubic spline interpolation, as for subsequent analyses) and, in a trialwise manner, aligned at the time point of interest. We then applied a GLM to each time point and computed one beta weight per time point, which resulted in a time course of beta weights for each regressor. We used a leave-one-out procedure to conduct significance tests on the beta-weight time courses. For this, in a predefined time window, we calculated the absolute peak of the time course (defined as the maximal deviation from zero, either positive or negative). We did this for all participants except a left-out participant. We then determined the beta weight of the left-out participant at the time of the peak of the remaining group. In this manner, we determined a beta weight for every participant, which, importantly, was independent of the participant's own data. We subsequently performed $t$-tests against zero on these beta weights.

We used two time-course designs, both time-locked to the end of the observation phase, which was on average 2 s before the onset of the first decision. All regressors were normalized to a mean of zero and a standard deviation of 1. ROI GLM1 comprised the following regressors: $b_1$, $b_2$, $b_3$, S performance, P performance, O1 performance, O2 performance, S-position (the sequence position of self, which can be can be 1, 2, 3 or 4) and P-position (the same, but for the partner).

ROI GLM2 comprised a similar set of regressors: $b_1$, $b_2$, $b_3$, null vector, S-position and P-position.

Note that the null vector from ROI GLM2 is the sum of the performance of all players and hence cannot be part of ROI GLM1. For the analysis of the effects, we used an analysis time window of 4–10 s after the observation phase offset. To distinguish those effects from even earlier effects linked to the observation phase itself, we conservatively used an earlier time window of 0–6 s after the offset of the last RDK (only used for S-position; see the main text). The significance of the basis function projections and S-position was tested in ROI GLM1, and the null vector was tested for significance in ROI GLM2.

**Choice simulations.** We simulated, analysed and visualized data using Matlab 2021a, Jasp v.0.16 and gramm[60].

We simulated choices in our experiment to examine the effects of the primary and secondary basis functions on decision making. For all these analyses, primary and secondary basis function projections are inverted (expressed in an agent-centric, social reference frame (see 'Sorting of basis functions and transformation to choice'). We consider self decisions, but all results hold when simulating partner decisions accordingly.

We simulated choices as a linear combination of primary basis projection, secondary basis projection and the non-social bonus (see the DV definitions in the beginning of the Methods section; $DV_{self} = S − Or + B$). As we have shown analytically above, the correct agent-centric decision variable is given by the linear combination of these three variables, because primary and secondary basis projections in combination result in the performance difference of $S$ and $Or$, ignoring the two other players. Therefore, simulated choices used a logistic link function and a set

of weights for the three predictors ($\mathbf{w}_{prim}$, $\mathbf{w}_{sec}$ and $\mathbf{w}_{bonus}$) to estimate choice probabilities, which were then binarized to an engage (1) or avoid (0) decision with a likelihood based on the choice probability. We simulated self decisions for all participants with 200 simulations per participant. We subsequently fitted an agent-centric logistic GLM using S performance, P performance, Or performance, Oi performance and the bonus. Finally, we averaged and plotted beta weights from this GLM and examined the qualitative effects of irrelevant players (P and Or) on self decisions.

We simulated choices under two regimes. For both, we chose weight vectors that resulted in beta weights of similar magnitudes to those observed in our behavioural analyses. The first regime was the 'balanced' regime, which used identical weights for primary and secondary basis functions, as one would optimally use to analytically derive the agent-centric decision variable ($\mathbf{w}_{prim} = 1.5$, $\mathbf{w}_{sec} = 1.5$ and $\mathbf{w}_{bonus} = 1.5$). The second regime used a relative 'overweighting' of the primary over the secondary basis function, as indicated by our previous analyses ($\mathbf{w}_{prim} = 1.7$, $\mathbf{w}_{sec} = 1.3$ and $\mathbf{w}_{bonus} = 1.5$).

**Behavioural data analyses.** We used logistic GLMs to capture the weights participants assigned to different pieces of information when making their decisions. We predicted participants' choices to engage (versus avoid) as a function of a normalized set of regressors (each regressor had a mean of zero and a standard deviation of 1). We applied the GLMs separately to self and partner decisions. The GLM comprised the performance of self (S) and partner (P) as well as the two opponents, separately coded as the relevant opponent (Or; the one whose performance was to be considered in the dyadic comparison) and the irrelevant opponent (Oi; the one whose performance was irrelevant for the dyadic comparison). The GLM also contained the non-social bonus.

**Drift diffusion modelling.** A time-varying drift diffusion model[43] (tDDM) was fitted to the choice outcome (engage or avoid) and reaction time data of our participants. The tDDM expands the standard DDM[60,61] by allowing for different onset times of the attributes that influence the evidence accumulation process. Specifically, our tDDM allowed for different onset times between the primary and the secondary basis function (but in agent-centric space). We estimated a total of seven free parameters separately for each participant and experimental condition using the differential evolution algorithm[62]. The free parameters were the weights of the primary and secondary basis function, the weight of the bonus, the difference in onset times between the primary and the secondary basis function, the decision threshold, the starting-point bias and the non-decision time. The difference in onset times was estimated relative to the onset of the primary basis function. Thus, a positive difference indicates that the secondary basis function entered the evidence accumulation process later than the primary basis function, whereas a negative difference indicates that the secondary basis function entered the evidence accumulation process earlier. The bonus always entered the accumulation process at the same time as the function with the earlier onset. We optimized the tDDM parameters by simulating 3,000 decision outcomes and reaction times per iteration for each unique combination of primary function, secondary function and bonus that the respective participant encountered during the experiment. For any given participant, this could be a subset of all possible combinations, and some combinations could have been encountered repeatedly. The parameters were adapted from iteration to iteration to maximize the likelihood of the empirical data, given the distributions generated from the simulated decisions over a total of 150 iterations.

## Study 2: behavioural experiment
**Study procedures and data acquisition.** We ran a behavioural experiment online using Prolific (www.prolific.com). The experiment took one hour, and participants were paid £9 for taking part. The ethics committee of the University of Oxford approved the study and all

participants provided informed consent (MSD reference number: R70000/RE001). The experiment was programmed using jspsych[63] and the random-dot-motion toolbox[64]. As inclusion criteria, we used the age range of 18–40, and fluent English speakers were recruited in the United States and the United Kingdom. As in the fMRI study, participants first performed a behavioural pre-experiment described as a performance assessment. This comprised, in quick succession, random dot-motion stimuli of varying coherence and took in total about 3 min. Participants were informed that the pre-experiment was relevant for the next part of the study, when they were shown samples of their performance in a group decision-making experiment. After the pre-experiment, the instructions for the decision-making experiment followed. This part, again, was modelled on the fMRI study. After the instructions to the decision-making experiment, participants passed a comprehension check that asked three questions about the task rules. The participants went on to do the experiment only if they responded correctly to all three questions. Otherwise, the experiment was aborted and participants were given a small amount for their time investment (about 5–10 min at this point). After the decision-making experiment, participants filled in some questionnaires unrelated to the purpose of this study. The study was conducted over a time period of three weeks. Data for all versions of the experiment were acquired in parallel. For participants who completed the study multiple times, we only considered their initial participation and discarded subsequent data sets.

Of the 805 data sets collected, we excluded participants who: took longer than 2 h to complete the experiment; took longer than 30 s to respond to decision trials in more than 10% of trials; and showed a choice repetition bias in the initial performance assessment or the decision-making part of the study. A choice repetition bias was defined as picking the same choice (left or right button) in more than 85% of trials. This led to the exclusion of 11 participants overall. The final sample comprised 795 participants.

**Initial performance assessment.** After starting the experiment, participants entered the performance assessment stage, which they were told was important for the second, main part of the study. Participants estimated the motion direction of an RDK stimulus and responded with left/right buttons to indicate the corresponding direction. The performance assessment comprised 120 RDK trials, lasting in total about 3 min. The motion coherences were set at 0.512 for 20% of the trials and 0.032 for 80% of the trials, with 0.512 being a higher coherence and therefore an easier decision. Each RDK stimulus was presented for a maximum duration of 1 s, requiring participants to make their decisions within that time frame. If they took longer, they would see a 'Missed!' message on the screen and the trial would be marked as incorrect. The performance assessment consisted of two sub-parts. In the first sub-part, participants received feedback on their decisions (yellow circle for correct or red cross for incorrect). Following that, the task continued without any feedback. The reason for this was to prevent participants from becoming fully aware of their performance levels before the main experiment, when they would be exposed to similar stimuli.

**Decision-making schedules and across-subject conditions.** We modelled the behavioural group decision-making experiment closely on the fMRI study. The behavioural experiment comprised 108 trials. As in the fMRI experiment, each trial consisted of an observation phase and two subsequent decisions (a total of 216 decisions). The only modifications made served to shorten and simplify the task slightly, to adjust it to the time frame and complexity of large-scale online studies. We used no temporal jitters between the observation phase and the first decision phase, and no temporal jitters between the first and the second decision phase of each trial. Furthermore, to slightly reduce the difficulty of the task, we extended the time that a performance cue was shown during the observation phase from 200 ms to 300 ms,

with 100 ms delay between cues. Otherwise, the trial timeline was the same as in the fMRI study. The non-social bonus that was symbolized by yellow and red coins during the decision phase in the fMRI experiment was now symbolized by different colours of the arrow that indicated the players to be compared. A positive bonus was indicated by a yellow arrow, and a negative bonus was indicated by a red arrow. This simple colour-coding scheme was possible because we used only two magnitudes of the bonus in the behavioural study: −0.5 and 0.5.

We used four versions of the experimental schedule, arranged in a 2 × 2 between-subjects design; the two schedules comprised self, partner and group decisions (the group condition) and two schedules comprised only self and partner decisions (the no-group condition). A schedule was defined by the assignment of performance scores to players for each trial during the observation phase, and by the assignment of the bonus and decision type to each decision in the study. The comparison of participants' choice behaviour between group and no-group conditions was the focus of this study. The group condition comprised 72 group decisions, 72 self decisions and 72 partner decisions. In half of the self decisions, participants compared their own scores with those of O1 (36 decisions), and in the other half, with O2 (36 decisions). This meant that O1 and O2 were the relevant opponent for the self equally often. The same was true in partner decisions, with O1 and O2 being the relevant opponent equally often. The schedules used in the no-group condition were generated by replacing group decisions with self and partner decisions in equal number (keeping the bonus for all decisions the same). This resulted in 108 self decisions and 108 partner decisions in total. Again, the relevant opponent for each decision type was O1 and O2 equally often. Note that we analysed only self and partner decisions that were identical across conditions (matched decisions). We discarded self and partner decisions in the no-group schedule that were replacements of the group decisions, because these decisions had no direct correspondence across conditions. This resulted in 72 matched self decisions and 72 matched partner decisions. In this way we were able to compare identical decisions in the two conditions, but the identical decisions took place in the context of group decisions in the group condition but not in the no-group condition.

Although the schedule defined precisely which performance score was presented for which player on which trial, it did not specify the order in which the players were shown. To determine this, and to avoid any possibility that idiosyncrasies of the player order confounded our results, we generated 1,000 shuffled player-order sequences. Each sequence presented each possible player order equally often. With four players, 4 × 3 × 2 × 1 = 24 different player orders are possible. We used those for 4 × 24 = 96 trials. For the final 12 trials (the experiment comprised 108 trials), the player orders were randomly selected for each of the 1,000 player sequences. Then, for each participant performing the behavioural experiment, one player order was selected randomly out of the 1,000.

Overall, we used four experimental schedules. For both group and no-group conditions, we used two schedules that differed only in the precise assignment of scores to players that were shown during the observation phase (schedule 1 and schedule 2). The behavioural differences between these schedules were not of interest for the study's research question. The two versions were used only to assess the stability of our experimental effects across numerical differences in the information that was to be remembered. For the same schedule version, group and no-group conditions differed only in the presence or absence of the group decisions. Therefore, we made sure to acquire a similar number of participants for each schedule version for both group and no-group conditions. This ensured an equal number of participants in the conditions that we meant to compare in our study as follows: schedule 1, 192 participants in the no-group condition and 190 in the group condition; and schedule 2, 207 participants in the no-group condition and 206 in the group condition.

**Data analysis by ridge regression.** We analysed the behavioural data using Matlab 2021a and Jasp v.0.16. We fitted a logistic GLM to the choice data that we had also used for the fMRI sample. All regressors were normalized (mean of 0, standard deviation of 1) and predicted the choice to engage (1) or avoid (0), that is, whether one's own team member was judged to be the better performer. As in the fMRI data set, we included S performance, P performance, Or performance, Oi performance and the bonus in the regression. Because the experiment, for timing reasons, comprised fewer trials per participant, we used ridge regression[65,66] to estimate the regressors' beta weights (the effect sizes). Ridge regression penalizes large beta weights according to a regularization coefficient $\lambda$ and thereby prevents overfitting and improves generalization. This is appropriate for cases such as ours in which there are many regressors and comparatively few trials. We applied the regression model to all sessions using Matlab lassoglm (setting $\alpha$ to a very small value) in the following way. First, we determined an appropriate regularization coefficient $\lambda$. To do so, we repeatedly fitted the GLM to each individual dataset while varying $\lambda$ between zero and $10^{-3}$ to $10^{-1}$ (log-spaced). During each fit, we used a three-fold cross-validation approach to determine the overall model deviance for each $\lambda$ for all datasets combined. We repeated this procedure twice. Finally, we selected the $\lambda$ that resulted in the lowest overall model deviance. This is the $\lambda$ with the best cross-validated model fit, which was then used to run the ridge GLM of interest. Importantly, the same best-fitting $\lambda$ was used for all participants, irrespective of condition, to enable fair statistical comparisons of beta weights within and across conditions.

**Calculation of players' decision relevance.** We calculated the decision relevance of each player over an experimental schedule in the following way. If the player was irrelevant for a decision (for example, the partner in self decisions), their relevance score was zero. In dyadic trials, both relevant players' relevance scores were set to 0.5 (that is, self and relevant opponent in self decisions). In group decisions, each player's relevance score was 0.25 (self, partner, O1 and O2 contribute equally). We determined the relevance scores for all players for all decisions in an experimental schedule, added them up and divided by the number of trials.

### Study 3: control fMRI experiment
**Participants.** There were 36 participants. One participant did not complete the scanning session and two participants could not perform the experiment owing to problems with the button box. One participant moved extensively in the MRI scanner and during melodic preprocessing[57], and we discovered that very few fMRI components were noise free. These four participants were removed from the sample. The final sample contained 32 participants (age range 19–39 years, 22 female). Participants received £70 for taking part in in the experiment and received extra earnings that were allocated according to their task performance, mirroring the social fMRI experiment. The ethics committee of the University of Oxford approved the study and all participants provided informed consent (MSD reference number: R60547/RE001).

**Experimental procedures.** As in the main social fMRI experiment, participants performed a behavioural pre-experiment before entering the MRI scanner. However, the framing of the pre-experiment was very different from that of the social fMRI study. Instead of framing the task as a social decision-making experiment, we framed it as a motor task. Participants were told that the experiment was about learning and remembering motor sequences, in particular, sequences of finger taps. To this end, participants performed 'tap training' as pre-experiment preparation. They were shown sequences of finger taps and were asked to repeat these sequences. Four buttons were used in the pre-experiment, assigned to the index and middle fingers of the left and right hands. The screen showed the outlines of two hands with the four fingers highlighted (Fig. 5 and Extended Data Fig. 11). During the tap training, one finger at a time was highlighted and participants pressed

the corresponding buttons. Fingers were highlighted repeatedly, and participants were asked to press the corresponding buttons until the sequence ended. Incorrect button presses and long response times were counted as error trials and led to the sequence being repeated until it was completed without error. The pre-experiment comprised 15 trials. Participants performed the pre-experiment for approximately 15 min.

The pre-experiment helped create the cover story that the study was about motor sequences. However, the pre-experiment was designed to have clear analogies to the observation phase of the social fMRI experiment, which presented sequences of successful (indicated by yellow coins) and erroneous (indicated by red Xs) performance scores. The motor pre-experiment similarly comprised yellow coins to indicate that a button should be pressed and a red X to indicate that a button press should be omitted. The resulting sequence of finger taps in the control fMRI experiment resembled the sequence of four players' performance scores in the social task. In our social experiment, performance scores for four players had been shown in sequence, but now, the required number of presses for four fingers were shown in a sequence. There were always six yellow/red cues shown per finger (indicating either an executed button press or button press omission), just as six performance cues had been shown per player in our previous social experiment. Finally, in the same way that self and partner were one team, and O1 and O2 were another team in the social experiment, the four fingers naturally grouped together as the two fingers of the left hand and the two fingers of the right hand, so the two hands corresponded to the two teams in the main control fMRI experiment. Just as participants had compared performance scores across players in the social experiment, participants of the control fMRI experiment completed an fMRI experiment, in which they compared the number of finger taps between the fingers of the two hands. Participants never made decisions between fingers from the same hand, just as they had not made comparisons of players from the same team in the social task.

The behavioural pre-experiment that took place before scanning and the control fMRI experiment were programmed in Matlab using Psychtoolbox-3 (ref. 55; http://psychtoolbox.org) and in Psychopy (https://www.psychopy.org/).

The main experiment was designed to closely match the rationale and the statistics of the social fMRI experiment. The difference between the experiments comprised the framing of the task as a motor experiment versus a social experiment.

In the same way that there was continuity between the pre-experiment and the main experiment in the social study, there was a clear relationship between the pre-experiment and the main experiment here too. In both, participants were shown a display outlining their hands, highlighting the left middle (LM) and the left index finger (LI), as well as the right index (RI) and the right middle finger (RM). In the main control fMRI experiment, the hands were placed on the left and right sides of a button box. As in the pre-experiment, participants observed sequences of yellow and red cues indicating executed finger taps and finger-tap omissions. Their task was to remember the number of taps per finger (the number of yellow coin symbols per finger, ranging from 0 to 6) to make good decisions in the second part of the trial. However, importantly, participants did not concurrently press the highlighted buttons. They were purely observing them in the same way as they were observing the performance scores in the social fMRI study. After the observation phase, participants made decisions about the observed information (the number of presses per finger). These decisions were framed as decisions between the two hands (mimicking the two teams from the social fMRI experiment). We arbitrarily mapped the identities of the players from the social task on the control fMRI experiment. We identified LM as motor–self, LI as motor–partner, RI as motor–opponent 1 and RM as motor–opponent 2. This mapping was consequential, because we utilized exactly the same experimental schedules as we had used for the social experiment for the control fMRI experiment. Therefore, 33% of decisions that were previously self decisions were now LM/motor–self

decisions comparing the number of button presses between LM and one of the 'opponent fingers' RI or RM (both in equal number); 33% of decisions were partner decisions comparing LI/motor–partner with either RI or RM (both in equal number). The remaining 33% of group decisions were now decisions that asked participants to compare the overall number of button presses for the left hand with the overall number of button presses for the right hand. In all decision types, decisions in favour of the left or right hand were to be made by a press using the thumb of the congruent hand. As in the social fMRI experiment, every decision comprised a bonus that had a value of −1.5, −0.5, +0.5 or +1.5. The timing of all trial events was precisely the same as for the social fMRI experiment. We also used the same pay-off scheme as for the social version, meaning that decisions in favour of the left hand led to win or loss of points accumulated, and decisions in favour of the right hand led to no change in overall points accumulated during the experiment. Although this pay-off scheme may have seemed arbitrary in the context of the control fMRI experiment, it ensured that the behavioural and neural results could be compared across the two experimental versions.

**Experimental schedules.** As in the social fMRI study, this experiment comprised 144 trials; 288 decisions were made, which were distributed evenly between motor–self, motor–partner and both-hands/group decisions. Crucially, we used the same experimental schedules that we had created for the social fMRI experiment. As described above, the four players were mapped onto the fingers. This equivalence meant that precisely the same scores seen for the players in the social experiment were now seen for the corresponding fingers of the motor study. The scores were shown in precisely the same temporal position during the observation phase of the trial, in a manner corresponding exactly to the social version of the experiment on every single trial. And the same logic applied to the decision types (including the size of the bonus) that were precisely matched with the same choice being correct and the same pay-off being at stake for corresponding trials of the two experiments. Moreover, using identical schedules also streamlined the timings across both experiments. The control fMRI experiment therefore used identical timings to the social fMRI experiment for the observation phase, the decision phases and all temporal jitters. In short, from a numerical perspective, the experimental design, as well as the requirements for solving decisions in both experiments, were identical. The experiments differed only in their framing as a social experiment versus a motor sequence task.

**Behavioural and fMRI analyses.** All behavioural and neural analysis of the motor study closely resembled the analyses run for the original social fMRI study. Behaviourally, we analysed the percentage of correct motor–self and motor–partner decisions, as well as the median reaction types in motor–self and motor–partner decisions. We also calculated the same logistic GLM models for motor–self and motor–partner decisions predicting engage/avoid decisions as a function of the performance scores for motor–self, motor–partner, motor–Or, motor–Oi and the bonus.

We acquired neuroimaging data using the same acquisition protocol and implemented an identical preprocessing pipeline and parameters for the whole-brain analyses (see above for details). The fMRI whole-brain designs included the same set of regressors with identical timings to the social fMRI study (fMRI GLM1 and fMRI GLM2), ensuring that main and control task GLMs comprised the same degrees of freedom. We calculated one extra contrast for fMRI GLM2, which was the mean of all basis functions ($b_1$, $b_2$ and $b_3$ weighted by a [1, 1, 1] vector). We set up a new GLM (fMRI GLM4), which was identical to fMRI GLM2, except for one aspect: we replaced the parametric effects of motor–S, motor–P, motor–O1 and motor–O2 with the mean of these three parameters, which corresponded to the null vector. This mirrors to the way in which we tested the null vector in the social fMRI study, which was the purpose of this new GLM.

Masks for ROI analyses had a radius of three voxels (the same radius as in the original social fMRI experiment) and were centred on peak voxels of significant clusters. To guarantee statistical independence, we analysed only variables that were independent of ROI selection and only epochs that were temporally dissociated from the time period that served for ROI selection. Within ROIs, we extracted contrast of parameter image (COPE) values from the whole-brain design to assess significance[57,67]. In this way, we isolated the effects that correspond to S performance in self decisions and P performance in partner decisions (for both analysing only the second decision in a trial), as displayed in Fig. 1 for the social study (fMRI GLM1). This is how we assessed the neural effects of motor–S and motor–P. For fMRI GLM2, we extracted the COPEs for the three basis function projections as well as the contrast relating to their mean (see above). For fMRI GLM4, we extracted the null vector. Furthermore, we used pre-threshold masking to assess the whole-brain significance of the mean effect of $b_1$, $b_2$ and $b_3$ combined. The mask was centred on the pgACC peak from the social fMRI study (MNI = (−8, 42, 14)) and had a radius of 20 mm. We used a threshold of $z > 3.1$, $P = 0.05$ family-wise-error corrected[25,68].

## Study 4: supplementary behavioural study

**Study procedures and data acquisition.** We ran a second behavioural experiment online (Extended Data Fig. 5). The procedures for the study and the data acquisition were the same as for the first online experiment, study 2, and the initial performance assessment was identical. The same exclusion conditions applied, with the only difference to the other behavioural study relating to the choice repetition criterion, which we applied separately to the training and the test phase of the experiment (instead of applying it to the entire decision-making session). We excluded 29 participants on the basis of our exclusion criteria, resulting in a final sample of 1,022 participants.

**Decision-making schedules and across-subject conditions.** As before, we modelled the group decision-making experiment closely on the fMRI study. We implemented the same timing and complexity adjustments as for the other behavioural study. This experiment comprised 96 trials. As in the fMRI experiment, each trial consisted of an observation phase and two subsequent decisions (so there were 192 decisions). For each trial, one subsequent decision was a group decision, and one subsequent decision was a dyadic decision (either a self or a partner decision). This also meant that overall, in the entire decision-making experiment, 25% of the decisions were self decisions, 25% were partner decisions and 50% were group decisions. Each decision type was presented as often as the first and second decisions.

We used a between-subject design with four different experimental schedules. These schedules differed only with respect to the sequential order in which players were presented on a given trial. The performance schedule (which performance score was assigned to which player in a given trial) and the decision schedule (which decision was a self/partner/group decision on a given trial) were identical in all four versions. Therefore, we could be certain that any systematic difference in decision-making behaviour across schedules had to be caused by the sequential player position information.

The four experimental schedules were organized in a 2 (training) × 2 (test) design. All schedules comprised 64 initial training trials and 32 subsequent test trials. The difference between the training and test trials was exclusively related to the sequential order of the players in the observation phase in each trial. All decisions and performance scores were identical across schedules for each trial. However, there were two different training schedules: pre3 and pre4. For pre3, in all training trials, when a dyadic decision was cued, the dyadic decision took place between the second and the third player of the sequence. For pre4, all dyadic decisions took place between the second and the fourth player. As a result, for all four schedules, during the training phase, a dyadic decision would always concern the second player of

the four-player sequence. However, in pre3, the second player would always be compared with the third player, and in pre4, the second player would always be compared with the fourth player of the sequence. In other words, we changed the order in which players were presented in a systematic way between schedules, but we kept the assignment of performance scores to players constant. This also meant that, for a given schedule, the dyadic decision that would follow was entirely determined by the sequence in which the players were shown in the observation phase. After the 64 trials of the training phase, the test phase followed, comprising 32 trials. There were also two schedules for the test phase: post3 and post4, and they followed the same logic as before. In all test schedules, the first element of the sequence became relevant for dyadic decisions. Critically, the second relevant player was either the player in the third position (post3) or the player in the fourth position (post4). These schedule designs meant that the same player positions always remained relevant for dyadic decisions for long phases of the experiment. However, this was not easy for participants to notice because it was obscured by the fact that only half of the decisions were dyadic decisions, and the other half were group decisions. In group decisions, all four players were always relevant. As noted above, participants reported no awareness of this manipulation. We collected a similar number of participants for each of the four conditions: pre3/post3 had 270 participants; pre3/post4 had 252 participants; pre4/post3 had 250 participants; and pre4/post4 had 250 participants.

**Estimation of sequential decision weights.** For estimation of the decision weights associated with the sequential positions, we regressed the engage/avoid decisions from dyadic trials onto five predictors: the performance scores of the first, second, third and fourth player in the sequence, as well as the bonus. Again, we used the same ridge-regression procedure as for the other behavioural study (study 2). However, because we fitted fewer trials (only 64 decisions per participant from the testing phase), we adjusted the range in which we searched for the best-fitting $\lambda$ to range between $10^{-3}$ and $10^{0}$. As scores associated with the sequential positions, we used the performance scores of self and partner (numbers between 0 and 6). We inverted the performance scores of the opponents (that is, recoded 6 as 0, 5 as 1, 4 as 2, 3 as 3, 2 as 4, 1 as 5 and 0 as 6) because the performance range was symmetrical around 3.

### Reporting summary
Further information on research design is available in the Nature Portfolio Reporting Summary linked to this article.

## Data availability
The data used in this study are available at https://github.com/wittmannlab/BasisFunctions2024. Source data are provided with this paper.

## Code availability
The code used in this study is available via https://github.com/wittmannlab/BasisFunctions2024.

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

**Acknowledgements** We thank L. Hunt, J. O'Reilly and C. Summerfield for comments on the project; and T. Behrens for feedback and comments on an early draft of the manuscript. M.K.W. was supported by an MRC CDA (MR/Y010477/1). Y.L. was supported by the China Scholarship Council (grant 202206040126) and the Major Project of National Social Science Foundation (20&ZD153). L.S. was supported by an Oxford-Radcliffe Medical Sciences Graduate School Scholarship and the Medical Research Council (MR/N013468/1). N.K. was supported by a BBSRC Discovery fellowship (BB/W008947/1). P.L.L. was supported by a Medical Research Council fellowship (MR/P014097/1 and MR/P014097/2), a Jacobs Foundation research fellowship, a Leverhulme Prize (PLP-2021-196), a Wellcome Trust/Royal Society Sir Henry Dale fellowship (223264/Z/21/Z) and a UKRI EPSRC Frontiers Research Guarantee/ ERC starting grant (EP/X020215/1). M.N.B. was supported by the Deutsche Akademische Austauschdienst (German Academic Exchange Service, DAAD) and the Studienstiftung des deutschen Volkes (German Academic Scholarship Foundation). M.F.S.R. was supported by the Wellcome Trust (221794/Z/20/Z and BBSRC BB/W003392/1).

**Author contributions** M.K.W. and M.F.S.R. conceptualized the research. M.K.W., N.S.F., Y.L., P.L.L. and M.F.S.R. discussed the methodology. M.K.W., Y.L., D.P., C.D. and L.S. programmed the experiments. M.K.W., Y.L., S.L., C.H., A.A. and S.H. collected the data. M.K.W. analysed the data, except for the drift diffusion modelling, which was done by M.N.B. and N.K. M.K.W. wrote the original draft with feedback from M.F.S.R. All authors reviewed and edited the manuscript.

**Competing interests** The authors declare no competing interests.

**Additional information**
**Correspondence and requests for materials** should be addressed to Marco K. Wittmann.

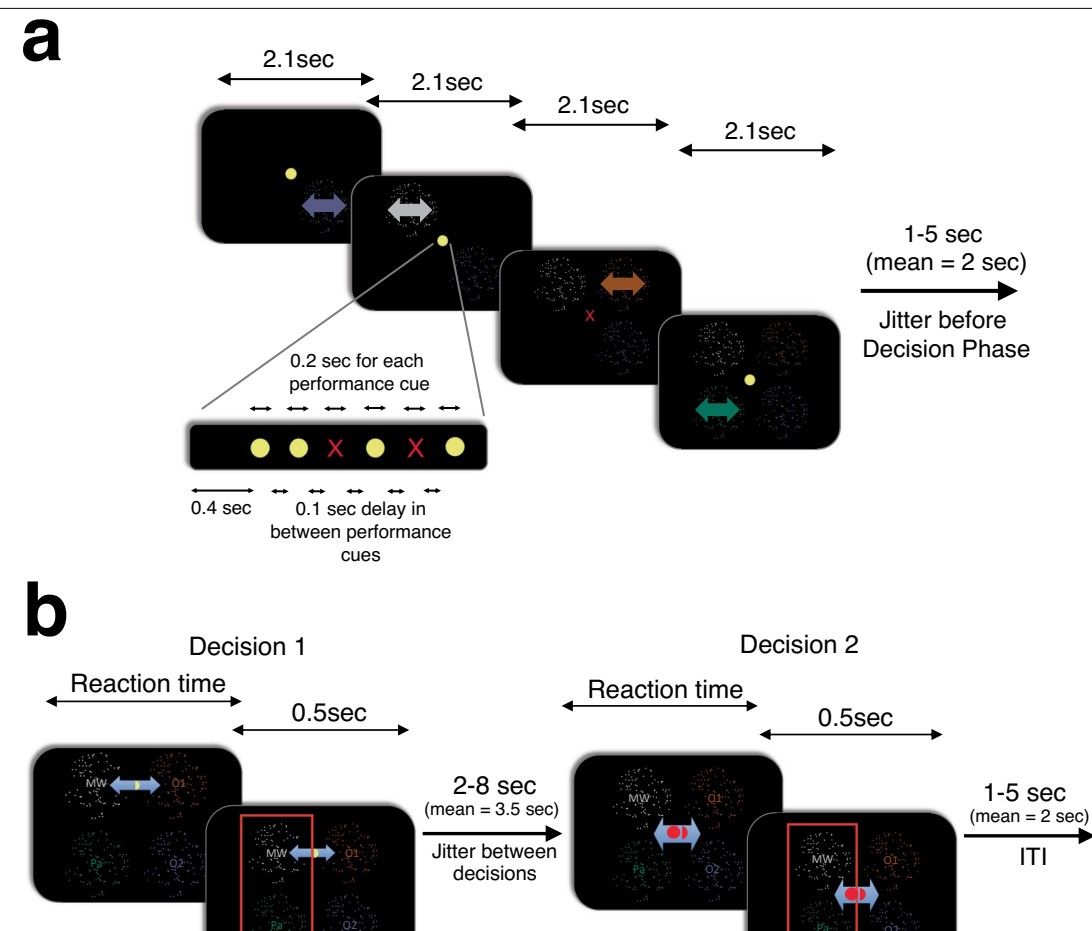

**Extended Data Fig. 1 | Detailed trial timeline of social fMRI experiment (study 1).** See Methods *fMRI experiment* section for study 1 for a detailed summary of the experimental timeline. Note that the four players are represented by random dot motion kinematograms (RDKs). This reflected the behavioral pre-experiment where performance was assessed via a perceptual RDK task and the information presented in the main social fMRI experiment mirrored this RDK task. In the main social fMRI experiment, the performance obtained in the pre-experiment was observed by participants in the observation phase (panel A) and participants made decisions about these performances in the performance phase (panel B). **(a)** Observation phase. In a pseudorandom order, the performance of each player was displayed. Performance cues were always presented centrally and the relevant player to which the performance related was indicated by its active RDK (the dots in that RDK were slowly moving). While the RDK was active, a sequence of six performance cues was presented centrally (yellow "coins" for successful performances, red "X"s for erroneous performances; shown for illustrated in the bottom left). After a player's performance was observed, the RDK became inactive (dots in the RDK became static) and another player's performance was shown until all players' performance was shown and the full set of four static RDKs was visible on screen. Participants knew identities associated with each RDK/player and their positions remained constant throughout the experiment but were randomized and balanced across participants. **(b)** Decision phase. The decision phase comprised two decisions about the performance scores (=sum of each player's successful performances) observed in the observation phase. Letters indicated player identities. Participants' initials (e.g., "MW" here) were used to indicate themselves. "Pa" indicated "partner" and "O1" and "O2" referred to the two opponents. An arrow cue indicated trial condition and which player's performance had to be compared (in this example: a self decision in decision 1 and a group decision in decision 2). A small "bonus" appeared on top of the arrow as either yellow or red coins and half-coins. The bonus had to be added to the relevant player's performances when making the decision. Yellow coins indicated a positive bonus that was added to one's own team (0.5 points in example decision 1) and red coins indicated a negative bonus that was added to the opponents' team (-1.5 points in example decision 2). The engage/avoid decision was made via a press of two buttons (as in[6,25]). An engage choice indicated the assessment that one's own relevant team member(s) had performed better (also factoring in the bonus) and resulted in a red box shown around one's own team (as in both examples here). The payoff from an engage choice corresponded to the real performance difference (including the bonus) of the relevant players and could be positive or negative. An avoid choice resulted in a box shown around the opponent team and resulted in a zero payoff. Participants task was to accumulate as many points as possible.

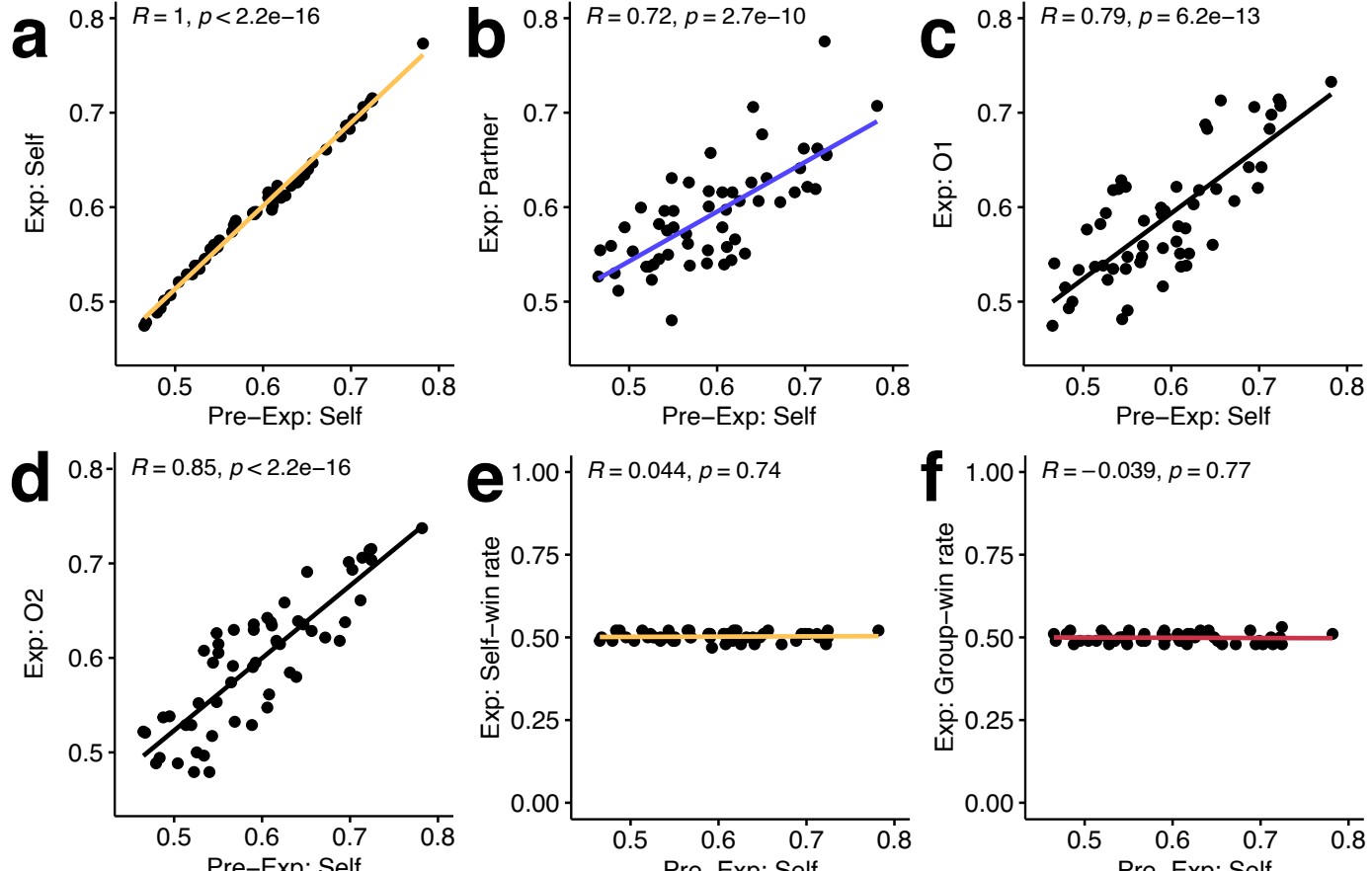

**Extended Data Fig. 2 | Pre-experimental performance was unrelated to winning over the other team in main social fMRI experiment (study 1).** As explained in the methods section for study 1 (sub section: *Experimental procedures* and *Behavioural pre-experiment. Performance assessment*), participants' decisions in the social fMRI experiment were about performance scores from themselves and three other players. Specifically, they decided whether their own and a partner's performance scores were better or worse than the scores from two opponents. These performance scores reflected veridical performance in a random-dot-motion (RDM) task from the participants themselves and three other participants who had performed the RDM task previously. However, the experimental schedule of the main experiment was carefully constructed and ensured that the schedules were comparable across participants, balanced, and finely calibrated with respect to several decision-theoretic aspects. One key consideration here was to ensure that participants could not use insight into their own pre-experimental RDM performance levels to predict whether they would perform better or worse than other players in the main experiment. The correlation plots above show correlations across participants between the pre-experiment RDM performance of the participant themselves (*Pre-Exp: Self*: the percentage of correct RDM responses) and measures related to the main task. They illustrate how we uncoupled performance in the pre-experiment from the rate of winning over the other team in the main experiment. **(a)** As the performance scores for participants were taken from the pre-experiment, there was a near 1 correlation between participants' own RDM performance pre-experiment and the performance scores for Self (*Exp: Self*; see insets for Pearson correlations and p-values).

Experimental performance scores are shown as the percentage of correct RDM scores. **(b,c,d)** However, we carefully selected which other participants the Self would be paired with and, therefore, what other participants' performances in the main experiment were. This resulted in a positive correlation between the participants' own pre-experimental performance scores and the scores of the other players in the main experiment (P, O1, O2 refer to the percentage correct RDM scores of the partner, opponent1, and opponent2). Again, all these correlations are highly significant. In sum, this means that participants who performed well in the pre-experiment were paired with other high-performing players. **(e)** The result of this was that we ensured that participant's win-rate in self decisions (*Exp: Self-win rate*, i.e. how often it was correct to indicate that oneself was the higher performer) was always around 50%. In other words, roughly half of the time the correct response to self decisions was to indicate that oneself performed better, and half of the time the correct response was to indicate that the relevant opponent performed better. A win-rate of roughly 50% was desirable, because it prevented floor/ceiling effects in participants' choice behaviour and ensured that choice analyses were sensitive to a broad range of potentially relevant behavioural variables. This also meant that there was no correlation between participants' pre-experimental RDM performance and their chance of winning over the opponent in self decisions (r = 0.044; p = 0.74). **(f)** The same was true for the relationship between pre-experimental RDM performance and winning in group decisions (*Exp: Group-win rate*; r = −0.039; p = 0.77). Note that partner decisions are not shown, as they were per definition independent of participants pre-experimental performance. (n = 56).

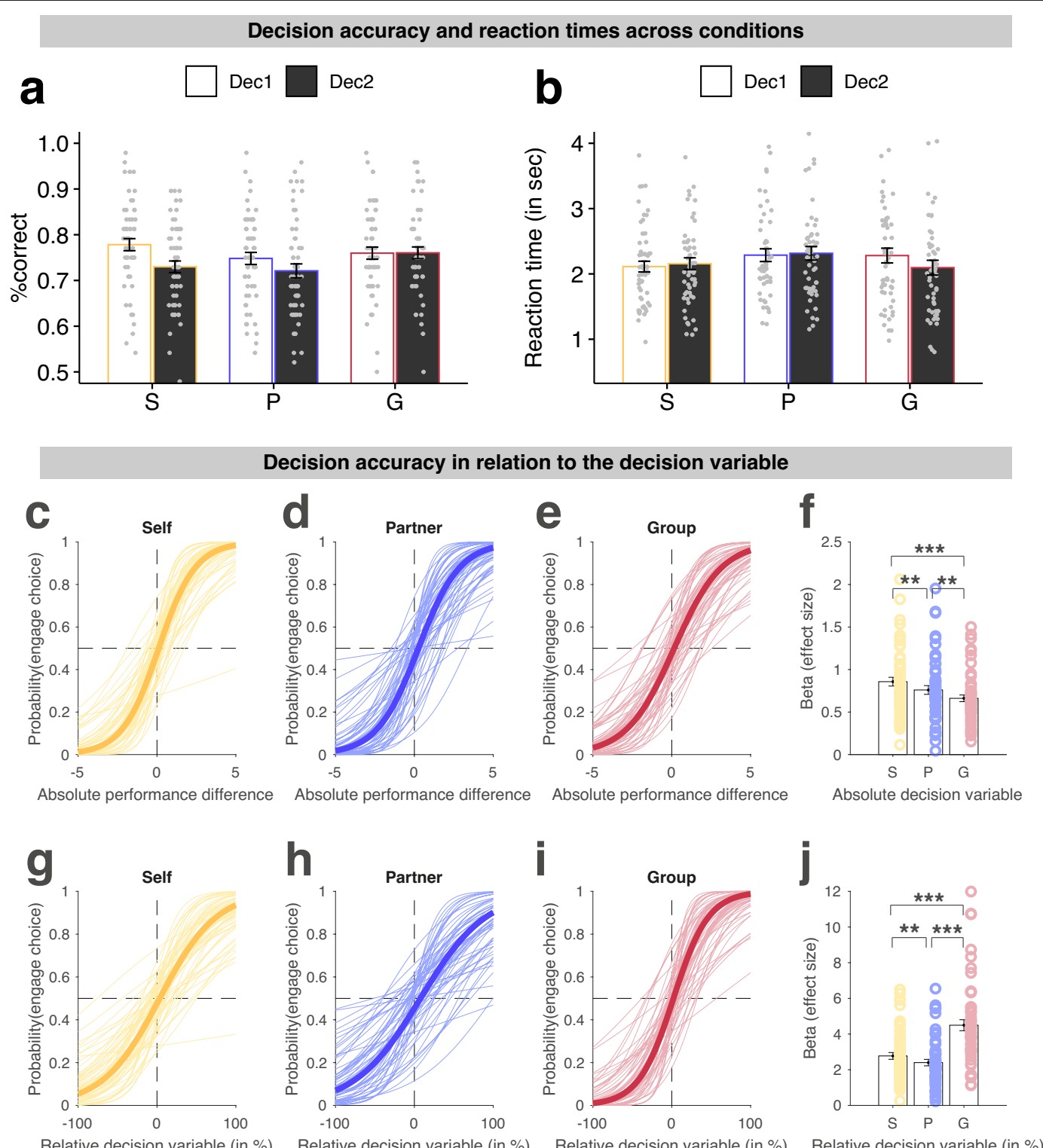

**Extended Data Fig. 3** | See next page for caption.

**Extended Data Fig. 3 | Supplementary behavioural analyses of study 1.**
**(a,b) Decision accuracy and reaction times across conditions. (a)** We analysed
decision accuracy (percentage of correct choices, %correct) per decision type
(S = Self decision, P = Partner decision, G = Group decision) and separately for
the first decision (Dec1) and the second decision (Dec2) in a trial. We especially
considered how well participants performed in group decisions compared to
dyadic decisions. In a three-way repeated measures ANOVA with the factors
decision number (Dec1 vs Dec2) and decision type (Self, Partner, Group), we
found significant main effects of decision number ($F_{1,55} = 13.08$, $p < 0.001$),
decision type ($F_{2,110} = 5.02$, $p = 0.008$), as well as a significant interaction
($F_{2,110} = 4.68$, $p = 0.011$). Interestingly, while the second decision was less
accurate than the first one for both dyadic decision types (Self: $t_{55} = 4.69$,
$p < 0.001$; Partner: $t_{55} = 2.37$, $p = 0.021$), decision accuracy was virtually the
same for the second compared to the first group decision ($t_{55} = 0.06$, $p = 0.954$).
One might expect that decisions about the group were less accurate because
they need to integrate (noisy) estimates of all four individuals rather than only
two. However, in contrast to that we found that, for second decisions, accuracy
in group decisions exceeded both self decisions ($t_{55} = 2.52$, $p = 0.015$) and
partner decisions ($t_{55} = 3.27$, $p = 0.002$). For initial decisions, self decisions were
significantly more accurate than partner decisions ($t_{55} = 2.43$, $p = 0.018$), but
not significantly more accurate than group decisions ($t_{55} = 1.68$, $p = 0.098$).
**(b)** We applied the same ANOVA to the reaction times (using the mean-RT per
participant). We identified a main effect of decision type ($F_{2,110} = 9.01$, $p < 0.001$)
and an interaction between decision type and decision number ($F_{2,110} = 10.25$,
$p < 0.001$). Decisions were made more quickly for self than for partner (Dec1:
$t_{55} = 3.816$, $p < 0.001$; Dec2: $t_{55} = 4.196$, $p < 0.001$). Group decisions were made
with roughly the same speed as partner decisions ($t_{55} = 0.12$, $p = 0.903$) and
slower than self decisions if they came first ($t_{55} = 2.76$, $p = 0.008$). By contrast,
group decisions were just as fast as self decisions ($t_{55} = 1.14$, $p = 0.258$), and
quicker than partner decisions if they came second ($t_{55} = 5.27$, $p < 0.001$).
Together, these results further support the finding that participants made the
decisions more accurately and quickly for self than for partner, confirming the
social nature of our task. In addition, participants performed well on the group
decisions, often at the same level as in self decisions and better than in partner
decisions. These results are consistent with our hypothesis that people's
decisions rely on a compressed code linked to the groupings of the two teams.

**(c-j) Decision accuracy in relation to the decision variable**. We analysed
decision accuracy in relation to the decision variable (DV) for self (S), partner
(P) and group (G) decisions. To this end, we fitted a logistic regression to the
choice data using ridge regression to account for outlying data points. We
predicted the choice to engage versus avoid (see Methods: *Data analysis. Ridge
Regression*). The logistic regressions comprised the DV as the only regressor.
Importantly, the DV was not normalised so that the logistic fit would retain the
original range of the DV. We fitted two variants of the DV. We fitted all decision
types together for each of the DV; this ensured that the beta weights were
comparable across decision types for the same DV (but not between DVs).
**(c,d,e,f)** We fitted the absolute DV (the performance difference between
relevant players plus the non-social), to participants' choices. Panels c,d,e show
the fitted logistic function and panel f shows the beta weights for the absolute
DV, which capture the slope of the logistic functions in c,d,e. Comparing the beta
weights using a one-way ANOVA, we found a significant main effect of decision
type ($F_{2,110} = 16.373$, $p < 0.001$). The effect of the absolute DV was strongest in self
decisions (vs partner: $t_{55} = 2.75$, $p = 0.008$; vs group: $t_{55} = 5.77$, $p < 0.001$) followed
by the partner condition (vs group: $t_{55} = 2.96$, $p = 0.005$). **(g,h,i,j)** Same as in
c,d,e,f, but shown for the relative DV. The relative DV captures the percent
difference in the performance of the relevant players including the bonus. This
helps to account for the fact that group decisions entail the comparison of larger
quantities. For example, while the decision between self and opponent may be
between a performance score of 4 and 5, a decision between the groups may
amount to a decision between 9 and 10. While the absolute DV is the same in both
cases (==1), the relative DV considers that 9 and 10 are closer together in relative
terms than 4 and 5. We calculated the relative DV by dividing the absolute DV
by the mean of the two relevant players in self and partner trials (e.g., for self
decisions: mean[self, relevant opponent], and by dividing it by the mean of the
sums of both teams in group trials (for group decisions: mean [self+partner,
opponent1+opponent2]). Considering the relative DV revealed again a main
effect in the corresponding one-way ANOVA ($F_{2,110} = 78.30$, $p < 0.001$).
Participants were more sensitive to the percentage difference in the
performance of groups than in the performance of individuals (vs self:
$t_{55} = 8.52$, $p < 0.001$; vs partner: $t_{55} = 10.67$, $p < 0.001$). The relative DV, again, had
a stronger effect in self compared to partner trials ($t_{55} = 2.93$, $p < 0.005$). (n = 56;
all tests were two-sided; ** $p < 0.01$; *** $p < 0.001$; error bars are S.E.M).

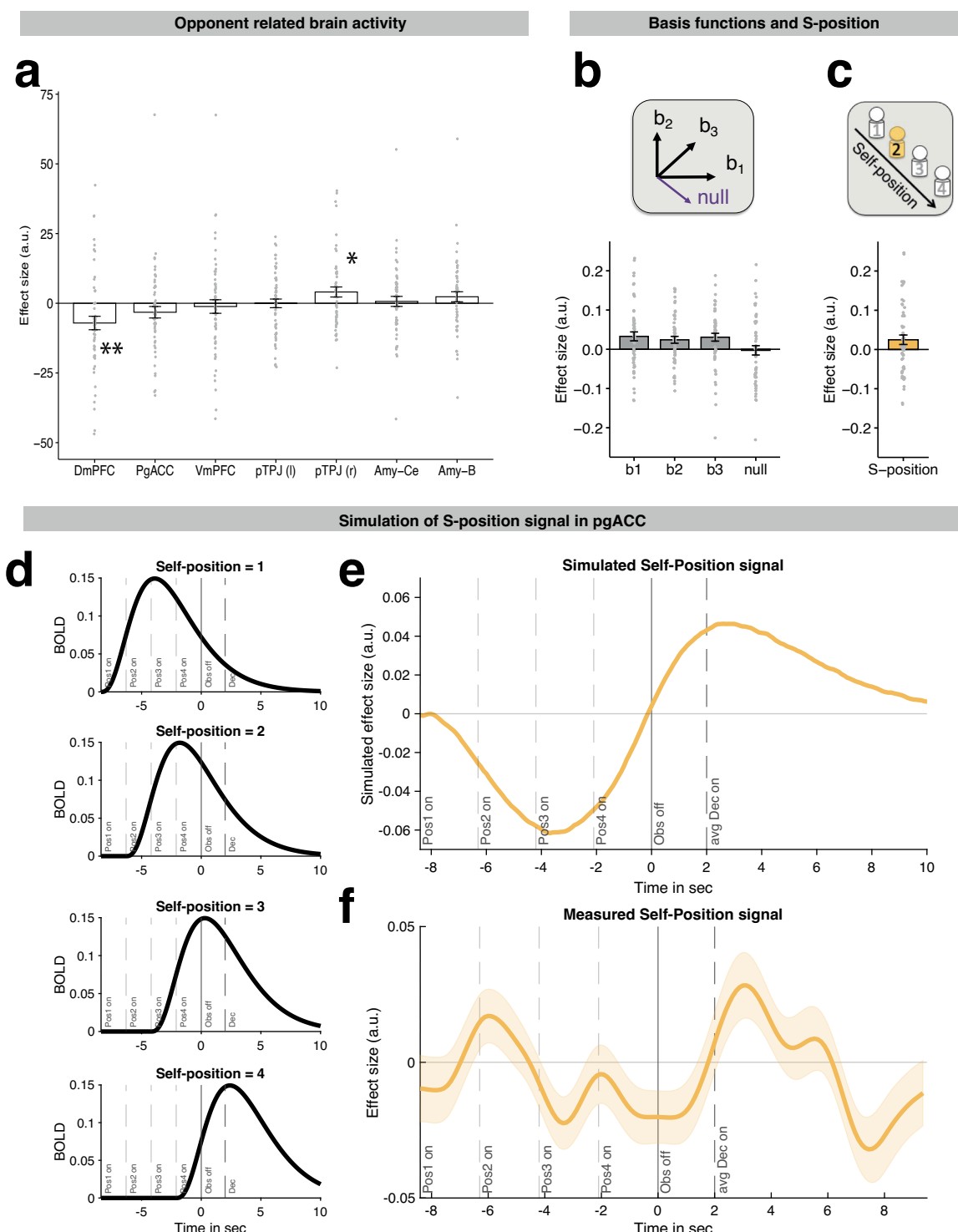

**Extended Data Fig. 4** | See next page for caption.

**Extended Data Fig. 4 | Neural activity in related to opponents, basis functions, and S-position (study 1). (a) Opponent related activity in regions of interest (ROIs).** Our results showed whole-brain corrected effects of S-performance in pgACC, and Partner performance in dmPFC. By contrast, we did not find whole-brain corrected effects related to the opponents' performance. We investigated whether opponent-related brain signals were present in ROIs. We examined activity in dmPFC (MNI = -2/28/44) and pgACC (MNI = -8/42/14), the two regions where we found Partner and Self signals (Fig. 1e). In addition, we inspected activity in ventromedial prefrontal cortex (vmPFC; MNI = -2/60/-12). We took coordinates from chosen value signals reported in Fig. 3g (red cluster, social DV). VmPFC is not typically associated with social cognition. In addition to spherical ROIs in these regions, we used masks of areas typically found in studies of social interactions. Based on recent anatomical parcellations, we inspected activity in left and right posterior temporoparietal junction (pTPJ)[69] as well as central and basal parts of the amygdala (Amy-Ce and Amy-B)[70]. We extracted FSLs COPE images[57] from *fMRI GLM 1* in these regions (i.e. images of fMRI effect sizes). We extracted Or-performance from the second decisions in a trial – the same time point where we had found Self and Partner related signals. We did this for Self and Partner decisions and averaged their effect sizes in these ROIs. We found significant activations in the right TPJ (one-sample t-test; $t_{55}$ = 2.249, p = 0.029) and dmPFC (one-sample t-test; $t_{55}$ = −2.935, p = 0.005). Both regions are often found in studies of social cognition. Note that the dmPFC region is the same one where we had found partner-related activity and so these results might suggest that dmPFC represents social agents more generally, in line with our previous findings regarding dmPFC[6]. Note that the opponent-related activity in dmPFC also survives Bonferroni correction for multiple comparisons for using 7 ROIs ($p_{Bonferroni}$ = 0.05/7 = 0.007). **(b,c) ROI analyses of neural activity linked to basis functions in medial prefrontal cortex (study 1). (b)** Effect sizes for the neural ROIs in panel C in Fig. 3, illustrated as bar plot. We extracted participant-specific neural effect sizes using a leave-one-out procedure to ensure that individual data points were independent (see Methods). **(c)** Effect sizes for the neural ROIs in panels D in Fig. 3, illustrated as bar plot. Again, we used the same leave-one-out procedure for the S-position variable in the early time window after observation phase offset. **(d-f) Simulation analysis of the Self-Position signal in pgACC.** In the main text, we reported a signal in pgACC indexing the position of oneself within the sequence of players in the observation phase (regressor with values 1,2,3 or 4 depending on one's own position in the sequence; Fig. 3d). We analysed whether the peak timing of this signal was related to the actual Self position in the sequence, i.e. whether the Self-position signal reflected a BOLD response that was simply the aggregate of several separable neural responses each of which is time-linked to a particular instance of "Self" score presentation rather than the reactivation of a representation of the Self position within the sequence prior to decision making. Our analyses suggested that this was unlikely to be the case and that the Self-position signal we observed was more likely a reactivation of previous position information that emerged later in time. We used simulations for these analyses as follows. **(d)** We considered a time period that included the observation phase and the onsets of the 4-player specific displays (see timing of vertical lines: Pos1 on, Pos2 on, Pos3 on Pos4 on). In addition, the analyses covered the time period analysed in the main manuscript figure (everything to the right of "Obs off", the end of the observation phase). We simulated a hemodynamic response function (Hrf) time-locked to a schedule in which the Self was either presented 1st, 2nd, 3rd of 4th, mirroring the balanced presenting of Self in our actual experiment. The Hrf was modelled using a gamma probability distribution with a mean of 6 and sigma of 3. These settings were taken to match the default Hrf response in FSL[57] and we used the gamma-function implemented in FSL. We simulated a schedule of 144 trials for 56 participants, in which the Self was equally often presented in each sequence position, matching the schedule of our experiment. We generated Gaussian distributed white noise (sigma of 0.05) and filtered it via a Savitzky-Golay filter to simulate autocorrelated noise as observed in the BOLD signal. Subsequently, we applied a neural GLM (FSL's ols) to the data using the ground-truth Self-Position as the only regressor. We extracted the neural effect size at every time point of the time series. **(e)** This gave the simulated effect size of Self-position shown in panel e. A Self-position signal caused by time-linked Hrfs would generate a large negative signal during the observation phase that turns slowly positive after the offset of the observation phase. The empirical effect sizes did not match this pattern and instead suggested a sharp rise of the signal at the time of observation phase offset. **(f)** The empirical effect-size for Self-Position, expanded to also cover the time period before observation phase offset (everything to the right of the "obs off" line corresponds to the main manuscript figure). In summary, the temporal evolution of the Self-position signal does not match up with a timecourse expected from a Self-position signal that is caused by separable neural responses each of which is time-linked to Self presentation in the observation phase. Instead, the signal rather seemed to reflect a reactivation of information linked to the sequential position of the Self in the sequence. Moreover, when analysing the same timecourse in our motor control experiment (see Fig. 5), there was never a significant deflection of the Motor-S-Position signal from zero, not even when tested without a stringent leave-one-out procedure (all p's>0.12 in time window 0–10 s after observation phase offset; two-sided one-sample t-tests). (n = 56; all t-tests were two-sided; *, p < 0.05; **, p < 0.005; error bars are S.E.M).

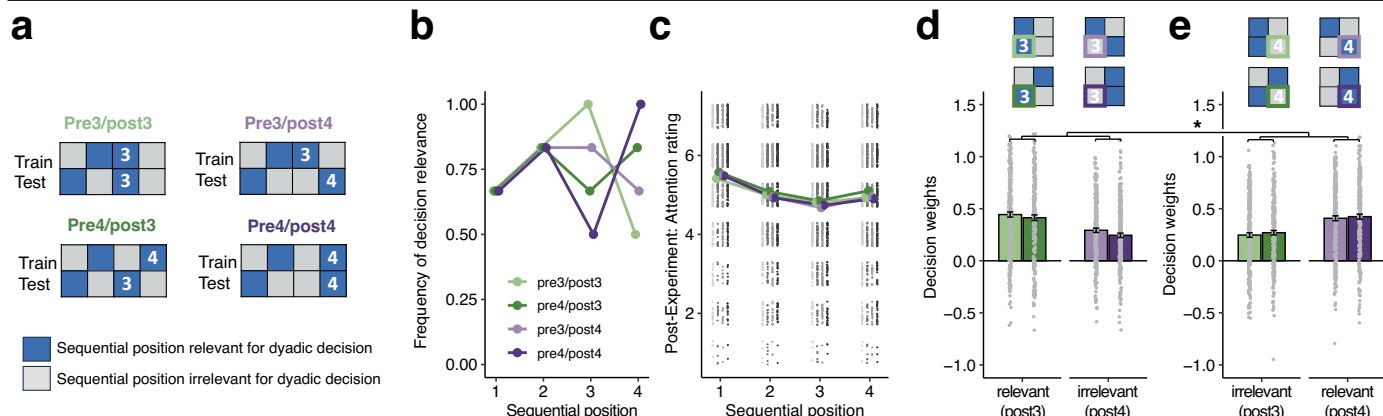

**Extended Data Fig. 5 | Supplementary behavioural study on the role of sequential expectations in social decision-making (study 4).** We conducted another between-subject behavioural experiment online to investigate how people adapt to sequential information in our experiment. We habituated participants to specific sequential positions in the 4-player sequence (64-trial training phase) and then abruptly changed the relevance of these positions (32-trial testing phase). The transition between training and testing phases was seamless and uncued. **(a)** We conducted a 2 training sequences [pre3/pre4] × 2 test sequences [post3/post4] between-subject design that assessed the impact of specific transitions on decision making performance. Importantly, during both training and test phases, dyadic decisions (both self and partner decisions) were always made between players in the same sequential positions. For instance, when participants were initially trained on the "pre3" sequence, they only made dyadic decisions entailing comparison between the player observed in the second sequence position and the player observed in the third position (hence the label "pre3") of the observation phase. However, players were assigned to different sequential positions in different trials. The rows of each graphic illustrate the relevant (blue) and irrelevant (grey) elements for dyadic decisions in training (top rows) and test phases (bottom rows). The white numbers highlight specific sequential elements that were relevant in different schedules, and that were used to define the schedule (e.g. the pre3/post3 schedule is highlighted by two white "3"s). We optimised the experimental schedule for the analysis of sequential positions 3 and 4. Note that that meant that only one dyadic decision was possible for each trial. In addition, because participants were asked to make two different decisions on each trial, each trial also included a group decision (across trials the order of the group and dyadic trials was balanced). For group decisions, all four players were relevant. The assignment of performance scores to player identities as well as the decision schedule was identical in all four conditions. Hence, differences in decision-making performance across conditions could be unequivocally attributed to the processing of sequential positions. **(b)** Decision-relevance of each element differed by schedule. Note, for example, that in pre3/post3, sequential position 3 is relevant in both training and test phase, and hence in 100% of trials. By contrast, in pre4/post4, position 3 is only relevant in group decisions (because all four positions are relevant in group decisions, and 50% of all decisions are group decisions). **(c)** However, in a post-experimental questionnaire, when participants self-reported their attention to each element of the four-player sequence, they showed no subjective awareness of the manipulation. Using the ratings, we tested whether participants directed their attention more towards frequently occurring sequence positions. We ran a 4 × 4 ANOVA with the factors "schedule" and "sequence position". However, we found no interaction between schedule and attended position rating was apparent ($F_{9,3054} = 1.103$, $p = 0.357$).

Participants did not report that they were directing their attention more towards more frequently relevant sequential positions. This suggests that participants were not aware of the underlying sequential structure of the task or the sequence positions that were most often relevant for making decisions. **(d,e)** We analysed the test phase of the experiment. An initial analysis simply looking at overall performance accuracy during dyadic decisions revealed no main effects, and no evidence of an interaction between the positions that had been relevant during the initial training phase and the positions that were relevant during the subsequent testing phase (all p's > 0.11). We followed this analysis up in more detail by analysing the decision weights associated with different sequential positions using a GLM approach. We focused on elements 3 (panel d) and 4 (panel e) of the sequence, which we had manipulated systematically across conditions to assess the fluency of transitions when a sequence element remained relevant (pre3/post3 and pre4/post4) versus when the relevance changed (pre3/post4 and pre4/post3). We expected larger decision weights in the test phase for elements that had been relevant before, independent of whether they were relevant now. We submitted the decision weights for elements 3 and 4 of the test phase to a 2 × 2 × 2 ANOVA with the factors decision weight [pos3/pos4], training [pre3/pre4] and test [post3/post4]. Indeed, we found a significant decision weight x training interaction when comparing the decision weights from the test phase ($F_{1,1018} = 5.913$, $p = 0.015$). **(d)** This meant that the decision weight for element 3 was increased when it had been relevant before in the training phase, regardless of whether it was currently relevant (left bars, the post3 condition) or irrelevant in the testing phase (right bars, the post 4 condition). Note the that the insets above illustrate the 3rd and 4th element of the four conditions. The colour coding is the same in all plots. Note that all bars in this panel reflect the decision weight of the third element of the sequence. Compare the green and purple bars respectively and note the increased decision weight for instances when the third element of the training sequence is relevant/blue compared to when it is not relevant/grey. **(e)** *Vice versa*, the weight for element 4 was increased if it had been relevant before (pre4/post3 and pre4/post4), again regardless of whether it was currently relevant or irrelevant in the testing phase. These results demonstrate the sequential position of a player in the sequence can impact how much weight that player has on the decisions that participants make and that this weight is impacted by experience of what has previously been relevant. People adapt to weighing sequential positions with a given weight, and while this may facilitate performance for as long as the relevant information continues to appear in the same sequential position, there is a cost once another position becomes relevant. The questionnaire analysis suggests that this process operates below the level of awareness. (n = 1022; error bars are S.E.M, *, p < 0.05).

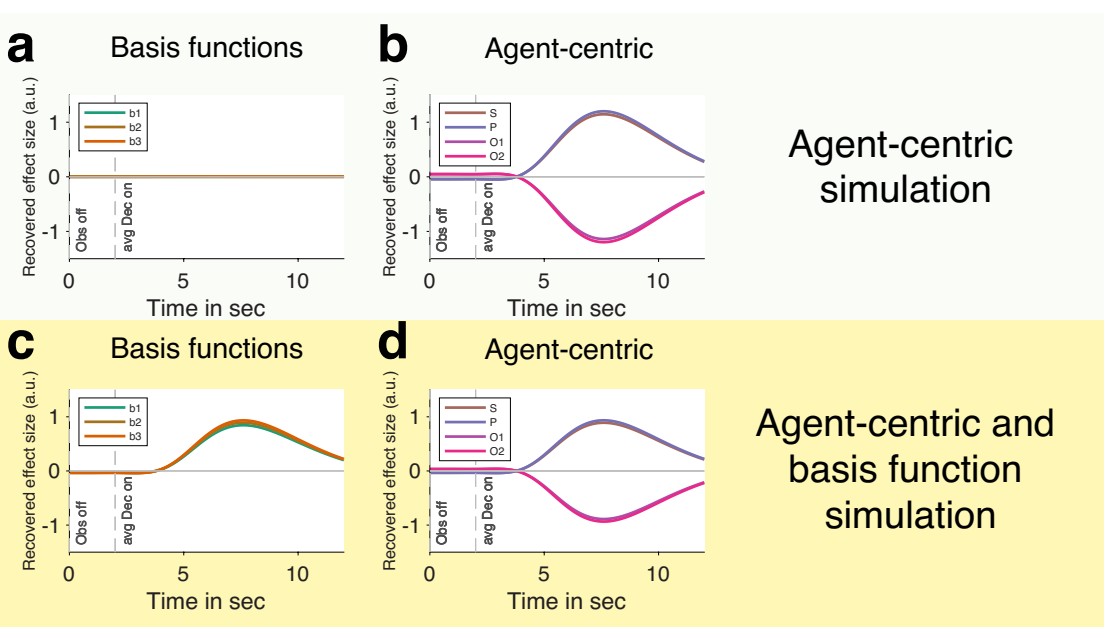

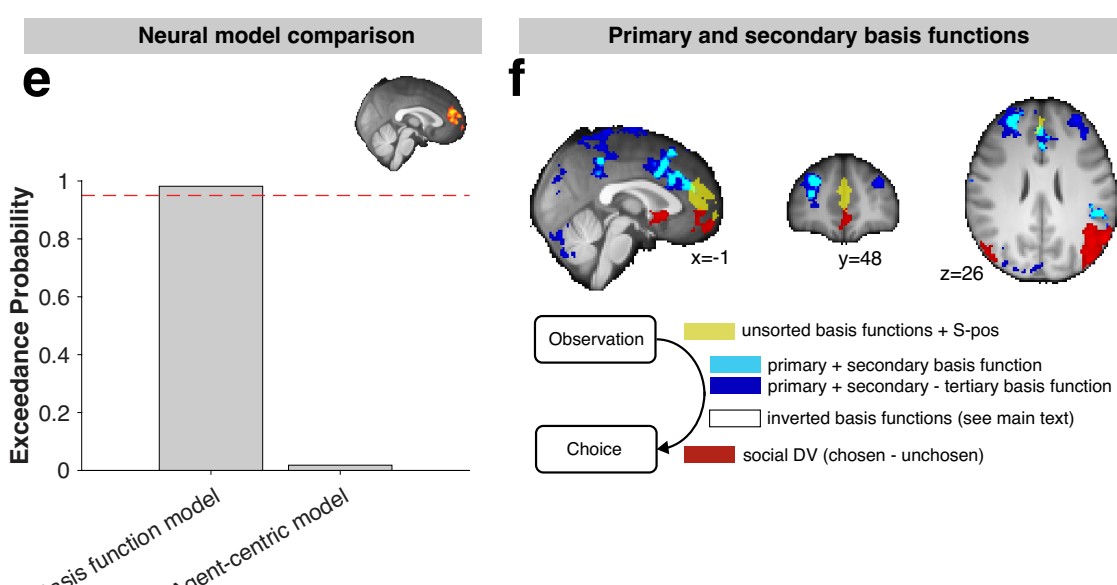

**Extended Data Fig. 6 |** See next page for caption.

**Extended Data Fig. 6 | Supplementary analyses of neural agent-centric and basis function signals. (a) BOLD Simulations demonstrate that basis function projections and agent-centric brain signals are dissociable (study 1).** We conducted BOLD simulation analyses to show that basis function related signals would not occur as a by-product of holding agent-centric representations. We simulated a BOLD time course by recreating the convolved design matrix[57] from *fMRI GLM2*. Afterwards, we applied the same time course analyses to the synthetic data that identified basis function projections (b1,b2,b3) in the first place (*ROI GLM1*). This time course GLM comprised regressors for both basis function-related activity and agent-centric signals. We tested whether basis function signals would be measurable in BOLD data that was (i) created from purely agent-centric representations or (ii) both agent-centric and basis function computations. **(a,b)** Recovered effect sizes for basis function-related activity (panel a) and agent-centric coding (S,P,O1, and O2 signals; panel b) in a timecourse GLM fitted on synthetic BOLD data using only agent-centric effects. We created the synthetic BOLD data by weighing the design matrix $\mathbf{X}$ from *fMRI GLM2* with a weight vector $\mathbf{w1}$ which was set to [1 1 -1 -1] for the agent-centric effects [S,P,O1,P2] of the GLM (see Methods). All other weights were set to zero. We used positive weights for one's own team and negative weights for the opponent team, because we had observed such positive/negative coding of the agents in other supplementary analyses of frontal cortical activity (see Extended Data Figs. 4, 7). We jittered the weights slightly so that the resulting lines in the timecourse plot do not fully overlap for visualisation. We created the synthetic BOLD time course by taking the simple dot product of $\mathbf{Xw_1}$. We then fed the synthetic BOLD data into precisely the same timecourse analysis (*ROI GLM1*) that we had used to identify b1,b2,b3 in the first place. We repeated the procedure for all participants and show the recovered effect sizes for b1, b2, and b3 in panel a. As expected from BOLD data created using purely agent-centric effects, no basis function-related signals were visible at any point of the time course for neither b1, b2, or b3 (all p > .1). By contrast, all four agent-centric effects could be reliably recovered in the expected directions (all p < 0.001 at expected peak time window 7–9 s). **(c,d)** We repeated the same simulation to show that basis function-related signals can be successfully recovered if they are simulated. Again, we used the same convolved design matrix and a weight matrix $\mathbf{w_2}$. $\mathbf{w_2}$ simulated the same agent-centric effects as $\mathbf{w_1}$, but, importantly, in addition simulated the parallel existence of basis-function related effects. $\mathbf{w_2}$ modelled the same agent-centric weights of [1 1 -1 -1] for [S,P,O1,P2], but added positive effects [1 1 1] for [b1,b2,b3]. Then we calculated synthetic BOLD data by calculating $\mathbf{Xw_2}$. Afterwards, we applied the same GLM (*ROI GLM1*) as before and inspected both basis function-related effect sizes and agent-centric effects in the same GLM model. As expected, now b1, b2 and b3 can be successfully recovered in addition to agent-centric effects (all p < 0.001 at expected peak time window 7–9 s; two-sided one-sample t-tests). **(e) Bayesian model comparison suggests that dmPFC/ACC BOLD signal is better explained by basis function model compared to agent-centric model**. We performed a Bayesian model comparison[65] investigating how well the neural signal in dmPFC/ACC is explained by our basis function model compared to agent-centric effects. We focused on the same ROI as the main text (MNI coordinates =(−8, 42, 14); relating to Fig. 3b). We used the convolved design matrices from *fMRI GLM2* to run an ROI analysis on this region. We tested two models: (1) a basis function GLM and (2) an agent-centric GLM. Both models were derived from *fMRI GLM2*, which contained both basis function-related regressors ($b_1$, $b_2$, $b_3$, S-position) and agent-centric regressors (S-performance, P-performance, O1-performance, O2-performance) time-locked to the end of the observation phase. We derived the basis function model of neural activity by removing the four above-mentioned agent-centric regressors from the convolved design. We also removed their associated temporal derivative regressors. This represents a model of neural activity without agent-centric influences on the BOLD signal. Next, we built the analogous agent-centric model from *fmri GLM2*. We kept in the four agent-centric regressors (S-performance, P-performance, O1-performance, O2-performance), but instead removed the four regressors related to the basis function representations ($b_1$, $b_2$, $b_3$, S-position) and their associated temporal derivatives. This agent-centric model captured the influences of agent-related performance effects but did not contain regressors related to the basis functions. Both resulting GLM models comprised the same set of additional regressors related to other points in the trial (see methods: *fMRI GLM2*), with the only difference relating to the changes described above. Both GLMs have the same number of regressors. We applied both the basis function GLM and the agent-centric GLM to the neural data in dmPFC/ACC and calculated the log model evidence for each GLM and each participant (note that there was no need to correct for the number of regressors in the design, as the number of regressors were identical in both GLMs). Afterwards we submitted the log model evidence into a Bayesian model selection random-effects analysis (using spm_BMS.m)[71], which computed the exceedance probability of the two GLMs. We found that that the exceedance probability greatly favoured the basis function model (exceedance probability=0.982), suggesting that relevant neural activity in dmPFC/ACC was better explained by basis functions compared to agent-centric representations. **(f) Cingulate cortex and frontal pole encode basis function projections using a sequential frame of reference (light and dark blue effects)**. Areas representing the combination of primary and secondary basis function projections (light blue activation), and the same contrast minus the projections onto the tertiary basis function (dark blue activation). The current analysis were time-locked to the first decision. Frontal pole activity might therefore reflect information pertinent to future decisions. Social decision variable effects and unsorted basis function related effects are the same as in the main text and are shown for reference. (n = 56; MRI results cluster-corrected at Z > 3.1; p = 0.05 FWE-corrected).

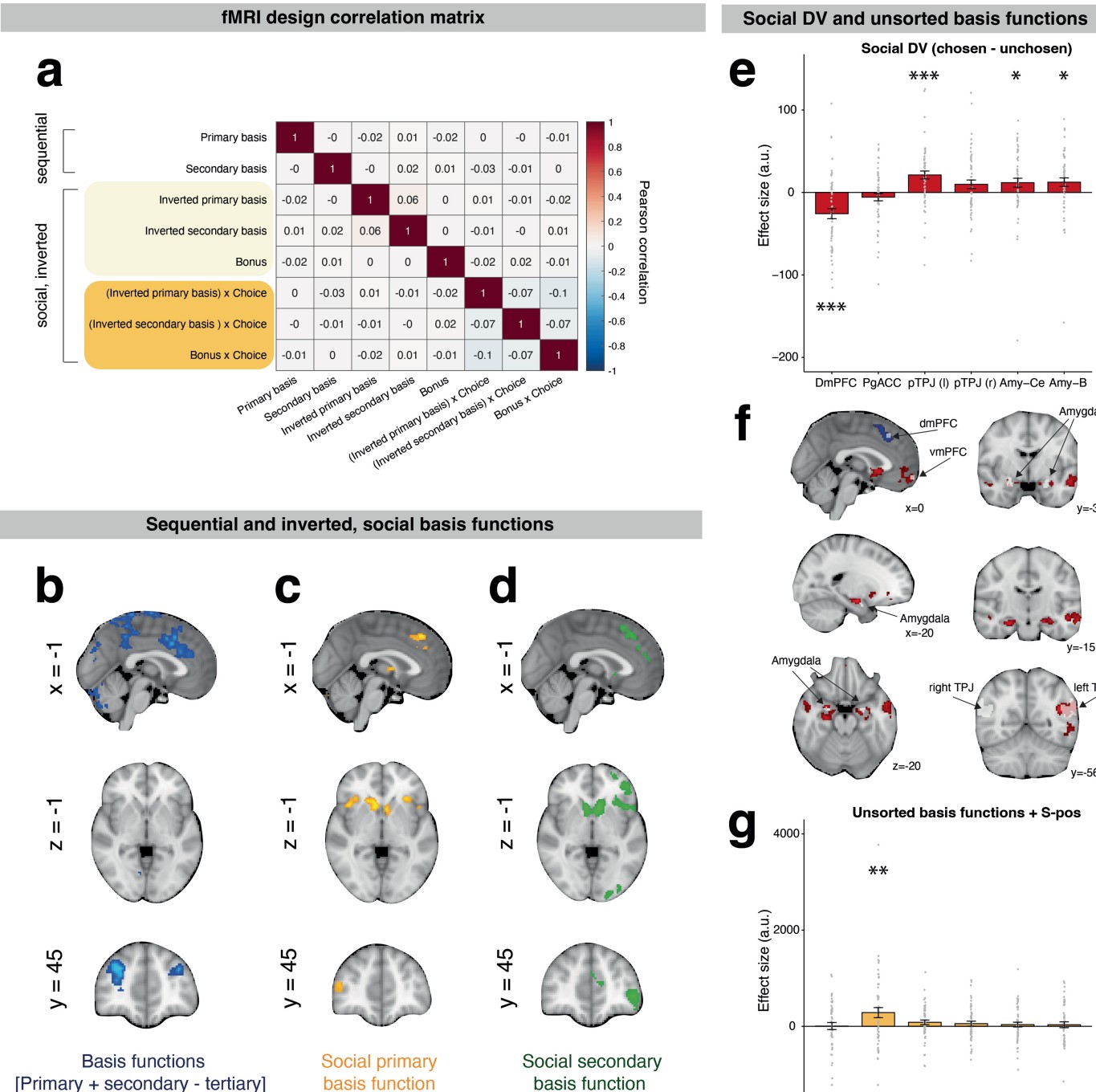

**Extended Data Fig. 7** | See next page for caption.

Extended Data Fig. 7 | Supplementary neural analysis of basis functions and the social decision variable. (a) Correlation matrix of fMRI design assessing effects of primary and secondary basis function (fMRI GLM3). The matrix shows correlation coefficients for relevant regressors averaged over all participants. All $|r| < 0.1$. Note that whereas primary and secondary basis functions are defined in terms of sequential positions, this is different for all other variables that are defined in an agent-centric way (positive and negative weights are related to one's own versus the opponent team). Note that the social agent-centric decision variable is a linear combination of the inverted primary and inverted secondary basis projections as well as the bonus (light-yellow box). Correlations also remain low when considering the variables in interaction with choice (dark yellow box). (b,c,d). Neural signals linked to sequential and inverted social basis functions. With the same GLM (fMRI GLM3), we analysed neural activity related to the projections onto the basis functions using two different frames of reference. All results shown in this figure are time-locked to the onset of dyadic decisions (both self and partner decisions) and derived from the same GLM model. (b) We identified activity related to the sequential basis functions (see main text) in posterior and anterior parts of the cingulate cortex as well as the bilateral frontal pole. Here, activity related to a combination of the basis function projections is shown (primary + secondary – tertiary). (c,d) In the same GLM, time-locked to the same point in time, we also identified neural activity related to the projections of the basis functions onto a social (or: inverted) frame of reference: A social primary basis function captures difference between one's own and the other team's performance and a social secondary basis function separates decision-relevant from decision-irrelevant agents (see methods: Sorting of basis functions and transformation to choice). In medial frontal cortex, we found activations related to both the sequential basis functions and the social basis functions. Notably, the social basis functions were represented in more dorsal and anterior parts of medial prefrontal cortex and did overlap with each other, but did not overlap with the sequential basis functions. While the former were represented in pre-SMA and anterior parts of ACC sulcus, activity related to the sequential basis functions were represented in the gyral part of ACC. However, sequential and social basis functions were represented not only in adjacent but also in clearly distinct brain networks. Both primary and secondary social basis function related activity was found in the anterior caudate nucleus, extending bilaterally into the ventral striatum (more strongly for the secondary social basis function). By contrast, the latter regions showed no response related to the sequential basis functions, also not below cluster-forming threshold (compare second row across panels b, c, d). In addition, social basis function related activity was found bilaterally in the intraparietal sulcus, the anterior insula, and parts of ventrolateral prefrontal cortex. By contrast, the sequential basis functions were represented in more dorsal parts of lateral prefrontal cortex, more consistent with the location

of the frontal poles (compare third row across panels b,c,d). (e,f,g) Activity related to the social decision variable and the unsorted basis functions. (e) We investigated the distribution of the social decision variable (chosen – unchosen) (red activity in Fig. 3g) in a selected set of regions of interest (see Extended Data Fig. 4 for a list of the regions). We examined activity in dmPFC (MNI = -2/22/28) and pgACC (MNI = -8/42/14), the two regions were we found Self and Partner related signals (see Fig. 1e). In addition, we used masks of areas typically found in studies of social interactions. Based on recent anatomical parcellations, we inspected activity in left and right posterior temporoparietal junction (pTPJ)[69] as well as central and basal parts of the amygdala (Amy-Ce and Amy-B)[70]. We extracted FSLs COPE images[57] from fMRI GLM 3 in these regions (i.e. images of fMRI effect sizes for the social DV (chosen-unchosen)). We found significant effects of the social decision variable in several regions associated with social decisions making including dmPFC (one-sample t-test; $t_{55} = -4.224$, $p < 0.001$), the left TPJ ($t_{55} = 4.385$, $p < 0.001$), and both central ($t_{55} = 2.182$, $p = 0.033$) and basal amygdala ($t_{55} = 2.402$, $p = 0.020$). One notable exception is pgACC ($t_{55} = -1.288$, $p = 0.203$), where we had found activity related to the basis functions. This is consistent with our proposal that pgACC is important for representing social information in a compressed and combinatorial code along the basis functions, while vmPFC has a relatively more important role in guiding decisions. (f) Whole brain cluster-corrected activation of the social decision variable (chosen – unchosen). We have overlaid these activation maps with the ROI masks we had used (see arrows and compare with panel e). VmPFC and dmPFC both reflect the decision variable, but with opposing weights. PgACC in between these two clusters did not show significant effects. Furthermore, these maps show that there is activity in or adjacent to the amygdala in both hemispheres (note that the two amygdala ROIs are close together, this is why we highlight them jointly). However, the centre of this activation was more posterior than the amygdala, and effects were most clear bilaterally in the anterior hippocampus. (g) We analysed activity related to the unsorted basis functions and S-position (same yellow activity as in main Fig. 3b). To this end, we extracted COPE images from fMRI GLM 2 relating to this contrast. In addition to the established ROIs, we also inspected activity in ventromedial prefrontal cortex (vmPFC; MNI: -2/60/-12). We took coordinates from the social DV contrast in panels e and f. (see Fig. 3g (red activity cluster, social DV (chosen-unchosen))). The only significant cluster was the one in vmPFC ($t_{55} = 2.735$, $p = 0.008$). Note that the effect remains significant after excluding the one apparent positive outlier. In contrast to the social DV, the unsorted basis functions plus S-position contrast only identified one large significant cluster centered on pgACC. (*p < 0.05; **p < 0.01; ***p < 0.001; t-tests are all two-sided one-sample t-test; error bars are S.E.M; n = 56, all MRI results are family-wise error (FWE) cluster-corrected at Z > 3.1, p = 0.05).

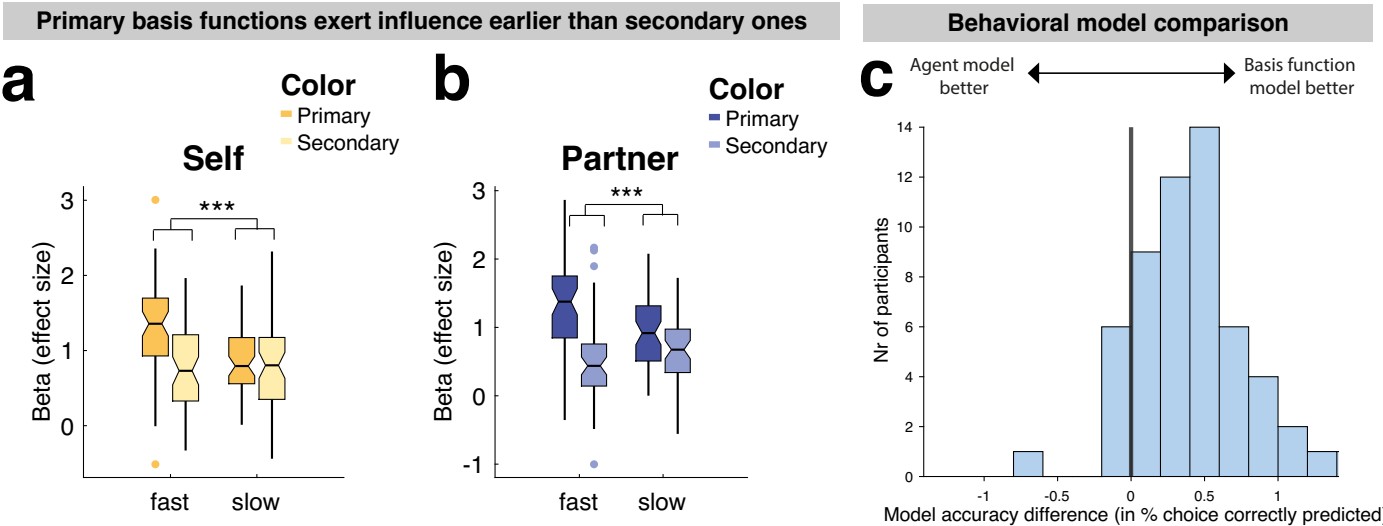

**Extended Data Fig. 8 | Supplementary analyses of basis function-related behavioural signatures 1 (study 1). (a,b) Primary basis functions exert their influence on choice earlier than secondary basis functions**. Using a model-free approach, we replicated our finding that primary basis functions exert their influence on choice earlier in the decision process compared to secondary basis functions, for both Self and Partner decisions. See the drift diffusion modelling results in the main text for model-based evidence for the same point (see also[43]). Again, the analyses are based on the two types of dyadic decisions, as only here decisions can be made as a combination of primary and secondary basis function (both projections already "inverted" to agent-centric, social space for the analysis here). We tested the following prediction: If primary basis functions exert their influence earlier in the decision process compared to secondary basis functions, then faster decisions should be relatively more driven by primary compared to secondary basis functions. By contrast, slower decisions allow the secondary basis that contribute to decisions later in time to increase in influence on choice. This meant we predicted an interaction effect with the predominance of primary over secondary basis function effects being more pronounced in trials with fast reaction times compared to trials with slow reaction times. We found precisely such an basis function x reaction time interaction in a 2 x 2 [basis function projection: primary/secondary; reaction time: fast/slow] within-participant analysis of variance for both Self decisions ($F_{1,55} = 16.991$; $p < 0.001$) and Partner decisions ($F_{1,55} = 24.363$; $p < 0.001$). The GLM underlying this analysis was constructed as follows. First, we split decisions by their median reaction time in trials with fast and slow reaction times. Then, we predicted choice as a linear combination of the primary and the secondary basis projection (translated to agent-centric space), and the bonus. To prevent overfitting given the reduced trial number for each GLM, we used L2 regularization. To do so, we used Matlab's *lassoglm* function (setting Alpha to Matlab's smallest possible value) and used a lambda of 0.015[65], thereby ensuring constrained decision weights. (n = 56; repeated measures ANOVA was used; boxplot middle line shows median, box is -+1.58 interquartile range [Tukey boxplot] and points are outliers relative to that). **(c) Model comparison of behavioural basis function and agent model**. Our results showed that irrelevant players in dyadic decisions impact choice in a manner consistent with their group membership. For example, during Self decisions, P and Oi are irrelevant. Nevertheless, our GLM analyses showed that P-performance had a significantly positive effect during Self decisions, and Oi-performance had a significantly negative one. This pattern of effects is predicted by our basis function model which suggests that participants first consider a primary basis function and then a secondary one. In support of our approach, we conducted

an additional model comparison. We compared two models. One model was the "basis function model" which predicted choices as a function of the performances of S, P, Or, Oi and the non-social bonus (same as in main manuscript). The second one was an "agent model" that predicted choices only as a function of the relevant players and the bonus (e.g., S, Or and bonus for Self decisions). The agent model captured the intuition that decisions are made only based on the relevant players in a decision. The basis function model does not negate this. However, it suggests that there is a necessary addition to the agent model that is needed to explain choice better. This addition is reflected in the weights of influence of the irrelevant players, which are predicted by the basis function model. We compared the two logistic GLM models using a cross-validation approach. We collapsed the data over both Self and Partner decisions. Then we used a leave-one-participant-out cross-validation. Cross-validation has the advantage that model performance can be directly compared across models without setting somewhat arbitrary penalisation criteria for the number of parameters in the model (like AIC or BIC). Overfitting is directly punished by worse generalisation performance in the held-out sample. We implemented the cross-validation in the following way. We gathered the data from all participants except a left-out participant. We ran the model GLM on the concatenated data of the participants and then applied the resulting decision weights to the left-out participants data. We predicted the choices of the left-out participants on the basis of these weights that had been determined without the influence of the left-out participant's data. We then calculated the model deviance and model accuracy for the left-out participant[72]. The model deviance is a simple function of the difference between actual choices and the models' choice probabilities. To derive an estimate of the model accuracy, we sampled 10.000 times from the predicted choice probability distribution to get model simulated binary choices. We calculated model accuracy by comparing these simulated choices with the actual choices and averaged the result to a single accuracy value. We repeated this procedure for all participants and aggregated the model deviances and model accuracies for both models. A smaller model deviance indicates a better model fit and we found that the basis function model indeed had significantly smaller deviance (paired t-test; $t_{55} = -2.981$, $p = 0.004$). The result from the estimated model accuracies also indicated that the basis function model significantly predicted a higher percentage of participants' decisions (paired t-test; $t_{55} = 8.111$, $p < 0.001$, Cohen's d = 1.084). A histogram of the difference in estimated model accuracies is shown in this figure. Note that the distribution is significantly shifted to the right, indicating better model fit for the basis function compared to the agent model. (n = 56; *, ***, $p < 0.001$ error bars are S.E.M; tests are ANOVAs and two-sided t-tests).

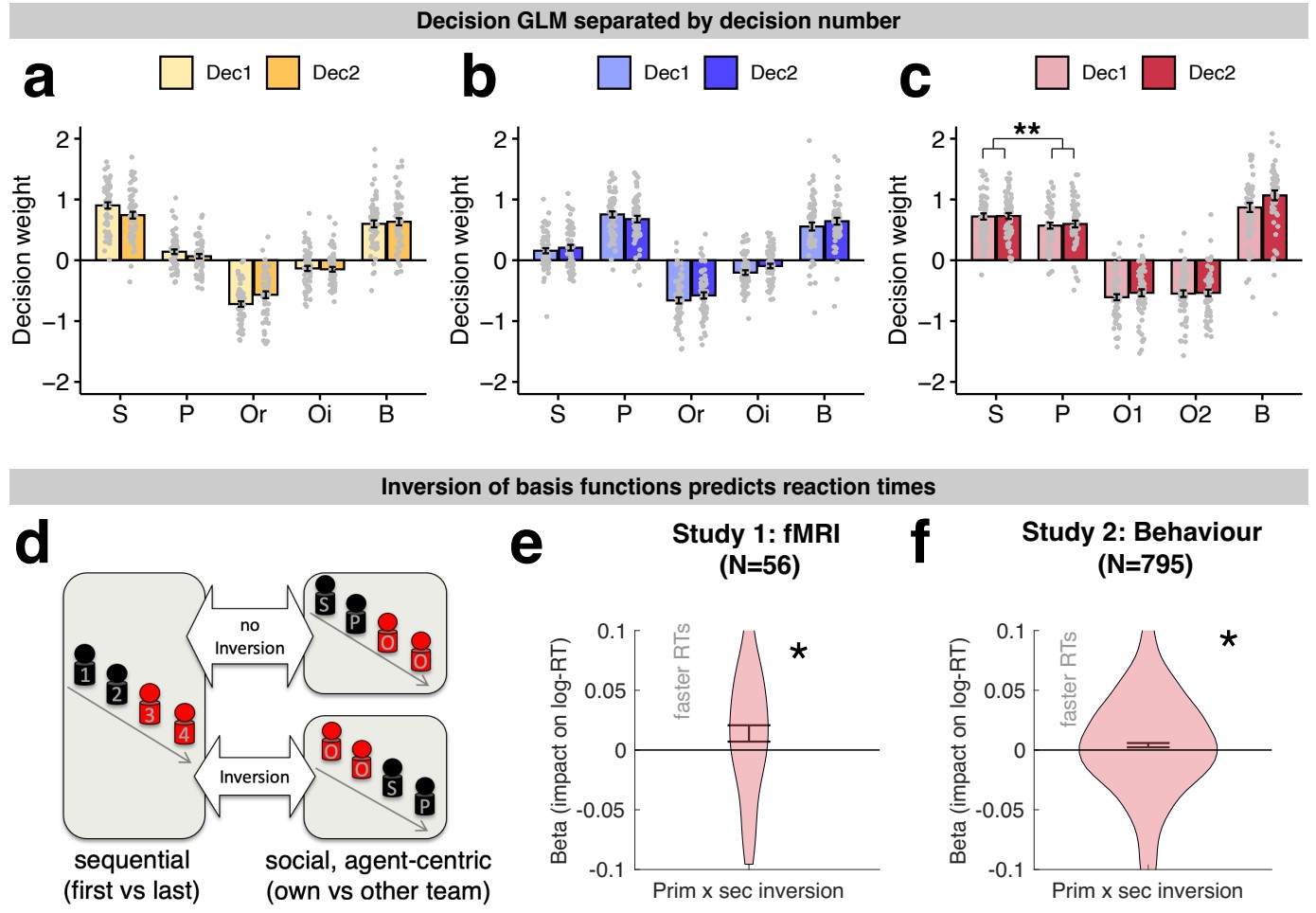

**Extended Data Fig. 9** | See next page for caption.

**Extended Data Fig. 9 | Supplementary analyses of basis function-related behavioural signatures 2 (study 1). (a,b,c) Decision GLM for all decision types binned by decision number (study 1).** We estimated the decision effect sizes for all players and the non-social bonus for all decision types (self, partner, group decisions). To this end, we fitted a logistic regression to the choice data (engage vs avoid) using ridge regression (see Methods: *Data analysis. Ridge Regression*). All regressors were normalised (mean of zero, standard deviation of 1). We fitted the regressions separately for decision type but combined decisions 1 and 2 to ensure comparable effect sizes. **(a) Self decisions.** Decision weights for Self-performance (S), Partner-performance (P), relevant opponent performance (Or), irrelevant opponent performance (Oi) and the non-social bonus (B). We ran 5 [variable: S,P,Or,Oi,B] × 2 [decision number: 1,2] repeated measures ANOVA on the decision weights and found a main effect of variable ($F_{4,220} = 195.7$, $p < 0.001$) and an interaction ($F_{4,220} = 5.18$, $p < 0.001$). Participants were more likely to judge themselves as better than Or if they performed well, Or performed badly, and the bonus was high. The interaction indicated that decision weights trended more towards zero for the second decision. **(b) Partner decisions.** The results in partner decisions were comparable. Note that S and P had reversed meaning in terms of their decision relevance. We ran 5 [variable: S,P,Or,Oi,B] × 2 [decision number: 1,2] repeated measures ANOVA on the decision weights and found a main effect of variable ($F_{4,220} = 154.3$, $p < 0.001$), a main effect of decision number ($F_{1,55} = 5.146$, $p = 0.027$). Again, similar changes between decision 1 and decision 2 were visible consistent with a decrease in accuracy. **(c) Group decisions.** In the group decision, all players are relevant. We ran a 5 [variable: Self (S), Partner (P), Opponent 1 (O1), opponent 2 (O2), bonus (B)] x 2 [decision number: 1,2] repeated measures ANOVA on the decision weights and found a main effect of variable ($F_{4,220} = 205.7$, $p < 0.001$), a main effect of decision number ($F_{1,55} = 4.856$, $p = 0.032$). In addition to these analyses, in group decisions, we analysed whether people place more weight on information related to themselves (S) compared to their partner (P), even though, based on the payoff scheme, S and P should be weighted exactly equally. We ran a 2 [player: S,P] x 2[decision number: 1,2] repeated measures ANOVA and indeed found a main effect of player ($F_{1,55} = 10.292$, $p = 0.002$), but no main effect of decision number and no interaction (all $p > 0.68$). Participants indeed relied more on S-performance compared to P-performance when weighing up which team performed better in group decisions. **(d,e,f) Inversion of basis functions correlates with reaction times (study 1 and study 2). (d)** Illustration of primary basis function inversion. Black and red indicate positive and negative weights, respectively. Note that the lower-right case involves a "flip" or inversion of the basis function, whereas the upper-right case does not. **(e)** We analysed the effects on choice of whether or not a basis function had to be inverted (see *Sorting of basis function and transformation to choice* section) in our fMRI data set, focusing on self and partner decisions. To detect effects of basis function inversion, we restricted the analyses to trials with only non-zero basis functions. A basis vector with a value of zero never needed to be inverted, as it cannot be sign-reversed. We used a linear regression to analyse reaction times (RT). We log-transformed RTs to account for skewness of the data. All regressors were normalized (mean of zero, standard deviation of one). As control regressors, we included absolute value difference (relevant own team member minus opponent team member plus bonus), the temporal positions of Self and Partner (could be 1, 2, 3, or 4th). The latter ensured that effects of primary function inversions were not driven in a simple way by temporal preference effects for players of one's own group. As we have shown that secondary basis functions should be integrated only after primary basis functions, we reasoned that secondary code inversions might impact RTs as a function of whether or not primary code inversions had also occurred. Hence, we considered the interaction term of primary and secondary code inversions (1 if both codes remained non-inverted or if both codes were inverted, 0 if one code was inverted and the other was not). We entered this interaction term as our variable of interest in the linear GLM predicting logRTs. In addition, we entered the component parts of the interaction: a regressor for primary code inversions (1 or 0) and secondary code inversions (1 or 0). We found indeed that the interaction term between primary and secondary basis function inversions predicted faster reaction times ($t_{55} = -2.026$, $p = 0.048$). **(f)** We replicated the above effect in our online study. We used the same GLM model as for the fMRI study. To account for the reduced trial number and increased noisiness of the data, we included all self and partner decisions in the analyses and did not restrict the analysis to trials with only non-zero basis functions. Also, we used the matlab function "robustfit" to account for outliers. We found, again, that the interaction term capturing whether or not primary and secondary basis functions were either both un-inverted or both inverted predicted faster reaction times ($t_{790} = 2.2718$, $p = 0.023$). Note that the effect is driven by a significant effect in the Group condition ($t_{393} = 2.1$, $p = 0.036$). The effect was not itself significant in the No-Group condition ($t_{396} = 1.05$, $p = 0.29$). Note that this result even persists when sub-selecting trials with non-zero basis functions as in the related fMRI analysis ($t_{791} = -1.82$, $p = 0.035$, one-sided test). Finally, we note that the GLMs failed to fit in a very small subset of participants (up to 4 out of the 795 participants) and so the analysis could not be performed with their data ($n = 56$; *, $p < 0.05$; **, $p < 0.005$; error bars are S.E.M; tests are two-sided t-tests).

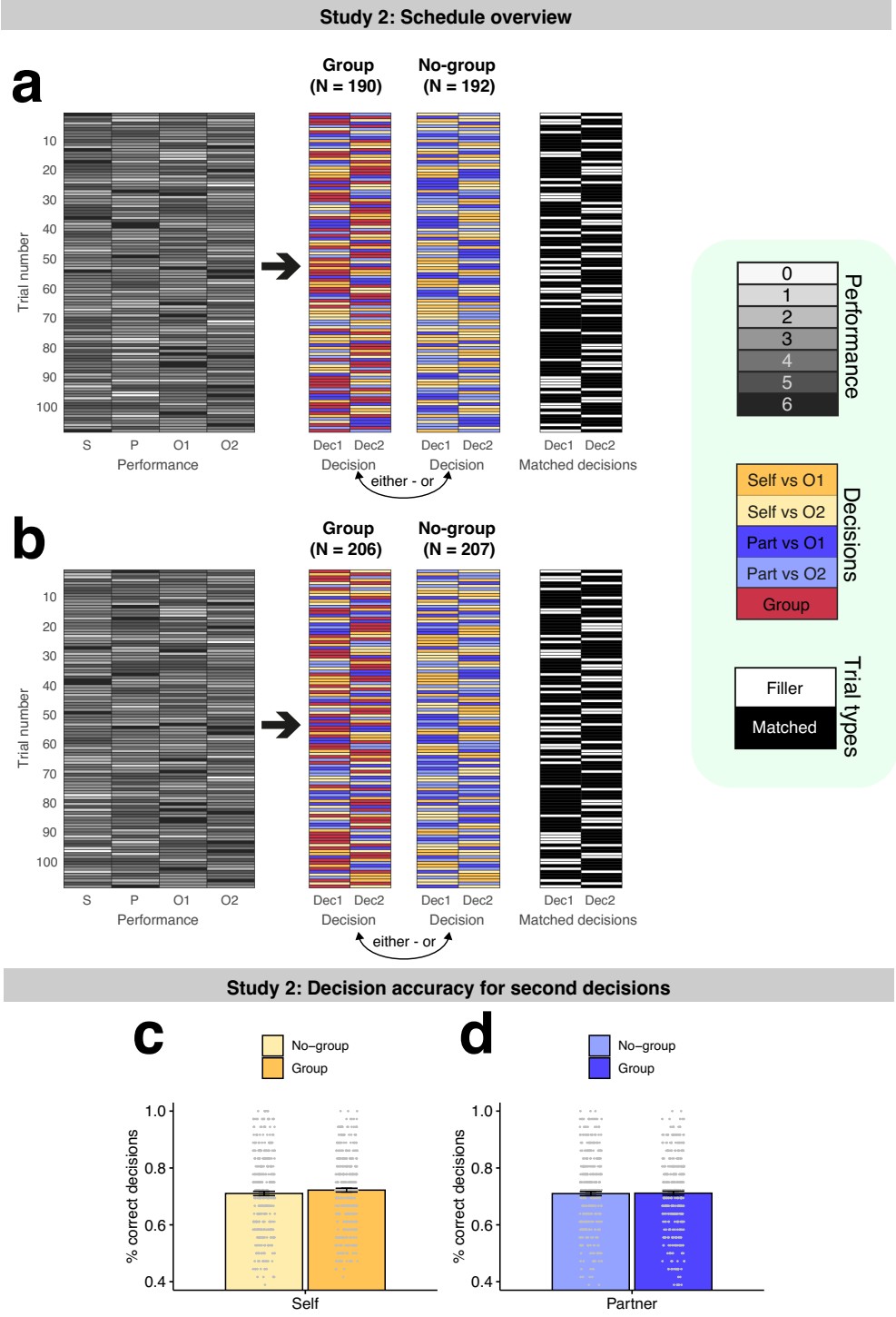

**Study 2: Schedule overview**

**Study 2: Decision accuracy for second decisions**

**Extended Data Fig. 10** | See next page for caption.

**Extended Data Fig. 10 | Supplementary schedule information and analyses of study 2. (a,b) Schedule properties and supplementary analyses controlling for schedule properties. (a)** Detailed illustration of one schedule we used in the online experiment. The left-most box illustrates the precise set of performances (colour coded in white-to-black) that were shown for all four players (S=Self, P=Partner, O1=Opponent 1, O2=Opponent 2) on each trial of the experiment (overall 108 trials). After observing the performances in the observation phase participants made two decisions about these performances (Dec1=Decision 1; D2=Decision 2; see Fig. 1 and Extended Data Fig. 1 for details of the trial timeline). The decisions differed based on condition, but the schedules of observed performances were the same. In the Group condition, participants performed self, partner or group decisions. In the No-Group condition, participants performed either self or partner decisions but no group decisions. Note that this plot illustrates the decisions in detail, separating them into those that involved O1 or O2 (the two shades of yellow/blue, respectively). The right-most box indicates *matched decisions* in black. We analysed exclusively choices on matched decisions that were identical across Group and No-Group condition. Note that, in the No-Group condition, group decisions were replaced by self and partner decisions, but these decisions were "filler" decisions. We did not analyse filler decisions because they were not precisely matched across conditions. **(b)** In addition to the "version 1" schedule above, we also used a second version of the schedule ("version 2") that differed only in the precise performance and decision assignments (see Methods). Note that behavioural differences between "version 1" and "version 2" were not of interest for our research question. We were interested in behavioural differences across Group and No-Group condition. The two versions were only used to assess the stability of our experimental effects across numerical differences in the to-be-remembered information. Note that, therefore, we made sure to acquire a similar number of participants for each schedule version for both Group and No-Group condition. Indeed, all our results persisted when controlling for schedule version. We did this in a detailed manner that followed up on each of our results: To examine participant accuracy we ran a two-way ANOVA with the factors schedule [version1 vs version2] and group [group vs No-group].

We applied this ANOVA separately to examine the percentage correct first self decisions and percentage correct first partner decisions. We used separate ANOVAs to specifically investigate each result in detail. The critical test was whether the main effect of Group vs No-Group condition persisted and whether it needed to be qualified by an interaction with schedule. In all cases, the main effect of group persisted and we did not observe an interaction with schedule. When we examined the percentage correct on first self decisions, we found that our main effect of group persisted ($F_{1,791} = 4.225$, $p = 0.04$) and there was no interaction with schedule ($F_{1,791} = 0.156$, $p = 0.693$). Similarly, the main effect of group persisted in first partner decisions ($F_{1,791} = 4.958$, $p = 0.026$) and there no interaction ($F_{1,791} = 0.402$, $p = 0.526$). We also applied the same rationale to our logistic GLM of self decisions: We ran two-way ANOVAs with the factors schedule [version1 vs version2] and group [Group vs No-Group] for each effect of interest. In all cases, we found that the main effect of group persisted while there was no interaction with schedule. The group effect of S-Or remained ($F_{1,791} = 9.219$, $p = 0.002$) in the absence of an interaction ($F_{1,791} = 0.231$, $p = 0.631$). The group effect of P-Oi remained ($F_{1,791} = 20.395$, $p < 0.001$) in the absence of an interaction ($F_{1,791} = 0.640$, $p = 0.424$). The bonus effect remained higher for the no-Group compared to the Group condition ($F_{1,791} = 18.142$, $p < 0.001$) in the absence of an interaction ($F_{1,791} = 0.480$, $p = 0.489$). Finally, we applied the same approach to partner decisions and obtained analogous results. The group effect of P-Or remained significant ($F_{1,791} = 9.045$, $p = 0.003$), while there was no interaction ($F_{1,791} = 0.048$, $p = 0.827$). The group effect of S-Oi remained ($F_{1,791} = 19.232$, $p < 0.001$), while there was no interaction ($F_{1,791} = 0.064$, $p = 0.80$). Again, the bonus had a bigger effect in the No-Group compared to the Group condition ($F_{1,791} = 21.984$, $p < 0.001$) in the absence of an interaction ($F_{1,791} = 0.263$, $p = 0.608$). **(c,d) Supplementary results for self decisions that were made second and partner decisions that were made second**. Note that, if anything, the mean tends to be higher in the Group compared to the No-Group condition. However, these effects are not significant in self decisions that came second ($t_{793} = 1.153$, $p = 0.249$) nor for partner decisions that came second ($t_{793} = 0.144$, $p = 0.886$). (two-sided t-tests and ANOVAs were used; error bars are S.E.M; $n_{total} = 795$; $n_{group} = 396$; $n_{no-group} = 399$).

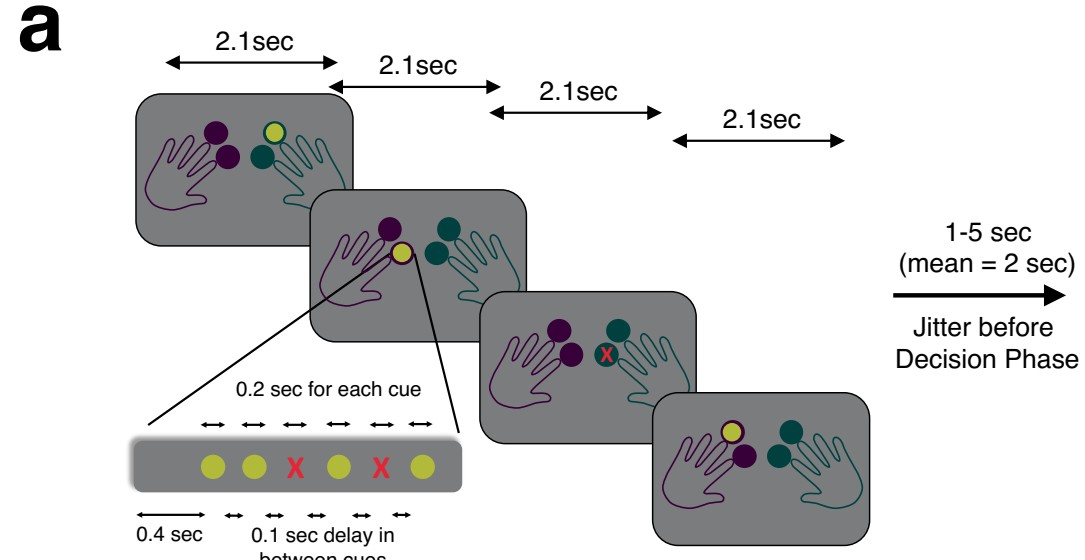

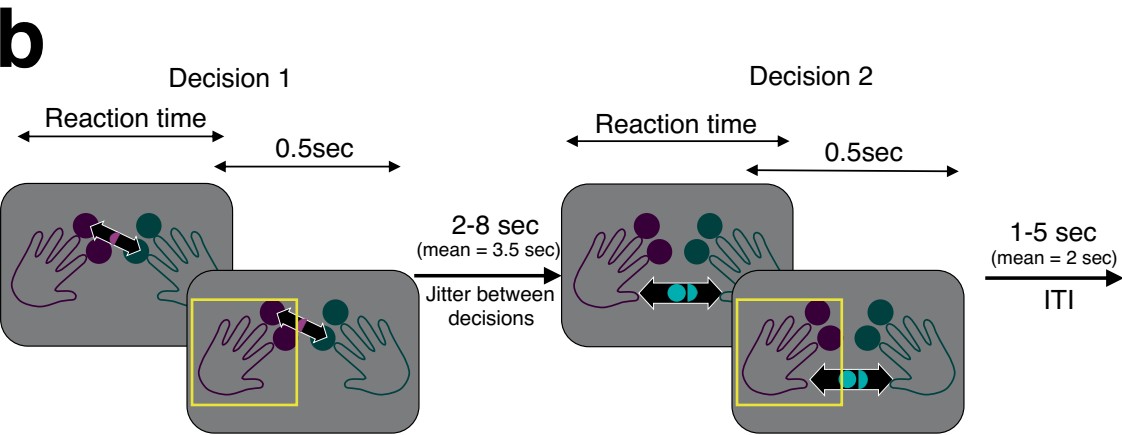

**Extended Data Fig. 11 | Detailed trial timeline of control fMRI experiment (study 3).** See Methods regarding study 3 (*Motor fMRI experiment* section) for a detailed summary of the experimental timeline. As in the social fMRI experiment, each trial in the control fMRI experiment comprised an observation phase and two decisions. **(a)** As in the social fMRI experiment, the control fMRI experiment comprised an observation phase. This phase was conceptually, and with respect to timings, identical to the social experiment. The difference lay only in the visual stimulus arrangement and the framing of the experiment. While in the social experiment, participants monitored success/failure (yellow coin/ red cross) outcomes for each player in a pseudorandom sequence, participants now monitored button presses/omissions (yellow coin/ red cross) for four fingers (left and right middle and index fingers) in pseudorandom order. Note that participants did not press buttons during this; this was just a cover story (see Methods). **(b)** Decision phase. The decision phase of the control fMRI experiment, too, differed from the social fMRI experiment exclusively in terms of the visual display and the framing of the decisions using the identical payoff matrix. Participants compared the number of assumed presses per finger (i.e., the number of yellow coins per finger) and an arrow indicated which fingers needed to be compared. As in the observation phase, all statistical contingencies of the experiment determining optimal choices were identical and matched with the social study trial-by-trial for every single participant.

Note that the roles of the two hands in the control fMRI experiment formally correspond to the roles of the two groups from the social fMRI experiment. Similarly, we arbitrarily assigned roles to specific fingers that had previously been assigned to specific players: left-middle finger was "Motor-Self", left index finger was "Motor-Partner", right-index finger was "Motor-Opponent 1" and right-middle finger was "Motor-Opponent 2". We used the same decision pairings as in the social fMRI experiment but mapped onto these relationships. Again, we also used a bonus represented, just as in the social fMRI study, by coloured coins displayed on the arrow indicating the to-be-compared fingers. The bonus was purple or turquoise and the colour indicated which hand the bonus favoured (purple for purple hand, turquois for turquois hand). The left hand mapped onto the self/partner group of the social study and the right hand mapped onto the O1/O2 group of the social study. Consequently, we defined engage decisions as picking the left hand and avoid decisions as picking the right hand (in the social fMRI experiment, engage decisions related to picking one's own team and avoid decisions related to picking the opponent team). The payoff from an engage choice corresponded to the difference in scores/button presses (including the bonus) of the relevant fingers and could be positive or negative. An avoid choice resulted in a box shown around the opponent team and resulted in a zero payoff. The participants' task was to accumulate as many points as possible during the experiment.

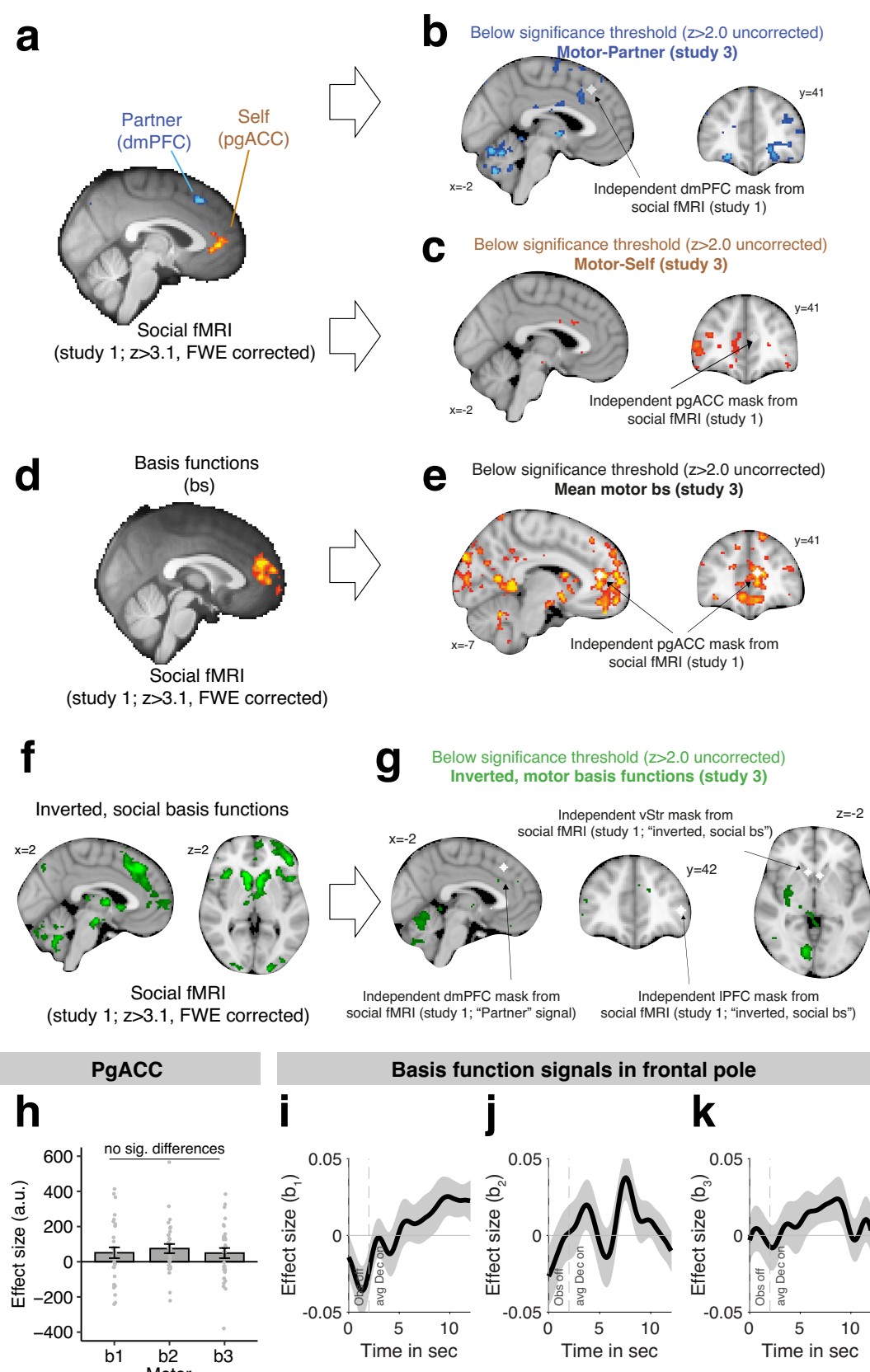

**Extended Data Fig. 12** | See next page for caption.

**Extended Data Fig. 12 | Supplementary neural analyses of study 3.**
**(a-g) Sub-threshold activation maps for social and basis function signals
in the control fMRI experiment. (a)** Signals relating to self and partner in the
social fMRI experiment (see main text). **(b,c)** The corresponding activation
maps for motor-P and motor-S thresholded at z > 2.0 for illustration only. Note
the small white spheres, which are ROI masks of dmPFC and pgACC used to
assess significance in an ROI for the motor study. Note that both ROIs are
unbiased and independent (i.e. they were derived from the social study and
applied to the motor study). Even when examining activity at this very low
threshold, nothing corresponding to the partner-related activity in dmPFC and
the self-related activity in pgACC could be seen in the control fMRI experiment.
**(d)** Whole brain effects of the three basis function projections and S-position in
the social experiment. **(e)** The corresponding activation map for the mean of
the basis function projections (not including S-position, because that variable
had lost its meaning in the context of the motor study) in the control fMRI
experiment, thresholded at z > 2.0 for illustration only. Note the small white
pgACC sphere used to assess significance in a ROI for the motor study. Again,
ROI selection was unbiased and independent (i.e. ROIs were derived from the
social fMRI experiment and applied to the control fMRI experiment. The
pattern of activity seen here is consistent with those shown in Fig. 5 and confirm
that analogous basis functions are coded in both tasks. Note that we did not find
any whole-brain significant negative effect of the combined combined or any
individual basis function projection; not in the social study (study 1) and not in
the motor study (study 3). **(f)** Cluster-significant inverted social basis functions
in the social study. **(g)** The corresponding activation maps for the inverted
motor basis functions (combining inverted primary and inverted secondary
basis function across both decisions). The activation map is thresholded at
z > 2.0 for illustration only. Note that the small white spheres, are unbiased
and independent masks from the social study related to the partner signal
observed there (dmPFC mask, same as above) and related to the aggregate
effects of the inverted social basis functions (lPFC = (−44, 42, 4), vStr ROI
combines (−6, 14, -2) and (8, 18, -2)). As can be seen, inverted basis functions
signals are clearly absent in and adjacent to these regions. Instead, the inverted
basis functions in the motor study captured significant effects in the motor
cortex, insular cortex and cerebellum. Note that these signals could indicate
that inverted basis functions in the motor domain are processed in effector-
related regions. However, they might also relate to the fixed response mapping
utilised in the motor study (the left hand was equivalent to one's own "team")
(n = 32). **(h) In the control study, basis function projections are uniformly
present in pgACC.** Inspection of the three basis function projections in
pgACC revealed that their effects were uniformly positive with no significant
differences between them (Fig. 5j; 1-way ANOVA: $F_{2,62} = 0.288$ p = 0.751,
$BF_{01} = 8.312$). Moreover, as expected, replicating previous results, the null-
vector reflecting variation in the BOLD signal that was not along the basis
function axes, had no significant effects in pgACC ($t_{31} = 1.448$, p = 0.158,
$BF_{01} = 2.056$). **(i,j,k) Basis function signals in frontal pole**. These plots display
the basis function signals in the frontal pole (MNI = (−14, 56, 24)). Note that we
refrain from testing their significance in these panels, because we had already
confirmed their average effect in our cluster-corrected whole brain analysis
(see Fig. 5). In addition, we also confirmed that there were no significant
differences between the three basis functions (absence of main effect in 1-way
ANOVA with $b_1, b_2$ and $b_3$) at any point of the timecourse (all $F_{2,62} < 1.479$; all
p > 0.236; n = 32; error bars are S.E.M).

# Reporting Summary

## Statistics

For all statistical analyses, confirm that the following items are present in the figure legend, table legend, main text, or Methods section.

| n/a | Confirmed | |
|---|---|---|
| ☐ | ☒ | The exact sample size (*n*) for each experimental group/condition, given as a discrete number and unit of measurement |
| ☐ | ☒ | A statement on whether measurements were taken from distinct samples or whether the same sample was measured repeatedly |
| ☐ | ☒ | The statistical test(s) used AND whether they are one- or two-sided<br>*Only common tests should be described solely by name; describe more complex techniques in the Methods section.* |
| ☐ | ☒ | A description of all covariates tested |
| ☐ | ☒ | A description of any assumptions or corrections, such as tests of normality and adjustment for multiple comparisons |
| ☐ | ☒ | A full description of the statistical parameters including central tendency (e.g. means) or other basic estimates (e.g. regression coefficient) AND variation (e.g. standard deviation) or associated estimates of uncertainty (e.g. confidence intervals) |
| ☐ | ☒ | For null hypothesis testing, the test statistic (e.g. *F*, *t*, *r*) with confidence intervals, effect sizes, degrees of freedom and *P* value noted<br>*Give P values as exact values whenever suitable.* |
| ☐ | ☒ | For Bayesian analysis, information on the choice of priors and Markov chain Monte Carlo settings |
| ☐ | ☒ | For hierarchical and complex designs, identification of the appropriate level for tests and full reporting of outcomes |
| ☐ | ☒ | Estimates of effect sizes (e.g. Cohen's *d*, Pearson's *r*), indicating how they were calculated |

*Our web collection on statistics for biologists contains articles on many of the points above.*

## Software and code

Policy information about availability of computer code

| Data collection | Stimuli were presented and choices recorded via Psychtoolbox and Matlab. Imaging data were acquired with a 3-Tesla Siemens MRI scanner by using a 64-channel head coil. Choices and response times were recorded with button boxes and keyboards. |
|---|---|
| Data analysis | We simulated, analysed and visualised data using Matlab 2021a, Jasp version 0.16 , and gramm. We used FSL for the neural data analysis. |

For manuscripts utilizing custom algorithms or software that are central to the research but not yet described in published literature, software must be made available to editors and reviewers. We strongly encourage code deposition in a community repository (e.g. GitHub). See the Nature Portfolio guidelines for submitting code & software for further information.

## Data

Policy information about availability of data

All manuscripts must include a data availability statement. This statement should provide the following information, where applicable:
- Accession codes, unique identifiers, or web links for publicly available datasets
- A description of any restrictions on data availability
- For clinical datasets or third party data, please ensure that the statement adheres to our policy

Data Availability. The data that support the findings of this study are available from the corresponding author upon reasonable request and will be made publicly available in an online repository upon publication.

## Human research participants

Policy information about studies involving human research participants and Sex and Gender in Research.

| | |
|---|---|
| Reporting on sex and gender | We conducted two functional magnetic resonance imaging (fMRI) experiments and two behavioural experiments. The final sample for the first fMRI (study 1) comprised 56 participants (age range 18 - 38 years, 33 female). The final sample for the second fmri study (study 3) comprised 32 participants (age range 19 - 39 years, 22 female). The final sample for the first behavioural study (study 2) comprised 795 participants and for the second behavioural study (study 4) 1022 participants. |
| Population characteristics | see above |
| Recruitment | Participants for the fMRI experiments were recruited via study advertisements, word-of-mouth and email. Participants for the behavioural studies were recruited online via Prolific. |
| Ethics oversight | The Ethics committee of the University of Oxford |

Note that full information on the approval of the study protocol must also be provided in the manuscript.

## Field-specific reporting

Please select the one below that is the best fit for your research. If you are not sure, read the appropriate sections before making your selection.

☒ Life sciences          ☐ Behavioural & social sciences          ☐ Ecological, evolutionary & environmental sciences

For a reference copy of the document with all sections, see nature.com/documents/nr-reporting-summary-flat.pdf

## Life sciences study design

All studies must disclose on these points even when the disclosure is negative.

| | |
|---|---|
| Sample size | Study 1 (fmri) comprised 56 participants, study 2 (behaviour) comprised 795 participants, study 3 (fmri) comprised 32 participants, study 4 (behaviour) comprised 1022 participants. |
| Data exclusions | We excluded a small number of participants from the fMRI and behavioural experiments according to standard criteria for fMRI and online studies. |
| Replication | We replicated neural effects of interest across the two fMRI studies. We replicated behavioural effects of interest across all four studies. |
| Randomization | For fMRI studies, we used a within subject design and so no randomization was performed. Online studies were conducted during a similar time period and participants assigned themselves to different conditions by choosing to participate in the study (without knowing which experiment version they signed up for) . |
| Blinding | For fMRI experiments, we used a within subject design and therefore blinding was not relevant. Online studies were double blinded in the sense that participants were unaware of their experimental condition and experimenters did not interact with participants, because the study was conducted online and required no direct contact between experimenter and participant. |

## Behavioural & social sciences study design

All studies must disclose on these points even when the disclosure is negative.

| | |
|---|---|
| Study description | *Briefly describe the study type including whether data are quantitative, qualitative, or mixed-methods (e.g. qualitative cross-sectional, quantitative experimental, mixed-methods case study).* |
| Research sample | *State the research sample (e.g. Harvard university undergraduates, villagers in rural India) and provide relevant demographic information (e.g. age, sex) and indicate whether the sample is representative. Provide a rationale for the study sample chosen. For studies involving existing datasets, please describe the dataset and source.* |
| Sampling strategy | *Describe the sampling procedure (e.g. random, snowball, stratified, convenience). Describe the statistical methods that were used to predetermine sample size OR if no sample-size calculation was performed, describe how sample sizes were chosen and provide a rationale for why these sample sizes are sufficient. For qualitative data, please indicate whether data saturation was considered, and what criteria were used to decide that no further sampling was needed.* |
| Data collection | *Provide details about the data collection procedure, including the instruments or devices used to record the data (e.g. pen and paper, computer, eye tracker, video or audio equipment) whether anyone was present besides the participant(s) and the researcher, and* |

| | |
|---|---|
| | *whether the researcher was blind to experimental condition and/or the study hypothesis during data collection.* |
| Timing | *Indicate the start and stop dates of data collection. If there is a gap between collection periods, state the dates for each sample cohort.* |
| Data exclusions | *If no data were excluded from the analyses, state so OR if data were excluded, provide the exact number of exclusions and the rationale behind them, indicating whether exclusion criteria were pre-established.* |
| Non-participation | *State how many participants dropped out/declined participation and the reason(s) given OR provide response rate OR state that no participants dropped out/declined participation.* |
| Randomization | *If participants were not allocated into experimental groups, state so OR describe how participants were allocated to groups, and if allocation was not random, describe how covariates were controlled.* |

# Ecological, evolutionary & environmental sciences study design

All studies must disclose on these points even when the disclosure is negative.

| | |
|---|---|
| Study description | *Briefly describe the study. For quantitative data include treatment factors and interactions, design structure (e.g. factorial, nested, hierarchical), nature and number of experimental units and replicates.* |
| Research sample | *Describe the research sample (e.g. a group of tagged Passer domesticus, all Stenocereus thurberi within Organ Pipe Cactus National Monument), and provide a rationale for the sample choice. When relevant, describe the organism taxa, source, sex, age range and any manipulations. State what population the sample is meant to represent when applicable. For studies involving existing datasets, describe the data and its source.* |
| Sampling strategy | *Note the sampling procedure. Describe the statistical methods that were used to predetermine sample size OR if no sample-size calculation was performed, describe how sample sizes were chosen and provide a rationale for why these sample sizes are sufficient.* |
| Data collection | *Describe the data collection procedure, including who recorded the data and how.* |
| Timing and spatial scale | *Indicate the start and stop dates of data collection, noting the frequency and periodicity of sampling and providing a rationale for these choices. If there is a gap between collection periods, state the dates for each sample cohort. Specify the spatial scale from which the data are taken* |
| Data exclusions | *If no data were excluded from the analyses, state so OR if data were excluded, describe the exclusions and the rationale behind them, indicating whether exclusion criteria were pre-established.* |
| Reproducibility | *Describe the measures taken to verify the reproducibility of experimental findings. For each experiment, note whether any attempts to repeat the experiment failed OR state that all attempts to repeat the experiment were successful.* |
| Randomization | *Describe how samples/organisms/participants were allocated into groups. If allocation was not random, describe how covariates were controlled. If this is not relevant to your study, explain why.* |
| Blinding | *Describe the extent of blinding used during data acquisition and analysis. If blinding was not possible, describe why OR explain why blinding was not relevant to your study.* |

Did the study involve field work?  ☐ Yes  ☐ No

## Field work, collection and transport

| | |
|---|---|
| Field conditions | *Describe the study conditions for field work, providing relevant parameters (e.g. temperature, rainfall).* |
| Location | *State the location of the sampling or experiment, providing relevant parameters (e.g. latitude and longitude, elevation, water depth).* |
| Access & import/export | *Describe the efforts you have made to access habitats and to collect and import/export your samples in a responsible manner and in compliance with local, national and international laws, noting any permits that were obtained (give the name of the issuing authority, the date of issue, and any identifying information).* |
| Disturbance | *Describe any disturbance caused by the study and how it was minimized.* |

# Reporting for specific materials, systems and methods

We require information from authors about some types of materials, experimental systems and methods used in many studies. Here, indicate whether each material, system or method listed is relevant to your study. If you are not sure if a list item applies to your research, read the appropriate section before selecting a response.

| Materials & experimental systems | | Methods | |
|---|---|---|---|
| n/a | Involved in the study | n/a | Involved in the study |
| ☒ | ☐ Antibodies | ☒ | ☐ ChIP-seq |
| ☒ | ☐ Eukaryotic cell lines | ☒ | ☐ Flow cytometry |
| ☒ | ☐ Palaeontology and archaeology | ☐ | ☒ MRI-based neuroimaging |
| ☒ | ☐ Animals and other organisms | | |
| ☒ | ☐ Clinical data | | |
| ☒ | ☐ Dual use research of concern | | |

# Antibodies

Antibodies used — *Describe all antibodies used in the study; as applicable, provide supplier name, catalog number, clone name, and lot number.*

Validation — *Describe the validation of each primary antibody for the species and application, noting any validation statements on the manufacturer's website, relevant citations, antibody profiles in online databases, or data provided in the manuscript.*

# Eukaryotic cell lines

Policy information about cell lines and Sex and Gender in Research

Cell line source(s) — *State the source of each cell line used and the sex of all primary cell lines and cells derived from human participants or vertebrate models.*

Authentication — *Describe the authentication procedures for each cell line used OR declare that none of the cell lines used were authenticated.*

Mycoplasma contamination — *Confirm that all cell lines tested negative for mycoplasma contamination OR describe the results of the testing for mycoplasma contamination OR declare that the cell lines were not tested for mycoplasma contamination.*

Commonly misidentified lines
(See ICLAC register) — *Name any commonly misidentified cell lines used in the study and provide a rationale for their use.*

# Palaeontology and Archaeology

Specimen provenance — *Provide provenance information for specimens and describe permits that were obtained for the work (including the name of the issuing authority, the date of issue, and any identifying information). Permits should encompass collection and, where applicable, export.*

Specimen deposition — *Indicate where the specimens have been deposited to permit free access by other researchers.*

Dating methods — *If new dates are provided, describe how they were obtained (e.g. collection, storage, sample pretreatment and measurement), where they were obtained (i.e. lab name), the calibration program and the protocol for quality assurance OR state that no new dates are provided.*

☐ Tick this box to confirm that the raw and calibrated dates are available in the paper or in Supplementary Information.

Ethics oversight — *Identify the organization(s) that approved or provided guidance on the study protocol, OR state that no ethical approval or guidance was required and explain why not.*

Note that full information on the approval of the study protocol must also be provided in the manuscript.

# Animals and other research organisms

Policy information about studies involving animals; ARRIVE guidelines recommended for reporting animal research, and Sex and Gender in Research

Laboratory animals — *For laboratory animals, report species, strain and age OR state that the study did not involve laboratory animals.*

Wild animals — *Provide details on animals observed in or captured in the field; report species and age where possible. Describe how animals were caught and transported and what happened to captive animals after the study (if killed, explain why and describe method; if released, say where and when) OR state that the study did not involve wild animals.*

Reporting on sex — *Indicate if findings apply to only one sex; describe whether sex was considered in study design, methods used for assigning sex.*

| Reporting on sex | *Provide data disaggregated for sex where this information has been collected in the source data as appropriate; provide overall numbers in this Reporting Summary. Please state if this information has not been collected.  Report sex-based analyses where performed, justify reasons for lack of sex-based analysis.* |
| Field-collected samples | *For laboratory work with field-collected samples, describe all relevant parameters such as housing, maintenance, temperature, photoperiod and end-of-experiment protocol OR state that the study did not involve samples collected from the field.* |
| Ethics oversight | *Identify the organization(s) that approved or provided guidance on the study protocol, OR state that no ethical approval or guidance was required and explain why not.* |

Note that full information on the approval of the study protocol must also be provided in the manuscript.

# Clinical data

Policy information about clinical studies

All manuscripts should comply with the ICMJE guidelines for publication of clinical research and a completed CONSORT checklist must be included with all submissions.

| Clinical trial registration | *Provide the trial registration number from ClinicalTrials.gov or an equivalent agency.* |
| Study protocol | *Note where the full trial protocol can be accessed OR if not available, explain why.* |
| Data collection | *Describe the settings and locales of data collection, noting the time periods of recruitment and data collection.* |
| Outcomes | *Describe how you pre-defined primary and secondary outcome measures and how you assessed these measures.* |

# Dual use research of concern

Policy information about dual use research of concern

## Hazards

Could the accidental, deliberate or reckless misuse of agents or technologies generated in the work, or the application of information presented in the manuscript, pose a threat to:

No | Yes
☐ ☐ Public health
☐ ☐ National security
☐ ☐ Crops and/or livestock
☐ ☐ Ecosystems
☐ ☐ Any other significant area

## Experiments of concern

Does the work involve any of these experiments of concern:

No | Yes
☐ ☐ Demonstrate how to render a vaccine ineffective
☐ ☐ Confer resistance to therapeutically useful antibiotics or antiviral agents
☐ ☐ Enhance the virulence of a pathogen or render a nonpathogen virulent
☐ ☐ Increase transmissibility of a pathogen
☐ ☐ Alter the host range of a pathogen
☐ ☐ Enable evasion of diagnostic/detection modalities
☐ ☐ Enable the weaponization of a biological agent or toxin
☐ ☐ Any other potentially harmful combination of experiments and agents

# ChIP-seq

## Data deposition

☐ Confirm that both raw and final processed data have been deposited in a public database such as GEO.

☐ Confirm that you have deposited or provided access to graph files (e.g. BED files) for the called peaks.

| Data access links | *For "Initial submission" or "Revised version" documents, provide reviewer access links.  For your "Final submission" document, provide a link to the deposited data.* |
| *May remain private before publication.* | |

| Files in database submission | *Provide a list of all files available in the database submission.* |
|---|---|

| Genome browser session (e.g. UCSC) | *Provide a link to an anonymized genome browser session for "Initial submission" and "Revised version" documents only, to enable peer review. Write "no longer applicable" for "Final submission" documents.* |
|---|---|

## Methodology

| Replicates | *Describe the experimental replicates, specifying number, type and replicate agreement.* |
|---|---|
| Sequencing depth | *Describe the sequencing depth for each experiment, providing the total number of reads, uniquely mapped reads, length of reads and whether they were paired- or single-end.* |
| Antibodies | *Describe the antibodies used for the ChIP-seq experiments; as applicable, provide supplier name, catalog number, clone name, and lot number.* |
| Peak calling parameters | *Specify the command line program and parameters used for read mapping and peak calling, including the ChIP, control and index files used.* |
| Data quality | *Describe the methods used to ensure data quality in full detail, including how many peaks are at FDR 5% and above 5-fold enrichment.* |
| Software | *Describe the software used to collect and analyze the ChIP-seq data. For custom code that has been deposited into a community repository, provide accession details.* |

# Flow Cytometry

## Plots

Confirm that:

☐ The axis labels state the marker and fluorochrome used (e.g. CD4-FITC).

☐ The axis scales are clearly visible. Include numbers along axes only for bottom left plot of group (a 'group' is an analysis of identical markers).

☐ All plots are contour plots with outliers or pseudocolor plots.

☐ A numerical value for number of cells or percentage (with statistics) is provided.

## Methodology

| Sample preparation | *Describe the sample preparation, detailing the biological source of the cells and any tissue processing steps used.* |
|---|---|
| Instrument | *Identify the instrument used for data collection, specifying make and model number.* |
| Software | *Describe the software used to collect and analyze the flow cytometry data. For custom code that has been deposited into a community repository, provide accession details.* |
| Cell population abundance | *Describe the abundance of the relevant cell populations within post-sort fractions, providing details on the purity of the samples and how it was determined.* |
| Gating strategy | *Describe the gating strategy used for all relevant experiments, specifying the preliminary FSC/SSC gates of the starting cell population, indicating where boundaries between "positive" and "negative" staining cell populations are defined.* |

☐ Tick this box to confirm that a figure exemplifying the gating strategy is provided in the Supplementary Information.

# Magnetic resonance imaging

## Experimental design

| Design type | within subject design |
|---|---|
| Design specifications | parametric fMRI design |
| Behavioral performance measures | choice and reaction times |

## Acquisition

| Imaging type(s) | functional magnetic resonance imaging |
|---|---|
| Field strength | 3T |
| Sequence & imaging parameters | 3-Tesla Siemens MRI scanner ,64-channel head coil. T1: TE= 3.97ms, TR = 1.9s,  1x1x1mm voxel size. Multiband T2*- |

| Sequence & imaging parameters | weighted echo planar imaging sequence with acceleration factor of two with TE= 30ms, TR= 1.2sec, 2.4x2.4x2.4mm voxel size, 60° flip angle, a 216 mm field of view and 60 slices per volume. Two fieldmap scans (sequence parameters: TE1, 4.92ms; TE2, 7.38ms; TR, 4482ms; flip angle, 46°; voxel size, 2 x 2 x 2 mm). |
|---|---|
| Area of acquisition | whole-brain |

Diffusion MRI ☐ Used ☒ Not used

## Preprocessing

| Preprocessing software | FSL |
|---|---|
| Normalization | MNI |
| Normalization template | MNI |
| Noise and artifact removal | standard preprocessing and artefact removal via FSL Melodic |
| Volume censoring | none |

## Statistical modeling & inference

| Model type and settings | 1st and 2nd level GLMs |
|---|---|
| Effect(s) tested | parametric regressors of interest |

Specify type of analysis: ☒ Whole brain  ☐ ROI-based  ☐ Both

| Statistic type for inference (See Eklund et al. 2016) | Z>3.1, p=0.05 FWE corrected |
|---|---|
| Correction | Z>3.1, p=0.05 FWE corrected |

## Models & analysis

| n/a | Involved in the study |
|---|---|
| ☒ ☐ | Functional and/or effective connectivity |
| ☒ ☐ | Graph analysis |
| ☒ ☐ | Multivariate modeling or predictive analysis |

| Functional and/or effective connectivity | *Report the measures of dependence used and the model details (e.g. Pearson correlation, partial correlation, mutual information).* |
|---|---|
| Graph analysis | *Report the dependent variable and connectivity measure, specifying weighted graph or binarized graph, subject- or group-level, and the global and/or node summaries used (e.g. clustering coefficient, efficiency, etc.).* |
| Multivariate modeling and predictive analysis | *Specify independent variables, features extraction and dimension reduction, model, training and evaluation metrics.* |

