## [Peer Review File · Nature]

Basis functions for complex social decisions in dorsomedial frontal cortex

Corresponding Author: Mr Marco Wittmann

Version 0:

Reviewer comments:

Referee #1

(Remarks to the Author)

In this paper, Mathew Rushworth and colleagues present an interesting and exciting set of studies aimed at understanding how the brain represents the behavior of multiple agents and how the behavior of other's may be uniquely encoded. To this end, they developed an innovative multi-person social decision-making task in which information about four distinct players were used by the participants to make subsequent decisions. They also developed a novel basis-function focused approach (analogous to that often used to create a vector space but for functions) to examine how neural activity in areas such as the ACC and dmPFC respond to specific elements that define the subjects' decisions. Using these approaches, the authors find that the basis functions aligned well with specific social interactions, more so than simple identities, and that these can be explained by a neural format similar to that previously described for other visual and motor domains. Taken together, this study provides novel findings and a new way to think about how the brain represents interactive social behavior. That said, there are also a number of methodological questions that would be helpful to address and points of clarification that could aid in improving the paper.

First, while the authors provide a number of elegant analyses which suggest that basis functions could explain the data, it wasn't clear to me whether or to what degree are their models have higher explanatory power than other simpler functions. It would be helpful for example to add formal information theoretic metrics such as AIC or BIC. Further, while they made important comparisons to other models such as a drift diffusion model, it would be helpful to provide further comparisons with other models or more basic analytic techniques to demonstrate that their results are optimally explained by their functions.

Second, while the authors suggest that they found signatures of agent-centric representations, it would be helpful to provide a non-social control to demonstrate the social specificity of their findings. While the 'self' and 'other' comparisons suggest a distinction, it wasn't clear whether similar findings could have been observed when playing against any other non-social agent. It also wasn't clear whether or not similar findings could have been observed if performing the above analyses on other non-social serial events/decision making sequences.

Third, the authors suggest that secondary basis projections exerted their influence on choice significantly later than primary basis functions. However, it would be helpful to provide further formal social metrics such differences in reputation or strategy for comparison. In a similar direction, on line 84, do the authors mean that mean that people tended to judge higher their performance as opposed to their opponent's?

Fourth, it would be helpful to provide some clarifications on the various figures and statistics below. Some of the main findings for example appeared to display relatively small differences in activity and it wasn't easily apparent whether they accounted for multiple comparisons.

1. For Figure 1A: There is potentially a typo: number 2) is a different character size than 1). For 1C: this seems to be a very small difference judging by the dot distribution, and by looking at the histogram itself. Was this difference corrected for multiple comparison? If so, indicate in the main text, or in the legend. Same goes for 1D (although there the difference is a little more visible). For 1E, please clarify what the MNI coordinates for the two clusters (medial prefrontal cortex and ACC), not only the X coordinate of the slice. Also, was the cluster in dmPFC bilateral?

2. Figure 2B: Clarify that $DV = pos1 - pos3$ is just an example. It may also help to change the numbers into x and y , so it's more straightforward than $x = 1$ and $y = 3$ if you are comparing S and O-green. Also, add here a number per position in light grey, similarly to figure 2D under the arrow - so if I understand well, it should be 3,5,2,5. It'd make it easier to unpack the figure.

3. Figure 2C & D: Figure 2D: maybe I am missing something, but I don't understand the order of the vectors in the space, so please either clarify it in the legend, or edit the figure. Why is it pos3, 2,1,4? It is a little confusing especially when trying to match it with the light grey example under the arrow (the order of the numbers still reflects the position order - pos1 = 3, pos2=5, pos3=2, pos5=5, but reading it in their order of appearance, it does not correspond, as it is pos3, pos2, pos1, pos4).

4. Figure 3H: following up on my previous comment, I wonder if Social DV is confounded by the fact that by doing chosen-unchosen you are automatically introducing some sort of likelihood of getting reward: for instance, if the subject is to judge performances, comparing a 1 vs a 5 (difference is 4) is not the same as comparing a 3 vs a 4 (difference is 1; i.e. likelihood of getting reward is higher in the first case as it should be easier to judge the performance). If my assumption is correct, wouldn't doing chosen-unchosen be effectively a measure of value in this case, rather than an agent specific measure? Authors should better clarify their view for this in the paper, and potentially think about ways of more directly disentangling chosen value from agent specific activations.

5. Figure 3F: add legend for red and black dots again.

Finally, there were a number of areas within the main text that needed additional clarification with respect to the regions of interest and how different areas may have responded to the task.

1. Lines 133-138: Text needs clarifications. Here, the authors state that "player positions during the observation phase (position1, position2, ...) were independent of player identity (self, partner)". If that's the case, then, why do the vectors for S and pos1 (2C and 2D) have the same length? Similarly, P, seems equal to pos3, O-grey to Pos2, and O-green to Pos4. I don't understand this result when taken together with the sentence at 138: "Analogously, the sequential perspective 136 entailed a 4-dimensional representation along the axes of the four sequential positions, where the length of each axis reflects the performance for the player occupying this sequence position independent of identity". Please, clarify figure and text.

2. Line 228: While it's clear that dmPFC and ACC are relevant areas, and therefore I agree with the idea of a preliminary ROI analysis there, I was wondering if the authors considered to also extend this to vmPFC. Given the task structure and the presence of a monetary reward at the end of the trials attributes a value to each choice, it'd make sense to me if you decided to repeat the same ROI analysis also in vmPFC. I think it'd be plausible to expect activations in vmPFC as well. Moreover, I wonder if that would have any correlations with agent-driven clusters. I would be interested to see results of such analysis specifically (i.e. comparisons between dmPFC and vmPFC chosen value vs agent specific).

3. Line 349: I suggest adding here reference to Figure 3H, and in general also expand this more (see also comment #6).

4. Line 434-447: Have authors observed clusters of activations in amygdala, and if those are also agent dependent? There's a cluster around there in Figure 3H, for social DV, but authors do not talk about it explicitly. The idea underlined at line 446-447 is very compelling, but it'd be nice to see if there're activations in this other traditional social area, and mainly quantify it, as much as the other clusters. It would be useful to add a supplementary figure with social DV areas in support of the discussion, focusing on the red clusters.

Referee #2

(Remarks to the Author)

Wittmann et al. investigated how several pieces of quantitative information about the performance of multiple human players might be initially encoded in the brain, and then later transformed flexibly into the decision variable that can be used during a decision about which individual or group performed better. The main hypothesis tested in this study is that the brain encodes multiple variables in the sequential order as they are presented during the "observation" period using a set of basis functions, rather than encoding them in the agent-centric frame of reference that can be used more directly to make the choice. Indeed, they found that the information about the performance of multiple players is first encoded in terms of the basis functions defined by the serial order, especially in the medial prefrontal cortex. In addition, the ways in which these basis functions must be combined to compute the decision variable depended on the nature of decision (e.g., individual vs. group performance) required in each trial and were reflected in the speed and accuracy of choices (as determined by fitting the drift-diffusion model to the data). Overall, although the behavioral task and analyses are sophisticated, the manuscript was written clearly and relatively easy to follow. The main findings are potentially of great significance, since they might indicate a common computational strategy (basis functions) across perceptual, motor, and social domains. On the other hand, some of the inferences presented in the manuscript for analyzing the neural data were not entirely convincing, and the authors might need to consider some alternative and potentially more parsimonious explanations. If these potential weaknesses are successfully addressed, this manuscript would be exciting for many researchers.

1. The authors should better clarify the relationship between the transformation based on basis functions and data compression. They seem to equate these two types of computations, but one can change the coordinate system without

losing any information or data compression. For task periods for which the authors show strong evidence for the use of basis functions (e.g., between observation and decision phases), the authors do not seem to provide much evidence for the compression.

2. It is not entirely clear and therefore perhaps needs to be explained better why the authors consider only the basis functions indexed by the serial order, but not those indexed by the identities of the agents. For example, in the fMRI GLMs, the information indexed by the serial order is represented in the basis functions, but the information about each agent is entered individually. Is it possible to test both types of information using similar basis functions?

3. One puzzling finding is that the decision accuracy was affected by whether the basis functions need to be flipped or not (page 12, Figure 4B) and by the interaction between the flipping of primary and secondary basis functions (Figure 4C). This is difficult to understand because the results from the analyses related to the basis functions would not change when you multiply any subset of the basis functions by minus one. Therefore, it is not clear why the flipping (or inversion) of any single basis function should have any effect on the behavior (or analysis of neural data).

4. It is interesting that the primary basis function (group contrast) had earlier and stronger effects on behavior than the secondary basis function (needed to be added to primary basis function to compute the decision variables in individual comparison). Is it possible that this is equivalently and alternatively explained as the spreading of individual scores within the same group?

Minor comments:

1. Figure 1B. The arrow indicating the decision type is shown in pink, although this is described as blue in the text.
2. Figure 1E needs a caption to explain the meaning of colors used.
3. Wouldn't it be better to refer to "flipped" and "sign-flipped" as "inverted" and "rotated", instead?
4. For fMRI GLM1, how are O1 and O2 defined for group trials?

Referee #3

(Remarks to the Author)

In this study, the authors examined how human participants represented the decision variables (DVs) in a complex social decision paradigm. In the task, participants first performed a random dot motion discrimination task as the pre-experiment. From this pre-experiment, the researchers defined performance score for each participant (i.e., the accuracy in the motion discrimination task). During the decision period, participants needed to use the performance score information to make decisions in social contexts. The experiment period included two phases. In the observational phase, participants were shown the performance scores for four players (including themselves), where two players formed a team resulting in two teams total. To avoid potential confounds, the participants were not directly told the scores, but rather they needed to infer this information in an evidence-accumulating manner. Importantly, the performance of the four players were shown in a random sequence during the observation phase. It, therefore, raised an interesting question on how participants represented these performance scores. The second phase was the decision phase, during which the participants decided whether to engage or avoid a game based on if their team had a higher score or not compared with the opposite team. The decision was separated into three conditions: 1) only comparing themselves with one of the members in the opposite team; 2) only comparing the partner in their team with one of the members in the opposite team; and 3) comparing the teams as a whole. This design allowed the researchers to dissociate the DV encoding the observation phase and the decision phase, and also dissociate the identities of the players (agents) and their orders in the sequence during the observation phase. In terms of the neural representation, the researchers mainly found that: 1) both agent-based and sequence-based representation in the brain, with stronger signals in the sequential frames of reference. The agent-based and sequence-based representation seemed to happen in different brain areas. 2) Brain regions did not simply encode the DVs for different orders (order-based), but they encoded specific patterns of sequences. 3) When representing such sequences, brain regions only encoded maximally 3 sequences instead of the exhaustive list of sequences, which seemed analogous to the computational mechanisms found in the sensory and motor systems. The authors termed these encoded sequences "basis functions".

Overall, the researchers examined both behavioral and neural bases of a novel computational framework for social decision making by adopting an established approach used for sensory and motor systems research. This is the major strength of this work – they examined how a basis function framework could be used for complex social decisions, in addition to agent-specific computations. However, their analyses and interpretations did not fully rule out some alternatives. Specifically, if their results could possibly be explained in an alternative way over the use of the proposed basis functions may require more exploration (i.e., agent encoding + task sequence encoding). Moreover, it remains unclear how the use of the basis function for computing social decisions could be generalized to other behavioral contexts beyond the specific experimental procedures used here. Despite some remaining clarifications and explanations, this is very exciting and novel work that provides a new framework for social decision making.

Major comments:

1. Generally, these results showed that, in this task, the brain encodes DVs in a hierarchical way? The brain first encodes

DVs for the teams, and then filters out useless information based on the need of the decision phase. Did the basis function encoding say anything more than this? Because such hierarchical encoding does not necessarily require the sequence-based frame of reference, the agent-based model can also explain this by first considering agent groupings, such as my team and other team, in-group and other-group, etc, before becoming more specific. That is, does basis function critical or is it just about encoding another task dimension in addition to the specific agents involved?

2. Line 140: Intuitively, the reason why only 3 dimensions were needed was because the frame of reference for the player sequence was 3 (4 minus 1, once the participants know the identities of the first three positions, they would know the fourth one for certain). Therefore, it is not very surprising that DVs are encoded in a 3D space. For example, if there were 6 players, the dimension would need be 5. Would the number of dimensions simply depend on the task, or is there something special about having 3 dimensions?

3. The authors only showed the comparison between Self and Partner decision conditions (Figure 1C and D). How about the Group condition? It is actually important to know how the Group condition's accuracy and RT looked like, because based on the later basis function hypothesis, the authors should expect higher accuracy and shorter RT for the Group condition. This is because only primary basis function is needed for the Group condition.

4. For the two consecutive decision phases in one trial, did they affect each other (Fig. 1)? For example, was the second decision more accurate? Was the second decision associated with shorter RT compared with the first one? Did the first decision improve or harm the second decision in any way (both behaviorally and neurally)? The authors should explore these topics more and if/how this might be related to the proposed basis function.

5. Is it possible that participants did better in the Self condition mostly because they inferred the performance score better for themselves during the observation score since they directly participated in these trials during pre-experiment (Figure 1C and D)? The authors might want to quantify and compare the accuracy for the participants to infer the scores for each player (including opponents in the other team). It is unclear if the behavioral effects were strongly influenced by the fact that the participants used their memorized experience from pre-experiment in the observation phase for the Self condition to improve the calculation of the scores for themselves. To put it another way, would the participants able to infer accuracy of all players?

6. For Figure 2, could the authors explain more how their sequence-based code can explain the better performance in the Self condition shown in Figure 1C, 1D?

7. In Figure 3D, is the positive peak timing related to the actual Self position in the sequence (that is, earlier peak if Self position was 1 compared to Self position was 4)?

8. Were the information about opponents encoded by any brain areas? (related to Figure 1E)

9. In relation to understanding Figure 3C, please explain explicitly what the null axis is.

Minor comments:

1. line 89: should be Figure 1E.

2. Figure 1E: Please show the color code as an inset to the panel.

Referee #4

(Remarks to the Author)

"Basis functions for complex social decisions in dorsomedial frontal cortex"

Witmann et al.

In this paper, the authors explore how the human brain represents variables relevant to playing a multi-agent social game, as a way to gain traction on the fundamental question of how the brain represents different agents and information pertaining to each in complex social situations. They measure fMRI responses from subjects while the subjects first view the performance of a series of four agents (including themselves) on an auxiliary task, and then answer questions about various combinations of performance outcomes (e.g., self + partner versus two others, self versus other agent # 1, etc.). The authors argue that that dmPFC and ACC represent basis functions capturing social interaction types that align with the task, rather than the performance of individual agents.

I found the paper rather dense and difficult to understand. Nevertheless, I thought the central question posed, what basis set does the brain use to represent a complex social situation, was novel and exciting. However, the evidence presented in favor of the author's claims did not seem very compelling to me. The amount of data presented was very limited and most of the signals were very small (e.g., Figure 1C, D, Figure 4B, C, F). The fMRI data involved averaging across a large number of subjects (N = 56) and no single subject data is shown, making it unclear whether the focal activations might largely be an artifact of intersubject averaging. As further detailed below, I think that a key comparison necessary to interpret the data is

missing, and the central claim re the social basis functions used by the brain is not well supported. Overall, the paper does not seem appropriate for the broad readership of Nature.

Major

1. The authors say multiple times that they think the brain is encoding basis functions related to the task at hand, and that this offers a way to “summarize possible social interactions in a compressed format.” I’m not entirely sure I understand what they mean by compression here. To encode the outcomes of four agents, one would need a 4d space. For their task, they need a 3d space. Is the 4d->3d transformation what they mean by compression? If so, in pure efficiency terms, it doesn’t seem like much is gained by such compression. Moreover, in a real-life social situation, one doesn’t generally know what decision one will be asked to make, and hence the compression strategy here doesn’t seem like it would apply in general—raising doubts about the significance of the findings outside this experiment.

2. The authors pit their finding against the intuitive notion that the brain tags performance of social agents by the agent’s identity separately for each individual agent, i.e., the brain is only representing the performance of the agents along the rotated axes (b1,b2,b3 in Figure 2E) and not agent-centric axes (S, P, O1, O2). But as far as I could tell, nowhere in the paper is it shown that when one simply regresses to the performance of the four individual agents (including self), the beta value is 0. Indeed, I would find this deeply surprising, since it seems that behaviorally, subjects should be able to answer re how each individual agent performed.

The only indirect piece of evidence I saw related to this is the finding that Beta = 0 for the null direction (Figure 3C). I don’t think it is possible to conclude that the brain is not coding the performance of individual agents based on this single indirect negative result. On page 9, the authors state, “none of these agent-centric or decision-centric variables shared more than 0.01 of variance with the basis function projections and all neural analysis controlled for these related variables (Methods)” and later on page 11 they state, “Again, this is striking because these variables encode sequential information and are unrelated to agent-centric or decision related variables, which we again control for in the analysis (Fig. 2E, Supplementary Fig. 2, Methods).” However, I was unable to glean from the Methods how this was controlled.

More fundamentally: within a given trial, the rotated axis (b1, b2, b3) is just a linear combination of the agent-centric axes (S, P, O1, O2). fMRI is simply not an appropriate technique to address what particular basis, amongst different bases related by a linear transform, an area is using, given that it is summing across many, many different cells. To take a simpler example, if an area were coding color, and one found evidence for modulation by red, green, and blue, this would certainly not be an argument that the area does not code yellow (red+green), cyan (green+blue), and magenta (red+blue). Indeed, fMRI would be simply unable to pinpoint whether individual cells were coding RGB or CMY, since this is a question about single cell coding mechanism. Furthermore, fMRI is terrible for making conclusions re what the brain is *not* coding, since finding an fMRI signal depends on many cells having clustered tuning functions that agree in sign, which is a very strong assumption. So I just don’t understand how the authors can be so certain, based on their results, that the brain is not coding the performance of individual agents, but is instead using an alternative basis.

If the claim of the authors is only that the brain is representing social information along task-specific basis functions *in addition* to agent-specific basis functions, this would be a much weaker claim. And it seems that this has to be true, given that performance of the group decision task *requires* computation of b1,b2,b3. So then, it seems the main value is in pinpointing where this representation of b1,b2,b3 occurs and demonstrating that it can be isolated with fMRI. While valuable, I think this is not a major conceptual advance.

Other

1. I don’t understand the social component in the setup of the task at all. Why is it necessary for actual people to be play the role of “P, O1, O2”? It seems the performances of the different agents could have been computer generated without changing the results of the paper.

2. In connection to the point above, I am not convinced that the experiments engaged social neural circuits at all, which is central to the paper. The task seems to involve counting and memorization rather than anything resembling social interaction. If the authors would like to claim the latter, then significantly more evidence should be presented supporting their claim. Figure 1E seemingly represents a small step towards that but is not convincing or sufficient.

3. The task design seems incredibly convoluted and badly presented, introducing a possibility that the results are influenced by severe confounds unrelated to the questions asked. For example, the task contained a bonus component in each trial, which was not described in the figures or the main text, where the counted number of correct trials had to be adjusted with a number shown on the screen. This number had to be added to the own team tally if positive or to the opponent team tally if negative. However, the participants could’ve easily employed a more intuitive strategy of only mentally manipulating the tally of one of the groups, for example own group, and adding positive and subtracting negative numbers. Any contrast might then reveal broad neural signatures of differing mental processes of manipulating numbers rather than any kind of social decisions. Furthermore, inexplicably, the entire structure of the task differed for the trials where own team was judged to perform better than the opponent one. Trials where the opponent team performed better had to be avoided and were not rewarded in any case whereas participants had to select to engage and were rewarded for the trials where their own team was better, possibly biasing fMRI data with activity irrelevant to the question posed but rather coding for task peculiarities. What is the reason for such large variations in the task, especially when the task was not interactive at all?

4. I am confused by the task design shown in Figure 1B. The authors point out in the figure legend that the numbers corresponding to performance were not shown directly, but instead subjects had to “aggregate” the performance. Why is this necessary?

5. The difference between self and other correct seems tiny (Figure 1C).

6. Figures 1E, 3B, 3H are averaged across 56 subjects. Please show single subject data (e.g., the same slice in all 56 subjects) so readers can get a sense for how broad the activations are within single subjects.

7. Please discuss the discrepancies in the fMRI data shown. The activations in Figure 1E seem non-overlapping with activations in Figures 3B and 3H to an unexpectedly large degree. Am I correct in assuming that regardless of analysis type, the brain should still be performing computations revealed in Figure 1E in Figure 3B/3H? In addition to that, the authors could discuss more in depth the broad activations seen in Figure 3H.

8. Figure 3E seems unnecessary; this seems obvious.

9. The effects of representing in a compressed basis on choice shown in Figure 4 seem quite small. It would be beneficial to see data in Figure 4B and 4C also presented in terms of actual RT and % correct, similar to Figures 1C and 1D. Further, Figures 4E and 4F lack appropriate controls. The irrelevant player signals are very small, and their matching direction could be explained by shared task variables, etc. (see 3. above).

Version 3:

Reviewer comments:

Referee #1

(Remarks to the Author)

I would like to thank Dr. Rushworth and team for providing such an extraordinarily comprehensive, detailed and well thought out response and for the truly herculean amount of work that went into revising the paper. The manuscript is much improved and provides a high degree of rigor across multiple modalities and controls that significantly strengthen the paper. In particular, with a new online platform, the authors have now provided additional controls to determine whether the observed social behaviors are better explained by the basis functions versus agent-based models (using nearly 800 new participants). They also examined whether their model better explains the players' scores and the effects of decision-irrelevant players as well as explain how their basis function control for agent-centric representations on fMRI. To further provide a non-social comparison, they recruited 32 additional participants who performed an analogous (motor-based) task, but which lacked social elements. Finally, they provided extensive revisions and clarifications to several of their figures as well as further streamlined their paper and results. As it currently stands, this is one of the most rigorously controlled and well-designed social studies that I have seen in a long time and I would be excited to see it published!

As I stated before and emphasize now again, the paper provides fundamental new insights into social behavior and an important new approach for studying group interactions in humans. Given the amount of new data and added results, my only remaining suggestion would be to trim down (with editorial guidance) some of the text to allow for easier readability. I have no doubt that this paper will be highly influential and well-cited in the field. I would strongly support its publication.

Referee #2

(Remarks to the Author)

The main conclusion of this manuscript is that when the information about the strength of multiple individuals in a social group is sequentially presented for a subsequent decision regarding the relative strengths of individuals or their groups, this information is encoded using a set of basis functions, while the behaviorally irrelevant information encoded in the "null" dimensions is largely ignored. The possibility of this data compression is supported by an array of behavioral and neural findings presented in this study. The authors have also conducted additional behavioral and fMRI experiments to address the issues raised by the reviewers of the original manuscript. These results lend additional support to the hypothesis about the basis functions in the paper, but at the same time, also reveal some limitations of the study and may require some adjustment in the main conclusions, as suggested below.

1. It is quite convincing that the sequential basis functions manifest in the behavioral data and are also reflected in the fMRI signals in the ACC. However, the results from the (motor) control task strongly suggest that the activity in this area correlated to the sequential basis functions is not specific to the social setting. By contrast, the results shown in Supplementary Figure 10 regarding the fMRI activity related to "social" (inverted) basis functions might be more specific to social decisions. It seems unlikely that similar activity would appear during the motor control task. Therefore, it would be quite important for the authors to test whether such inverted (hand-specific) basis functions appear in the same or different brain areas compared to those identified in Supplementary Figure 10.

2. Results shown in Figure 3B need a bit more explanation. First, the authors should include a color bar (scale). Second, what they mean by "the average effects of the projections on to 3 basis functions and S-position" is not clear. For example, does this refer to the average effect size of 3 basis functions, or the total variance accounted for by these 4 regressors? The latter might be more appropriate. This is because the fact that for both results illustrated in Figures 3B and 5J consist only of the positive correlation is very puzzling, given that the signs of the basis functions are (almost) completely arbitrary. For example, in the schematic shown in Figure 2F, the sign for the first serial position is positive for w_2 and w_3 , but negative for w_1 . How were these choices made? If the sign for the first serial position was made to be always positive, do the authors get the same results in Figure 3B? This depends on exactly what is shown in Figure 3B. Also, did the authors find any brain areas in which the average coefficient for any basis function was significantly negative? If not, how can this be explained?

3. Explanation of the results shown in Figures 3G is also somewhat confusing. In particular, the figure caption refers to the light blue region as "primary + secondary basis function". Since the signs for the basis functions that are combined must

change according to the type of dyadic decisions, the authors might have already inverted the signs of the primary and secondary basis functions when necessary to account for the specific type of decisions. If so, this should be mentioned explicitly.

Minor comments.

1. As in the original manuscript, in the Introduction, the authors draw an analogy between their findings and the single-neuron findings from the studies by Tanji and his colleagues during a sequential motor task. However, given that there are some major differences between these studies, this may mislead or confuse some readers. For example, the neurons characterized by Tanji et al. are good examples of “combinatorial” codes, since they respond to specific conjunctions of motor responses and their serial positions. However, they do not behave as the basis functions as described in this manuscript, since they do not encode signals related to the linear summation of multiple movements. In addition, there is no evidence of “compression” in the neurons studied by Tanji et al. A better example might be the use of the basis functions in Alex Pouget’s work to describe the response properties of the neurons in the primate posterior parietal cortex (e.g., LIP).

2. Page 16, line 552. The phrase “sparser code” does not appear to be relevant in this context. Compressed codes are not necessarily sparse.

3. Although the self-position is meaningless in the motor control task, it would be better to include its equivalent term in the GLM, so that the analyses for the main and control tasks have the same degrees of freedom.

Referee #3

(Remarks to the Author)

The authors have done an excellent job of addressing all of our original comments and substantially increased the depth of this fascinating work. In fact, the authors added three new experiments including a new behavioral and a new neuroimaging study to provide compelling evidence to fill various gaps pointed out by the reviewers through empirical data themselves. The new insights and clarifications coming from these added data and numerous data analyses have, in my opinion, really advanced the core concept behind the combinatorial code of social interaction based on the basis functions in the human brain. I also appreciated the time and care that the authors have put into making the response document that is so well-organized and comprehensive. Initially, the major concern was whether their basis function encoding was necessary - their new analyses and experiments clearly showed that basis function encoding is better than other “simpler” kinds of encodings. I am very enthusiastic about this work, and my enthusiasm only increased after seeing all the added analyses and clarifications provided by the authors in this revision.

Signed below to opt in for Nature’s transparent peer review scheme:

Reviewed by Steve W. C. Chang with assistance from Weikang Shi, a postdoctoral fellow in his lab.

Referee #4

(Remarks to the Author)

Basis functions for complex social decisions in dorsomedial frontal cortex – re-review

This manuscript investigates the neural representation of social information during decision-making, proposing a model based on basis functions that compress and sequence social information. I remain convinced that the general problem of understanding how the brain encodes social situations is extremely exciting and important. However, I remain unconvinced of the robustness and generalizability of the particular findings reported in this paper. My concerns extend to both experimental and conceptual aspects of the revised manuscript. Despite the authors’ efforts to address the initial reviewers’ concerns, I believe this resubmission falls short of convincingly demonstrating the proposed basis function model for social decision-making. Most concerning for me are the tiny effect sizes (point #1 below) and the lack of clear applicability of the proposed sequence-specific basis functions to real world decision making (see point #2 below).

Major issues:

1. Small effect sizes of fMRI and behavioral results: While the authors acknowledge the small effect sizes, they argue that the effects are replicable and consistent with previous findings. They provide a new behavioral study and additional analyses to support their claims. However, the new behavioral study *still shows very small effect sizes*. I believe the key result remains Figure 3C: this is the most direct assay of whether the brain codes these basis functions, and the effect size remains very small.

2. Real world applicability: In response to the criticism that their basis scheme is artificial and unrelated to real world decision making, the authors argue that understanding social situations in terms of teams is advantageous. I fully agree with this. However, I have trouble understanding how, e.g., $b_1 = [-1 \ 1 \ -1 \ 1]$, defined as a fixed sequence of weightings, represents team structure—*on some trials, this particular basis function correlates with team structure and on other it does not*. As the authors themselves point out, what is relevant is the “primary” basis function—the one that aligns with team structure on any trial. But the claim that the brain encodes social situations in terms of the primary basis function, i.e., team A performance – team B performance, is a much weaker claim—it simply says that the brain should represent a decision variable directly

related to the current decision that needs to be made ("which team won?"). As I pointed out in previous review, the brain *must represent this* in order for the subject to do the task, and it could readily do so by *simply linear combining individual agent performance*. I understand the essential novel claim of the paper to be that the brain represents sequential basis functions b1, b2, b3. After reading the revised paper, this framework still feels contrived to me. I remain unconvinced that these functions have general relevance to social decision-making, and the small effect sizes (see point #1) make me doubt whether the brain actually uses this scheme.

3. Compression: While the authors claim that basis functions offer a "compressed" representation, the demonstration of compression remains weak. As acknowledged by the authors, the 4D to 3D transformation presented as compression offers minimal efficiency gain. Indeed, the authors clarify in their rebuttal that they believe these basis functions are represented in addition to agent-specific information (page 150, "We take care to avoid claiming that there is no representation of individual agents. Indeed, there is abundant evidence for representations of individuals. Our claim is that there is an additional representation that relates to the patterns of interaction that can occur between individuals"). If so, it is not clear to me how these basis functions accomplish any kind of meaningful compression.

4. New online experiment: The new online experiment (Study 2) attempts to provide evidence for compression by showing improved performance in the "Group condition" compared to the "No-Group condition". However, the interpretation that this is due to compression is debatable and only tested indirectly. The "No-Group" condition is simpler with fewer decision states and more training on relevant trial types, suggesting potential alternative explanations for the observed performance difference: in particular, the improved performance in the "Group condition" could be due to other factors, such as increased attention or motivation, rather than the use of a compressed code. Furthermore, the effect sizes presented in Fig 4D-I are very small. While the presented comparison (Supplementary Fig. 13) using cross-validation is useful, it doesn't alleviate these concerns, but instead exacerbates them: *the improvement in explanation by the sequential basis function model compared to agent model is on order of a fraction of a percent*.

5. We thank the reviewer for clarifying that their variables b1-3 are not linear combinations of S, P, O1, and O2 because they are tied to sequential positions, and appearance of different agents at different sequential positions was balanced. But how can the authors be certain that their subjects didn't simply zone out on some trials (given that behavioral performance was far from perfect), which would disrupt their careful balancing, and lead to the small effects they attribute to sequential basis function encoding?

6. Related to point #5: Even if the subject is paying careful attention on all trials, it seems that given the finite # of trials, the balancing cannot be perfect, and the tiny effects observed could arise from agent tuning plus imperfections in balancing. The authors should simulate brain activity driven solely by agent identity and repeat their time course analysis on the simulated neural data (Fig. 3C). Are the simulated effect sizes for the basis function projections really zero? Or instead, is it possible that they will be small (as reported in Fig. 3C) but still diverge significantly from zero at multiple time points?

7. Basis Function Sets: The authors focus on a specific set of basis functions without exploring alternative sets that might be equally or more efficient for representing the task structure. For instance, could a basis set that directly reflects team membership and player relevance for each decision type provide a better account of the data? Investigating alternative basis function sets would provide a stronger argument for the chosen set and enhance the understanding of the underlying representational mechanisms.

8. Social Specificity: The authors attempt to address the concerns about the social specificity of their findings with a non-social control experiment (Study 3). However, the results of this experiment are far from conclusive. If anything, Study 3 provides further evidence that the authors' findings are not socially specific, since they still find basis function representation (Figure 5). To facilitate comparison with Study 1, the authors should show the results of Figure 5 in the format in Figure 3C.

9. fMRI issues: The authors were requested to show single subject fMRI data. It is disappointing that they do not do so in the revision. Instead, they simply state that this is "not consistent with best practice." I strongly disagree. If one averages noisy or variable signals, then the common signals will sum and the variable signals will divide. So a study that only shows the group average is assuming, in effect, that there is no individual variability in brain activation patterns that is not merely due to noise. This is almost certainly a false assumption. Furthermore, there is information loss when mapping from individual brains to template space. The mapping algorithms are not perfect, and they perform worse in brain regions where there is relatively larger individual anatomical variability (e.g. *in the prefrontal cortex*). So any study that only shows the group average and not the individuals is effectively assuming that the mapping from individual brain space to the common template space is accurate. This is clearly a false assumption.

Ideally, authors should do the analysis in each individual subject and then report how many subjects show the reported pattern. The authors state, "We agree with R4 that [not showing individual subject data] means that there is a possibility that there may be additional patterns of neural activity that we do not describe because they are present in a small number of participants." Due to individual variability, it could be the case that *all* the subjects show a pattern of activity that is quite different and much more diffuse from the average pattern that is shown currently. We cannot know without seeing the single subject data. Indeed, the highly variable behavioral results (e.g., Fig. 1C, D) could be due such variability across subjects.

10. dmPFC: While the manuscript now primarily focuses on pgACC, the role of dmPFC remains unclear. The authors initially report agent-specific signals in dmPFC (Fig. 1E) and later claim that dmPFC activity reflects social information without basis function activity (Supplementary Fig. 5 & 11). This inconsistency is confusing and requires clarification. It

would be helpful to explicitly address the relationship and interactions between agent-specific representations in dmPFC and the basis function model in pgACC.

Minor issues:

1. Figure 3b: there is no color bar.
2. Is Figure 3B showing just activity in dmPFC or were all voxels across the whole brain included?
3. There is no color bar in Figure 3G
4. In Figure 3I, J: the model predicts a specific sign of deviation (Fig 3I), while the data seems to go in both directions (Fig 3J), which could be easily explained by noise.

Version 4:

Reviewer comments:

Referee #1

(Remarks to the Author)

I remain highly supportive of the paper and would like to see it published. I believe that it will become a highly influential, seminal paper in the field.

Referee #2

(Remarks to the Author)

The reviewers have addressed all the concerns raised by this reviewer previously.

Referee #3

(Remarks to the Author)

I did not have any additional comments from the last submitted revision. I am still enthusiastic about this work for proposing and providing compelling evidence for social-cognitive operations based on basis functions, akin to what has been proposed in the past for sensorimotor transformation for visually guided movements and multi-sensory integration. From my perspective, reading the authors' responses to the additional comments from Rev 2 and Rev 4, I believe the authors have done a great job.

Referee #4

(Remarks to the Author)

Rebuttal response - Basis functions for complex social decisions in dorsomedial frontal cortex

I thank the authors for their extensive responses to my previous comments. The authors have clearly put a considerable effort into addressing the concerns raised, providing additional analyses and clarifications and conducted additional experiments. While I still have some reservations about certain aspects of the study, particularly regarding the broader interpretation and significance of the findings, I acknowledge that the authors have made a good faith effort to address the critiques raised. After carefully considering their rebuttal, I'd like to offer the following response:

1. Effect sizes: I acknowledge the clarification regarding the effect sizes, particularly the Cohen's d values. While many of these do indeed fall into the medium to large range according to statistical metrics, I maintain reservations about effect sizes. It would be helpful for me to know how the *magnitude* of beta weight in Fig. 3b compares to the magnitude of beta weight for the response of this region to its most effective driving stimulus. I think that absolute signal magnitudes matter when considering effect size, not just statistical significance, which is what Cohen's d is measuring. My impression is that beta weight of < 0.05 is very weak; when further combined with the huge scatter across participants evident in Fig. S7, with many subjects showing correlations of opposite sign, and a spread across subjects $\sim 10x$ the size of the reported effect, I personally remain unconvinced that this is a real and robust phenomenon.

2. Compression and basis functions: The additional explanations and analyses regarding compression and the use of basis functions are informative. However, while I agree that the brain likely employs a flexible set of basis functions tailored to the specific task and context, I still have some concerns about the generalizability of this specific set of basis functions to real-world social decision-making scenarios.

3. Online experiment: I acknowledge the additional clarifications provided for this study. While the results do support the authors' model to some extent, the interpretation of improved performance in the "Group condition" as evidence for compression remains debatable. Furthermore, the relatively small changes in behavioral performance observed raise questions about the practical implications of this compression. As I previously mentioned, factors beyond compression, such as attention or motivation, could potentially contribute to the observed differences in performance between the "Group" and "No-Group" conditions, and they have not been conclusively ruled out.

4. Social specificity: The additional analyses comparing the social and motor studies are helpful. However, the presence of

basis function representation in the motor study still raises questions about the social specificity of the findings. While the authors demonstrated some differences between the social and motor domains, the overall similarity in processing might suggest a more domain-general mechanism than initially proposed.

5. fMRI: I appreciate the inclusion of single-participant data and apologize for overlooking this previously. However, I maintain that a more comprehensive presentation of individual variability would strengthen the authors' conclusions, particularly given the high variability observed in behavioral results and in effect sizes across participants.

6. dmPFC: Clarification regarding the different activations in dmPFC is helpful. However, the overall picture of dmPFC function in the proposed model remains somewhat unclear, particularly in how it integrates with the basis function model in pgACC.

Nonetheless, I believe the manuscript has reached a point where it can make a contribution to the field, even though some issues and uncertainties remain. Therefore, I at this point recommend accepting this manuscript for publication, allowing the broader scientific community to evaluate and build upon this work. I hope that future studies will continue to explore and refine our understanding of the neural basis of social decision-making, potentially addressing some of the remaining questions raised during this review process.

Dear reviewers,

Thank you for your comments on our manuscript "Basis functions for complex social decisions in dorsomedial frontal cortex". We are addressing your points in this rebuttal. Following the advice from the editor and to ensure a succinct rebuttal, we respond by aggregating the points made by the four reviewers and by answering them together, as many of the points and our answers are shared between the reviewers. However, we have also provided the editor with a detailed point-by-point response to every single reviewer question. Please refer to this supplementary document if needed. We have answered every single point carefully and in depth. This rebuttal is part of an appeal that we initiated. As far as we can see, our manuscript was met with enthusiasm and support by three of four reviewers. They all comment on the novelty but also on the clarity with which the ideas and data are presented.

To address the reviewers' concerns, we conducted three new experiments including a new behavioural experiment and a new neuroimaging study in humans (total N=1,905, up from N=56 in our initial submission). Our reanalyses and new results confirm, replicate, and substantially extend the main claim made in our original manuscript – a claim that all four reviewers described as "a new way to think about how the brain represents interactive social behavior" (R1), "of great significance" (R2), and "exciting and novel work that provides a new framework for social decision making" (R3). R4 was more negative but even R4 noted that "the central claim (...) was novel and exciting" (R4).

The major claim of our manuscript is that, when faced with social interaction problems, the brain does not solve them by solely relying on an agent-centric code – separate representations for all social partners involved in an interaction. Instead, the brain keeps track of the relationships between agents. Neural activity tracks social information in a compressed, and combinatorial format, projected along the basic axes of interactions that are possible in a social group in a given setting.

In our revised manuscript, we have addressed the methodological and conceptual points from the reviewers in great detail. We have replicated both the neural and the behavioural signatures of our basis function model in new experiments that we have conducted for the purpose of this revision. We have run non-social control experiments that clarify the social and domain-general component mechanisms of our basis function model on a behavioural but importantly also on a neural level. We show that the use of the basis functions enables a decision-maker to improve their performance in our task in a manner that cannot be explained by an agent-centric model. We have done that in several large-scale behavioural studies conducted online (N=1,817) and in a new, non-social fMRI control experiment (N=32). We have substantially revised our manuscript and added 2 new main figures (while moving 1 to the supplements) and 15 new supplementary figures.

To keep this rebuttal concise, we utilise the following presentation strategy. First, we address the three most frequent and/or severe concerns raised by the reviewers. We address them using new

behavioural and neural data that we have acquired, and this new data has led to the most fundamental changes in our manuscript. To this end, we group remarks from the reviewers together into broad categories. Initially we address the most frequent and substantial concerns, which are also the ones that have led to the most substantial changes of our manuscript. Subsequently we go to concerns that were specific to individual reviewers. This is the broad organisation of the sections:

- 1) **Ruling out simpler explanations of our data. Compression.** We rule out simpler alternatives to our basis function model. We conducted a new behavioural experiment online that demonstrates a striking effect that is predicted by our basis function model but cannot be explained by simpler agent-centric models (study 2, N=795, Fig.4, Fig.S16,S17). We show that the use of the basis functions enables a decision-maker to improve their performance in a manner that cannot be explained by an agent-centric model. We were able to improve people's memory by using a method that requires no specific instructions, capitalises on generalisation across decision types, and improves performance more than repeated training. Our findings suggest new ways to think about the benefits of group membership by demonstrating that grouping of information can in fact lead to a more precise (instead of less precise) representation of specific elements. This study uncovers a new principle of information grouping that cannot be explained by existing theories in cognitive neuroscience or memory research in general. Note that we will subsequently also address other aspects of simpler explanations that have been raised by reviewers individually (e.g. R3's point about sequenceness).
- 2) **Social specificity of our effects and of the basis function model.** We conducted a non-social control experiment to clarify the social and domain-general component mechanisms of our basis function model on a behavioural but importantly also on a neural level (study 3, N=32, Fig.5, Fig.S18,S19). This investigation not only provided clarity on the mechanisms involved but also underscored the versatility of our findings across various tasks and situations. In this control fMRI experiment, we found that (1) removing the social framing eliminates signatures of socially specific representations, but that (2) our proposed basis functions still guided choices and were measurable in the brain and behaviour. Our findings shed light on situations where decision-makers grapple with complex scenarios requiring the dynamic integration of multiple pieces of information. Such scenarios characterise complex social situations. However, our findings also highlight the need for studies in cognitive neuroscience to consider similar scenarios where more than three pieces of information must be tracked and dynamically integrated – so far, a dramatically understudied phenomenon in our field.
- 3) **Replication of key effects and concerns about statistical analysis methods.** In this section, we specifically address concerns that our behavioural and neural effects are small, and in particular R4's major and very unusual concern about our analysis approach that *"The fMRI data involved averaging across a large number of subjects (N = 56) and no single subject data is shown, making it unclear whether the focal activations might largely be an artifact of intersubject averaging"*. We do take concerns about the strengths of our effects seriously and hence detail below the various ways in which our neural and behavioural key effects replicate across the new experiments. We replicate key behavioural results across two more experiments (%correct, reaction times, GLM weight patterns; Fig.1C,1D,3K,4H,4I,5D, Fig.S6,S15) and key neural results in a new fMRI study (neural basis function effects; Fig.5I,5J5L, Fig.S18,S19). We also add the information-theoretic measures supporting our model (Fig.S13). However, we firmly oppose R4's idea that our large number of participants obscure effects of interest. This comment goes against the ethos of the field in improving the quality of neuroimaging research and is clearly at odds with statistical best practice. It is not correct that brain activity patterns that are shown are simply the result of averaging. They are the result of parametric statistical analysis that takes into account both the mean and variance of effects. Significant effects are identified when participants consistently show the same pattern

of results. This approach is ubiquitous not just in whole brain neuroimaging but in many areas of science.

- 4) **Requests for additional neural analyses.** Based on the points raised by the reviewers, we have conducted additional neural analyses supporting and extending our results. We have thoroughly analysed basis function effects and social signals in regions of interests comprising ventromedial prefrontal cortex, amygdala, TPJ and dorsomedial prefrontal cortex (Fig.S5, Fig.S7, Fig.S11). We have clarified the relationships between the neural representations of basis functions in a sequential and agent-centric frame of reference and explored the nature of the self-position signal (Fig.S8, Fig.S10)
- 5) **Requests for additional behavioural analyses.** Based on the points raised by the reviewers, we have also conducted additional behavioural analyses supporting and extending our results. We have analysed and replicated sequential effects of basis functions (Fig.S6, Fig.S15) and supplemented exhaustive analyses of decision making behaviour in our task (Fig.S2, Fig.S3, Fig.S4, Fig.S14).
- 6) **Conceptual clarifications and questions about framing.** We have addressed various conceptual concerns and adjusted main text, figures and supplements.
- 7) **Minor clarification question and typos.** Finally, we are grateful for your detailed feedback and have clarified minor points and corrected typos.

In the following, we will address the reviewers' comments grouped in this way and in this order. We also make clear when we address points that have been raised by several reviewers. We refer to the points of the reviewers abbreviated as, for example, "R3|2" meaning "reviewer 3 point 2", or R3|2,3 meaning reviewer 3, point 2 and 3 etc. Please refer to the following outline for an overview. Please also refer to the additional supplementary document outlining a point-by-point response to all reviewer's individually if you feel an important point has not been recognised sufficiently in this document. We have answered all points in depth. We thank you for reviewing our manuscript.

Outline

1) Ruling out simpler explanations of our data. Compression (R1,R2,R3,R4) 5

1.1) *New behavioural experiment demonstrating compression of social information (R1|1, R2|1, R3|4, R4|1) 5*

1.2) *Implications of the new behavioural results for related reviewer points (R1|3 and R2|4) 11*

2) Social specificity of our effects and of the basis function model (R1|2, R2|2, R3|6, and R4|O1,O2) 13

3) Replication of key effects and concerns about analysis methods (R1,R2,R3,R4) 19

3.1) *Replication of behavioural effects: Self-bias and basis function effects (R1|4.1, R1|1, R4|O5, R4|O9, R4|Intro) 20*

3.2) *Replication of neural effects: Social signals and basis function signals 24*

4) Requests for additional neural analyses (R1,R2,R3,R4) 25

4.1) *Region of interest analyses of dmPFC, vmPFC and amygdala in relation to basis function effects and social processing (R1|5.2, R1|5.4, R3|8, R4|6) 25*

4.2) *Sequential vs agent-centric basis function signals (R2|2, R3|1) 28*

4.3) *Additional analyses of the self-position signal (R3|7) 32*

5) Requests for additional behavioural analyses (R2,R3) 34

5.1) Behavioural effects of basis function “flipping” (R2|3, R3|1) 34

5.2) Additional analyses of behavioural decision making performance (R3|3, R3|4) 36

5.3) Behavioural results cannot be explained by pre-experiment performance differences (R3|5) 41

6) Conceptual clarifications and questions about framing (R1,R3,R4) 44

6.1) Reviewer 1 (R1|1) 44

6.2) Reviewer 3 (R3|2, R3|9) 45

6.3) Reviewer 4 (R4|1, R4|2, R4|O1, R4|O4, R4|O8, R4|O9) 46

7) Minor clarification questions and typos (R1,R2,R3) 54

7.1) Reviewer 1 (R1|4, R1|5.1, R1|5.3) 54

7.2) Reviewer 2 (R2|Minor) 59

7.3) Reviewer 3 (R3|Minor) 59

1) Ruling out simpler explanations of our data. Compression (R1,R2,R3,R4)

1.1) New behavioural experiment demonstrating compression of social information (R1|1, R2|1, R3|4, R4|1)

Several of you raised the possibility that our data could be explained by simpler models' than our basis function model. We appreciate that this is an important point and many of you put it at the start of their review. In particular, many of you were asking about our claim that basis function offer a "compressed" representation. R1 notes in their first point (R1|1) that **"while the authors provide a number of elegant analyses which suggest that basis functions could explain the data, it wasn't clear to me whether or to what degree are their models have higher explanatory power than other simpler functions. It would be helpful for example to add formal information theoretic metrics such as AIC or BIC."** R2 notes in their first main point (R2|1) **"The authors should better clarify the relationship between the transformation based on basis functions and data compression. They seem to equate these two types of computations, but one can change the coordinate system without losing any information or data compression"**. Reviewer 3 points out the relevance of the group decisions in our experiment, as we argue that those are one source of data compression in our task (R3|3) **"How about the Group condition? It is actually important to know how the Group condition's accuracy and RT looked like, because based on the later basis function hypothesis, the authors should expect higher accuracy and shorter RT for the Group condition. This is because only primary basis function is needed for the Group condition."** Also, reviewer asks about the relevance and evidence for compression in our task in their first point (R4|1) **"I'm not entirely sure I understand what they mean by compression here. To encode the outcomes of four agents, one would need a 4d space. For their task, they need a 3d space. Is the 4d->3d transformation what they mean by compression? If so, in pure efficiency terms, it doesn't seem like much is gained by such compression."**

In response to these points, we have conducted many more control analyses (see below), but also, importantly, a new behavioural study conducted online (N=795). In this new behavioural study ("study 2") we show in a particularly striking and direct way that our basis function model has higher explanatory power than an agent-centric model (see R1|1). This online study provides strong evidence that participants use data compression, as suggested by our basis function model, to solve our task. By data compression, we mean that reducing the pieces of information necessary to solve the task. We followed R2|1's suggestion to clarify the relationship between the transformation based on basis functions and data compression. We agree that these two types of computations are not equivalent, and we have revised our manuscript to make this point clearer. Specifically, we have added a section in the paper that explains the differences between data compression and data transformation and how they relate to our findings, see below. Our new behavioural study clearly demonstrates that using a compressed representation is an efficient way to solve our experiment. Our results are incompatible with the view that participants merely use 4 independent, agent-centric, representations (one for each player, an agent-centric code). Instead, we show that people are better able to solve the task if they are incentivised to represent the task in a compressed code along the basis functions.

We performed the additional study in the following way. We devised an online version of our task and tested 795 participants on the online platform Prolific. The task was identical to the fMRI experiment with very minor changes related mostly to trial numbers and timings (see new methods section on the behavioural study below). The study was designed to test whether our task could be better solved by (a) holding 4 agent-centric representations or (b) by forming compressed representations.

The rationale was the following: the basis functions correspond to the three possible combinations in which the two groups can be slotted into the 4-player sequence that people observe. The primary basis function corresponds to the difference in performance between both teams and can be immediately used to guide group decisions. The presence of the group condition is hence likely to incentivise people to break up the 4-player sequence into the basis functions. Therefore, we reasoned that by removing the group condition from the experiment, people would be less likely to use the basis functions to solve the task. Note that this follows up on R3|3's request to show if and how group decisions are crucial for the cognitive strategy that participants employ to solve the task. If the basis functions indeed are an efficient code for solving the task, then we can make a simple prediction that would be very counterintuitive from any perspective other than that offered by the basis function approach; we predicted that decision performance in the dyadic trials would decline as a consequence of removing group trials, and that this would be caused by a worse memory for relevant performances. In other words, if the task is apparently (from a superficial, intuitive perspective) made easier by reducing the number of decision types (only dyadic decisions rather than a mixture of dyadic and group decisions) then the basis function account predicts that performance will be worse because participants are discouraged from assembling the full, simple set of basis functions. In other words, we hypothesized that removing the group condition makes the task more difficult in dyadic trials:

(illustration and reminder of the experimental task. Our claim is that if we remove the group decisions in red, performance decreases in the blue and yellow dyadic conditions)

We confirmed this hypothesis in this online experiment as follows. The online experiment comprised 795 participants divided up into two groups of 396 and 399 participants. We compared the decision-making behaviour of the two groups of participants in two closely matched schedules with the only difference being that one schedule comprised group trials (“Group condition” schedule) and the other one did not (“No-Group condition” schedule). Both schedules comprised the same set of observed performance scores for all players. Hence, the information to be memorized was identical. Both conditions comprised self and partner trials that were placed at exact identical positions within the schedule. For example, if on trial 39, participants observed $S=5$, $P=2$, $O1=3$, $O2=6$, followed by a comparison between S and $O1$, then all this was exactly identical in the other schedule as well. The only difference was that 1/3 of trials in the “Group condition” were group trials (72 decisions). The remaining 144 decisions were, in equal numbers, self and partner trials placed in identical positions within the schedule. We refer to these 144 dyadic decisions as “matched decisions” and these were the only (!) decisions that we compared across the two conditions in our analyses. In the “No-Group” condition, the group trials were in equal numbers replaced with partner trials and self trials resulting in 108 partner trials and 108 self trials in the “No-Group” condition. We analysed decision making only on the comparable “matched decisions” ignoring the group trials and the “filler” self and partner decisions in our analyses. Our prediction was that decision accuracy on the matched trials would be better in the “Group condition” compared to the “No-Group condition”. This is because the presence of group decisions is likely to incentivise people to break up the 4-player sequence pairwise into the basis functions.

We want to highlight that this is a highly counterintuitive prediction. By all standard measures of task complexity that we usually consider in our field, the “No-Group condition” is easier to solve: there are fewer decision states to prepare for (only self and partner decisions, no group decisions) and participants have overall more training in the relevant decisions (216 dyadic decisions in the “No-Group” condition compared to 144 dyadic decisions in the Group condition). Any model that suggests that the task can be best solved by four independent agent-centric representations (one for each player) would clearly predict either no difference between conditions, or that the “No-Group” participants perform better. Note, for instance, that all four players have a decision relevance of 25% in both conditions, which we now illustrate in the main text:

Nevertheless, we find that participants perform better in the “Group condition”. This provides clear-cut evidence that participants using a compressed code solve the task better, despite less training with relevant trial types and an apparently more complex state space. We analysed the data as follows.

First, as pointed out, we only analysed matched decisions at identical positions within the “Group” and “No-Group” condition. We first looked at the broadest and most general measure of performance, the percentage of correctly made decisions (“% correct”). We expected effects to be quite small, given that they would be purely driven by the context within which these matched decisions would be encountered. We also considered that the contextual effects of the “Group condition” might vary depending on whether participants responded to the first or second decisions after the observation phase. As a reminder, each trial comprised two different decisions after the observation phase. This is important because it required participants to represent the performance scores in a flexible format for a longer time. Our fMRI analyses suggested the use of basis functions particularly between the observation phase and decision 1, when the need for a flexible, compressed representation was greatest. Hence, we ran a three-way ANOVA with the factors Decision number [2 levels: first vs second] x player [2 levels: self trial vs partner trial] x Group [2 levels: Group vs No-Group] and the percentage of correctly made decisions as the dependent variable. While we found no significant main effect of group ($F_{1,793} = p=0.11$), we found a significant group x decision number

interaction ($F_{1,793} = p=0.003$). Following up on this interaction revealed the participants in the “Group” condition performed significantly more correctly on both self ($t_{793} = p=0.038$), and partner trials ($t_{793} = p=0.024$), if these were the first decisions encountered – precisely at the time where we identify the basis functions in fMRI.

Comparison of percent correct decisions in matched decisions for decision nr 1:

We subsequently analysed where the improvement in decision performance came from, why the participants performed better on matched dyadic decisions if they encountered them within the context of group decisions in the “Group” condition. The large amount of data collected allowed us to simply plot the social decision variable for self and partner trials – the difference in scores between the S/P and the relevant opponent. Participants should choose self if they had scored higher than Or. Hence, we expected a sigmoid modulation of choice along the decision variables. This was indeed what the data revealed, but with a bigger modulation in the “Group condition” in self trials. This was apparent as a significant interaction in a two-way ANOVA with the factors Performance difference [9 levels: -4,-3,-2,-1,0,1,2,3,4] x Group [2 levels: Group vs No-Group]: $F_{8,6344} = 9.339$, $p < 0.001$. We replicated this result in partner trials: $F_{8,6344} = 10.036$, $p < 0.001$.

We included this result in the main manuscript (note that error bars are included in the below figure; they are just smaller than the data points indicating the mean):

This meant that participants were better at remembering the correct difference in scores between the relevant players in the “Group condition”. We further tested this effect of significance using a regression approach (see below)

This result shows that participants had a *clearer* representation of individual players’ scores in the “Group condition”. So rather than blurring or spreading of scores within a group, people became

more accurate in correctly deciding according to the to-be-remembered performance scores. This result cannot be explained by assuming that people solve this task optimally by holding four independent representations of the agents. Such a model cannot explain the improved decision accuracy in the "Group condition". However, this result follows directly from our theory that representations in the "Group condition" are formed using an efficient and compressed code. This result is evidence that if incentivised to use such compressed representations (by including group decisions), people indeed remember relevant information better.

Next, we fitted the regression model we had used in the fMRI to explain participants' decisions as a function of all players' performances. This model allowed us to examine the effects of the relevant players' scores as well as the effects of the decision-irrelevant players – the effects that we maintain are a signature of compressed coding (via the basis functions). Following our argument from above, we expect self-other-mergence effects (effects of irrelevant players' scores on decisions according to their group membership, the ones that we quantified using the new information theoretic metrics explained above) if people use a compressed representation that integrates over players along the basis functions. Hence, we expect mergence effects to be higher in the "Group condition" compared to the "No-Group" condition, although both conditions similarly require participants to hold performances from all players in memory and to compare them in terms of membership of their own vs the opponents' team. In the fMRI study, we observed mergence effects in self decisions as positive effects of the Partner, P, and simultaneous negative effects of the irrelevant opponent, Oi. We aggregated the effects of the irrelevant players in self trials by calculating the difference between P and Oi beta weights. We expected an increase in the P-Oi effect in the "Group" condition compared to the "No-Group" condition, reflecting increased mergence. This is precisely what we observed in our data: $t_{793} = 4.471$, $p < 0.001$. At the same time, the effect of irrelevant players in partner decisions was similarly increased in the Group condition: effect of S-Oi, $t_{793} = 4.317$, $p < 0.001$. These effects occurred in parallel to an increase in decision-relevant effects in the "Group" condition compared to the "No-Group" condition: S-Or, the social decision variable for self decisions, increased in effect size ($t_{793} = 3.058$, $p = 0.002$) and so did the effect of P-Or in partner decisions ($t_{793} = 3.021$, $p = 0.003$). The latter two effects confirm the results demonstrated in the sigmoid plots above.

(left panel: self decisions, right panel: partner decisions)

Note that in the Group condition of the behavioural study, we could replicate all four effects of the irrelevant players that we had identified in the fMRI study:

- P in S trials: $t_{395} = 5.143$, $p < 0.001$
- Oi in S trials: $t_{395} = -5.313$, $p < 0.001$
- S in P trials: $t_{395} = 6.440$, $p < 0.001$
- Oi in P trials: $t_{395} = -6.450$, $p < 0.001$

Furthermore, we show that the improved decision performance is linked to increased memory performance, and not linked to an increase in the use of non-memory related, but task-relevant information. Every decision involved a “bonus” – a small score increment that is either added to the score of one’s own team or the other team. This is a feature of the original fMRI experiment as well as the new behavioural study. People should increasingly choose one’s own team if the bonus is positive, but they should do so less if the bonus is negative. This is indeed what we found in both conditions. However, participants from the “No-Group” condition rely more on the bonus than participants from the “Group” condition. Again, this result was obtained for both the self trials ($t_{793} = 4.239$, $p < 0.001$) and the partner trials ($t_{793} = 4.658$, $p < 0.001$). This suggests that the “Group” condition does not uniformly improve performance in all aspects of the task, but instead, specifically, performance that is linked to a better memory of agent scores. Moreover, it suggests that participants in the “No-Group” condition might rely on the bonus to compensate for poorer memory of agent scores.

We hope that this additional evidence convinces the reviewers that people in our fMRI experiment use a compressed representation that is organised along the basis functions. The basis functions coincide with all possible group configurations. We show that, if people expect group decisions in their schedule, then they show improved accuracy on dyadic decision trials. This is a highly counterintuitive prediction that cannot be explained by an agent-centric model with four elements. This new behavioural study therefore demonstrates the explanatory power of our basis function model: It can account for new and counterintuitive findings that the standard agent-centric model cannot account for. We find this effect although the “No-Group condition” appears to have lower task demands because of fewer decision states and increased training with decisions of the same decision type. Nevertheless, people perform better if group decisions are included in the schedule. This suggests a mechanism that relies on a compressed representation. Our results also suggest that the merge effects are signatures of an efficient, compressed code, rather than suboptimal features of blurring or spreading of scores within groups. Although group decisions require participants to combine information across players, they paradoxically improve accuracy for individualised, dyadic decisions. This can be explained by our basis function model assuming data compression, but not by conventional 4-agent models.

Moreover, we want to highlight the importance of the findings of this behavioural study beyond the scope of the reviewers’ questions. We present a study where we were able to improve people’s memories by using a method that requires no specific instructions, capitalises on generalisation across decision types, and improves performance more than repeated training of a specific decision type. It furthermore suggests new ways to think about the benefits of group membership for individual cognition by demonstrating that grouping of information can in fact lead to a more precise (instead of less precise) representation of specific elements within the remembered groupings.

We have integrated the new behavioural study fully in our revised manuscript including profound changes in main text, methods, and supplements:

Main text: Page 14 (line 448) to page 16 (line 552); note the section on data compression vs transformation in the first paragraph, incorporating R2|1’s remark

Methods: Page 34 (line 1351) to page 36 (line 1480)

Figures: Figure 4

Supplements: Supplementary Figs. 16,17

1.2) Implications of the new behavioural results for related reviewer points (R1|3 and R2|4)

The new experiment suggests that people compress social information in line with our basis function model. As highlighted above, the existence of group decisions is crucial for this compression. We argue that basis functions enable people to compress data and to decompose the decision-making process serially in two steps. First people represent the primary basis function. The primary basis function is sufficient to solve group decisions (decisions that compare the joint performance of self and partner, S+P, with the joint performance of the Opponents, O1+O2). For dyadic decisions – “self trials” in which one’s own performance is compared with a relevant opponent and “partner trials” in which the partner’s performance is compared with a relevant opponent – two basis functions are needed; the same primary one that is relevant for the group decision, plus a secondary one. Regarding this, reviewer 2 raises in their point 4 (R2|4) “It is interesting that the primary basis function(group contrast) had earlier and stronger effects on behavior than the secondary basis function(needed to be added to primary basis function to compute the decision variables in individual comparison). Is it possible that this is equivalently and alternatively explained as the spreading of individual scores within the same group?”. Similarly, reviewer 1 asks about this aspect of our model (R1|3) “Third, the authors suggest that secondary basis projections exerted their influence on choice significantly later than primary basis functions. However, it would be helpful to provide further formal social metrics such differences in reputation or strategy for comparison. In a similar direction, on line 84, do the authors mean that mean that people tended to judge higher their performance as opposed to their opponent’s?”

It is a fair description to say that that participants appear to spread the scores within the same group (R2|4). Our critical point, however, is that these mergence effects do *not* just reflect a spreading of individual scores within the group. Our point is that they are behavioural consequences of representing information efficiently along the basis functions. According to our model, the secondary basis function exerts their influence earlier than the secondary one. We contend that this is a key characteristic of decisions when participants compress the information along the axes of the basis functions in our task. We note that in the revised manuscript we replicate this effect several times and we are able to show it both using a variant of a drift diffusion model and in a “model-free” way by using a general linear model. As we explain more clearly in the main text now, the effects of the irrelevant players are a direct consequence from this sequential integration of information during decision making. Moreover, we show that just as we can predict when and how these effects will appear we can also predict occasions when they will be absent. For example, in the no-group condition (see figures above).

(I) Computational modelling simulations examine the impact of primary and secondary basis functions on decisions. Simulated decisions are analyzed with a GLM predicting engage/avoid decisions in Self trials. Optimally, with equal weighting of primary and secondary basis function (left), choices should be guided by the performance scores of oneself (S) and the relevant opponent (Or), but not the partner (P) or the irrelevant opponent (Oi). The positive S-weight in in this subplot indicates that with higher S-performance, participants are increasingly likely to make “engage” choices and indicate that one’s own performance was better than Or’s. The negative Or effect similarly means that “engage” choices are likely after bad Or performance. An overweighting of primary basis functions compared to secondary basis functions predicts a characteristic deviation from this optimality (right): Despite being irrelevant for choice, P is expected to have a slight positive effect on choice, and

O_i is expected to have a small negative effect on choice. **(J,K)** As predicted, we found small but highly significant effects of irrelevant players in Self trials and also in Partner trials in precisely the predicted directions (note that in Partner trials, S and O_i are the irrelevant players).

Our new behavioural study clearly demonstrates that using a compressed representation is an efficient way to solve our experiment. Our results are incompatible with the view that participants merely use 4 independent, agent-centric, representations (one for each player, an agent-centric code). Instead, we show that people are better able to solve the task if they are incentivised to represent the task in a compressed code along the basis functions. Furthermore, our results clearly indicate that the above merge effects increase as a consequence of this efficient and compressed representation, rather than a mere spread of information (R2|4). In our new behavioural study, we identify one key prerequisite that this effect depends upon: the existence of group decisions, which incentivise participants to compress information along the basis functions.

To summarize the relevant part of the revised manuscript that addresses this point: We fitted the regression model we had used in the fMRI to explain participants' decisions as a function of all players' performances. This model allowed us to examine the effects of the relevant players' scores as well as the effects of the decision-irrelevant players – the effects that we maintain are a signature of compressed coding (or, a formal metric of this effect as requested by R1|3). Following our argument outlined above, we expect self-other-merge effects (effects of irrelevant players on choice according to their group membership) if people use a compressed representation that integrates over players along the basis functions. Hence, we expect merge effects to be higher in the “Group condition” compared to the “No-Group condition”, although both conditions similarly require participants to hold performances from all players in memory and to compare them in terms of membership of their own vs the opponents' team. In the fMRI study, we observed merge effects in self decisions as positive effects of the Partner, P, and simultaneous negative effects of the irrelevant opponent, O_i. We aggregated the effects of the irrelevant players in self trials by calculating the difference between P and O_i beta weights. We expected an increase in the P-O_i effect in the Group condition compared to the No-Group condition, reflecting increased merge. This is precisely what we observed in our data: $t_{793} = 4.471$, $p < 0.001$. At the same time, the effect of irrelevant players in partner decisions was similarly increased in the Group condition: effect of S-O_i, : $t_{793} = 4.317$, $p < 0.001$. These effects occurred in parallel to an increase in decision-relevant effects: S-Or, the social decision variable for self decisions, increased in effect size ($t_{793} = 3.058$, $p = 0.002$) and so did the effect of P-Or in partner decisions ($t_{793} = 3.021$, $p = 0.003$). The latter two effects confirm the results demonstrated in the sigmoid plots above.

Note that in the Group condition of the online study, we could replicate all four effects of the irrelevant players that we had found in the fMRI study:

- P in S trials: $t_{395} = 5.143$, $p < 0.001$
- Oi in S trials: $t_{395} = -5.313$, $p < 0.001$
- S in P trials: $t_{395} = 6.440$, $p < 0.001$
- Oi in P trials: $t_{395} = -6.450$, $p < 0.001$

By contrast, only the effect of S in P trials was significant in the No-Group condition: $t_{398} = 3.813$, $p < 0.001$ (all other $p > 0.21$). Note the exceptional effect of the self-scores in partner trials in the “No-Group condition” is better understood as an instance of a previously documented “egocentricity bias” – the fact that we find it hard to avoid our own perspective and beliefs about ourselves in our assessment of the world. By contrast, the effects of the various irrelevant players in the “Group condition” manifested to a similar degree regardless of whether the irrelevant player was the self, the partner, or one or other of the players on the opposing team. This suggests that the effects indicate the usage of a compressed group representation that is more efficient in guiding decisions than individualised agent-centric codes.

Furthermore, regarding the last sentence in the R1|3’s point, we clarify that our results do not show that people rated their own performances as better than their partners but rather they tended to rate their own performances more accurately than the partner’s performances and they rate their own performances more quickly than the partner’s performances. We have revised the manuscript to make this as clear as possible (page 4, paragraph 2 of manuscript).

2) Social specificity of our effects and of the basis function model (R1|2, R2|2, R3|6, and R4|O1,O2)

We conducted a non-social control experiment to clarify the social and domain-general component mechanisms of our basis function model on a behavioural but importantly also on a neural level. Several of you have raised this point. Reviewer 1 says this most directly **“Second, while the authors suggest that they found signatures of agent-centric representations, it would be helpful to provide a non-social control to demonstrate the social specificity of their findings. While the ‘self’ and ‘other’ comparisons suggest a distinction, it wasn’t clear whether similar findings could have been observed when playing against any other non-social agent. It also wasn’t clear whether or not similar findings could have been observed if performing the above analyses on other non-social serial events/decision making sequences.”** (R1|2). Similarly, reviewer 2 asks about the relationships between our basis functions and social, agent-centric coding (note that we will respond to the “serial order” aspect in a separate point): **“It is not entirely clear and therefore perhaps needs to be explained better why the authors consider only the basis functions indexed by the serial order, but not those indexed by the identities of the agents.”** (R2|2). Reviewer 3 also raises the point about the social specificity of the basis functions, by asking whether they explain our self-referential effects in behaviour and neural signals: **“could the authors explain more how their sequence-based code can explain the better performance in the Self condition shown in Figure 1C, 1D?”** (R3|6). And finally reviewer 4 says **“I don’t understand the social component in the setup of the task at all.”** (R4|O1) and **“In connection to the point above, I am not convinced that the experiments engaged social neural circuits at all, which is central to the paper. The task seems to involve counting and memorization rather than anything resembling social interaction. If the authors would like to claim the latter, then significantly more evidence should be presented supporting their claim. Figure 1E seemingly represents a small step towards that but is not convincing or sufficient.”** R4|O2).

We agree that it is important to understand the social specificity of our observed effects. Our proposed basis functions reflect a mechanism to efficiently organise complex information and to

prepare adaptive courses of actions. We argue that this is a necessary computational prerequisite in complex multi-person social interactions. Representing social information along the basis functions allows a decision-maker to simplify decision problems and solve them sequentially. However, it is possible that such a representational format is not unique to social situations. Instead, it might guide decisions in a variety of domains where multiple pieces of information need to be tracked and strong priors exist about which combinations of these pieces are possible and which not. Therefore, we conducted a non-social fMRI control experiment (N=32). We predicted that (1) removing the social framing would eliminate signatures of socially specific, socially anchored representations, but that (2) an underlying combinatorial code would still guide choices and be measurable in behaviour in the same way, and that (3) neural signatures of basis functions would persist in medial prefrontal cortex, albeit with slightly different anatomical centres of gravity. We tested and confirmed these predictions in the following way.

We have conducted the additional fMRI experiment with a further 32 participants who performed a formally very similar task, but which lacked any social element. In the new task, which we refer to as the motor task, participants made judgements about finger movements that could be made with two fingers on each hand. The two fingers on one hand correspond to the two players on the participants own team in the original social task. The two fingers on the other hand correspond to the two players on the opposing team in the original social task. The number of times each finger was moved corresponded to the points that the four players accumulated in the social task. We used a numerically identical sequence of scores/movements in the original social task and the new motor task (note that the participants did not make sequences of motor movements in the fMRI scanner in order to not add motor confounds; instead cues that participants observe signified motor movements according to an elaborate cover story which we explain in the methods). The task is summarized as shown below:

Figure 5. Control fMRI experiment revealed behavioural and neural signatures of basis functions (study 3).
 (A) In our social fMRI experiment, participants tracked performance scores for four players and compared them.
 (B) [FIGURE REDACTED]

More details about the new task are provided below:

[FIGURE REDACTED]

[REDACTED]

In summary, there were three important findings obtained with this new task. First, we found that there were, indeed, no social agent-centric representations in the motor task. Behavioural signatures of social processing were absent. While participants had performed more accurately and quicker in Self decisions compared to partner decisions, this was no longer the case for the corresponding trial types in the motor task (Note that we have also added Bayesian statistics citing evidence in favour of the null hypothesis whenever we make a claim to argue for the null). This is not just the case for the behavioural expressions of social processing, but also for its neural signatures. On the left the figure below shows the social agent specific representations in the original social study. When the same areas were interrogated in the motor task, there were no corresponding effects in the motor task (panels F and G shown to the right below):

These figures are panels taken from a new figure added to the revised manuscript. However, in the supplementary information in supplementary Fig.19, we show that this absence of effect is not simply a consequence of the precise region of interest examined or the precise threshold set for testing if effects are significant. We show that even when we set the threshold at a very low level and make no correction for multiple comparisons we still see no effect:

Supplementary Fig.19. Sub-threshold activation maps for social and basis function signals in the control fMRI experiment (study 3). (A) Signals relating to self and partner in the social fMRI experiment (see main text). (B,C) The corresponding activation maps for Motor-P and Motor-S thresholded at $z > 2.0$ for illustration only. Note the small white spheres, which are ROI masks of dmPFC and pgACC used to assess significance in an ROI for the motor study. Note that both ROIs are non-biased and independent (i.e. they were derived from the social study and applied to the motor study. Even when examining activity at this very low threshold, nothing corresponding to the partner-related activity in dmPFC and the self-related activity in pgACC could be seen in the control fMRI experiment. (D) Whole brain effects of the three basis function projections and S-position in the social experiment. (E) The corresponding activation map for the mean of the basis function projections (not including S-position, because that variable had lost its meaning in the context of the motor study) in the control fMRI experiment, thresholded at $z > 2.0$ for illustration only. Note the small white pgACC sphere used to assess significance in a ROI for the motor study. Again, ROI selection was non-biased and independent (i.e. ROIs were derived from the social fMRI experiment and applied to the control fMRI experiment. The pattern of activity seen here is consistent with those shown in Figures 5I,J,K,L and confirm that analogous basis functions are coded in both tasks ($N = 32$).

Second, we found that when participants made judgements about finger movements, their decision making patterns suggested that they used sets of basis functions that bore similarities with those used in our social task. This was, for example, apparent in the way in which aspects of the decision weights resembled those associated with the social task; participants' judgements about one finger's motor performances reflected the impact of a certain weight given to the performance of the other finger on the same hand (note that these weights were precisely the ones that were abolished in the behavioural study that did NOT contain group decisions). When this finger's performances were compared with one of the fingers of the other hand, then again, the irrelevant finger exerted an influence on the decisions that were made. This was true both when participants in the new experiment made judgements about the finger that formally corresponded with "self" in the social task and when they made judgements about the finger that formally corresponded with "partner" in the original social task.

However, other aspects of behaviour in the new motor task were profoundly different to that recorded in the original social task. For example, the bias for making decisions more accurately and quickly when they involved the self was not found for the formally identical decisions that were matched for their numerical features in the motor task:

The third key finding is that at the neural level we were able to identify neural correlates of the basis functions found in the motor task but they were not anchored to a representation of the self as had been the case in the social task. We found that the basis functions themselves in the social task and the motor task overlapped but that the social effects were especially prominent in pgACC extending into the adjacent medial frontal pole while the motor basis functions were more prominent in a part of frontal pole bordering with the dorsolateral frontal cortex (perhaps consistent with the stronger anatomical connectivity between this region and ventral premotor and parietal motor regions (Yeterian et al., Cortex, 2012)).

(H) Next, we applied the fMRI design that identified whole-brain effects of basis functions in the social fMRI experiment to the control fMRI experiment. We used the same contrast (only omitting Self-position, as it had become meaningless) in the same pgACC ROI as above. (I) We found a significantly positive effect of this regressor in the control fMRI experiment, replicating our previous effects of basis function projections in pgACC. (J) Using a 1-way ANOVA, we confirmed that all three basis function projects were uniformly positive (no main effect of basis function) (K) We identified a whole-brain significant effect of the mean basis function contrast in frontal pole (pre-threshold masked in 20mm-radius sphere around pgACC, $z > 3.1$, $p = 0.05$ FWE). (L) Basis function projects were uniformly positive in this region, too (1-way ANOVA; no main effect of basis function). (motor-S, motor-P, motor-Or, motor-Oi correspond to analogous variables in the social study, the abbreviations stand for: S=Self, P=Partner, Or=relevant opponent, Oi=irrelevant opponent; all data from control fMRI experiment [study 3]; *, $p < 0.05$; **, $p < 0.005$; ***, $p < 0.001$, error bars are S.E.M; $N = 32$).

In summary, we have followed the reviewers' suggestion and conducted a non-social control experiment. In the motor control task, neural signatures of social processing were abolished with no evidence of the social anchoring of basis functions. However, motor basis function related activity was retained with a complementary anatomical centre of gravity within medial prefrontal cortex, consistent with the idea that connectivity with motor regions of the brain was relatively enhanced. Simultaneously, behavioural signatures of self-processing were abolished while signatures of basis function related choice remained. We conclude that basis functions capture a general mechanism used by the brain to compress information in combinatorial patterns. This process is likely to be implemented in slightly different ways and parts of the brain depending on the content and social nature of the relevant information.

In the manuscript, we have added a new main figure and text explaining the motor task and their results. We have also added supplementary materials relating to it and a detailed writeup of the procedures in the Methods section:

Main text: Page 16 (line 554) to page 19 (line 663)

Methods: Page 36 (line 1482) to page 39 (line 1613)

Supplements: Supplementary Figs. 18, 19

3) Replication of key effects and concerns about analysis methods (R1,R2,R3,R4)

As it hopefully has become clear by this point, we have extended but also replicated key effects of interest over several new data sets. This strengthens our argument and shows that behavioural and neural effects of interest are replicable. We would like to emphasize this particularly, because concerns about the strength of our effects were particularly prominent in the most negative review from R4. For instance, **"The difference between self and other correct seems tiny (Figure 1C)"** (R4|O5) or **"The effects of representing in a compressed basis on choice shown in Figure 4 seem quite small"** (R4|O9). Also other reviewers ask about the strengths of our effects, albeit in a much less negative way: **"For 1C: this seems to be a very small difference judging by the dot distribution, and by looking at the histogram itself."** (R1|4.1). R1 in particular suggests, among other comments in their major point 1 that **"It would be helpful for example to add formal information theoretic metrics such as AIC or BIC"** (R1|1). Note, however, that reviewer 4's concern extends beyond the question about the strengths of our effects: **"... the evidence presented in favor of the author's claims did not seem very compelling to me. The amount of data presented was very limited and most of the signals were very small (e.g., Figure 1C, D, Figure 4B, C, F). The fMRI data involved averaging across a large number of subjects (N = 56) and no single subject data is shown, making it unclear whether the focal activations might largely be an artifact of intersubject averaging."** (R4|IntroductorySummaryComment).

We do take concerns about the strengths of our effects seriously and hence detail below the various ways in which our neural and behavioural key effects replicate (addressing R1|4.1, R4|IntroductorySummaryComment, R4|O5, R4|O9). We also add the formal information-theoretic measure requested by R1 (R1|1). However, we firmly oppose R4's idea that somehow our large number of participants obscure effects of interest. This comment goes against the ethos of the field in improving the quality of neuroimaging research and is clearly at odds with statistical best practice. It is not correct that brain activity patterns that are shown are simply the result of averaging. They are the result of parametric statistical analysis that takes into account both the mean and variance of effects. Significant effects are identified when participants consistently show the same pattern of

results. This approach is ubiquitous not just in whole brain neuroimaging but in many areas of science. We agree with R4 that this means that there is a possibility that there may be additional patterns of neural activity that we do not describe because they are present in a small number of participants. However, the results that are included are the ones that we can be confident are replicable because they are found in the majority of participants and exhibit little variance. In summary, averaging data points and assessing significance based on measures of central tendency and spread of data points is a ubiquitous approach in neuroscience research. We would be concerned if we were to take another approach. Nevertheless, we followed R4's suggestion add show single subject data points to our plots, which is obviously very easy to do and we would be happy if this could alleviate R4's concern. ,

3.1) Replication of behavioural effects: Self-bias and basis function effects (R1|4.1, R1|1, R4|O5, R4|O9, R4|Intro)

In the new behavioural experiment outlined above ("study 2"), we replicate the self-bias in accuracy and reaction times. Note that this effect was the one that both R1 and R4 referred to above and that was shown in our previous Fig.1C (R1|4.1, R4|O5). In our large behavioural study, we precisely repeated the underlying analysis, and, as expected, participants were again more accurate and quicker in responding to self-decisions compared to partner decisions. We have hence added this replication to the main text:

Clearly, behavior and neural signals revealed signatures of agent-centric representations. Participants did not rate their own performances as better than their partner's but they did **rate their own performances** more accurately ($t_{55}=2.094$, $p=0.041$; Fig.1C) **than the partner's performances and they made these ratings more quickly than they did for the partner** ($t_{55}=5.299$, $p<0.001$; Fig.1D). This was true even though self and partner exhibited comparable performance, were equally often decision-relevant, and had the same influence on reward payoff. **We replicated both effects in a follow-up behavioural study (study 2; accuracy: $t_{794}=4.369$, $p<0.001$; reaction time: $t_{794}=10.447$, $p < 0.001$; Fig.1C,D).**

Fig.1 ... (C,D) Choice accuracy and reaction times. Participants were more correct in Self decisions compared to Partner decisions while simultaneously responding quicker. **We took the median reaction time per participant as reaction time in the online data were skewed.** The left-hand side shows the results from our fMRI experiment and the right-hand side shows the replication of these effects in an independent behavioural experiment (study 1 and study 2). ($n=56$ for social fMRI experiment [study 1]; $N=795$ for behavioural experiment [study 2] all MRI results cluster-corrected at $Z>3.1$, $p=0.05$; *, $p < 0.05$; ***, $p < 0.001$).

In addition, we also now provide a conceptual replication of the self-bias effect. We do this in our analysis of group decisions in the fMRI study. We reasoned that the above self-bias indicate that participants are better able to retain self-related information (as opposed to partner-related

information) when making decisions. We tested this prediction in group decisions by comparing how the degree to which participants base group decisions on self- vs partner related information. Indeed, a self-bias was again present in group decisions as laid out in this new supplementary figure:

Supplementary Fig.14. ... (C) Group decisions. ... In addition to these analyses, in group decisions, we analysed whether people place more weight on information related to themselves (S) compared to their partner (P). This analysis is related to our observation that participants decided more accurately and quickly in self decisions compared to partner decisions. We tested whether we could replicate the finding of increased reliance on self-relevant information in the group decisions, even though, based on the payoff scheme, S and P should be weighted exactly equally. We ran a 2 [player: S,P] x 2[decision number: 1,2] repeated measures ANOVA and indeed found a main effect of player ($F_{1,55} = 10.292$, $p = 0.002$), but no main effect of decision number and no interaction (all $p > 0.68$). Participants indeed relied more on S-performance compared to P-performance when weighing up which team performed better in group decisions. ($n=56$; **, $p < 0.005$; error bars are S.E.M).

The behavioural effects in our data are therefore highly replicable and consistent. Moreover, our basis function model makes precise predictions when effects should be present in another experiment, and not. For example, we predicted that basis function related signatures of decisions making replicate in the motor fMRI study, but not the socially specific effects (the ones showed in Fig.1C,D above). And this is precisely what we find. Here we show the above accuracy and reaction time plots for the motor fMRI study:

In contrast to this, and in line with our basis function model, basis function related effects of irrelevant players are highly replicable in the motor task (insets in both figures are from the social fMRI task)

Please refer to the full explanation of the motor fMRI task above for full details. Moreover, we have also already shown this in this response, in one of the online studies we made predictions regarding the same type of irrelevant player effect shown here. We predicted that in a No-group condition, participants would be less likely to use basis functions to solve the task and hence the irrelevant players' effects should be significantly reduced in the No-Group condition compared to the Group condition. Again, this is precisely what we found in Self decisions in the behavioural study, and what we replicated in Partner decisions in the same behavioural study (again, for full details, please see behavioural study above):

In addition, we followed R1|1's suggestion and now also employed a model comparison approach to derive information theoretic measures of behavioural model performance. To do so, we used a cross-validation approach. Cross-validation has the advantage that model performance can be directly compared across models without setting somewhat arbitrary penalisation criteria for the number of parameters in the model (like AIC or BIC). Overfitting is directly punished by worse generalisation performance in the held-out sample (see Farshahi et al., Neuron, 2017). We calculated the model deviance and model accuracy. A smaller model deviance indicates a better model fit and we found that the basis function model indeed had significantly smaller deviance (paired t-test; $t(55)=-2.981$, $p=0.004$). The result from the estimated model accuracies also indicated that the basis function model predicted a significantly higher percentage of participants' decisions (paired t-test; $t(55)=8.111$, $p<0.001$). We think that this additional analysis directly addresses the reviewer's question and so we have added it to the revised manuscript in Supplementary Fig.13 as follows:

Supplementary Fig.13. Model comparison of behavioural basis function and agent model (study 1). Our results showed that irrelevant players in dyadic decisions impact choice in a manner consistent with their group membership. For example, during Self decisions, P and Oi are irrelevant. Nevertheless, our GLM analyses showed that P-performance had a significantly positive effect during Self decisions, and Oi-performance had a significantly negative one. This pattern of effects is predicted by our basis function model which suggests that participants first consider a primary basis function and then a secondary one. In support of our approach, we conducted an additional model comparison. We compared two models. One model was the “basis function model” which predicted choices as a function of the performances of S, P, Or, Oi and the non-social bonus (same as in main manuscript). The second one was an “agent model” that predicted choices only as a function of the relevant players and the bonus (e.g., S, Or and bonus for Self decisions). The agent model captured the intuition that decisions are made only based on the relevant players in a decision. The basis function model does not negate this. However, it suggests that there is a necessary addition to the agent model that is needed to explain choice better. This addition is reflected in the weights of influence of the irrelevant players, which are predicted by the basis function model. We compared the two logistic GLM models using a cross-validation approach. We collapsed the data over both Self and Partner decisions. Then we used a leave-one-participant-out cross-validation. Cross-validation has the advantage that model performance can be directly compared across models without setting somewhat arbitrary penalisation criteria for the number of parameters in the model (like AIC or BIC)⁹. Overfitting is directly punished by worse generalisation performance in the held-out sample. We implemented the cross-validation in the following way. We gathered the data from all participants except a left-out participant. We ran the model GLM on the concatenated data of the participants and then applied the resulting decision weights to the left-out participants data. We predicted the choices of the left-out participants on the basis of these weights that had been determined without the influence of the left-out participant’s data. We then calculated the model deviance and model accuracy for the left-out participant¹⁰. The model deviance is a simple function of the difference between actual choices and the models’ choice probabilities. To derive an estimate of the model accuracy, we sampled 10.000 times from the predicted choice probability distribution to get model simulated binary choices. We calculated model accuracy by comparing these simulated choices with the actual choices and averaged the result to a single accuracy value. We repeated this procedure for all participants and aggregated the model deviances and model accuracies for both models. A smaller model deviance indicates a better model fit and we found that the basis function model indeed had significantly smaller deviance (paired t-test; $t_{55}=-2.981$, $p=0.004$). The result from the estimated model accuracies also indicated that the basis function model significantly predicted a higher percentage of participants’ decisions (paired t-test; $t_{55}=8.111$, $p<0.001$). A histogram of the difference in estimated model accuracies is shown in this figure. Note that the distribution is significantly shifted to the right, indicating better model fit for the basis function compared to the agent model (N=56).

We also explicitly refer to this result in the main text:

We examined the consequences of this in computational simulations and found that such an overweighting predicts a characteristic pattern of choices, in which players irrelevant to dyadic decisions still have an inappropriate effect on decisions in line with their team membership (Fig.3I). Indeed, analyzing player specific effects on choice revealed precisely these effects of irrelevant players in Self decisions (P: $t_{55}=3.093$, $p=.003$; Oi: $t_{55}=-5.131$, $p < 0.001$; Fig.3J) and Partner decisions (S: $t_{55}=4.867$; $p < 0.001$; Oi: $t_{55}=-5.104$; $p < 0.001$; Fig.3K). Formal information theoretic metrics confirmed the higher model accuracy of the basis function model over the agent-centric model (paired t-test on model accuracy; $t_{55}=8.111$, $p < 0.001$; Supplementary Fig.13; see Supplementary Fig.14 for a detailed decision analysis of all trial types).

3.2) Replication of neural effects: Social signals and basis function signals

As we have laid out in detail above, the results of the new motor fMRI study further confirmed our basis function model on a neural level. We replicated our basis function related brain signals in pgACC using the analogous fMRI design. We saw again significant effects in pgACC:

By contrast, and also as predicted by our model, neural signatures of social processing (self signal in pgACC, partner signal in dmPFC) were indeed absent in the motor study:

In summary, we have shown two types of neural effects and two types of behavioural effects – related to basis function processing and social processing, respectively. We have shown that these effects replicate across another behavioural study and an fMRI study in precisely the conditions where we expect them to replicate. We have also shown that these signals are significantly weaker or absent in conditions where our model predicts them to be. This is strong evidence that the effects we identify and that we base our conclusions on are reliable and replicable. We now also provide an information-theoretic metric to demonstrate them behaviourally.

4) Requests for additional neural analyses (R1,R2,R3,R4)

Several reviewers have asked additional questions about our fMRI data analysis and requested additional analyses. We present these analyses in this section. Note that sometimes reviewers shared concerns about related questions. However, we have identified less convergence among reviewers on these issues and will therefore separate them in different questions. While until the current point, we deem our responses relevant for every single reviewer, the following answers address questions that were not asked by all reviewers.

4.1) Region of interest analyses of dmPFC, vmPFC and amygdala in relation to basis function effects and social processing (R1|5.2, R1|5.4, R3|8, R4|6)

To give a fuller picture of our data, we address points of reviewers 1,3 and 4 related to our neural analysis. Reviewer 1 requests ROI analyses in vmPFC, dmPFC, and the amygdala in relation to social signals in particular: **“there were a number of areas within the main text that needed additional clarification with respect to the regions of interest and how different areas may have responded to the task.”** (R1|5). **“While it’s clear that dmPFC and ACC are relevant areas, and therefore I agree with the idea of a preliminary ROI analysis there, I was wondering if the authors considered to also extend this to vmPFC.... I would be interested to see results of such analysis specifically (i.e. comparisons between dmPFC and vmPFC chosen value vs agent specific).”** (R1|5.2) **“Have authors observed clusters of activations in amygdala, and if those are also agent dependent?”** (R1|5.4). We have conducted these additional analyses. We have combined them with a request from reviewer 3 asking for the existence of a specific type of other related signal, namely brain signals related to the opponents: **“8. Were the information about opponents encoded by any brain areas? (related to Figure 1E)”** (R3|8). When conducting and presenting these additional analyses, we kept in mind the request from reviewer 4 to show individual data points: **“6. Figures 1E, 3B, 3H are averaged across 56 subjects. Please show single subject data (e.g., the same slice in all 56 subjects) so readers can get a sense for how broad the activations are within single subjects.”** (R4|6).

We have followed the suggestions of the reviewers and now present detailed ROI analyses of (a) basis function related social activity (our main contrast of the sum of the three basis function projections plus S-Position) and (b) the social DV. These signals correspond to the yellow red cluster in the main figure).

There are indeed dissociations apparent along the lines suggested by R1|5.2, R1|5.4. dmPFC, in particular, does not carry basis function related activity but is instead sensitive to the social DV. By contrast, vmPFC also reflects the social DV, but in addition encodes information about the basis functions. We complement these analyses with ROIs in more traditional social areas such as the amygdala and the TPJ, where we also examine social basis function and social DV related activity. To be thorough, we have taken ROIs in two regions of the amygdala (a more basolateral region and a more dorsal region in and adjacent to the central nucleus). We find that while the social DV is processed in a wide network of social regions including the TPJ, amygdala, and dmPFC, the basis function related activity is confined to medial prefrontal cortex with its centre of gravity in pgACC, but

extending towards vmPFC (but not towards dmPFC). We have added the following supplementary figure to convey these findings:

Supplementary Fig.11. Activity related to the social decision variable and the unsorted basis functions (study 1). (A) We investigated the distribution of the social decision variable (chosen – unchosen) (red activity in Fig.3G) in a selected set of regions of interest (see Supplementary Fig.5). We examined activity in dmPFC (MNI: -2/22/28) and pgACC (MNI: -8/42/14), the two regions where we found Self and Partner related signals (see Fig.1E). In addition, we used masks of areas typically found in studies of social interactions³. Based on recent anatomical parcellations, we inspected activity in left and right posterior temporoparietal junction (pTPJ)⁴ as well as central and basal parts of the amygdala (Amy-Ce and Amy-B)⁵. We extracted FSLs COPE images⁶ from *fMRI GLM 3* in these regions (i.e. images of fMRI effect sizes for the social DV (chosen-unchosen)). We found significant effects of the social decision variable in several regions associated with social decisions making including dmPFC (one-sample t-test; $t_{55}=-4.224$, $p<0.001$), the left TPJ ($t_{55}=4.385$, $p<0.001$), and both central ($t_{55}=2.182$, $p=0.033$) and basal amygdala ($t_{55}=2.402$, $p=0.020$). One notable exception is pgACC ($t_{55}=-1.288$, $p=0.203$), where we had found activity related to the basis functions. This is consistent with our proposal that pgACC is important for representing social information in a compressed and combinatorial code along the basis functions, while vmPFC has a relatively more important role in guiding decisions. (B) Whole brain cluster-corrected activation of the social decision variable (chosen – unchosen). The whole brain maps confirm that a wide network of regions carried information about the social decision variable. We have overlaid these activation maps with the ROI masks we had used in our region of interest analyses. Arrows point towards their location to aid interpretation of the ROI results in panel A. It is

apparent that vmPFC and dmPFC both reflect the decision variable, but with opposing weights. PgACC in between these two clusters did not show significant effects. Furthermore, these maps show that there is activity in or adjacent to the amygdala in both hemispheres (note that the two amygdala ROIs are close together, this is why we highlight them jointly). However, the centre of this activation appeared to be more posterior than the amygdala, and effects were most clear bilaterally in the anterior hippocampus (see middle row). **(C)** We analysed activity related to the unsorted basis functions and S-position (same yellow activity as in main Fig.3B). To this end, we extracted COPE images from *fMRI GLM 2* relating to this contrast. In addition to the established ROIs, we also inspected activity in ventromedial prefrontal cortex (vmPFC; MNI: -2/60/-12). We took coordinates from the social DV contrast in panels A and B. (see main Fig.3G (red activity cluster, social DV (chosen-unchosen))). The only significant cluster was the one in vmPFC ($t_{55}=2.735$, $p=0.008$). Note that the effect remains significant even after excluding the one apparent positive outlier (above the significance stars). This is consistent with the relatively restricted distribution of this contrast apparent in the whole brain maps. In contrast to the social DV, the unsorted basis functions plus S-position contrast only identified one large significant cluster centered on pgACC. (* $p < 0.05$; ** $p < 0.01$; *** $p < 0.001$; error bars are S.E.M; $N=56$, MRI results cluster-corrected at $Z>3.1$, $p=0.05$).

Moreover, we have conducted additional analyses suggested by R3|8, who was asking which brain regions represent opponent-related activity. We have supplemented this analysis as well and their results are very consistent with this point raised before. We did not observe any whole-brain corrected signals relating to the opponents. However, opponent-related activity is apparent in dmPFC, precisely the region where that was identified as carrying information about the performance of the partner. Opponent-related activity was not apparent in pgACC or vmPFC. The dmPFC activity by contrast was so strong that it even survived multiple comparison correction for the number of ROIs used. This result is further evidence for the contention that dmPFC is relatively specialised for representing social information, while basis function related activity is not apparent in this region:

Supplementary Fig.5. Opponent related activity in regions of interest (study 1). Our results showed whole-brain corrected effects of S-performance in pgACC, and Partner performance in dmPFC. By contrast, we did not find whole-brain corrected effects related to the opponents' performance (neither Or nor Oi). We investigated whether opponent-related brain signals were present in a series of regions of interest (ROIs). We examined activity in dmPFC (MNI: -2/28/44) and pgACC (MNI: -8/42/14), the two regions where we found Partner and Self related signals (see Fig.1E). In addition, we inspected activity in ventromedial prefrontal cortex (vmPFC; MNI: -2/60/-12). We took coordinates from chosen value signals that we report in Fig.3G (red activity cluster, social DV). VmPFC is a region not typically associated with social cognition. In addition to spherical ROIs in these regions, we used masks of areas typically found in studies of social interactions³. Based on recent anatomical parcellations, we inspected activity in left and right posterior temporoparietal junction (pTPJ)⁴ as well as central and basal parts of the amygdala (Amy-Ce and Amy-B)⁵. We extracted FSLs COPE images⁶ from *fMRI GLM 1* in these

regions (i.e. images of fMRI effect sizes). We extracted Or-performance from the second decisions in a trial – the same time point where we had found Self and Partner related signals. We did this for Self and Partner decisions and averaged their effect sizes in these ROIs. We found significant activations in the right TPJ (one-sample t-test; $t_{55}=2.249$, $p=0.029$) and dmPFC (one-sample t-test; $t_{55}=-2.935$, $p=0.005$). Both regions are often found in studies of social cognition. Note that the dmPFC region is the same one where we had found partner-related activity and so these results might suggest that dmPFC represents social agents more generally, in line with our previous findings regarding dmPFC¹. Note that the opponent-related activity in dmPFC also survives Bonferroni correction for multiple comparisons for using 7 ROIs ($p_{\text{Bonferroni}} = 0.05/7 = 0.007$). (*, $p < 0.05$; **, $p < 0.005$; error bars are S.E.M; $N=56$).

Note that, as requested by R4|6, we display single subject data in all the above plots. We also replot our main results illustrating single subject data points in Fig.S7:

4.2) Sequential vs agent-centric basis function signals (R2|2, R3|1)

Another point concerning the neural analysis that was raised concerns their relationship to agent-centric coding. Reviewer 2 notes “It is not entirely clear and therefore perhaps needs to be explained better why the authors consider only the basis functions indexed by the serial order, but not those indexed by the identities of the agents. For example, in the fMRI GLMs, the information indexed by the serial order is represented in the basis functions, but the information about each agent is entered individually. Is it possible to test both types of information using similar basis functions?” (R2|2). In a similar manner, reviewer 3 says “Generally, these results showed that, in this task, the brain encodes DVs in a hierarchical way? The brain first encodes DVs for the teams, and then filters out useless information based on the need of the decision phase. Did the basis function encoding say anything more than this? Because such hierarchical encoding does not necessarily require the sequence-based frame of reference, the agent-based model can also explain this by first considering agent groupings, such as my team and other team, in-group and other-group, etc, before becoming more specific.” (R3|1).

We address these points both on a neural and on a behavioural level (see below). It is very accurate that we argue that basis functions enable people to compress the observed player information and to decompose the decision-making process hierarchically in two steps (R3|1). First people represent the primary basis function. The primary basis function – when transposed in a reference frame of own team vs other team – provides the DV for the teams and is hence sufficient to solve group decisions (decisions that compare the joint performance of self and partner, S+P, with the joint performance of the Opponents, O1+O2). For dyadic decisions – “self trials” in which one’s own performance is compared with a relevant opponent and “partner trials” in which the partner’s performance is compared with a relevant opponent – two basis functions are needed; the same primary one that is relevant for the group decision, plus a secondary one. We propose that rather than merely using 4

independent, agent-centric representations of the group, people build compressed representations along the basis functions to represent player information efficiently.

However, in doing this we consider basis functions on two levels. First, on the sequential level. We highlight them because we believe that they reflect a fundamental aspect of multi-person decision making that has been overlooked so far. We also highlight them because they are the most general way in which the task space can be compressed in our experiment. The social basis functions (e.g., the DV between the teams in the reference frame of “my team vs other team”) directly follow from the sequential basis functions. When describing how they are used to facilitate decision making, we referred to them in the previous draft of our manuscript as being “flipped” or as “flipped basis functions”. We used the word “flipped” to indicate that they were transposed into a social reference frame, one that is encoded the frame of reference of one’s own team versus the other team. [Note that following R2’s suggestion, we now refer to the “flipped” basis functions as “inverted” or “social” basis functions, because they reflect differences in player-linked scores in an agent-centric frame of reference. “3. Wouldn’t it be better to refer to “flipped” and “sign-flipped” as “inverted” and “rotated”, instead?” (R2|Minor3)]

We present several pieces of evidence in our manuscript that the basis functions exist in a sequence-based reference frame, in particular the brain results in Fig.3 that emphasize sequential basis functions at the time before choice. Note also, that we below show a replication of sequential basis functions. In a new fMRI study (N=32) we find sequential basis functions in the motor domain in medial prefrontal cortex. Therefore, we show strong and repeated evidence for sequential basis functions in medial prefrontal cortex (below a panel from the new study):

Following the point raised by the reviewers, however, we have now also added new analyses to further elaborate on the relationship between sequential and agent-centric basis functions. In the manuscript, we refer to the agent-centric basis functions in a supplementary figure showing that they were included in the same fMRI design as the sequential basis functions. The regressor set related to the light-yellow box below refers to the inverted – or social – primary and secondary basis functions. Those are the two that are needed to estimate dyadic decisions. The primary social basis function

reflects the score difference between one's own team versus the other team, and the secondary social basis function reflects a combination across players that points at performance differences of relevant versus irrelevant players (those that are compared in a dyadic decision, and those that are ignored).

We have included new results that compare sequential and social primary and secondary basis function. Importantly, both results stem from the same neural GLM meaning that we can show the presence of social and sequential basis functions simultaneously while controlling for variance associated each other type of basis function. We have included this result in the supplement, and added a clear reference to the social basis functions and the supplementary figure to the main text: Main text:

Indeed, a large region along the cingulate sulcus, posterior to our previously identified contrast, encoded the combination of primary and secondary basis function projections (Fig.3G). Again, this is striking because these variables encode sequential information and are unrelated to agent-centric or decision-related variables, which we again control for in the analysis (Fig.3A, Supplementary Fig.9; Methods). For example, for the primary basis function encoding the difference between pairs of sequential positions, we included a control regressor indexing the same difference but in the **social** frame of reference of “own team vs other team”. We refer to the latter agent-centric primary and secondary basis functions as “inverted” basis functions and we found that these inverted basis functions captured brain activity in distinct regions of frontal cortex and subcortical regions such as the ventral striatum – importantly, in the same GLM (Supplementary Fig.9,10). Therefore, even after the observation phase had ended, the cingulate sulcus carried a sequential code of the projections onto the primary and secondary basis function.

Supplementary Fig.10. Neural signals linked to sequential and inverted social basis functions (study 1). With the same GLM (*fMRI GLM3*, see Methods), we analysed neural activity related to the projections onto the basis functions using two different frames of reference. All results shown in this figure are time-locked to the onset of dyadic decisions (both self and partner decisions) and derive from the same GLM model. **(A)** We identified activity related to the sequential basis functions (see main text; same activation is reproduced in this figure) in posterior and anterior parts of the cingulate cortex as well as the bilateral frontal pole. Here, activity related to a combination of the basis function projections is shown (primary + secondary – tertiary). **(B,C)** In the same GLM time-locked to the same point in time, we also identified neural activity related to the projections of the basis functions onto a social frame of reference. We also refer to this frame of reference as “inverted”, because it anchors the basis functions along the difference between one’s own and the other team’s performance difference (social primary basis function) and along an axis separating decision-relevant from decision-irrelevant agents (social secondary basis function). The calculation of and transformation between these frames of reference is explained in the Methods (*Sorting of basis functions and transformation to choice*). Importantly, due to the design of our experiment and in particular the fact that sequential positions within the 4-player observation phase were counterbalanced, it was possible to include projections onto both types of basis functions in the same GLM model. The sequential and the inverted (social) basis function projections were statistically independent ($R^2 < 0.004$; see Supplementary Fig.9). In medial frontal cortex, we found activations related to both the sequential basis functions and the social basis functions. Notably, the social basis functions were represented in more dorsal and anterior parts of medial prefrontal cortex and did overlap with each other, but did not overlap with the sequential basis functions (compare medial sections in first row across panels A, B, C). While the former were represented in pre-SMA and anterior parts of ACC sulcus, activity related to the sequential basis functions were represented in the gyral part of ACC. However, sequential and social basis functions were represented not only in adjacent but also in clearly distinct brain networks. Both primary and secondary social basis function related activity was found in the anterior caudate nucleus, extending bilaterally into the ventral striatum (more strongly for the secondary social basis function). By contrast, the latter regions showed no response related to the sequential basis functions, also not below cluster-forming threshold (compare second row across panels A, B, C). In addition, social basis function related activity was found bilaterally in the intraparietal sulcus, the anterior insula, and parts of ventrolateral prefrontal cortex. By contrast, the sequential basis functions were represented in more dorsal parts of lateral prefrontal cortex, more consistent with the location of the frontal poles (compare third row across panels A, B, C). ($n=56$; all MRI results cluster-corrected at $Z>3.1$, $p=0.05$).

We have also added a reference to the social basis functions in one of the main figures linked to the main text. Note that the dark-blue activity pattern in the supplementary figure (shown also above) is apparent in several brain regions and at the same time there is activity linked to the social basis functions that can also be identified. Sometimes it is in the same brain area but sometimes in different brain areas.

4.3) Additional analyses of the self-position signal (R3|7)

Reviewer 3 asks “7. In Figure 3D, is the positive peak timing related to the actual Self position in the sequence (that is, earlier peak if Self position was 1 compared to Self position was 4)?” (R3|7)

This is an interesting point. We addressed it using simulation analyses. In the main text, we reported a signal in pgACC indexing the position of oneself within the sequence of players in the observation phase (regressor with values 1,2,3 or 4 depending on one’s own position in the sequence). In our new analyses, we investigated whether the peak timing of this signal was related to the actual Self position in the sequence, i.e. whether the Self-position signal reflected a BOLD response that was simply the aggregate of several separable neural responses each of which is time-linked to a particular instance of “Self” score presentation rather than the reactivation of a representation of the Self position within the sequence prior to decision making. Our analyses suggested that this was unlikely to be the case and that the Self-position signal we observed was more likely a reactivation of previous position information that emerged later in time. We used simulations for these analyses as explained in the following new supplementary figure:

Supplementary Fig. 8. Simulation analysis of the Self-Position signal in pgACC (study 1). In the main text, we reported a signal in pgACC indexing the position of oneself within the sequence of players in the observation phase (regressor with values 1,2,3 or 4 depending on one’s own position in the sequence; Fig.3D). We analysed whether the peak timing of this signal was related to the actual Self position in the sequence, i.e. whether the Self-position signal reflected a BOLD response that was simply the aggregate of several separable neural responses each of which is time-linked to a particular instance of “Self” score presentation rather than the reactivation of a representation of the Self position within the sequence prior to decision making. Our analyses suggested that this was unlikely to be the case and that the Self-position signal we observed was more likely a reactivation of previous position information that emerged later in time. We used simulations for these analyses as follows. **(A)** We considered a time period that included the observation phase and the onsets of the 4 -player specific displays (see timing of vertical lines: Pos1 on, Pos2 on, Pos3 on, Pos4 on). In addition, the analyses covered the time period analysed in the main manuscript figure (everything to the right of “Obs off”, the end of the observation phase). We simulated a hemodynamic response function (Hrf) time-locked to a schedule in which the Self was either presented 1st, 2nd, 3rd or 4th, mirroring the balanced presenting of Self in our actual experiment. The Hrf was modelled using a gamma probability distribution with a mean of 6 and sigma of 3. These settings were taken to match the default Hrf response in FSL⁶ and we used the gamma-function implemented in FSL. We simulated a schedule of 144 trials for 56 assumed participants, in which the Self was equally often presented in each sequence position, matching the schedule of our experiment. We generated Gaussian distributed white noise (sigma of 0.05) and filtered it via a Savitzky-Golay filter to simulate autocorrelated noise as observed in the BOLD signal. Subsequently, we applied a neural GLM (FSL’s ols) to the data using the ground-truth Self-Position as the only regressor. We extracted the neural effect size at every time point of the time series. **(B)** This gave us the simulated effect size of Self-position shown in panel B. It can be seen that a Self-position signal caused by time-linked Hrf’s would generate a large negative signal during the observation phase that turns slowly positive after the offset of the observation phase. The empirical effect sizes did not match this pattern and instead suggested a sharp rise of the signal at the time of observation phase offset. **(C)** The empirical effect-size for Self-Position, expanded to also cover the time period before observation phase offset (everything to the right of the “obs off” line corresponds to the main manuscript figure). In summary, the temporal evolution of the Self-position signal does not match up with a timecourse expected from a Self-position signal that is caused by separable neural responses each of which is time-linked to Self presentation in the observation phase. Instead, the signal rather seemed to

reflect a reactivation of information linked to the sequential position of the Self in the sequence. Moreover, when analysing the same timecourse in our motor control experiment (see main Fig.5, Supplementary Fig.18), there was never a significant deflection of the Motor-S-Position signal from zero, not even when tested without a stringent leave-one-out procedure (all p 's > 0.12 in time window 0-10 sec after observation phase offset). (*, $p < 0.05$; **, $p < 0.005$; error bars are S.E.M; N=56).

5) Requests for additional behavioural analyses (R2,R3)

In the same manner as in the last section, we here present additional behavioural analyses that we conducted upon the reviewer's requests.

5.1) Behavioural effects of basis function "flipping" (R2|3, R3|1)

Related to the neural questions about the sequential basis functions, we now answer to questions from R2 and R3 about the sequential nature of the basis functions on a behavioural level. Reviewer 2 notes that **"One puzzling finding is that the decision accuracy was affected by whether the basis functions need to be flipped or not (page 12, Figure 4B) and by the interaction between the flipping of primary and secondary basis functions (Figure 4C). This is difficult to understand because the results from the analyses related to the basis functions would not change when you multiply any subset of the basis functions by minus one. Therefore, it is not clear why the flipping (or inversion) of any single basis function should have any effect on the behavior (or analysis of neural data)."** (R2|3) We respond to this point now, and we also further respond to reviewer 3's comment that we already discussed on a neural level in the previous response: **" Because such hierarchical encoding does not necessarily require the sequence-based frame of reference, the agent-based model can also explain this by first considering agent groupings, such as my team and other team, in-group and other-group, etc, before becoming more specific."** (R3|1)

First of all, we agree with R2|3 that the signs of the basis functions are arbitrary in the model and therefore one would not necessarily expect that they impact behaviour. However, we do note that the basis function flip influences behaviour is a unique prediction derived from our model, which, to our knowledge, does not follow from any other model of decision making. This is one way in which our basis function model relies on a sequential frame of reference. However, as suggested by R3|1, **we have** moved these results to the supplements. Furthermore, in the supplements we now also report that one of the two effects is replicated in our behavioural study, providing further evidence for this counterintuitive effect:

We repeated the analysis investigating the effect of both primary and secondary basis function sign inversions on log-transformed reaction times. We used the same GLM model as for the fMRI study. To account for the reduced trial number and increased noisiness of the data, we changed the following. First, we analysed all trials per participant and did not subsample (in our related fMRI analysis, we had focused on trials of particularly high dimensionality, where no basis function was zero). Second, we used the matlab function "robustfit" to account for outliers. We found, again, that the interaction term between primary and secondary basis function flips predicted faster reaction times ($t(790)=2.2718$, $p=0.023$). Note that the effect is driven by a significant effect in the Group condition ($t(393)=2.1$, $p=0.036$). The effect was not itself significant in the No-Group condition ($t(396)=1.05$, $p=0.29$). Note that this result even persists when sub-selecting trials with non-zero basis functions as in the related fmri analysis ($t(791)=-1.82$, $p=0.035$, one-sided test). Finally, we note that the GLMs failed to fit in a very small subset of participants (up to 4 out of the 795 participants) and so the analysis could not be performed with their data.

Supplementary Fig.15. Inversion of basis functions correlates with reaction times (study 1 and study 2). (A) Illustration of primary basis function inversion. Basis functions are defined sequentially, e.g., as shown, the basis function w_2 ([1 1 -1 -1]) contrasts the first two with the second two performances (see left side). Black and red indicate positive and negative weights, respectively. The right-hand-side illustrates the same basis function but inverted to an agent-centric perspective (note the letters on the figurines now indicate player identities). The latter social basis functions weigh one's own team members positively and performances from the opponent's team negatively. Note that the lower-right case involves a "flip" or inversion of the basis function, whereas the upper-right case does not. (B) We analysed the effects on choice of whether or not a basis function had to be inverted (see *Sorting of basis function and transformation to choice* section) in our fMRI data set. We focused on self and partner decisions as only those decisions entailed the possibility of inverting both primary and secondary basis functions. Inverting meant that the sequential basis function were inverted relative to the team groupings. For example, for primary basis functions, positive weights were assigned to the opponents and negative ones for one's own team. In such a case, the basis function projection had to be inverted (multiplied by "-1") to translate it in an agent-centric reference frame with positive weights for one's own team and negative weights for the other team. No inversion was necessary if the sequential basis function already coincided with the positive weights of the basis function. To detect effects of basis function inversion, we restricted the analyses to trials with only non-zero basis functions. A basis vector with a value of zero never needed to be inverted, as it cannot be sign-reversed. We used a linear regression to analyse reaction times (RT). We log-transformed RTs to account for skewness of the data. All regressors were normalized (mean of zero, standard deviation of one). As regressors, we first took account of the fact that choices should be more likely to be correct when value differences were either strongly in favor of an engage choice or strongly against it. Moreover, as an additional control, we also entered the temporal positions of Self and Partner (could be 1, 2, 3, or 4th position in the sequence of performance presentations for both players) as two additional regressors. This ensured that effects of primary function inversions were not driven in a simple way by temporal preference effects for players of one's own group. As we have shown that secondary basis functions should be integrated only after primary basis functions, we reasoned that secondary code inversions might impact RTs as a function of whether or not primary code inversions had also occurred. Hence, we considered the interaction term of primary and secondary code inversions (1 if both codes remained non-inverted or if both codes were inverted, , 0 if one code was inverted and the other was not). We entered this interaction term in a linear GLM predicting logRTs. In addition, we included as control variables: absolute value difference (own team member minus opponent team member plus bonus), self position, and partner position, and the component parts of the interaction: a regressor for primary code inversions (1 or 0) and secondary code inversions (1 or 0). We found indeed that the interaction term between primary and secondary basis function inversions predicted faster reaction times ($t_{55}=-2.026$, $p=0.048$). (C) We replicated the above effect in our online study. We used the same GLM model as for the fMRI study. To account for the reduced trial number and increased noisiness of the data, we included all self and partner decisions in the analyses and did not restrict the analysis to trials with only non-zero basis functions. Also, we used the matlab function "robustfit" to account for outliers. We found, again, that the interaction term capturing whether or not primary and secondary basis functions were either both un-inverted or both inverted predicted faster reaction times ($t_{790}=2.2718$, $p=0.023$). Note that the effect is driven by a significant effect in the Group condition ($t_{393}=2.1$, $p=0.036$). The effect was not itself significant in the No-Group condition ($t_{396}=1.05$, $p=0.29$). Note that this result even persists when sub-selecting trials with

non-zero basis functions as in the related fMRI analysis ($t_{791}=-1.82$, $p=0.035$, one-sided test). Finally, we note that the GLMs failed to fit in a very small subset of participants (up to 4 out of the 795 participants) and so the analysis could not be performed with their data (*, $p<0.05$; error bars are S.E.M).

Finally, we have also conducted a new behavioural demonstration of specifically sequential effects on choice in a new large-scale behavioural study conducted online. We incorporated it in the supplementary material: Supplementary Fig.6

5.2) Additional analyses of behavioural decision making performance (R3|3, R3|4)

Reviewer 3 raises two related points about the way participants make decisions in our experiment. “3. The authors only showed the comparison between Self and Partner decision conditions (Figure 1C and D). How about the Group condition? It is actually important to know how the Group condition’s accuracy and RT looked like, because based on the later basis function hypothesis, the authors should expect higher accuracy and shorter RT for the Group condition. This is because only primary basis function is needed for the Group condition.” (R3|3). We respond to this point in conjunction with the next point raised by the same reviewer which tackles a similar issue: “4. For the two consecutive decision phases in one trial, did they affect each other (Fig. 1)? For example, was the second decision more accurate? Was the second decision associated with shorter RT compared with the first one? Did the first decision improve or harm the second decision in any way (both behaviorally and neurally)? The authors should explore these topics more and if/how this might be related to the proposed basis function.” (R3|4)

Both points are very interesting. Thank you for raising them. Note that we respond to one important aspect of question R3|3 already in our very first point of this rebuttal. We highlight that the inclusion of group decisions incentivise participants to use efficient, compressed representations to make decisions. We find that participants perform better in a version of the task that comprises group decisions versus a version that does not, but that is in all other aspects closely matched. This highlights, as implicitly suggested by the reviewer, the importance of the group decisions within our experiment.

Moreover, we have now in addition thoroughly investigated the group decisions and the relationship between the first and the second decision in our experiment. We outline the results below. We conducted these analyses also and in particular in relationship to the group condition. As the reviewer rightly points out, the group condition plays a crucial role in our experiment and in our theoretical model.

Firstly, it is correct that group decisions are special in that only the primary basis function is needed to make a decision. By contrast, two basis functions are needed for dyadic decisions. It therefore is indeed a very interesting question how accurately and how quickly people respond to the group decisions opposed to the dyadic decisions. We agree with the reviewer that our model predicts that participants perform well in group decisions as only the primary basis function is needed. However, it is not clear whether our model would necessarily predict that people perform better in group decisions compared to dyadic decisions. Our neural evidence shows that at the time before the first decision, all three basis functions are represented. Indeed, the brain does need to hold on to all three basis functions until it is clear which decision is asked of the participant. It follows that dyadic decisions and group decisions pose unequal demands regarding the use of the basis functions: in dyadic decisions, a primary basis function is combined with a secondary one, whereas in group decisions the primary basis function needs to be used and the two others need to be discarded. Therefore, it is not clear whether participants necessarily should perform group decisions better and quicker than dyadic decisions. On the other hand, we do agree that our model predicts that participants should do well in group decisions, and they should do better than predicted by an agent-

centric model with four representations. The latter seems to predict that decision accuracy and reaction times should get worse the more players need to be assessed in the decision. This would predict that group decisions are worse than dyadic decisions. However, this is not the case in our data. Examining the %correct decisions in self, partner, and group decisions clearly shows that people do well in group decisions. In fact, for the second decision, participants perform better (!) in group decisions than in self and partner decisions, and for the first decision, participants perform better, although not significantly better, for the self decisions compared to the group decisions. The reaction time analyses tell a similar story. Together, these analyses confirm the presence of a self-bias outlined in our original submission (the fact that people respond more quickly and more accurately for self than for partner), but also highlight that participants perform exceptionally well in group decisions – even to a degree that can match – or even exceed – the known processing benefits of self-related information. We summarize the findings in a new supplementary figure as follows.

Supplementary Fig. 3. Decision accuracy and reaction times across conditions (study 1). (A) We analysed decision accuracy (percentage of correct choices, %correct) per decision type (S=Self decision, P=Partner decision, G=Group decision) and separately for the first decision (Dec1) and the second decision (Dec2) in a trial. We especially considered how well participants performed in group decisions compared to dyadic decisions. In a three-way repeated measures ANOVA with the factors decision number (Dec1 vs Dec2) and decision type (Self, Partner, Group), we found significant main effects of decision number ($F_{1,55}=13.08$, $p < 0.001$), decision type ($F_{2,110} = 5.02$, $p = 0.034$), as well as a significant interaction ($F_{2,110} = 4.68$, $p = 0.011$). Interestingly, while the second decision was less accurate than the first one for both dyadic decision types (Self: $t_{55} = 4.69$, $p < 0.001$; Partner: $t_{55} = 2.37$, $p = 0.021$), decision accuracy was virtually the same for the second compared to the first group decision ($t_{55} = 0.06$, $p = 0.954$). One might expect that decisions about the group were less accurate because they need to integrate (noisy) estimates of all four individuals rather than only two. However, in contrast to that we found that, for second decisions, accuracy in group decisions exceeded both self decisions ($t_{55} = 2.52$, $p = 0.015$) and partner decisions ($t_{55} = 3.27$, $p = 0.002$). For initial decisions, self decisions were significantly more accurate than partner decisions ($t_{55} = 2.43$, $p = 0.018$), but not significantly more accurate than group decisions ($t_{55} = 1.68$, $p = 0.098$). (B) We applied the same ANOVA to the reaction times (using the mean-RT per participant). We identified a main effect of decision type ($F_{2,110} = 9.01$, $p < 0.001$) and an interaction between decision type and decision number ($F_{2,110} = 10.25$, $p < 0.001$). Decisions were made more quickly for self than for partner (Dec1: $t_{55} = 3.816$, $p < 0.001$; Dec2: $t_{55} = 4.196$, $p < 0.001$). Group decisions were made with roughly the same speed as partner decisions ($t_{55} = 0.12$, $p = 0.903$) and slower than self decisions if they came first ($t_{55} = 2.76$, $p = 0.008$). By contrast, group decisions were just as fast as self decisions ($t_{55} = 1.14$, $p = 0.258$), and quicker than partner decisions if they came second ($t_{55} = 5.27$, $p < 0.001$). Together, these results further support the finding that participants made the decisions more accurately and quickly for self than for partner, confirming the social nature of our task. In addition, participants performed well on the group decisions, often at the same level as in self decisions and better than in partner decisions. This might be surprising if one assumes that people solve the task by using four independent, agent-centric representations where the estimation noise accumulates with the number of players that are included in a decision. However, these results are highly consistent with our hypothesis that people use compressed representations to summarize the observed information and

that the way the information is compressed is linked to the groupings of the two teams. (n=56, error bars are S.E.M)

We went on to examine the difference in decision accuracy between self, partner, and group decisions even more closely. Although %correct and RT are clearly central markers of decision performance, they may not be the best ways to compare the different decision types. One other established decision-theoretic approach is to examine decision accuracy in relation to the decision variable (DV). The DV for each decision type is calculated as:

- a) Self decisions: $S - Or + B$ (S = Self, Or = relevant opponent, B = Bonus)
- b) Partner decisions: $P - Or + B$ (P = Partner)
- c) Group decisions: $(S + P) - (O1 + O2) + B$

As a reminder, the DV here also considers the non-social Bonus, B. The bonus is a small additional increment that is presented as a yellow (for positive values) or red (for negative values) cue at the time of decision. The bonus must be added to the DV to result in the correct decision. We examined decision accuracy in relation to the DV by fitting a logistic function to people's choices. We did this in two ways: by plotting the absolute and the relative DV (== DV in percentage). The absolute DV (the DV as expressed in a,b,c above) is a clear and transparent index of good decisions: positive values indicate that choosing one's own team is better, and negative values indicate that choosing the opponent team is better. The absolute DV is well suited to compare self and partner decisions but may not be the best way to compare these two decision types with group decisions. Group decisions contrast the combined performances of both teams and hence the quantities to compare are much bigger than the quantities that are compared in dyadic decisions. Hence, it may be fairer to compare decision performance based on the relative DV, the percentage difference between the two performance scores of the teams. This may become clearer by considering the following numeric example, omitting the bonus for simplicity (assuming S = 3, P = 6, O1 = 4, O2 = 4):

- a) Self decisions: $3 - 4 \rightarrow$ absolute DV = -1
- b) Partner decisions: $6 - 4 \rightarrow$ absolute DV = 2
- c) Group decision: $(3 + 6) - (4 + 4) = 9 - 8 \rightarrow$ absolute DV = 1

In this example, the absolute DV between self and group decisions is the same (a difference of [minus] "1"). It does not consider that the numbers 8 and 9 are much closer together, in relative terms, than the numbers 3 and 4. Hence, we now supplement analyses of decision accuracy based on both measures, the absolute and the relative DV. These new analyses show that, again, higher decision accuracy for self than for partner decisions. In addition, they show that group decisions are less accurate than the former two trial types in terms of the absolute DV, but they are in fact much more accurate when considering the percentage DV. We added the following supplementary figure:

Supplementary Fig. 4. Decision accuracy in relation to the decision variable (study 1). We analysed decision accuracy in relation to the decision variable (DV) for self (S), partner (P) and group (G) decisions. To this end, we fitted a logistic regression to the choice data using ridge regression to account for outlying data points. We predicted the choice to engage (1) versus avoid (0). We followed the same procedure as described in the Methods section regarding the online experiment (see Methods: *Data analysis. Ridge Regression*). The logistic regressions comprised the DV as the only regressor. Importantly, the DV was not normalised so that the logistic fit would retain the original range of the DV. We fitted two variants of the DV. We fitted all decision types together for each of the DV; this ensured that the beta weights were comparable across decision types for the same DV (but not between DVs). (A,B,C,D) We fitted the absolute DV, defined as the performance difference between relevant players plus the non-social bonus (see Methods section for study 1: *fMRI experiment. Decision phase*), to participants' choices. Panels A, B, C show the fitted logistic function and panel D shows the beta weights for the absolute DV, which captures the slope of the logistic functions in A, B, C. Comparing the beta weights for the absolute DVs using a one-way ANOVA, we found a significant main effect of decision type ($F_{2,110} = 16.373$, $p < 0.001$). The effect of the absolute DV was strongest in self decisions (vs partner: $t_{55} = 2.75$, $p = 0.008$; vs group: $t_{55} = 5.77$, $p < 0.001$) followed by the partner condition (vs group: $t_{55} = 2.96$, $p = 0.005$). (E,F,G,H) Same as in A, B, C, D, but shown for the relative DV. The relative DV captures the percent difference in the performance of the relevant players including the bonus. While the absolute DV is well suited to compare self and partner decisions but may not be the best way to compare these two decision types with group decisions. Group decisions contrast the combined performances of both teams and hence the quantities to compare are much bigger than the quantities that are compared in dyadic decisions. Therefore, it may be fairer to compare decision accuracy based on the relative DV, the percentage difference between the two performance scores of the teams. For example, while the decision between self and opponent may be between a performance score of 4 and 5, a decision between the groups may amount to a decision between 9 and 10. While the absolute DV is the same in both cases ($=1$), the relative DV considers that 9 and 10 are closer together in relative terms than 4 and 5. We calculated the relative DV by dividing the absolute DV by the mean of the two relevant players in self and partner trials (e.g., for self decisions: $\text{mean}[\text{self}, \text{relevant opponent}]$), and by dividing it by the mean of the sums of both teams in group trials (for group decisions: $\text{mean}[\text{self}+\text{partner}, \text{opponent1}+\text{opponent2}]$). Considering the relative DV revealed again a main effect in the corresponding one-way ANOVA ($F_{2,110} = 78.30$, $p < 0.001$). Participants were more sensitive to the percentage difference in the performance of groups than in the performance of individuals (vs self: $t_{55} = 8.52$, $p < 0.001$; vs partner: $t_{55} = 10.67$, $p < 0.001$). The relative DV, again, had a stronger effect in self compared to partner trials ($t_{55} = 2.93$, $p < 0.005$). ($n=56$; **, $p < 0.01$; ***, $p < 0.001$; error bars are S.E.M).

In summary, considering several measures of decision making accuracy shows that participants perform exceptionally well in group decisions. The finding that people perform better in self compared to partner decisions is also stable across several accuracy modalities. While it is not trivial to find the most appropriate measure to compare decision accuracy between group and dyadic trials, most of the measures indicate that people perform the same, if not better, in group trials compared to dyadic ones. This is difficult to reconcile with an agent-centric perspective that assumes that the error in decision making accumulates the more players are considered in the choice. However, it easily follows from our basis function model that suggests that holding in memory the group DV is a critical component to maintain a compressed representation of all four players. We now acknowledge this explicitly in the text and add a reference to the supplementary figures:

Clearly, behavior and neural signals revealed signatures of agent-centric representations. Participants did not rate their own performances as better than their partner's but they did **rate their own performances** more accurately ($t_{55}=2.094$, $p=0.041$; Fig.1C) **than the partner's performances and they made these ratings more quickly than they did for the partner** ($t_{55}=5.299$, $p<0.001$; Fig.1D). This was true even though self and partner exhibited comparable performance, were equally often decision-relevant, and had the same influence on reward payoff. **We replicated both effects in a follow-up behavioural study (study 2; accuracy: $t_{794}=4.369$, $p<0.001$; reaction time: $t_{794}=10.447$, $p < 0.001$; Fig.1C,D).**

Further decision analyses suggested that participants performed similarly well, if not better, in group decisions compared to dyadic ones (Supplementary Fig.3,4). This is difficult to explain for agent-centric decision models that assume that decision errors accumulate the more elements are to be considered. However, it is consistent with our model that assumes that decisions are guided by combinatorial representations such as the overall difference in performance between groups.

Lastly, we supplement detailed GLM analyses that compare how people weight decision-relevant information in the first compared to the second decision. We show this data for all three decision types (self, partner, and group decisions).

Supplementary Fig. 14. Decision GLM for all decision types binned by decision number (study 1). We estimated the decision effect sizes for all players and the non-social bonus for all decision types (self, partner, group decisions). To this end, we fitted a logistic regression to the choice data using ridge regression to account for outlying data points. We predicted the choice to engage (1) versus avoid (0). Positive effect sizes of the decisions weights mean a positive relationship between the variable and the likelihood of making an engage choice. We followed the same procedure as described in the Methods section regarding the online experiment (see Methods: *Data analysis. Ridge Regression*). All regressors were normalised (mean of zero, standard deviation of 1). Note that we included decision 1 and decision 2 in the same ridge regression to ensure that beta weights were comparable between the two decisions. However, we fitted each decision type (self, partner, group decisions) separately to estimate the best-fitting model per decision condition. Note that some similar analyses are reported elsewhere in this manuscript and these analyses focused especially on changes in decision weights across decision 1 and decision 2. **(A) Self decisions.** Decision weights for Self-performance (S), Partner-performance

(P), relevant opponent performance (Or), irrelevant opponent performance (Oi) and the non-social bonus (B). We ran a 5 [variable: S,P,Or,Oi,B] x 2 [decision number: 1,2] repeated measures ANOVA on the decision weights and found a main effect of variable ($F_{4,220} = 195.7, p < 0.001$) and an interaction ($F_{4,220} = 5.18, p < 0.001$). Participants based their engage decisions on the relevant information in self decisions. They were more likely to judge themselves as better than Or if they performed well (positive effect size of S), Or performed badly (negative effect size of Or), and the bonus was high (positive effect size). The interaction indicated that decision weights trended more towards zero for the second compared to the first decision. This is consistent with the decreased decision accuracy in the second decision reported in Supplementary Fig.3. **(B) Partner decisions.** The results in partner decisions were similar, except that here P was relevant instead of S and hence the decision weights of P were big and the weights of S were close to zero. We ran a 5 [variable: S,P,Or,Oi,B] x 2 [decision number: 1,2] repeated measures ANOVA on the decision weights and found a main effect of variable ($F_{4,220} = 154.3, p < 0.001$), a main effect of decision number ($F_{1,55} = 5.146, p = 0.027$). Again, similar changes between decision 1 and decision 2 were visible consistent with a decrease in accuracy. **(C) Group decisions.** In the group decision, all players are relevant. We ran a 5 [variable: Self (S), Partner (P), Opponent 1 (O1), opponent 2 (O2), bonus (B)] x 2 [decision number: 1,2] repeated measures ANOVA on the decision weights and found a main effect of variable ($F_{4,220} = 205.7, p < 0.001$), a main effect of decision number ($F_{1,55} = 4.856, p = 0.032$). In group decisions, all players are relevant and as expected decision weights for S and P were positive, and negative for O1 and O2. The trend towards decision weights that were closer to zero in the second decision was less clear in group decisions, consistent with the observation that decision accuracy in group decisions does not decrease across decisions (see Supplementary Fig.3). In addition to these analyses, in group decisions, we analysed whether people place more weight on information related to themselves (S) compared to their partner (P). This analysis is related to our observation that participants decided more accurately and quickly in self decisions compared to partner decisions. We tested whether we could replicate the finding of increased reliance on self-relevant information in the group decisions, even though, based on the payoff scheme, S and P should be weighted exactly equally. We ran a 2 [player: S,P] x 2 [decision number: 1,2] repeated measures ANOVA and indeed found a main effect of player ($F_{1,55} = 10.292, p = 0.002$), but no main effect of decision number and no interaction (all $p > 0.68$). Participants indeed relied more on S-performance compared to P-performance when weighing up which team performed better in group decisions. ($n=56$; **, $p < 0.005$; error bars are S.E.M).

5.3) Behavioural results cannot be explained by pre-experiment performance differences (R3|5)

Reviewer 3 raises another interesting point regarding the behavioural results, specifically in relation to the pre-experiment: “5. Is it possible that participants did better in the Self condition mostly because they inferred the performance score better for themselves during the observation score since they directly participated in these trials during pre-experiment (Figure 1C and D)? The authors might want to quantify and compare the accuracy for the participants to infer the scores for each player (including opponents in the other team). It is unclear if the behavioral effects were strongly influenced by the fact that the participants used their memorized experience from pre-experiment in the observation phase for the Self condition to improve the calculation of the scores for themselves. To put it another way, would the participants be able to infer accuracy of all players?” (R3|5)

This is an interesting point. Thanks for raising it. We are very confident that we can rule out this alternative explanation for why participants performed better and faster in self compared to partner decisions. We will first carefully explain why knowledge of one's performance levels in the pre-experiment could not help participants to perform well in the decision making experiment. In short, even though participants who performed better in the pre-task experienced more positive scores for themselves in the main experiment, we paired them with other players who had had similarly high performance levels. That led to a situation where the win-rate (the % of times that participants' own scores were higher than the opponents') was roughly 50% for all participants, irrespective of their pre-experiment performance. This meant that knowing about the pre-experiment performance could

not help participants in their self choices. Secondly, we refer to the results of our new behavioural study conducted online, in which, again, participants performed more accurately in self decision compared to partner decisions. Crucially, in this behavioural study, all participants performed the same decision making schedule irrespective of the performance in the pre-experiment. Therefore, this experiment rules out that the effect is driven by knowledge about the performance in the pre-experiment.

First, regarding the fMRI experiment and the relationship between pre-experiment performance and decision making in the main task: We have summarized our argument in the below supplementary figure that we have added to the manuscript. It clearly shows that the rate of successful pre-experiment performance was unrelated to the rate of winning over the other team in the main experiment. Therefore, insight into self-performance from the pre-task could not have helped participants to decide more accurately for themselves compared to the partner. Also note that even though we based the experimental schedule on the veridical performance of the participants, we scrambled the order in which the performance scores were shown. So, if a participant, for example, was bad in the pre-experiment early on and good later, then this did not mean that they received low self-performance scores early in the main task and better ones later on.

Supplementary Fig. 2. Pre-experimental performance was unrelated to winning over the other team in main social fMRI experiment (study 1). As explained in the methods section for study 1 (sub section: *Experimental procedures* and *Behavioural pre-experiment. Performance assessment*), participants' decisions in the social fMRI experiment were about performance scores from themselves and three other players. Specifically, they decided whether their own and a partner's performance scores were better or worse than the scores from two opponents. These performance scores reflected veridical performance in a random-dot-motion (RDM) task from the participants themselves and three other participants who had performed the RDM task previously. However, the experimental schedule of the main experiment was carefully constructed and ensured that the schedules were comparable across participants, balanced, and finely calibrated with respect to several decision-theoretic aspects. One key consideration here was to ensure that participants could not use insight into their own pre-experimental RDM performance levels to predict whether they would perform better or worse than other players in the main experiment. The correlation plots above show correlations across participants between the pre-experiment RDM performance of the participant themselves (*Pre-Exp: Self*: the percentage of correct RDM responses) and measures related to the main task. They illustrate how we uncoupled performance in the pre-experiment from the rate of winning over the other team in the main experiment. **(A)** As the performance scores for participants were taken from the pre-experiment, there was a near 1 correlation between participants' own RDM performance pre-experiment and the performance scores for Self (*Exp: Self*; see insets for Pearson correlations and p-values). Experimental performance scores

are shown as the percentage of correct RDM scores. **(B,C,D)** However, we carefully selected which other participants the Self would be paired with and, therefore, what other participants' performances in the main experiment were. This resulted in a positive correlation between the participants' own pre-experimental performance scores and the scores of the other players in the main experiment (P, O1, O2 refer to the percentage correct RDM scores of the partner, opponent1, and opponent2). Again, all these correlations are highly significant. In sum, this means that participants who performed well in the pre-experiment were paired with other high-performing players. **(E)** The result of this was that we ensured that participant's win-rate in self decisions (*Exp: Self-win rate*, i.e. how often it was correct to indicate that oneself was the higher performer) was always around 50%. In other words, roughly half of the time the correct response to self decisions was to indicate that oneself performed better, and half of the time the correct response was to indicate that the relevant opponent performed better. A win-rate of roughly 50% was desirable, because it prevented floor/ceiling effects in participants' choice behaviour and ensured that choice analyses were sensitive to a broad range of potentially relevant behavioural variables. This also meant that there was no correlation between participants' pre-experimental RDM performance and their chance of winning over the opponent in self decisions ($r = 0.044$; $p = 0.74$). **(B)** The same was true for the relationship between pre-experimental RDM performance and winning in group decisions (*Exp: Group-win rate*; $r = -0.039$; $p = 0.77$). Note that partner decisions are not shown, as they were per definition independent of participants pre-experimental performance. ($n=56$).

In summary, knowing one's RDM performance levels from the pre-experiment cannot be used to infer one's chance of winning in the main experiment. Hence it is not helpful for participants to infer the likely correct decision in self trials more than partner trials. This is one key reason why we argue the faster and more accurate responses for self compared to partner reflect a psychological effect, a "self-bias". Another argument to support this conclusion comes from the new behavioural study that we conducted since our last submission. In that study, participants performed the same type of pre-experiment before going on to the main decision making task. However, in that study they were told very little about the relationship between pre-experiment and main task. They were only informed that it was important to do the pre-task, that their performance would be recorded and that they would see their performance in the main task. Indeed, they observed the positive and negative performance cues in the main task that they knew from the pre-experiment. However, the exact sequence of performance scores that was shown for each player reflected a pre-determined schedule that was derived from our fmri participants' schedules. This was necessary, because the behavioural study utilised a between-subject design that tested the effects on very specific schedule differences on decision making. However, this procedure meant that the performance in the pre-experiment was unrelated to the performance scores that were observed for oneself in the main experiment. Therefore, any insight into the performance in the pre-experiment was not useful to infer one's approximate level of performance scores in the main experiment. Despite that, we nevertheless observed the same self-bias in accuracy and reaction times. We present this result now prominently in figure 1 and in the corresponding part of the main text:

Main text:

Clearly, behavior and neural signals revealed signatures of agent-centric representations. Participants did not rate their own performances as better than their partner's but they did rate their own performances more accurately ($t_{55}=2.094$, $p=0.041$; Fig.1C) **than the partner's performances and they made these ratings more quickly than they did for the partner** ($t_{55}=5.299$, $p<0.001$; Fig.1D). This was true even though self and partner exhibited comparable performance, were equally often decision-relevant, and had the same influence on reward payoff. **We replicated both effects in a follow-up behavioural study (study 2; accuracy: $t_{794}=4.369$, $p<0.001$; reaction time: $t_{794}=10.447$, $p < 0.001$; Fig.1C,D).**

Figure and legend:

Figure 1. Group decision task and agent-centric representations... (C,D) Choice accuracy and reaction times. Participants were more correct in Self decisions compared to Partner decisions while simultaneously responding quicker. We took the median reaction time per participant as reaction time in the online data were skewed. The left-hand side shows the results from our fMRI experiment and the right-hand side shows the replication of these effects in an independent behavioural experiment (study 1 and study 2). ... (n=56 for social fMRI experiment [study 1]; N=795 for behavioural experiment [study 2] all MRI results cluster-corrected at $Z > 3.1$, $p = 0.05$; *, $p < 0.05$; ***, $p < 0.001$).

6) Conceptual clarifications and questions about framing (R1,R3,R4)

Several reviewers raised a couple of questions about the framing and questions about conceptual clarifications. We respond to them here, in the order of the reviewers.

6.1) Reviewer 1 (R1|1)

We have discussed in detail why we argue that our basis function model has a higher explanatory power than simpler models. However we would like to clarify one more aspect regarding the reviewer's first point "1) First, while the authors provide a number of elegant analyses which suggest that basis functions could explain the data, it wasn't clear to me whether or to what degree are their models have higher explanatory power than other simpler functions. It would be helpful for example to add formal information theoretic metrics such as AIC or BIC. Further, while they made important comparisons to other models such as a drift diffusion model, it would be helpful to provide further comparisons with other models or more basic analytic techniques to demonstrate that their results are optimally explained by their functions." (R1|1)

We have conducted the additional behavioural experiment showing compression, and we have conducted many additional analyses including one that tests information-theoretic metrics comparing our basis function model with simpler models. However, we also want to emphasize that our key neural measure of the basis functions controls for agent-centric representations by including them in the same analysis. Therefore, all neural signatures of basis functions explain variance in the neural signal beyond mere agent-centric representations. We explain that more clearly in the main text now. We also want to note that we do not claim that agent-centric signals do not exist. By contrast, we report agent-centric representations prominently in Fig.1E.

In the revised manuscript we, therefore, try to clarify that our contention is that while the basis function model is the new model and provides the best account of pgACC activity, both of the models provide explanations of different aspects of neural activity and in combination they can explain behaviour. In particular, dmPFC appears to encode social information about the partner but also the opponents, but it does so in the absence of basis function-related activity (see elaboration below presented in response to point 5.2 and 5.4 of this reviewer’s comments). Moreover, do note that in our non-social fMRI control study (presented in response to the reviewer’s next point) we show that, without a social framing of the task, agent-centric signatures of brain activity disappear, while activity related to non-socially anchored basis functions is still significantly measurable but with the anatomical centres of their most prominent effects in adjacent subsections of medial prefrontal cortex. This, in addition to the previous arguments, also rules out that basis function related activity can in a simple way be reduced to agent-centric representations.

We have implemented the outlined changes in the following way in the manuscript. The new behavioural study is included in the main text. We made textual changes in the main text addressing the explanatory power of basis functions in behaviour (the information theoretic metric) and fMRI analysis. We added supplementary figures relating to the information theoretic metrics as well as details of the behavioural study in methods and supplements (see previous points). In addition, we have clarified the legend of the corresponding figure 3:

(A) Correlation matrix summarizing the fMRI analysis approach used. Our model included both agent-centric regressors (S, P, O1, O2 refer to performance scores linked to specific players) and the sequential basis functions (b_1 , b_2 , b_3) as well as S-position. **The basis function related regressors therefore captured variance in the data that was not explained by agent-centric regressors.**

6.2) Reviewer 3 (R3|2, R3|9)

2. Line 140: Intuitively, the reason why only 3 dimensions were needed was because the frame of reference for the player sequence was 3 (4 minus 1, once the participants know the identities of the first three positions, they would know the fourth one for certain). Therefore, it is not very surprising that DVs are encoded in a 3D space. For example, if there were 6 players, the dimension would need be 5. Would the number of dimensions simply depend on the task, or is there something special about having 3 dimensions? (R3|2)

This is a good point. Yes, indeed the number of dimensions should depend on the task. Our task comprises four players, but the DVs can be constructed using the three basis functions only. Depending on the decisions that people face, more or fewer basis functions might be needed.

Our argument is that people represent the information they observe about the players in a format that prepares them for the decisions they encounter later in the second phase of the trial. If the set of decisions they encounter changes, so should the set of basis functions. Our argument is therefore that the basis functions that are used by participants reflect the combinatorial possibilities encountered in the environment. This is analogous to the way in which the basis functions in the visual system reflect the statistics of the visual environment and basis functions in the motor system reflect the task in hand and its statistical structure. We now clarify this in the part of the main text highlighted by the reviewer:

Critically, all sequential DVs lay within a 3-dimensional subspace of the 4-dimensional sequential performance space (Fig.2E). **This was a consequence of the team structure and the decisions of our task.** The subspace was defined by three basis functions. The four sequential performance scores could be projected onto the three basis functions via a linear combination (dot product) of the performance scores and the basis functions. For example, one basis function, $w_2 = [1 \ 1 \ -1 \ -1]$, weights the first two performances positively and the last two negatively and combining w_2 with the performance sequence results in the performance difference between these pairs of positions (Fig.2F). The subspace was independent of a “null axis”¹¹ that reflected variation in performance scores that was irrelevant for the task and not useful for responding to any decision that could be encountered. **The null axis reflected the sum of performances across players. Knowledge about the absolute sum was irrelevant in our task as all decisions involved relative comparisons between a member or members of one team versus the other.** The three comparison vectors for the sequential group decisions (Fig.2F) form an ideal set of basis functions for the transformation into the subspace: Pairwise, linear combinations of these vectors, which specify all possible dyadic decision weight vectors, when combined with the sequential performance scores, can guide any decisions about dyadic interactions (Fig.2G). Just one of the three basis function is sufficient, when combined with the sequential performance scores, to guide decisions about any groupwise interaction that may occur.

9. In relation to understanding Figure 3C, please explain explicitly what the null axis is. (R3|9)

The null axis relates to information about the observed performance sequence that was irrelevant for the task and not useful for responding to any decision that could be encountered. The null axis is orthogonal to the basis functions (just as the basis functions themselves are orthogonal to each other). In our task, the null axis is the sum of the performances of all players. Because each decision requires a comparison of performances between players, knowledge about the sum across all players is not needed. Consistent with the idea that medial prefrontal cortex encodes task-relevant but not task-irrelevant axes, we identify signals related to all three basis functions in this brain region, but not signals related to the null axis. We previously explained that the null axis amounts to sum of performance scores in the methods section. We now explain this more explicitly in the main text as well:

The subspace was independent of a “null axis”¹¹ that reflected variation in performance scores that was irrelevant for the task and not useful for responding to any decision that could be encountered. **The null axis reflected the sum of performances across players. Knowledge about the absolute sum was irrelevant in our task as all decisions involved relative comparisons between a member or members of one team versus the other.**

6.3) Reviewer 4 (R4|1, R4|2, R4|O1, R4|O4, R4|O8, R4|O9)

1. The authors say multiple times that they think the brain is encoding basis functions related to the task at hand, and that this offers a way to “summarize possible social interactions in a compressed format.” I’m not entirely sure I understand what they mean by compression here. To encode the outcomes of four agents, one would need a 4d space. For their task, they need a 3d space. Is the 4d->3d transformation what they mean by compression? If so, in pure efficiency

terms, it doesn't seem like much is gained by such compression. Moreover, in a real-life social situation, one doesn't generally know what decision one will be asked to make, and hence the compression strategy here doesn't seem like it would apply in general—raising doubts about the significance of the findings outside this experiment. (R4|1)

Note that we have addressed the point about compression in our very first response in this rebuttal. Now, we would like to address the reviewer's second point about generalisability to the real world and then discuss some additional points re compression raised by the reviewer..

We disagree with the assessment that a compression strategy such as the one we identified would not work in the real world, because **“in a real-life social situation, one doesn't generally know what decision one will be asked to make”**. In our experiment, just as in real-life situations, participants do not know in advance which decision they will be asked to make next. They observe 4 player's performances and are subsequently asked to compare some of them. At the beginning of each trial they do not know in advance which players they will be asked to compare and this is precisely the reason why they need to keep information about all players in memory – something which we claim they do efficiently via compression.

Of course, there are constraints on the social interactions that are possible in our experiment (only members of opposing teams are compared). However, a great many social situations have features that are very similar to these. For example, many games and sports contain an element that is similar to the interactions in our behavioural task. This is apparent when two football teams compete but also in the way in which in certain stages of the game two individuals drawn from each team might compete. Such individual contests within the context of a group competition are codified more explicitly in other games such as cricket and baseball when a bowler/ pitcher confronts a batter, drawn from each of teams, in certain phases of the games. This is approximately what happens in our task. Sometimes one whole team competes with the other but sometimes two individuals, each drawn from a different team, compete with one another.

Our experiment not only captures the way in which **“one doesn't generally know what decision one will be asked to make”**, but it also reflects the way in which our social interactions are often structured based on the cooperative and competitive relationships that exist in the social world; players of opposing teams are much more likely to compete for possession of the ball than players from the same team. We argue that to act adaptively and precisely in these sorts of situations, basis functions like the ones we identified are used by the brain to flexibly represent the social information that is potentially relevant, and, later-on, the information that is needed to make a specific social decision. Therefore, our basis function approach generalises well to situations in which one does not generally know exactly which decision one will be asked to make next.

While games provide useful examples, the same is true in less explicitly codified situations. For example, a cognitive neuroscientist over the last week is likely to have been part of an analogous set of social alliances that will remain constant in one setting but shift between contexts. For example, if you have been conducting an experiment with one or more collaborators in your laboratory and have been reviewing different ideas about how to proceed in the next step then you may have been reviewing the evidence put forward for one approach by one or two individuals with for example, especially strong data analysis skills, and weighing it against evidence put forward by one or two individuals with especially strong data collection skills. What is at stake, the decision to take – how to proceed next -- is clear even if there is disagreement about which of the choices should be made. The individuals in this scenario will reassemble in different ways entailing different patterns of alliance and opposition as they, for example, go with you to an ethical review board to explain the benefits

and costs of an experimental approach, prepare a class for students, or prepare a paper for publication. In other words, social relationships change dynamically and our basis function model proposes a compressed representation of a social group that allows a decision-maker to act adaptively in a variety of decision situations that might ensue.

We now clarify how our approach translates to real-life situations in the following way:

We devised a multi-person social decision-making task to test whether the brain employs a combinatorial or agent-centric frame of reference to guide social decisions (Fig.1; Supplementary Fig.1). To this end, on each trial of the experiment, we presented information about four players in randomized orders that participants subsequently made decisions about. **The scenario approximately corresponds to real life situations in which it is clear that several people are potentially relevant but exactly which decision will need to be made about which subset of individuals only becomes clear later. For example, like in a game of football where players from different teams are likely to compete, but it is unclear which individuals have to engage with one another until a certain situation plays out**

Second, we discuss the related point about compression. The compression from a 4D to a 3D format is indeed what we mean by compression. In some perceptual domains such as, for example, face processing, it is the case that if representation occurs in a compressed manner, perhaps through the use of a limited basis set, then one would expect the same form of neural representation to be present whenever a face is perceived. However, in the motor domain the code will not be constant but adapted to the task in hand and indeed, empirically, this is what has been found (Hennig et al., Nature Neuroscience, 2021; Sadtler et al., Nature 2014; Shima et al., Nature, 2007). Arguably basis functions in the social cognitive domain should be expected to be similar to those in the motor domain and suited to the task in hand.

Perhaps another way of thinking about this point is that the task in hand during face perception remains constant and does not change. The prediction, however, is that were individuals to be exposed, in separate periods, to two different sets of face stimuli then the basis sets most relevant to representing each group of face stimuli would become prominent during the relevant testing period.

In response to the reviewer's comments we acknowledge in the revised manuscript that the efficiency of the data compression in the current experimental setting may not be high but we explain how the key observations are that it happens, that it happens automatically, that it is associated with distinctive and reliable effects on behaviour that cannot be explained in other ways, its impact on behaviour is to make it more accurate even if it leaves a tell-tale record in aspects of behaviour, and that it is associated with reliable neural activity patterns. To do this, we now present an additional online experiment, partly inspired by the reviewer's comments in which we show how contextual factors determine when and how such a basis set emerges (see point 1 of this rebuttal).

2. The authors pit their finding against the intuitive notion that the brain tags performance of social agents by the agent's identity separately for each individual agent, i.e., the brain is only representing the performance of the agents along the rotated axes (b1,b2,b3 in Figure 2E) and not agent-centric axes (S, P, O1, O2). But as far as I could tell, nowhere in the paper is it shown that when one simply regresses to the performance of the four individual agents (including self), the beta value is 0. Indeed, I would find this deeply surprising, since it seems that behaviorally, subjects should be able to answer re how each individual agent performed.

The only indirect piece of evidence I saw related to this is the finding that $\text{Beta} = 0$ for the null direction (Figure 3C). I don't think it is possible to conclude that the brain is not coding the performance of individual agents based on this single indirect negative result. On page 9, the authors state, "none of these agent-centric or decision-centric variables shared more than 0.01 of

variance with the basis function projections and all neural analysis controlled for these related variables (Methods)” and later on page 11 they state, “Again, this is striking because these variables encode sequential information and are unrelated to agent-centric or decision related variables, which we again control for in the analysis (Fig. 2E, Supplementary Fig. 2, Methods).” However, I was unable to glean from the Methods how this was controlled. (R4|2)

We take care to avoid claiming that there is no representation of individual agents. Indeed, there is abundant evidence for representations of individuals. Our claim is that there is an additional representation that relates to the patterns of interaction that can occur between individuals. We note that this argument was clear to the other reviewers but accept that the reviewer’s comments suggest the need for clarification in the revised manuscript. In the revised manuscript we emphasize that analysis of the univariate BOLD activity reveals identity-linked neural activity relating to either the self or to others (figure 1e in the previous manuscript). In addition, we make clear that multivariate analyses have been used to show both in our laboratory but more frequently and prominently in many other laboratories that individual identities can be decoded from face processing regions. We explain that our argument is not that these do not exist but that they are not sufficient to represent the interaction patterns between individuals.

These codes can be thought of as complementing a suite of temporal lobe brain structures enabling primates to process visual information to recognize faces and to retrieve individual-specific identity information^{11,40}. **Representations of faces and individuals are essential for social cognition. However, such representations in isolation are insufficient to encode the relationships between individuals and it is knowledge both of the individuals and of their interrelationships that is needed to guide adaptive social decisions.** Here we suggest that representations of the social environment in the form of a set of basis functions define the structure of a ‘space’ for social interaction.

More fundamentally: within a given trial, the rotated axis (b1, b2, b3) is just a linear combination of the agent-centric axes (S, P, O1, O2). fMRI is simply not an appropriate technique to address what particular basis, amongst different bases related by a linear transform, an area is using, given that it is summing across many, many different cells. To take a simpler example, if an area were coding color, and one found evidence for modulation by red, green, and blue, this would certainly not be an argument that the area does not code yellow (red+green), cyan (green+blue), and magenta (red+blue). Indeed, fMRI would be simply unable to pinpoint whether individual cells were coding RGB or CMY, since this is a question about single cell coding mechanism. Furthermore, fMRI is terrible for making conclusions re what the brain is *not* coding, since finding an fMRI signal depends on many cells having clustered tuning functions that agree in sign, which is a very strong assumption. So I just don’t understand how the authors can be so certain, based on their results, that the brain is not coding the performance of individual agents, but is instead using an alternative basis.

If the claim of the authors is only that the brain is representing social information along task-specific basis functions *in addition* to agent-specific basis functions, this would be a much weaker claim. And it seems that this has to be true, given that performance of the group decision task *requires* computation of b1,b2,b3. So then, it seems the main value is in pinpointing where this representation of b1,b2,b3 occurs and demonstrating that it can be isolated with fMRI. While valuable, I think this is not a major conceptual advance.

We understand that the reviewer is raising an important issue here that goes to the heart of our manuscript’s claims. However, crucially the reviewer’s opening statement that the rotated axis (bi, b2, b3) is a linear combination of agent-centric axes (S, P, O1, O2) is not correct. The basis functions (b1,b2,b3) are defined with respect to the sequence of performance scores that have been observed in the initial phase of a trial. Clearly, knowing the scores for S,P,O1,O2 is not enough information to

know what the score is for the player that was seen in, say, the second position of the sequence. We emphasize that a key manipulation that allows us to dissociate between the effect of sequential position and the effect of player identity is that player's position in the sequence is balanced across trials. We make this clear very prominently in Fig.2A to really bring across that our experiment enables us to dissociate player position and player identity.

If we had not done this important manipulation, then the reviewer would have been correct that the basis functions (b_1, b_2, b_3) are a linear transformation of S, P, O_1, O_2 . However, this is precisely why we balanced players across sequential positions. It enabled us to use a GLM approach controlling for players scores and uncovering the effects of the sequence-linked basis functions. We note that the other reviewers agreed with this basic logic of our approach. In summary, the distribution of agent identities along each basis is controlled across trials making it unambiguous that the activity patterns we report in a medial frontal cortex are unambiguously related to player performance bases in relation to the basis sets proposed rather than agent identities. We make this statement clearly in the revised manuscript.

Legend Fig.2H:

(H) The basis function projections are unrelated to agent-centric and decision-related variables. Correlation plots of b_1, b_2, b_3 (rows) with agent-centric and decision-related (columns) variables. Data is collapsed over all participants. Lightness indicates number of data points. Team difference refers to both teams' performances when including all members of one's own team versus the other team regardless of their relevance for the current decision. The other variables are explained in detail in the main text. Note the absence of any meaningful relationship between the basis projections and the agent-centric and decision-related variables. This is what allowed us to capture independent components of the BOLD signal by using corresponding parametric regressors in the fMRI analyses. This is what allowed us in the fMRI analyses to capture independent components of the BOLD signal by using corresponding parametric regressors. Therefore, some of the activity clusters shown below can be partly overlapping but still capture independent components of the fMRI signal (all data from study 1).

Main text:

Using fMRI, we interrogated the brain for evidence of a sequential neural code in our task. As described above, such a code can be useful to compress social information and prepare a decision-maker for possible decisions. **The fact that our design allowed us to statistically disentangle sequential and agent-centric signals was a necessary prerequisite for this analysis.**

Like the other three reviewers we do not think that the existence of these basis functions for describing social interactions is trivial just because there is also a representation of the identities of individual agents. For example, nothing in the many decades of publication of face-related neurons even begins to address how the owners of those faces might be interacting. Just as our current findings do not diminish demonstrations of identity encoding in face areas, so is the converse true.

These codes can be thought of as complementing a suite of temporal lobe brain structures enabling primates to process visual information to recognize faces and to retrieve individual-specific identity information^{11,40}. **Representations of faces and individuals are essential for social cognition. However, such representations in isolation are insufficient to encode the relationships between individuals and it**

is knowledge both of the individuals and of their interrelationships that is needed to guide adaptive social decisions. Here we suggest that representations of the social environment in the form of a set of basis functions define the structure of a 'space' for social interaction.

Finally, we note that in response to the reviewer's point Other-2 below we have performed an additional fMRI experiment further demonstrating the social specificity of the effects we observe which we hope might address some of the reviewer's concerns.

Other

1. I don't understand the social component in the setup of the task at all. Why is it necessary for actual people to be play the role of "P, O1, O2" It seems the performances of the different agents could have been computer generated without changing the results of the paper. (R4|O1)

Again, we already responded to core aspects about the social nature of our task above in the second major point of this rebuttal.

The reviewer may potentially be correct, the social context in which the task is conducted may have been principally induced by: 1) the cover story; 2) the manner in which the participants themselves experienced really performing the dot motion task prior to making competitive social decisions about task while in the fMRI experiment; 3) the visual presentation of performance scores from other agents that were presented similarly to one's own performance scores. There are, however, some scientists often from a behavioural economics background who argue that it is important that the stimuli used are ones that were truly generated by other people if that is what participants are told was the case. By using real participant performances in this way we able to comply with this convention and avoid misleading our participants in any way. While the degree to which we would have misled our participants had we acted otherwise might be trivial, it is, by convention, not something that is done without strong justification in human experiments.

3. The task design seems incredibly convoluted and badly presented, introducing a possibility that the results are influenced by severe confounds unrelated to the questions asked. For example, the task contained a bonus component in each trial, which was not described in the figures or the main text, where the counted number of correct trials had to be adjusted with a number shown on the screen. This number had to be added to the own team tally if positive or to the opponent team tally if negative. However, the participants could've easily employed a more intuitive strategy of only mentally manipulating the tally of one of the groups, for example own group, and adding positive and subtracting negative numbers. Any contrast might then reveal broad neural signatures of differing mental processes of manipulating numbers rather than any kind of social decisions. (R4|3)

First, we note that the three other reviewers found our description of the task clear as have many audiences when we have presented our work. Nevertheless, we must concede that the reviewer has misunderstood some parts of our manuscript and we have revised it in the hope of avoiding similar misunderstanding in the future. Perhaps most critically, we hope that the inclusion of the new experiment described in response to the major point 2 above allows us to directly address the reviewer's concern that the task is simply a matter of manipulating numbers rather than representing social interactions. We were able to find socially specific behavioural and neural signatures in our social task that were both absent in the motor fMRI task.

On the left the figure below shows the social agent specific representations in the original social study. When the same areas were interrogated in the motor task, there were no corresponding effects in the motor task (panels F and G shown to the right below):

The bias for making decisions more accurately and quickly when they involved the self was not found for the formally identical decisions that were matched for their numerical features in the motor task:

We also made the following textual changes for clarity:

The presented information comprised abstract performance scores that ranged between 0 and 6 for each player. **To pre-empt the use of simple heuristics** we presented performance scores as brief sequences of six binary cues per player (indicating either successful performance or erroneous performance) that participants aggregated. The performance scores reflected veridical task performances by each player and were recorded prior to scanning in an arbitrary two-alternative forced choice task (Fig.1A,B; Supplementary Fig.1; **note that the task also comprised a non-social bonus that we controlled for in all analyses**). **Note that the experimental schedule was constructed from the pre-recorded performance using a procedure that ensured that insight into one's pre-recorded performance was not helpful for correctly responding in the main experiment (Supplementary Fig.2).**

Furthermore, inexplicably, the entire structure of the task differed for the trials where own team was judged to perform better than the opponent one. Trials where the opponent team performed better had to be avoided and were not rewarded in any case whereas participants had to select to engage and were rewarded for the trials where their own team was better, possibly biasing fMRI data with activity irrelevant to the question posed but rather coding for task peculiarities. What is the reason for such large variations in the task, especially when the task was not interactive at all?

This is incorrect. The task structure did not differ for trials where one's own teams was judged to perform better than the opponent one. The task structure was identical for both engage and avoid trials – precisely as the reviewer suggests it should be. We have made this clearer in the revised manuscript.

However, rewards could only be obtained if participants detected that their group was better and made a decision to “engage” in the trial. By contrast, losses on trials with better opponent-performances could be avoided by making the decision to “avoid” the trial. **Note that “engage” and**

“avoid” were simply labels for the binary decisions that participants made – participants progressed through the experiment in the same manner independent of which choice they made.

4. I am confused by the task design shown in Figure 1B. The authors point out in the figure legend that the numbers corresponding to performance were not shown directly, but instead subjects had to “aggregate” the performance. Why is this necessary? (R4|O4)

We are not certain that it is necessary but we felt that if this were to happen in a fast manner then participants would have an approximate sense of player’s performance levels rather than a precise numerical score. This is ensured by using aggregates of performances that exceed the range of subitizing for most participants. In other words, this helps us avoid what the reviewer senses might be a concern were the task to become simply an addition and calculation task. We have expressed this in the main manuscript as follows:

To pre-empt the use of simple heuristics we presented performance scores as brief sequences of six binary cues per player (indicating either successful performance or erroneous performance) that participants aggregated.

7. Please discuss the discrepancies in the fMRI data shown. The activations in Figure 1E seem non-overlapping with activations in Figures 3B and 3H to an unexpectedly large degree. Am I correct in assuming that regardless of analysis type, the brain should still be performing computations revealed in Figure 1E in Figure 3B/3H? In addition to that, the authors could discuss more in depth the broad activations seen in Figure 3H. (R4|7)

In the revised manuscript we have explained that the activity in figure 1 should vary as a function of the performance levels associated with either the self or another player but that activity patterns shown in figure 3 should not be expected to overlap. The activations shown in the two figures are fundamentally different from one another and they relate to different times in the trial. As explained above, they are not simply a linear function of one another. We hope that the revisions made in response to the reviewer’s points above sets the scene for this analysis more clearly. In addition, in the revised manuscript we note explicitly how the activation should be expected to be different by stating:

Using fMRI, we interrogated the brain for evidence of a sequential neural code in our task. As described above, such a code can be useful to compress social information and prepare a decision-maker for possible decisions. The fact that our design allowed us to statistically disentangle sequential and agent-centric signals was a necessary prerequisite for this analysis. We focused our analyses on the end of the performance observation phase (note that Fig.1E above related to a different phase of the trial) and searched for neural correlates of the projections onto the basis functions (as in all subsequent analyses, we correlated the neural signals with the trialwise variation in the projections onto the basis functions). Critically, we performed this analysis while simultaneously controlling for agent-centric representations in the same general linear model (Fig.3A).

8. Figure 3E seems unnecessary; this seems obvious. (R4|O8)

Our understanding is that this has been at the heart of misunderstandings between the reviewer and the authors in relation to the several of the reviewer’s points. While we hope we have shown that we have taken the reviewer’s points seriously, we think that this figure is best retained. We have, however, tried to set in context more clearly by making a number of changes to the legend, and by placing the figure earlier in the manuscript to foreshadow the rationale of the fMRI analyses (now Fig.2H)

(H) The basis function projections are unrelated to agent-centric and decision-related variables. Correlation plots of b_1 , b_2 , b_3 (rows) with agent-centric and decision-related (columns) variables. Data is collapsed over all participants. Lightness indicates number of data points. Team difference refers to both teams' performances when including all members of one's own team versus the other team regardless of their relevance for the current decision. The other variables are explained in detail in the main text. Note the absence of any meaningful relationship between the basis projections and the agent-centric and decision-related variables. This is what allowed us to capture independent components of the BOLD signal by using corresponding parametric regressors in the fMRI analyses. This is what allowed us in the fMRI analyses to capture independent components of the BOLD signal by using corresponding parametric regressors. Therefore, some of the activity clusters shown below can be partly overlapping but still capture independent components of the fMRI signal (all data from study 1).

9. The effects of representing in a compressed basis on choice shown in Figure 4 seem quite small. It would be beneficial to see data in Figure 4B and 4C also presented in terms of actual RT and % correct, similar to Figures 1C and 1D. Further, Figures 4E and 4F lack appropriate controls. The irrelevant player signals are very small, and their matching direction could be explained by shared task variables, etc. (see 3. above). (R4|O9)

First, to address the reviewer's concerns we have replicated the results shown in figure 4 and show that they are robust as might have been expected given their statistical significance. We have also since moved them to the supplementary material. We have not presented it in terms of RT and %correct, since this analysis is the result of a GLM analysis. Second, as highlighted multiple times now, we ensured that our task variables are independent and we show this multiple times in different figure panels such as the one present in response to the previous point of the reviewer, but also in Fig.3A (see below).

Third, even though we ensured that shared task variables cannot explain our effects, we have ruled out this possibility experimentally in the behavioural study. We used precisely the same schedule for matched Self and Partner decisions in the Group and No-Group condition of the behavioural study, and we find that the small effects in (formerly) panels E,F (now Fig.3J,K) are significantly decreased in the No-Group condition where participants are deterred from using the basis functions. Since we used the same schedule, this means that all shared task variables were precisely identical in both versions of the task and hence cannot explain why the effects of irrelevant players are present in one the Group-Condition but not in the No-group condition.

7) Minor clarification questions and typos (R1,R2,R3)

In this section, we list clarifying changes we have made to our manuscript. These comprise very straightforward and simple suggestions. We are grateful that the reviewer have raised them. We list these changes in the order of the reviewers

7.1) Reviewer 1 (R1|4, R1|5.1, R1|5.3)

The below points are listing R1|4's points, which have several sub-points. Note that we address the concern about relatively small effect sizes in our major response 3 above.

4) Fourth, it would be helpful to provide some clarifications on the various figures and statistics below. Some of the main findings for example appeared to display relatively small differences in activity and it wasn't easily apparent whether they accounted for multiple comparisons.

We have addressed each of the points that the reviewer has made below. However, we note that all the statistical results are corrected for multiple comparisons, and we have made this clear in the revised manuscript.

1. For Figure 1A: There is potentially a typo: number 2) is a different character size than 1). For 1C: this seems to be a very small difference judging by the dot distribution, and by looking at the histogram itself. Was this difference corrected for multiple comparison? If so, indicate in the main text, or in the legend. Same goes for 1D (although there the difference is a little more visible). For 1E, please clarify what the MNI coordinates for the two clusters (medial prefrontal cortex and ACC), not only the X coordinate of the slice. Also, was the cluster in dmPFC bilateral?

Thank you for examining our results and figures in such detail. There was indeed a typo. We have made the following adjustments to the figure:

- In panel 1A, we have changed the character size for 2) and now both 1) and 2) have the same font size
- In panel 1E, we have added the peak MNI coordinates for both clusters and clarified that both the dmPFC and the pgACC cluster extended bilaterally

Regarding the sizes of the effects in panels 1C/D: We report that participants decide more accurately and quickly when making decisions about themselves compared to their partner. In our experiment, being correct in either decision type is equally valuable in terms of the financial payoff that participants get from the experiment. Both self and partner decisions occur equally frequently, have a similar difficulty, and are in no way different except the psychological framing. For both self and partner, participants memorise performance scores with cues that they make decisions about. Therefore, while we agree that the reported effects are small in absolute terms, the fact that they can be measured at all in our experiment documents the social nature of our task. These effects are not smaller than the previous effects of increased accuracy and faster responding than reported in the past (e.g. Lockwood et al., 2018, Nat Commun), and they are also comparable to the ones documented in a recent review about the self-bias (Sui & Humphrys, 2015, TiCS). We now acknowledge that our hypotheses are based on this literature more explicitly in the main text. In our view, the size of these effects also translates well to our experiences in daily life, where we are able to learn and remember information about many people almost equally well, but where there is a slight advantage of learning about oneself.

Both the reaction time and the accuracy effects were predicted *a priori* and therefore not corrected for multiple comparisons. However, we have now conducted a large-scale behavioural study conducted online in which we replicated both effects in a new and independent data set. We report these results now next to the fMRI results to further demonstrate the robustness of our effects. Note that, regarding the reaction time analysis, we take the median reaction time per person in order to control for the skewness of the reaction time distribution in the online study. In addition to this replication, we also replicate the above effect conceptually within our fMRI experiment. We have now added a supplementary figure that shows that also in group decisions, when participants compare how well their team performed compared to the other team, participants relied more on performance scores related to themselves than on performance scores from their partner, even though both should be weighted equally based on the payoff scheme of the task.

Overall, we have updated the figure, the corresponding part in the main text, and the figure legend (Main text: page 4, line 96 – 103)

2. Figure 2B: Clarify that $DV = pos1 - pos3$ is just an example. It may also help to change the numbers into x and y , so it's more straightforward than $x = 1$ and $y = 3$ if you are comparing S and O-green. Also, add here a number per position in light grey, similarly to figure 2D under the arrow - so if I understand well, it should be 3,5,2,5. It'd make it easier to unpack the figure.

Thanks. This is really helpful. We implemented these suggestions and changed the figure. We also incorporated the suggestions in the reviewer's next point. We show the revised figure below.

3. Figure 2C & D: Figure 2D: maybe I am missing something, but I don't understand the order of the vectors in the space, so please either clarify it in the legend, or edit the figure. Why is it pos3, 2,1,4? It is a little confusing especially when trying to match it with the light grey example under the arrow (the order of the numbers still reflects the position order - pos1 = 3, pos2=5, pos3=2, pos5=5, but reading it in their order of appearance, it does not correspond, as it is pos3, pos2, pos1, pos4).

Thanks for raising this. We tried to make the panels more intuitive by ordering the players in the order of occurrence in the sequence. Based on the reviewer's previous suggestion, we also added light-grey numbers to panels 2C and 2D to enable the reader to match up the numbers across figure panels. The revised panels in Figure 2 addressed the two last points. We have also updated the legend to make clearer that B shows an example case. Please refer to the main document to see these changes: Figure 2.

4. Figure 3H: following up on my previous comment, I wonder if Social DV is confounded by the fact that by doing chosen-unchosen you are automatically introducing some sort of likelihood of getting reward: for instance, if the subject is to judge performances, comparing a 1 vs a 5 (difference is 4) is not the same as comparing a 3 vs a 4 (difference is 1; i.e. likelihood of getting reward is higher in the first case as it should be easier to judge the performance). If my assumption is correct, wouldn't doing chosen-unchosen be effectively a measure of value in this case, rather than an agent specific measure? Authors should better clarify their view for this in the paper, and potentially think about ways of more directly disentangling chosen value from agent specific activations.

As a reminder, this is the relevant (and now updated) figure panel:

What we want to convey when we refer to this variable "social DV" is, in fact, the difference in value that is associated with one choice rather than another. What we wanted to convey with the red region in panel 3H is that there is an activity pattern in ventromedial prefrontal cortex (vmPFC) that is

very different from that shown in dorsomedial prefrontal cortex and pgACC in panel 3B (and again in yellow in panel 3H). The pgACC activity reflects player performance information projected onto the basis functions which are referenced with respect to one's own position in the sequence but vmPFC activity reflects how much more valuable one choice is than the other. In other words, precisely as the reviewer suggests there is a need to "disentangling chosen value from agent specific activations" and this need is met by showing agent-related activations projected on basis functions in dmPFC (panel 3B, and yellow area in panel 3H) and a transformation of this information into a format specifying the relative values of the different choices for the participant performing the experiment regardless of the agent-related performances that informed the choice valuations. We also clarify the relationship between chosen value and our basis function model in the subsequent points of the reviewer below. We ran a series of ROI analyses, as suggested by the reviewer, to clarify the overlap of signals in different brain regions.

However, we acknowledge that there is a need to make this section of the manuscript clearer and on re-reading we felt that it could indeed be made clearer. In the revised manuscript we make sure that the DV that is reflected in the activity of vmPFC is a value-related DV as follows:

These signals are in strong contrast with activation in vmPFC, a very different prefrontal region robustly linked to decision making regardless of whether there is a social dimension ³⁸. **In contrast to pgACC, which reflects player performance information projected onto the basis functions which are referenced with respect to one's own position in the sequence, vmPFC activity during dyadic decisions reflects the value to the participant of taking one choice (the chosen option) as opposed to another (the unchosen option) regardless of the fact that these values originally related to the performances of different teams (Supplementary Fig.11).**

5. Figure 3F: add legend for red and black dots again.

Thanks for the suggestion. We have done that.

Furthermore, in their point 5, reviewer 1 also raises a number of clarification questions:

In their points 5|1: "5) Finally, there were a number of areas within the main text that needed additional clarification with respect to the regions of interest and how different areas may have responded to the task.

1. Lines 133-138: Text needs clarifications. Here, the authors state that "player positions during the observation phase (position1, position2, ...) were independent of player identity (self, partner)". If that's the case, then, why do the vectors for S and pos1 (2C and 2 D) have the same length? Similarly, P, seems equal to pos3, O-grey to Pos2, and O-green to Pos4. I don't understand this result when taken together with the sentence at 138: "Analogously, the sequential perspective 136 entailed a 4-dimensional representation along the axes of the four sequential positions, where the length of each axis reflects the performance for the player occupying this sequence position independent of identity". Please, clarify figure and text. "

Thanks for pointing this out. Here is the figure in question.

It is very true that the sequential position scores have a direct correspondence in agent-specific scores in this example. The important thing is to see this example together with the information that is provided in panel A. Panel A shows that all sequence positions are occupied by each player equally often and in all possible sequence combination. Therefore, it becomes possible to analyse signals related to sequence position – or our sequential basis functions - irrespective of player identity, because we experimentally balanced identity across positions. We clarify this in the text now as follows:

A critical design feature allowed us to dissociate the two formats of representation: player positions during the observation phase (position1, position2, ...) were independent of player identity (self, partner, ...; Fig.2A). **Because all sequence positions are occupied by each player equally often and in all possible sequence combination, it becomes possible to analyse signals related to sequence position – or our sequential basis functions - irrespective of player identity.**

And in the respective panel of the figure we have added:

(D) The same performance scores can be displayed in a sequential space along axes that represent the parametric performance score for each player position irrespective of identity (pos1, pos2, ...). **As sequence positions and player identities were balanced across all trials (see panel A), it became possible to dissociate those two parameters**

And point 5 | 3: 3. Line 349: I suggest adding here reference to Figure 3H, and in general also expand this more (see also comment #6).

Thanks. We have added the respective section and refer to the supplementary figure where we explain the relationships between pgACC and vmPFC in more detail:

These signals are in strong contrast with activation in vmPFC, a very different prefrontal region robustly linked to decision making regardless of whether there is a social dimension³⁸. **In contrast to pgACC, which reflects player performance information projected onto the basis functions which are referenced with respect to one's own position in the sequence, vmPFC activity during dyadic decisions reflects the value to the participant of taking one choice (the**

chosen option) as opposed to another (the unchosen option) regardless of the fact that these values originally related to the performances of different teams (Supplementary Fig.11).

7.2) Reviewer 2 (R2|Minor)

Minor comments:

1. Figure 1B. The arrow indicating the decision type is shown in pink, although this is described as blue in the text.

Thanks for making us aware of this. We have resolved the inconsistency.

2. Figure 1E needs a caption to explain the meaning of colors used.

Thanks. We have supplemented a colour explanation now together with the peak coordinate of the significant cluster.

3. Wouldn't it be better to refer to "flipped" and "sign-flipped" as "inverted" and "rotated", instead?

We understand that the "flipped" can appear unclear. That is why, in line with this comment from the reviewer, we now refer to them as "inverted" or "social" basis functions. We have changed the wording throughout the manuscript and in all figures.

4. For fMRI GLM1, how are O1 and O2 defined for group trials?

Thanks for raising this. O1 and O2 were identifiable because the letters "O1" and "O2" were overlaid on the cues signalling their identity. We make this clearer now in the methods section where we describe the GLM:

Note that O1 and O2, the two opponents, were clearly identifiable because the letters "O1" and "O2" were overlaid over their cues. We coded the fMRI regressors in line with these identities, even though other features of the opponents, such as their position on screen and colour were randomized across participants (see Fig.S1 for details of the visual presentation).

7.3) Reviewer 3 (R3|Minor)

Minor comments:

1. line 89: should be Figure 1E.

Thank you for spotting this. We have adjusted the reference:

Moreover, neural data revealed a previously characterized distinction of self- and other-related activity in medial prefrontal cortex in which perigenual ACC and dmPFC activity respectively reflected self and partner performances (Fig.1E).

2. Figure 1E: Please show the color code as an inset to the panel.

Thanks for making us aware of this. We have supplemented a colour explanation now by adding the name of the relevant contrast to the figure.

Referee #1 (Remarks to the Author):

I would like to thank Dr. Rushworth and team for providing such an extraordinarily comprehensive, detailed and well thought out response and for the truly herculean amount of work that went into revising the paper. The manuscript is much improved and provides a high degree of rigor across multiple modalities and controls that significantly strengthen the paper. In particular, with a new online platform, the authors have now provided additional controls to determine whether the observed social behaviors are better explained by the basis functions versus agent-based models (using nearly 800 new participants). They also examined whether their model better explains the players' scores and the effects of decision-irrelevant players as well as explain how their basis function control for agent-centric representations on fMRI. To further provide a non-social comparison, they recruited 32 additional participants who performed an analogous (motor-based) task, but which lacked social elements. Finally, they provided extensive revisions and clarifications to several of their figures as well as further streamlined their paper and results. As it currently stands, this is one of the most rigorously controlled and well-designed social studies that I have seen in a long time and I would be excited to see it published!

As I stated before and emphasize now again, the paper provides fundamental new insights into social behavior and an important new approach for studying group interactions in humans. Given the amount of new data and added results, my only remaining suggestion would be to trim down (with editorial guidance) some of the text to allow for easier readability. I have no doubt that this paper will be highly influential and well-cited in the field. I would strongly support its publication.

We thank the reviewer for their positive evaluation and their recognition of our comprehensive revisions, and describing our study as "one of the most rigorously controlled and well-designed social studies in a long time". We have invested considerable effort in addressing all previous concerns, and we are pleased that these improvements have significantly strengthened the manuscript.

Regarding the suggestion to enhance readability, we concur with the reviewer's recommendation. We have revised the manuscript once again and we are ready to polish the final version further.

Referee #2 (Remarks to the Author):

The main conclusion of this manuscript is that when the information about the strength of multiple individuals in a social group is sequentially presented for a subsequent decision regarding the relative strengths of individuals or their groups, this information is encoded using a set of basis functions, while the behaviorally irrelevant information encoded in the “null” dimensions is largely ignored. The possibility of this data compression is supported by an array of behavioral and neural findings presented in this study. The authors have also conducted additional behavioral and fMRI experiments to address the issues raised by the reviewers of the original manuscript. These results lend additional support to the hypothesis about the basis functions in the paper, but at the same time, also reveal some limitations of the study and may require some adjustment in the main conclusions, as suggested below.

Thank you for the detailed comments and for highlighting the range of evidence that we present, the work that has gone in our revisions and noting that they have addressed the issues raised by the reviewers. We address the remaining points below and have adjusted our main conclusions as appropriate.

1. It is quite convincing that the sequential basis functions manifest in the behavioral data and are also reflected in the fMRI signals in the ACC. However, the results from the (motor) control task strongly suggest that the activity in this area correlated to the sequential basis functions is not specific to the social setting. By contrast, the results shown in Supplementary Figure 10 regarding the fMRI activity related to “social” (inverted) basis functions might be more specific to social decisions. It seems unlikely that similar activity would appear during the motor control task. Therefore, it would be quite important for the authors to test whether such inverted (hand-specific) basis functions appear in the same or different brain areas compared to those identified in Supplementary Figure 10.

Thanks for raising this. This is a very interesting – and, as it turns out, also correct – point. We have done the analyses to test this suggestion and we indeed found, as predicted by the reviewer, that the basis functions, once inverted to the social, agent-centric frame of reference, appear to be specific to the social setting in dmPFC and some other brain regions highlighted by the reviewer. We test this rigorously using whole-brain analyses comparing the above effects across our two studies, and using an array of ROI analyses of the motor study including the use of Bayesian statistics to provide evidence for the null.

We believe that this is an important finding that further clarifies the relationship between domain general and socially specific signals in the brain. We have therefore substantially reworked the neural results sections of the social and the motor study. We have moved some aspects in the supplements and instead now show inverted basis function signals for the social study (Fig3) and for the contrast of social and motor study (Fig.5). In addition to this evidence of a dissociation between agent-centric and motor frame of reference, we show that decision-related vmPFC activity by contrast can be found in both motor and social domains, suggesting a common final pathway for decision-making across domains. We believe that these findings provide a nuanced explanations about the interplay of social and

domain general mechanisms in decision making at multiple stages of processing. We have therefore added them prominently to the main manuscript as follows:

Relating to the social study (study 1) shown in figure 3:

Main text, line 432-442:

We tested whether the brain organizes the basis function in this way and performed a whole-brain analyses with relevance-sorted basis functions during dyadic decisions. Indeed, we found evidence that a large region along the cingulate sulcus encoded the combination of the basis function projections using a **sequential frame of reference (Supplementary Fig.11)**. **These signals exist in parallel to the “inverted” basis function projections that utilize an agent-centric frame of the two teams and that we include in the same fMRI GLM (Supplementary Fig.12)**. For example, the inverted primary basis function indexed the same difference between one’s own versus the other team. We found that both primary inverted basis functions captured brain activity in distinct regions of frontal cortex and subcortical regions such as the dmPFC, lateral prefrontal cortex and the ventral striatum (Supplementary Fig.13). The combined effect of both primary and secondary inverted basis functions is shown in Fig.3G.

Figure 3. Basis functions in medial prefrontal cortex (study 1). ... (G) Neural signals in dmPFC, lateral prefrontal cortex, and the ventral striatum signaled a combination of the inverted, social basis function projections, an intermediate stage of information processing between observation and choice. The green contrast shows the average effect of inverted primary and secondary basis function across both decisions. By contrast, the final social decision variable in dyadic decisions, encoded in the reference frame of choice, is encoded in vmPFC. See supplements for evidence of uninverted basis functions.

Relating to the motor study (study 3) shown in figure 5:

Main text, line 716 - 731:

We tested for neural effects of inverted basis functions in the control fMRI experiment, hypothesizing that dissociations in activity patterns compared to those seen in study 1 might emerge after inversion to agent-centric versus motor frames of reference (*fMRI GLM 3*). Indeed whole-brain analyses comparing the combined inverted basis function representations (see Fig.3G, green activity) in an independent samples t-test, revealed significantly stronger activations in the social study (Fig.5J; $z > 3.1$; $p = 0.05$ FWE-corrected) in the ventral striatum and lateral prefrontal cortex, and also in dmPFC ($z > 3.1$; $p = 0.05$ FWE-corrected, pre-threshold masked in sphere around partner-related activity from Fig.1E). Corresponding signals were

absent in each of the three regions in the motor study (all $t_{31} < 0.575$, all $p > 0.570$, all $BF_{01} > 4.544$; Fig.3K). By contrast, activation was stronger in lateral primary motor regions in the motor study (Supplementary Fig.22). This suggests similar basis function-related computations across domains with differing neural implementations⁸, particularly at intermediate stages of processing. However, decision-related activity in vmPFC was present in both experiments (motor DV in frame of reference of choice; Fig.5L; $z > 3.1$; $p = 0.05$ FWE-corrected; compare with social DV in Fig.3G), even using the same ROI (Fig.5M). This indicates a common final pathway for decision-making across domains.

Figure 5. Control fMRI experiment revealed behavioural and neural signatures of basis functions (study 3). ... (J) Whole-brain analyses show that inverted, agent-centric basis functions in study 1 were significantly more strongly represented in dmPFC, ventral striatum (vStr), and lateral prefrontal cortex (IPFC) compared to inverted motor basis functions. The dmPFC cluster was pre-threshold masked ($z > 3.1$; $p = 0.05$, FWE) in a 20mm sphere around our independent dmPFC peak (partner signal in Fig.1, MNI: [-2 28 44]). (K) In ROI analyses of the motor study, inverted basis function signals in a motor frame of reference were absent in dmPFC, vStr and the IPFC. IPFC and bilateral vStr ROIs were taken from independent peaks for the same contrast in the social study (IPFC: [-44 42 4], vStr ROI combines [-6 14 -2] and [8 18 -2]; dmPFC: [-2 28 44]). (L) By contrast, the final DV (chosen – unchosen) was reflected in vmPFC activity in the same way in both motor and social tasks (compare Fig.3G) (M) These effects were also significant using an ROI derived from the social study (MNI: [-2 60 -12])

New supplementary figure panel detailing the inverted basis function effects in the motor study:

Supplementary Fig.22. Sub-threshold activation maps for social and basis function signals in the control fMRI experiment (study 3). (F) Cluster-significant inverted social basis functions in the social study. (G) The corresponding activation maps for the inverted motor basis functions (combining inverted primary and inverted secondary basis function across both decisions). The activation map is thresholded at $z > 2.0$ for illustration only. Note that the small white spheres are unbiased and independent masks from the social study related to the partner signal observed there (dmPFC mask, same as above) and related to the aggregate effects of the inverted social basis functions (IPFC: [-44 42 4], vStr ROI combines [-6 14 -2] and [8 18 -2]). As can be seen, inverted basis functions signals are clearly absent in and adjacent to these regions. Instead, the inverted basis functions in the motor study captured significant effects in the motor cortex, insular cortex and cerebellum. Note that these signals could indicate that inverted basis functions in the motor domain are processed in effector-related regions. However, they might also relate to the fixed response mapping utilised in the motor study (the left hand was equivalent to one's own "team") (N = 32).

2. Results shown in Figure 3B need a bit more explanation. First, the authors should include a color bar (scale). Second, what they mean by “the average effects of the projections on to 3 basis functions and S-position” is not clear. For example, does this refer to the average effect size of 3 basis functions, or the total variance accounted for by these 4 regressors? The latter might be more appropriate. This is because the fact that for both results illustrated in Figures 3B and 5J consist only of the positive correlation is very puzzling, given that the signs of the basis functions are (almost) completely arbitrary. For example, in the schematic shown in Figure 2F, the sign for the first serial position is positive for w_2 and w_3 , but negative for w_1 . How were these choices made? If the sign for the first serial position was made to be always positive, do the authors get the same results in Figure 3B? This depends on exactly what is shown in Figure 3B. Also, did the authors find any brain areas in which the average coefficient for any basis function was significantly negative? If not, how can this be explained?

Thanks for the suggestion. We have now explained the result shown in Fig.3B in more detail and hope that this made the results more accessible. We have now added a colour bar as follows:

And thanks for asking us to clarify what exactly the results in Fig.3B show and how the signs of the basis functions were determined. To calculate the basis functions, we used the weight vectors in Fig.2F that we show when we introduce the basis functions, i.e. w_1, w_2, w_3 , see here:

We then calculate the projections onto the basis functions by taking the dot product of weight vectors and the specific sequence of values observed in the current trial. We find in an independent ROI analysis of pgACC significant effects of all three basis function projections. Note that this test is not biased by the signs selected for the basis functions, as the statistical tests allows for effects to be significantly positive or significantly negative, both of which, as the reviewer points out, would be evidence that the brain encodes the respective basis function. Fig.3B then, is a test that examines whether the basis function

related activity, which we had tested in pgACC, is indeed strongest in this part of the brain, or whether similar signals are observed elsewhere. To do so, we calculate a contrast that is the linear combination of the three basis functions and S-position. We find whole-brain corrected effects in medial prefrontal cortex, but in no other parts of the brain. We did not find any negative whole-brain effects of the combined or any individual basis function projection; not in the social study and not in the motor study. We have added this information to the supplementary description of this contrast (see below).

The reviewer is correct that a reliably negative basis function effect might also be interesting given the almost arbitrary nature of their direction and, had that been the case, then a slightly different test would have been needed. However, while such alternative analyses might not be strictly necessary in the current study, they can, of course be performed and we present an example in a new figure and analysis in figure S10 and summarized below. Essentially, this new analysis examines the variance in neural activity explained by the basis functions compared to that explained by agent-centric regressors. Such an approach would be appropriate were basis function effects both positive and negative. This analysis is summarized at the end of our response to this question below.

Importantly, as pointed out by the reviewer, in our motor control study, we find neural activity in the same region of interest that is related to the basis function and signed in the same way. For this replication analysis, we took the most parsimonious approach possible and simply used the basis function weight vectors signed in the same way as in the independent social experiment. As noted, the signs of the basis functions are empirically determined rather than theoretically derived. However, as our results show, the basis functions can be measured as independent effects in our social experiment (all three are significantly different from zero, no matter the sign), and the direction of the effects replicate in our independent motor control study. As shown, below, the basis functions in the motor control study are similar to those in the social study, on aggregate, they are significantly positive and do not differ significantly from one another:

Following the advice of the reviewer we have edited our manuscript to clarify our analysis rationale. In addition, we have also conducted a Bayesian model comparison following a suggestion from another reviewer. This model comparison provides evidence for our basis function model that is also independent of the choice of specific basis functions signs, as suggested by the reviewer above. We have made the following changes to the main text and added the following supplementary analysis related to the Bayesian model comparison.

Main text describing the social study (study 1)

We discovered that activity in dmPFC extending into adjacent ACC gyrus (Fig.3B) covaried with such a sequential neural code in a whole brain analysis. The location is consistent with previous reports of grid cells in humans^{35,36}, but also close to areas engaged during theory of mind tasks^{4,9,10} and reports of a specific sub-region of ACC that may be especially important for social cognition^{1,37,38}. We discovered this activation using a ROI approach based on independent neural effects time-locked to a different trial phase discovered in Fig.1 (yellow activation in Fig.1E, MNI coordinates: [-8 42 14]). In this independent ROI, we found significant effects of all three basis function projections (using the weight vectors from Fig.2F, see methods): b_1 ($t_{55}=2.886$, $p=0.006$), b_2 ($t_{55}=2.749$, $p=0.008$) and b_3 ($t_{55}=3.081$, $p=0.003$; aggregate mean effect of all three projections: $t_{55}=5.02$, $p<0.001$; Cohen's $d = 0.671$), but, as predicted, not by the null-vector $t_{55}=0.229$, $p=0.820$; Bayesian evidence for the null: $BF_{01}=6.684$; Fig.3C; Supplementary Fig.7). S-position showed a negative deflection at the same time ($t_{55}=-2.379$, $p=0.021$) that was preceded by a positive signal ($t_{55}=2.051$, $p=0.045$; Fig.3D; Supplementary Fig.8). The whole-brain analysis subsequently confirmed that these effects are most prevalent in the dmPFC/ACC region and this is shown in Fig.3B. Additional neural simulations showed that these signals were clearly dissociable from agent-centric signals (Supplementary Fig.9) and Bayesian model comparison confirmed that at this point in the task, neural activity in dmPFC/ACC was better explained by our basis function model compared to agent-centric signals (Supplementary Fig.10). This suggests that dmPFC represents performance along all task-relevant dimensions in the flexible, temporally structured format prior to choice.

We added to the explanation of the motor study (study 3) that we did indeed use the same weight vectors:

Nevertheless, the same neural signatures of basis functions were still visible in the same pgACC region where we found them in the social study (Fig.5G). We looked for an average effect of the three basis function projections, as we had done in study 1, the social fMRI experiment (using the same weight vectors for their calculation as in study 1).

Supplementary Fig.22 now refers to absent negative effects of the basis functions:

Supplementary Fig.22. Sub-threshold activation maps for social and basis function signals in the control fMRI experiment (study 3). ... (E) Again, ROI selection was unbiased and independent (i.e. ROIs were derived from the social fMRI experiment and applied to the control fMRI experiment). The pattern of activity seen here is consistent with those shown in Figures 5 and confirm that analogous basis functions are coded in both tasks. Note that we did not find any whole-brain significant negative effect of the combined combined or any individual basis function projection; not in the social study (study 1) and not in the motor study (study 3).

New neural Bayesian model comparison:

Supplementary Fig.10. Bayesian model comparison suggests that dmPFC/ACC BOLD signal is better explained by basis function model compared to agent-centric model. We performed a Bayesian model comparison^{2,3} investigating how well the neural signal in dmPFC/ACC is explained by our basis function model compared to agent-centric effects. We focused on the same ROI as the main text (MNI coordinates: [-8 42 14]; relating to Fig.3B, reproduced here on the right). We used the convolved design matrices from *fMRI GLM2* to run an ROI analysis on this region. We tested two models: (1) a basis function GLM and (2) an agent-centric GLM. Both models were derived from *fMRI GLM 2*, which contained both basis function-related regressors (b_1, b_2, b_3 , S-position) and agent-centric regressors (S-performance, P-performance, O1-performance, O2-performance) time-locked to the end of the observation phase. We derived the basis function model of neural activity by removing the four above-mentioned agent-centric regressors from the convolved design. We also removed their associated temporal derivative regressors. This can be thought of as a model of neural activity without agent-centric influences on the BOLD signal at this time in the trial. Next, we built the analogous agent-centric model from *fMRI GLM2* using a similar approach. We kept in the four agent-centric regressors (S-performance, P-performance, O1-performance, O2-performance), but instead removed the four regressors related to the basis function representations (b_1, b_2, b_3 , S-position) and their associated temporal derivatives. This agent-centric model captured the influences of agent-related performance effects but did not contain regressors related to the basis functions. Both resulting GLM models, the basis function model and the agent-centric model, comprised the same set of additional regressors related to other points in the trial. They were identical to *fMRI GLM2* as described in the Methods, with the only difference relating to the changes described just above. Both GLMs have the same number of regressors. We applied both the basis function GLM and the agent-centric GLM to the neural data in dmPFC/ACC and calculated the log model evidence for each GLM and each participant (note that there was no need to correct for the number of regressors in the design, as the number of regressors were identical in both GLMs). Afterwards we submitted the log model evidence into a Bayesian model selection random-effects analysis (using the `spm_BMS.m` routine from SPM)⁴, which computed the exceedance probability of the two GLMs. We found that that the exceedance probability greatly favoured the basis function model (exceedance probability=0.982), suggesting that neural activity in dmPFC/ACC at this time in the experiment is better explained by basis functions compared to agent-centric representations.

3. Explanation of the results shown in Figures 3G is also somewhat confusing. In particular, the figure caption refers to the light blue region as “primary + secondary basis function”. Since the signs for the basis functions that are combined must change according to the type of dyadic decisions, the authors might have already inverted the signs of the primary and secondary basis functions when necessary to account for the specific type of decisions. If so, this should be mentioned explicitly.

Thanks for inquiring about this. The issue at stake is very closely related to the previous point and, as explained above, the approach that we have taken is simply that we have used the

same weight vectors for the basis functions that we had already introduced in the beginning in Fig.2F. The primary and secondary basis functions are determined based on the specific trial that participants experience currently and the signs of the primary and secondary basis functions are not inverted in this analysis. The light-blue contrast referred to by the reviewer then shows just that, a linear [1 1] contrast that averages the effects of the primary and the secondary basis function.

We do understand that the specific analysis shown in this panel may be confusing and may distract from our main findings. Because of this, and because of concerns about the length of the main text, we have therefore moved this panel to the supplements, where we have more space to explain these results in detail as follows. We have replaced this panel with the analyses concerning the inverted basis functions, which we have conducted in response to the reviewer’s point 1. As noted above, we believe that this result about the inverted basis functions is very valuable in explaining the socially specific and the domain general processes that operate in the context of our experiments. We have moved the other analysis to the supplements with an extended explanation as follows.

Supplementary Fig.11. Cingulate cortex and frontal pole encode basis function projections using a sequential frame of reference. As explained in the main text, the basis functions can be sorted in primary, secondary, and tertiary basis function to aid decision making. With primary and secondary basis functions, we refer to the two decision-relevant basis functions that are present in dyadic decisions with the primary basis function becoming apparent during the observation phase, and the secondary one becoming apparent when the specific decision is cued. Mathematically, the regressor values for these variables always correspond to the values for the unsorted basis functions: b_1 , b_2 and b_3 . The primary basis function then is equivalent to the value of the one of the three that is given by the observation phase when the positions linked to the self’s own group and the opponent group become apparent. The value for the secondary basis function is given by the value of the additional basis function projection that becomes apparent when the specific dyadic decision is cued. And the tertiary basis function is then simply defined as the remaining basis function projection. We found evidence that a large region along the cingulate sulcus, posterior to our previously identified contrast, encoded the combination of primary and secondary basis function projections (light blue activation). This is striking because these variables encode sequential information and are unrelated to agent-centric or decision-related variables, which we again control for in the analysis. For example, for the primary basis function encoding the difference between pairs of sequential positions, we included a control regressor indexing the same difference but in the social frame of reference of “own team vs other team”. This effect became even more prominent after contrasting this combination with projections onto the tertiary basis function (dark blue activation). This comparison additionally engaged bilateral frontopolar cortex as computing the difference in relevant and currently irrelevant basis function. It is important to recall that participants performed two

different decisions after seeing the performances. The current analysis was carried out at the time of the first of the two decisions following the observation phase. Hence, the tertiary basis projection, while not currently relevant for the decision in hand might become relevant for the second decision. This might explain why this contrast was linked to activity in frontal pole; frontal pole activity reflects information pertinent to future decisions⁴⁰. Importantly, these activity patterns, similar to previous analyses, reflected compressed, sequentially organized information. We controlled for corresponding agent-centric activations as well as any social decision-related activations in these analyses. By contrast, the social decision variable, encoded in the reference frame of choice, is encoded in vmPFC (same contrast as shown in main manuscript). The yellow activation is added for reference and reflects the contrast shown in Fig.3B. All activations (except the two blue ones) are non-overlapping. (n=56; study 1, social fMRI experiment; all MRI results cluster-corrected at $Z>3.1$; $p=0.05$ FWE).

Minor comments.

1. As in the original manuscript, in the Introduction, the authors draw an analogy between their findings and the single-neuron findings from the studies by Tanji and his colleagues during a sequential motor task. However, given that there are some major differences between these studies, this may mislead or confuse some readers. For example, the neurons characterized by Tanji et al. are good examples of “combinatorial” codes, since they respond to specific conjunctions of motor responses and their serial positions. However, they do not behave as the basis functions as described in this manuscript, since they do not encode signals related to the linear summation of multiple movements. In addition, there is no evidence of “compression” in the neurons studied by Tanji et al. A better example might be the use of the basis functions in Alex Pouget’s work to describe the response properties of the neurons in the primate posterior parietal cortex (e.g., LIP).

Thank you for engaging so deeply with the theoretical underpinnings of our study and for making these very insightful points. We agree that the Pouget studies are excellent examples of basis function coding and we thank the reviewer for making us aware of them in this context. Since the reviewer points out that the Tanji work is a good example of a combinatorial code, and we agree, we have moved this reference one paragraph up into a paragraph discussing combinatorial codes in general. We have then rewritten the following paragraph where we introduce the concept of basis functions, highlighting the similarity with the Pouget work. We would also be happy to remove the Tanji reference entirely if the reviewer feels the current version still offers room for misunderstanding. Textual changes are as follows (page2, lines 52 - 75)

Encoding the potential combinatorial patterns afforded by a given situation may constitute the flexible solution that is needed. Combinatorial patterns are widely utilized in sensory, motor, and spatial domains. For example, single neurons in the pre-supplementary motor area fire when a macaque prepares a movement, but only when the movement comprises a particular sequence of elements in a particular order, for example push→pull→turn¹². The neuron is silent when single elements of this sequence are prepared or when the elements are executed in a different order. More generally, planning of motor sequences and perception of sequences of visual cues are accompanied by abstract coding of sequential positions (1st cue, 2nd cue, etc.) independent of specific stimuli and specific actions^{17,18}. Computational theories suggest that such combinatorial and sequential codes are abstractions from sensory information that can act as a scaffold for the acquisition of new information^{13,19}.

This concept of combinatorial patterns in neural coding aligns with and extends to the idea of "basis functions". For example, Pouget and colleagues suggest that to solve the problem of visually guided motor actions, parietal neurons compress multiple input variables (e.g., retinal position, eye position, object orientation) into single neuronal responses. These compressed responses can be understood as "basis functions" that can flexibly be combined in a linear manner to guide motor actions²¹. Basis functions can effectively reduce the dimensionality of the representation by representing a limited set of feature dimensions that efficiently summarize a behavioral repertoire or task¹³.

2. Page 16, line 552. The phrase "sparser code" does not appear to be relevant in this context. Compressed codes are not necessarily sparse.

Thanks for pointing this out. We agree. We have changed the wording of the respective sentence as follows. Note that we ensured that the word "sparse" appears at no other point in the manuscript:

These findings cannot be explained by a four-element agent-centric code. Instead, they suggest that information is compressed and that this enables more efficient decision-making.

3. Although the self-position is meaningless in the motor control task, it would be better to include its equivalent term in the GLM, so that the analyses for the main and control tasks have the same degrees of freedom.

We agree with this, and we had, in fact, performed the analysis in precisely in the way the reviewer describes. We have now, however, tried to make this clearer in the main text and in the methods section:

Main text:

We went on to look for socially specific representations and signatures of basis functions in the brain. We applied the same fMRI GLMs as before (*fMRI GLM1* and *fMRI GLM2*), importantly this ensured that the analyses had the same degrees of freedom. Note that, due to our tightly matched design, these models were not just conceptually but also numerically identical to our social GLMs.

Methods:

The fMRI whole-brain designs included the same set of regressors with identical timings as the social fMRI study (*fMRI GLM 1* and *fMRI GLM 2*), ensuring that main and control task GLMs comprised the same degrees of freedom.

Referee #3 (Remarks to the Author):

The authors have done an excellent job of addressing all of our original comments and substantially increased the depth of this fascinating work. In fact, the authors added three new experiments including a new behavioral and a new neuroimaging study to provide compelling evidence to fill various gaps pointed out by the reviewers through empirical data themselves. The new insights and clarifications coming from these added data and numerous data analyses have, in my opinion, really advanced the core concept behind the combinatorial code of social interaction based on the basis functions in the human brain. I also appreciated the time and care that the authors have put into making the response document that is so well-organized and comprehensive. Initially, the major concern was whether their basis function encoding was necessary - their new analyses and experiments clearly showed that basis function encoding is better than other "simpler" kinds of encodings. I am very enthusiastic about this work, and my enthusiasm only increased after seeing all the added analyses and clarifications provided by the authors in this revision.

Signed below to opt in for Nature's transparent peer review scheme:

Reviewed by Steve W. C. Chang with assistance from Weikang Shi, a postdoctoral fellow in his lab.

Thank you for the feedback. We are very pleased that we have successfully addressed the reviewers concerns and that the reviewers highlight that our revisions were conducted rigorously and comprehensively, and that they strengthened the very core of our argument – that humans employ combinatorial basis functions during social interactions.

Referee #4 (Remarks to the Author):

Basis functions for complex social decisions in dorsomedial frontal cortex – re-review

This manuscript investigates the neural representation of social information during decision-making, proposing a model based on basis functions that compress and sequence social information. I remain convinced that the general problem of understanding how the brain encodes social situations is extremely exciting and important. However, I remain unconvinced of the robustness and generalizability of the particular findings reported in this paper. My concerns extend to both experimental and conceptual aspects of the revised manuscript. Despite the authors' efforts to address the initial reviewers' concerns, I believe this resubmission falls short of convincingly demonstrating the proposed basis function model for social decision-making. Most concerning for me are the tiny effect sizes (point #1 below) and the lack of clear applicability of the proposed sequence-specific basis functions to real world decision making (see point #2 below).

We note R4's main concerns, point 1 and point 2. We respond in more detail below.

However, in brief, R4's point 1 is factually incorrect. The effect sizes that we report are not "very small", as the reviewer argues in their point 1. In most cases, they are not even "small" but instead they are "medium" or "large" according to the most standardly used metric for effect sizes, Cohen's d .

R4's second point is misleading. Our claim is not that the brain uses a specific set of basis functions, b_1 , b_2 , and b_3 in all situations but that it uses a set of sequential basis functions that are appropriate for a given task and, in this particular case, the relevant sequential basis functions are b_1 , b_2 , and b_3 . We believe that this point is clear to other reviewers and no other reviewer has raised this point.

Major issues:

1. Small effect sizes of fMRI and behavioral results: While the authors acknowledge the small effect sizes, they argue that the effects are replicable and consistent with previous findings. They provide a new behavioral study and additional analyses to support their claims. However, the new behavioral study *still shows very small effect sizes*. I believe the key result remains Figure 3C: this is the most direct assay of whether the brain codes these basis functions, and the effect size remains very small.

The reviewer is factually incorrect. The effect sizes are not, were not, and remain not, "very small". As in our previous response to the reviewer's points, we strongly insist that statements about the significance of effects and effect sizes should be based on empirical evidence and established methods of statistical inference.

The most common estimate of effect size, Cohen's d , suggests that (Cohen 1988; Sawilowsky, 2009):

- $d = 0.01$ very small effects
- $d = 0.2$ small effects

- $d = 0.5$ medium effects
- $d = 0.8$ large effects

Our effect sizes are factually not very small or in the range below 0.2. In most cases they are above 0.5 or even reach 0.8, and classify as medium or large effects.

- The neural effect sizes for basis function related signals in Fig.3C, referred to by the reviewer in Fig.3C above, are $d = 0.67$
- The neural effect sizes in the replication study for the same effect are $d=0.56$
- The behavioural effect sizes for basis function related activity in our fMRI study, shown in Fig.3J/K are $d=.78$ and $d=0.81$. These are the ones that the reviewer describes in a point below (Minor point 4) as “could be easily explained by noise”, without having any empirical basis for this assessment. Again, we argue that statements about significance and effect sizes need to be based on empirical evidence and sound statistical inference – which strongly support our conclusions.
- We replicate these behavioural effects of basis functions with $d=0.36$ and $d=.47$ in our online study in precisely the predicted directions
- We then again replicate these behavioural effects in the replication study with $d=0.818$ and $d=0.736$ in precisely the direction predicted by our theoretical model
- In a model comparison comparing agent-centric model with our basis functions model, the effect size of improvement that comes with our basis function model is $d=1.084$ (Suppl Fig.16, previously Suppl Fig.13). This result is the one that in their point 4 below “exacerbates the reviewer’s concerns” of our effects having “small effect sizes”. As we show here, these concerns are based on the reviewer’s factually incorrect assessment of our effect sizes.

Our effects are not only not very small according to the standard measures categorization of effect sizes but they are not very small in a recently proposed revision of the standard categorization. A recent study empirically determined small, medium and large effects sizes in the field of social psychology by reviewing over 6,447 reported effect sizes from studies in 134 meta-analyses (Lovakov & Agadullina, 2019). They conclude that in fact the empirically more appropriate categorisation of effect sizes in this field should be:

- $d = 0.15$ small effects
- $d = 0.36$ medium effects
- $d = 0.65$ large effects

This demonstrates that our neural and behavioural effects are not only replicable, as the reviewer already accepts, but they are also in no way small; not according to any established metric of effect sizes employed across disciplines nor when assessed against the standards of studies of social cognition.

We have added effect sizes to the relevant parts of our manuscript:

Neural effects in social fmri study, line 340:

In this independent ROI, we found significant effects of all three basis function projections (using the weight vectors from Fig.2F, see methods): b_1 ($t_{55}=2.886$, $p=0.006$), b_2 , ($t_{55}=2.749$,

$p=0.008$) and b_3 ($t_{55}=3.081$, $p=0.003$; **aggregate mean effect of all three projections: $t_{55}=5.02$, $p<0.001$; Cohen's $d = 0.671$**), but, as predicted, not by the null-vector $t_{55}=0.229$, $p=0.820$; Bayesian evidence for the null: $BF_{01}=6.684$; Fig.3C; Supplementary Fig.7).

Behavioural effects and model comparison in social fmri study, line 471:

Indeed, analyzing player specific effects on choice revealed precisely these effects of irrelevant players in Self decisions (P: $t_{55}=3.093$, $p = .003$; Oi: $t_{55}=-5.131$, $p < 0.001$; **combined P – Oi effect: $t_{55}=5.803$, $p < .001$, Cohen's $d=0.775$** ; Fig.3J) and Partner decisions (S: $t_{55}=4.867$; $p < 0.001$; Oi: $t_{55}=-5.104$; $p < 0.001$; **combined S – Oi effect: $t_{55}=6.027$, $p < .001$, Cohen's $d=0.805$** ; Fig.3K). Formal information theoretic metrics confirmed the higher model accuracy of the basis function model over the agent-centric model (paired t-test on model accuracy; $t_{55}=8.111$, $p<0.001$, **Cohen's $d=1.084$** ; Supplementary Fig.16; see Supplementary Fig.17 for a detailed decision analysis of all trial types).

Behavioural effects in online study, line 571:

This resulted in the same characteristic pattern of irrelevant player effects in the Group condition that we had observed in the fMRI study, which also contained group decisions (**P-Oi in self decisions: $t_{395} = 7.241$, $p < 0.001$, Cohen's $d=0.364$; S-Oi in partner decisions: $t_{395} = 9.253$, $p < 0.001$, Cohen's $d=0.465$**).

Behavioural effects in motor replication study, line 633:

We found precisely this pattern in the control fMRI experiment: In Motor-Self decisions, Motor-P had positive (Fig.5D; $t_{31}=2.783$, $p=0.009$) and Motor-Oi had negative effects ($t_{31}=-4.490$, $p<0.001$; **combined Motor-P – Motor-Oi: $t_{31}=4.629$, $p<0.001$, Cohen's $d=0.818$**). The pattern of effects in Motor-P decisions provided further evidence for this claim (Motor-S effect: $t_{31}=3.564$, $p=0.001$; Motor-Oi effect: $t_{31}=-3.844$, $p<0.001$; **combined Motor-S – Motor-Oi: $t_{31}=4.166$, $p<0.001$, Cohen's $d=0.736$**).

Neural effects in motor replication study, line 702:

There was a strongly positive mean effect of all three basis function projections in pgACC (Fig.5G; same ROI as in social study, MNI: [-8 42 14], $t_{31}=3.168$, $p=0.003$, **Cohen's $d=0.560$** ; Supplementary Fig.23).

2. Real world applicability: In response to the criticism that their basis scheme is artificial and unrelated to real world decision making, the authors argue that understanding social situations in terms of teams is advantageous. I fully agree with this. However. I have trouble understanding how, e.g., $b_1 = [-1 \ 1 \ -1 \ 1]$, defined as a fixed sequence of weightings, represents team structure—*on some trials, this particular basis function correlates with team structure and on other it does not*. As the authors themselves point out, what is relevant is the “primary” basis function—the one that aligns with team structure on any trial. But the claim that the brain encodes social situations in terms of the primary basis function, i.e., team A performance – team B performance, is a much weaker claim—it simply says that the brain should represent a decision variable directly

related to the current decision that needs to be made (“which team won?”). As I pointed out in previous review, the brain *must represent this* in order for the subject to do the task, and it could readily do so by *simply linear combining individual agent performance*. I understand the essential novel claim of the paper to be that the brain represents sequential basis functions b_1 , b_2 , b_3 . After reading the revised paper, this framework still feels contrived to me. I remain unconvinced that these functions have general relevance to social decision-making, and the small effect sizes (see point #1) make me doubt whether the brain actually uses this scheme.

The essential claim is that the brain represents sequential basis functions that are appropriate for the task in hand. In this task, the appropriate sequential basis functions are b_1 , b_2 , and b_3 . Our argument is that just as the brain adopts a basis set that is appropriate for the motor tasks that occur in a given context, so it adopts a social basis set. If the social situation were to change then the basis set would change. Moreover, we demonstrate just such a change already in the comparison of the group and no-group conditions. We have clearly explained this in our manuscript and make it even clearer in our revised manuscript as follows:

Introduction, line 77:

Here we demonstrate that an analogous, limited set of basis functions is computed in dmPFC and adjacent ACC during social cognition (study 1: social fMRI experiment). The basis functions summarize the possible social interactions in a compressed format tailored to the decision task at hand. **As such, basis functions are specific to the decision problems that are expected in a given context.**

Discussion of the behavioural study 2, line 585:

Overall, participants relied more on correct information about relevant players in the Group condition. This is strong evidence that a compressed code along the basis functions improves decision-making **and it illustrates how the use of basis function changes depending on decision context.**

3. Compression: While the authors claim that basis functions offer a “compressed” representation, the demonstration of compression remains weak. As acknowledged by the authors, the 4D to 3D transformation presented as compression offers minimal efficiency gain. Indeed, the authors clarify in their rebuttal that they believe these basis functions are represented in addition to agent-specific information (page 150, “We take care to avoid claiming that there is no representation of individual agents. Indeed, there is abundant evidence for representations of individuals. Our claim is that there is an additional representation that relates to the patterns of interaction that can occur between individuals”). If so, it is not clear to me how these basis functions accomplish any kind of meaningful compression.

The key observations are that: 1) as even R4 acknowledges, there is a degree of compression and so the experiment demonstrates the principle that it is possible for the brain to identify a lower dimension compressed representation of the social interaction.; 2) when this compression is hampered, as in the “no-group” experiment, performance is worse even if the task is made simpler in other respects.

4. New online experiment: The new online experiment (Study 2) attempts to provide evidence for compression by showing improved performance in the "Group condition" compared to the "No-Group condition". However, the interpretation that this is due to compression is debatable and only tested indirectly. The "No-Group" condition is simpler with fewer decision states and more training on relevant trial types, suggesting potential alternative explanations for the observed performance difference: in particular, the improved performance in the "Group condition" could be due to other factors, such as increased attention or motivation, rather than the use of a compressed code. Furthermore, the effect sizes presented in Fig 4D-I are very small. While the presented comparison (Supplementary Fig. 13) using cross-validation is useful, it doesn't alleviate these concerns, but instead exacerbates them: *the improvement in explanation by the sequential basis function model compared to agent model is on order of a fraction of a percent*.

This fraction of a percent referred to by the reviewer in their last sentence represents a better fit for our basis function model that has an effect size of $d=1.084$. Cohen's effect size categorisation posits very small ($d=0.01$), small ($d=0.2$), medium ($d=0.5$) and large ($d=0.8$) effects. The effect referred to by the reviewer clearly deserves the classification of having a large effect size. It is factually not correct that this result exacerbates concerns about very small effect sizes.

The important feature of "no-group" versus "group" experiment is, indeed, that the "no-group" condition is "simpler with fewer decision states and more training on relevant trial types" than is the case in the group experiment yet the result is that performance is better in the group condition where basis functions are easier to employ. R4's argument that performance is better in the group condition because there is "increased attention and motivation" is unsupported by any evidence and the logic of the argument is weak. The reviewer's argument is, therefore, akin to saying that the difficult condition becomes less difficult because it is difficult and so participants try harder.

Also, again, the effect sizes we find in our online study are not "very small". As in the previous experiments, we measure the impact of basis function coding on behaviour in terms of the GLM effects of the irrelevant players. For instance, in self decisions, our theoretical model predicts positive effects of P and negative effects of Oi. We therefore measure the effect size of the aggregate effect of P-Oi to estimate the behavioural effect sizes of basis functions. In the group condition of the online study, where we posit that basis functions are used, we find effect sizes of Cohen's $d=0.364$ and $d=0.465$, for self and partner decisions respectively. This represents a significant increase relative to the No Group condition, where we hypothesized that basis functions impact behaviour to a lesser degree, with Cohen's d of $d=0.317$ and $d=0.306$. While these effect sizes are admittedly smaller than the ones in previous analyses, they are still larger than "very small" Cohen's d s. They also relate to the effects that replicate across studies, and that test very specific and directed hypotheses motivated by our theoretical basis function model.

However, simultaneously, the effects of the irrelevant players increased in the Group condition relative to the No-Group condition (P-Oi in self decisions: $t_{793} = 4.471$, $p < 0.001$, **Cohen's $d=0.317$** ; S-Oi in partner decisions: $t_{793} = 4.317$, $p < 0.001$, **Cohen's $d=0.306$**). This resulted in the same characteristic pattern of irrelevant player effects in the Group condition

that we had observed in the fMRI study, which also contained group decisions (P-Oi in self decisions: $t_{395} = 7.241$, $p < 0.001$, Cohen's $d=0.364$; S-Oi in partner decisions: $t_{395} = 9.253$, $p < 0.001$, Cohen's $d=0.465$).

5. We thank the reviewer for clarifying that their variables b1-3 are not linear combinations of S, P, O1, and O2 because they are tied to sequential positions, and appearance of different agents at different sequential positions was balanced. But how can the authors be certain that their subjects didn't simply zone out on some trials (given that behavioral performance was far from perfect), which would disrupt their careful balancing, and lead to the small effects they attribute to sequential basis function encoding?

First, as noted already, the effects are not small but in Cohen's medium to large range. Second, were participants to "zone out" on some trials, this would not disrupt the trial balancing because zoning out should be expected only to introduce noise rather than bias in the relationships between specific agents and sequential positions. Bias in the relationships could only occur if participants were able to divine the trial type before it began and then chose to zone out when agents were in certain positions. This is not a reasonable suggestion.

6. Related to point #5: Even if the subject is paying careful attention on all trials, it seems that given the finite # of trials, the balancing cannot be perfect, and the tiny effects observed could arise from agent tuning plus imperfections in balancing. The authors should simulate brain activity driven solely by agent identity and repeat their time course analysis on the simulated neural data (Fig. 3C). Are the simulated effect sizes for the basis function projections really zero? Or instead, is it possible that they will be small (as reported in Fig. 3C) but still diverge significantly from zero at multiple time points?

First, as elsewhere the effect sizes reported in figure 3c are not small. We have made this clear in the revised manuscript by reporting Cohen's d for the effects reported in figure 3c, as described above. Second, we have, as suggested by R4, simulated neural data in which there is solely an effect of agent identity but, with such simulated data, it is not possible to obtain effects similar to those that we report in figure 3c. This suggests that the effects in figure 3c cannot, as suggested by R4, be explained away as a consequence of agent-specific activations. We have added a new supplementary figure detailing the procedure and results of these simulations. We have also conducted a Bayesian model comparison on neural data comparing our basis function model with a purely agent-centric model. These analyses, again, confirm our basis function model:

Supplementary Fig.9. BOLD Simulations demonstrate that basis function projections and agent-centric brain signals are dissociable. Our proposed basis function model suggests that the brain represents projections onto the basis functions (b_1 , b_2 , b_3) that are dissociable from agent-centric representations (performance scores of Self (S), partner (P), opponent 1 (O1) and opponent 2 (O2)). We conducted BOLD simulation analyses to show that this is indeed the case and that basis function related signals would not occur as a by-product of holding agent-centric representations. We simulated a BOLD time course by recreating the convolved design matrix¹ from *fMRI GLM 2*. We used the convolved design matrix to create synthetic BOLD data. Afterwards, we applied the same time course analyses to it that allowed us to identify basis function projections (b_1, b_2, b_3) in the first place (*ROI GLM 1*). This time course GLM comprises regressors for both basis function-related activity and agent-centric signals. We tested whether basis function signals would be measurable in BOLD data that was (i) created from purely agent-centric representations or (ii) both agent-centric and basis function computations. We find strong evidence that basis function signals do not emerge from solely agent-centric data. **(A,B)** The plots show recovered effect sizes for basis function-related activity (panel A) and agent-centric coding (S,P,O1, and O2 signals) in a timecourse GLM that was purely fitted on synthetic BOLD data that was created using only agent-centric effects. We created the synthetic BOLD data by weighing the design matrix X from *fMRI GLM 2* with a weight vector w_1 which was set to [1 1 -1 -1] for the agent-centric effects [S,P,O1,P2] of the GLM (see Methods). All other weights were set to zero. We used positive weights for one's own team and negative weights for the opponent team, because we had observed such positive/negative coding of the agents in other supplementary analyses of frontal cortical activity (see Supplementary Fig.14) and hence reasoned that it is the most plausible agent-centric coding scheme we could assume. Note that we jittered the weights slightly so that the resulting lines in the timecourse plot do not fully overlap on one another but, instead, are each individually visible. We created the synthetic BOLD time course by taking the simple dot product of Xw_1 . We then fed the synthetic BOLD data into precisely the same timecourse analysis (*ROI GLM 1*) that we had used to identify b_1, b_2, b_3 in the first place. We repeated the procedure for all participants and show the recovered effect sizes for b_1 , b_2 , and b_3 in panel A. As expected from BOLD data created using purely agent-centric effects, no basis function-related signals were visible at any point of the time course for neither b_1 , b_2 , or b_3 (all $p > .1$). By contrast, all four agent-centric effects could be reliably recovered in the expected directions (all $p < 0.001$ at expected peak time window 7-9sec). **(C,D)** We repeated the same simulation to show that basis function-related signals can be successfully recovered if they are simulated. Again, we used the same convolved design matrix and a weight matrix w_2 . w_2 simulated the same agent-centric effects as w_1 , but, importantly, in addition simulated the parallel existence of basis-function related effects. w_2 modelled the same agent-centric weights of [1 1 -1 -1] for [S,P,O1,P2], but added positive effects [1 1 1] for [b_1, b_2, b_3]. Then we calculated synthetic BOLD data by calculating Xw_2 . Afterwards, we applied the same GLM (*ROI GLM 1*) as before and inspected both basis function-related

effect sizes and agent-centric effects in the same GLM model. As expected, now b_1 , b_2 and b_3 can be successfully recovered in addition to agent-centric effects (all $p < 0.001$ at expected peak time window 7-9sec).

Supplementary Fig.10. Bayesian model comparison suggests that dmPFC/ACC BOLD signal is better explained by basis function model compared to agent-centric model. We performed a Bayesian model comparison^{2,3} investigating how well the neural signal in dmPFC/ACC is explained by our basis function model compared to agent-centric effects. We focused on the same ROI as the main text (MNI coordinates: [-8 42 14]; relating to Fig.3B, reproduced here on the right). We used the convolved design matrices from *fMRI GLM2* to run an ROI analysis on this region. We tested two models: (1) a basis function GLM and (2) an agent-centric GLM. Both models were derived from *fMRI GLM 2*, which contained both basis function-related regressors (b_1, b_2, b_3 , S-position) and agent-centric regressors (S-performance, P-performance, O1-performance, O2-performance) time-locked to the end of the observation phase. We derived the basis function model of neural activity by removing the four above-mentioned agent-centric regressors from the convolved design. We also removed their associated temporal derivative regressors. This can be thought of as a model of neural activity without agent-centric influences on the BOLD signal at this time in the trial. Next, we built the analogous agent-centric model from *fMRI GLM2* using a similar approach. We kept in the four agent-centric regressors (S-performance, P-performance, O1-performance, O2-performance), but instead removed the four regressors related to the basis function representations (b_1, b_2, b_3 , S-position) and their associated temporal derivatives. This agent-centric model captured the influences of agent-related performance effects but did not contain regressors related to the basis functions. Both resulting GLM models, the basis function model and the agent-centric model, comprised the same set of additional regressors related to other points in the trial. They were identical to *fMRI GLM2* as described in the Methods, with the only difference relating to the changes described just above. Both GLMs have the same number of regressors. We applied both the basis function GLM and the agent-centric GLM to the neural data in dmPFC/ACC and calculated the log model evidence for each GLM and each participant (note that there was no need to correct for the number of regressors in the design, as the number of regressors were identical in both GLMs). Afterwards we submitted the log model evidence into a Bayesian model selection random-effects analysis (using the *spm_BMS.m* routine from SPM)⁴, which computed the exceedance probability of the two GLMs. We found that that the exceedance probability greatly favoured the basis function model (exceedance probability=0.982), suggesting that neural activity in dmPFC/ACC at this time in the experiment is better explained by basis functions compared to agent-centric representations.

7. Basis Function Sets: The authors focus on a specific set of basis functions without exploring alternative sets that might be equally or more efficient for representing the task structure. For instance, could a basis set that directly reflects team membership and player relevance for each decision type provide a better account of the data? Investigating

alternative basis function sets would provide a stronger argument for the chosen set and enhance the understanding of the underlying representational mechanisms.

This criticism is misplaced. We have already reported the results of tests for a number of alternative representations (for example bottom left of figure 3c) but perhaps most importantly we test precisely the combination suggested by R4 – activity that reflects team membership and player relevance. This combination corresponds exactly to what we have termed the combination of the primary and secondary basis functions in the anterior cingulate sulcus and dorsolateral prefrontal cortex (dark blue and light blue areas in figure 3G in the previous version, now supplementary figure S11 in the most recent revision) as opposed to the dmPFC in which information is encoded according to the social basis set (green area in figure 3G).

8. Social Specificity: The authors attempt to address the concerns about the social specificity of their findings with a non-social control experiment (Study 3). However, the results of this experiment are far from conclusive. If anything, Study 3 provides further evidence that the authors' findings are not socially specific, since they still find basis function representation (Figure 5). To facilitate comparison with Study 1, the authors should show the results of Figure 5 in the format in Figure 3C.

This is incorrect. We have provided clear-cut evidence that socially specific signals are present in the social study but not in the motor study. We have shown that signals relating to one's own and the partner's performance specifically recruit pgACC and dmPFC, respectively, but that these signals are absent in the motor study. This is and was clearly presented in figure 5, panel F:

(F) Neurally, in the control experiment, signatures of motor-S and motor-P in regions where we had found socially specific activations were absent. ROIs were centered on whole-brain cluster-corrected activation for self (pgACC, MNI: [-8 42 14]) and partner (dmPFC, MNI: [-2 28 44]) from the social study.

In addition, following the suggestions of reviewer 2, we have conducted further analyses that highlight which basis function-related signals are shared between social and motor task, and which are unique, focusing on the basis functions inverted into a social frame of reference:

Main text:

We tested whether the brain organizes the basis function in this way and performed a whole-brain analyses with relevance-sorted basis functions during dyadic decisions. Indeed, we found evidence that a large region along the cingulate sulcus encoded the combination of the basis function projections using a **sequential frame of reference** (Supplementary Fig.11). These signals exist in parallel to the “inverted” basis function projections that utilize an agent-centric frame of the two teams and that we include in the same fMRI GLM (Supplementary Fig.12). For example, the inverted primary basis function indexed the same difference between one’s own versus the other team. We found that both primary inverted basis functions captured brain activity in distinct regions of frontal cortex and subcortical regions such as the dmPFC, lateral prefrontal cortex and the ventral striatum (Supplementary Fig.13). The combined effect of both primary and secondary inverted basis functions is shown in Fig.3G.

Figure 3. Basis functions in medial prefrontal cortex (study 1). ... (G) Neural signals in dmPFC, lateral prefrontal cortex, and the ventral striatum signaled a combination of the inverted, social basis function projections, an intermediate stage of information processing between observation and choice. The green contrast shows the average effect of inverted primary and secondary basis function across both decisions. By contrast, the final social decision variable in dyadic decisions, encoded in the reference frame of choice, is encoded in vmPFC. See supplements for evidence of uninverted basis functions.

Relating to the motor study (study 3) shown in figure 5:

Main text:

We tested for neural effects of inverted basis functions in the control fMRI experiment, hypothesizing that dissociations in activity patterns compared to those seen in study 1 might emerge after inversion to agent-centric versus motor frames of reference (*fMRI GLM 3*). Indeed whole-brain analyses comparing the combined inverted basis function representations (see Fig.3G, green activity) in an independent samples t-test, revealed significantly stronger activations in the social study (Fig.5J; $z > 3.1$; $p = 0.05$ FWE-corrected) in the ventral striatum and lateral prefrontal cortex, and also in dmPFC ($z > 3.1$; $p = 0.05$ FWE-corrected, pre-threshold masked in sphere around partner-related activity from Fig.1E). Corresponding signals were absent in each of the three regions in the motor study (all $t_{31} < 0.575$, all $p > 0.570$, all $BF_{01} > 4.544$; Fig.3K). By contrast, activation was stronger in lateral primary motor regions in the motor study (Supplementary Fig.22). This suggests similar basis function-related computations across domains with differing neural implementations⁸, particularly at intermediate stages of processing. However, decision-related activity in vmPFC was present in both experiments (motor DV in frame of reference of choice; Fig.5L; $z > 3.1$; $p = 0.05$ FWE-corrected; compare with social DV in Fig.3G), even using the same ROI (Fig.5M). This indicates a common final pathway for decision-making across domains.

Figure 5. Control fMRI experiment revealed behavioural and neural signatures of basis functions (study 3). ... (J) Whole-brain analyses show that inverted, agent-centric basis functions in study 1 were significantly more strongly represented in dmPFC, ventral striatum (vStr), and lateral prefrontal cortex (IPFC) compared to inverted motor basis functions. The dmPFC cluster was pre-threshold masked ($z > 3.1$; $p = 0.05$, FWE) in a 20mm sphere around our independent dmPFC peak (partner signal in Fig.1, MNI: [-2 28 44]). (K) In ROI analyses of the motor study, inverted basis function signals in a motor frame of reference were absent in dmPFC, vStr and the IPFC. IPFC and bilateral vStr ROIs were taken from independent peaks for the same contrast in the social study (IPFC: [-44 42 4], vStr ROI combines [-6 14 -2] and [8 18 -2]; dmPFC: [-2 28 44]). (L) By contrast, the final DV (chosen – unchosen) was reflected in vmPFC activity in the same way in both motor and social tasks (compare Fig.3G) (M) These effects were also significant using an ROI derived from the social study (MNI: [-2 60 -12])

New supplementary figure panel detailing the inverted basis function effects in the motor study:

Supplementary Fig.22. Sub-threshold activation maps for social and basis function signals in the control fMRI experiment (study 3). (F) Cluster-significant inverted social basis functions in the social study. (G) The corresponding activation maps for the inverted motor basis functions (combining inverted primary and inverted secondary basis function across both decisions). The activation map is thresholded at $z > 2.0$ for illustration only. Note that the small white spheres are unbiased and independent masks from the social study related to the partner signal observed there (dmPFC mask, same as above) and related to the aggregate effects of the inverted social basis functions (IPFC: [-44 42 4], vStr ROI combines [-6 14 -2] and [8 18 -2]). As can be seen, inverted basis functions signals are clearly absent in and adjacent to these regions. Instead, the inverted basis functions in the motor study captured significant effects in the motor cortex, insular cortex and cerebellum. Note that these signals could indicate that inverted basis functions in the motor domain are processed in effector-related regions. However, they might also relate to the fixed response mapping utilised in the motor study (the left hand was always equivalent to one's own “team”) (N = 32).

Finally, as requested, to facilitate comparison with Study 1, we show the results of Figure 5 in the format in Figure 3C in a new supplementary figure:

Supplementary Fig.24. Basis function signals in frontal pole. These plots display the basis function signals in the frontal pole (MNI: [-14 56 24]). Note that we refrain from testing their significance in these panels, because we had already confirmed their average effect in our cluster-corrected whole brain analysis (see Fig.5). In addition, we also confirmed that there were no significant differences between the three basis functions (absence of main effect in 1-way ANOVA with b_1, b_2 and b_3) at any point of the timecourse (all $F_{2,62} < 1.479$; all $p > 0.236$).

9. fMRI issues: The authors were requested to show single subject fMRI data. It is disappointing that they do not do so in the revision. Instead, they simply state that this is “not consistent with best practice.” I strongly disagree. If one averages noisy or variable signals, then the common signals will sum and the variable signals will divide. So a study that only shows the group average is assuming, in effect, that there is no individual variability in brain activation patterns that is not merely due to noise. This is almost certainly a false assumption. Furthermore, there is information loss when mapping from individual brains to template space. The mapping algorithms are not perfect, and they perform worse in brain regions where there is relatively larger individual anatomical variability (e.g. *in the prefrontal cortex*). So any study that only shows the group average and not the individuals is effectively assuming that the mapping from individual brain space to the common template space is accurate. This is clearly a false assumption.

Ideally, authors should do the analysis in each individual subject and then report how many subjects show the reported pattern. The authors state, “We agree with R4 that [not showing individual subject data] means that there is a possibility that there may be additional patterns of neural activity that we do not describe because they are present in a small number of participants.” Due to individual variability, it could be the case that *all* the subjects show a pattern of activity that is quite different and much more diffuse from the average pattern that is shown currently. We cannot know without seeing the single subject data. Indeed, the highly variable behavioral results (e.g., Fig. 1C, D) could be due such variability across subjects.

It is factually incorrect to say that we omitted single participant data. It was included in supplementary figure 7 in the previous submission where it remains in the current submission.

10. dmPFC: While the manuscript now primarily focuses on pgACC, the role of dmPFC remains unclear. The authors initially report agent-specific signals in dmPFC (Fig. 1E) and later claim that dmPFC activity reflects social information without basis function activity (Supplementary Fig. 5 & 11). This inconsistency is confusing and requires clarification. It would be helpful to explicitly address the relationship and interactions between agent-specific representations in dmPFC and the basis function model in pgACC.

The dorsomedial prefrontal activation in figure 1E (blue area) is clearly far from activity in figure 3B. In case this is not obvious to a reader we point it out in the revised manuscript that the blue area of activity in 1E is over 3cm from the orange area in 3B and to emphasize the difference we note that the activity in 1E is in a dorsal part of dmPFC while the activity in 3B extends across dmPFC and pgACC.

Fig 1E:

Fig.3B

Figure 3. Basis functions in medial prefrontal cortex (study 1). (A) (B) dmPFC encoding the average effects of the projections onto the three basis functions and S-position in a region extending across dmPFC and pgACC. The signal peak in MNI=[2 48 18] was over 3 centimeter away from the dmPFC region in Fig.1

Minor issues:

1. Figure 3b: there is no color bar.

Colour bar is now added.

2. Is Figure 3B showing just activity in dmPFC or were all voxels across the whole brain included?

Figure 3B shows activity across the whole brain related to this contrast but only activity in the mPFC passes the cluster-corrected threshold. Note that we have already stated clearly in the manuscript that all whole-brain results are cluster-corrected using the most stringent significance criteria used in the field, unless otherwise stated. To remove any potential ambiguity, we have now also clarified at the end of the figure legend that we did not use masking methods for this analysis:

(n=56; study 1, social fMRI experiment; all MRI results cluster-corrected at $Z > 3.1$, **no masking was used and all voxels across the brain were included in the analysis**; $p = 0.05$; *, $p < 0.05$; **, $p < 0.01$; ***, $p < 0.001$).

3. There is no color bar in Figure 3G

As can be seen in Fig.3G, we are showing several cluster-significant effects on the same brain image in order to illustrate the distribution of effects. Note that we are using block colours, not colour gradients in the below image, for clarity. Below the brain images, the labels corresponding to the colours are clearly indicated. We are therefore very puzzled by the reviewer's request to provide a colour bar for a figure with no colour gradients. Nevertheless, we have now clarified in the figure legend what the blue colours mean. Note that we have moved the panel to the supplementary materials following suggestions from reviewer 2:

Supplementary Fig. 11. Cingulate cortex and frontal pole encode basis function projections using a sequential frame of reference. As explained in the main text, the basis functions can be sorted in primary, secondary, and tertiary basis function to aid decision making. With primary and secondary basis functions, we refer to the two decision-relevant basis functions that are present in dyadic decisions with the primary basis function becoming apparent during the observation phase, and the secondary one becoming apparent when the specific decision is cued. Mathematically, the regressor values for these variables always correspond to the values for the unsorted basis functions: b_1 , b_2 and b_3 . The primary basis function then is equivalent to the value of the one of the three that is given by the observation phase when the positions linked to the self's own group and the opponent group become apparent. The value for the secondary basis function is given by the value of the additional basis function projection that becomes apparent when the specific dyadic decision is cued. And the tertiary basis function is then simply defined as the remaining basis function projection. We found evidence that a large region along the cingulate sulcus, posterior to our previously identified contrast, encoded the combination of primary and secondary basis function projections (light blue activation). This is striking because these variables encode sequential information and are unrelated to agent-centric or decision-related variables, which we again control for in the analysis. For example, for the primary basis function encoding the difference between pairs of sequential positions, we included a control regressor indexing the same difference but in the social frame of reference of "own team vs other team". This effect became even more prominent after contrasting this combination with projections onto the tertiary basis function (dark blue activation). This comparison additionally engaged bilateral frontopolar cortex as computing the difference in relevant and currently irrelevant basis function. It is important to recall that participants performed two different decisions after seeing the performances. The current analysis was carried out at the time of the first of the two decisions following the observation phase. Hence, the tertiary basis projection, while not currently relevant for the decision in hand might become relevant for the second decision. This might explain why this contrast was linked to activity in frontal pole; frontal pole activity reflects information pertinent to future decisions⁴⁰. Importantly, these activity patterns, similar to previous analyses, reflected compressed, sequentially organized information. We controlled for corresponding agent-centric activations as well as any social decision-related activations in these analyses. By contrast, the social decision variable, encoded in the reference frame of choice, is encoded in vmPFC (same contrast as shown in main manuscript). The yellow activation is added for reference and reflects the contrast shown in Fig.3B. All activations (except the two blue ones) are non-overlapping. (n=56; study 1, social fMRI experiment; all MRI results cluster-corrected at $Z > 3.1$; $p = 0.05$ FWE).

4. In Figure 3I, J: the model predicts a specific sign of deviation (Fig 3I), while the data seems to go in both directions (Fig 3J), which could be easily explained by noise.

This is not correct. Our theoretical models predicts positive effects for one variable (“P”) and negative effects for the other variable (“Oi”). We find significant effects of both effects in precisely the predicted direction. When taking a measure of this overall effect (by calculating “P-Oi”), we find an effect size of Cohen’s $d=0.775$, which is close to the classification of $d=0.8$ being a large effect size. Below the plot in question. Given the evidence, this pattern of results cannot be explained by noise.

The strength of the effects in question can be seen more directly when zooming in (we predicted the second bar to be positive and the fourth to be negative).